# Large Language Model-Brained GUI Agents: A Survey

**Chaoyun Zhang**[1]*, **Shilin He**[1]*, **Jiaxu Qian**[1], **Bowen Li**[2], **Liqun Li**[1], **Si Qin**[1], **Yu Kang**[1], **Minghua Ma**[1], **Guyue Liu**[4], **Qingwei Lin**[1], **Saravan Rajmohan**[1], **Dongmei Zhang**[1], **Qi Zhang**[1]
*[1]Microsoft*    *[2]Shanghai Artificial Intelligence Laboratory*    *[3]Peking University*

**Reviewed on OpenReview:** *https://openreview.net/forum?id=xChvYjvXTp*

## Abstract

Graphical User Interfaces (GUIs) have long been central to human-computer interaction, providing an intuitive and visually-driven way to access and interact with digital systems. Traditionally, automating GUI interactions relied on script-based or rule-based approaches, which, while effective for fixed workflows, lacked the flexibility and adaptability required for dynamic, real-world applications. The advent of Large Language Models (LLMs), particularly multimodal models, has ushered in a new era of GUI automation. They have demonstrated exceptional capabilities in natural language understanding, code generation, task generalization, and visual processing. This has paved the way for a new generation of "LLM-brained" GUI agents capable of interpreting complex GUI elements and autonomously executing actions based on natural language instructions. These agents represent a paradigm shift, enabling users to perform intricate, multi-step tasks through simple conversational commands. Their applications span across web navigation, mobile app interactions, and desktop automation, offering a transformative user experience that revolutionizes how individuals interact with software. This emerging field is rapidly advancing, with significant progress in both research and industry.

To provide a structured understanding of this trend, this paper presents a comprehensive survey of LLM-powered GUI agents, exploring their historical evolution, core components, and advanced techniques. We address critical research questions such as existing GUI agent frameworks, the collection and utilization of data for training specialized GUI agents, the development of fine-tuned models tailored for GUI tasks, and the evaluation metrics and benchmarks necessary to assess their effectiveness. Additionally, we examine emerging applications powered by these agents. Through a detailed analysis, this survey identifies key research gaps and outlines a roadmap for future advancements in the field. By consolidating foundational knowledge and state-of-the-art developments, this work aims to guide both researchers and practitioners in overcoming challenges and unlocking the full potential of LLM-powered GUI agents. We anticipate that this survey will serve both as a practical cookbook for constructing LLM-powered GUI agents, and as a definitive reference for advancing research in this rapidly evolving domain.

The collection of papers reviewed in this survey will be hosted and regularly updated on the GitHub repository: https://github.com/vyokky/LLM-Brained-GUI-Agents-Survey. Additionally, a searchable webpage is available at https://aka.ms/gui-agent for easier access and exploration.

## 1 Introduction

Graphical User Interfaces (GUIs) have been a cornerstone of human-computer interaction, fundamentally transforming how users navigate and operate within digital systems Jansen (1998). Designed to make

---

*Chaoyun Zhang and Shilin He are corresponding authors: chaoyun.zhang@microsoft.com, shilin.he@microsoft.com

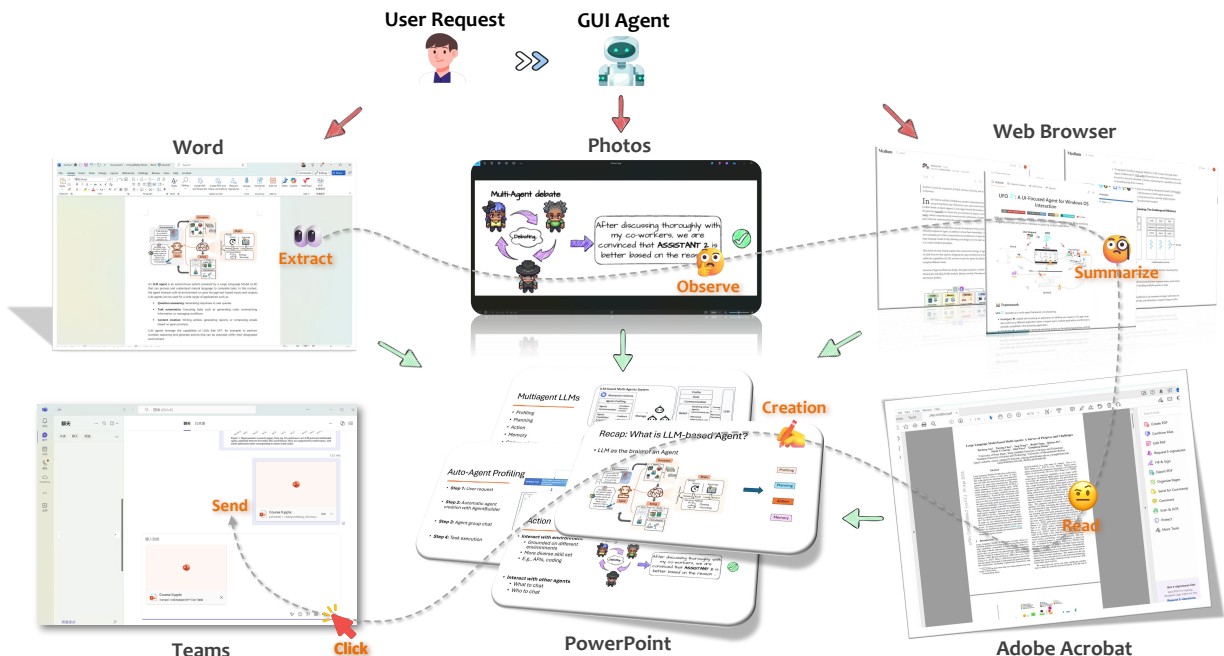

Figure 1: Illustration of the high-level concept of an LLM-powered GUI agent. The agent receives a user's natural language request and orchestrates actions seamlessly across multiple applications. It extracts information from Word documents, observes content in Photos, summarizes web pages in the browser, reads PDFs in Adobe Acrobat, and creates slides in PowerPoint before sending them through Teams.

computing more intuitive and accessible, GUIs replaced command-line interfaces (CLIs) Sampath et al. (2021) with visually driven, user-friendly environments. Through the use of icons, buttons, windows, and menus, GUIs empowered a broader range of users to interact with computers using simple actions such as clicks, typing, and gestures. This shift democratized access to computing, allowing even non-technical users to effectively engage with complex systems. However, GUIs often sacrifice efficiency for usability, particularly in workflows requiring repetitive or multi-step interactions, where CLIs can remain more streamlined Michalski et al. (2006).

While GUIs revolutionized usability, their design, primarily tailored for human visual interaction, poses significant challenges for automation. The diversity, dynamism, and platform-specific nature of GUI layouts make it difficult to develop flexible and intelligent automation tools capable of adapting to various environments. Early efforts to automate GUI interactions predominantly relied on script-based or rule-based methods Hellmann & Maurer (2011); Steven et al. (2000). Although effective for predefined workflows, these methods were inherently narrow in scope, focusing primarily on tasks such as software testing and robotic process automation (RPA) Ivančić et al. (2019). Their rigidity required frequent manual updates to accommodate new tasks, changes in GUI layouts, or evolving workflows, limiting their scalability and versatility. Moreover, these approaches lacked the sophistication needed to support dynamic, human-like interactions, thereby constraining their applicability in complex or unpredictable scenarios.

The rise of Large Language Models (LLMs)[1] Zhao et al. (2023b); Naveed et al. (2023), especially those augmented with multimodal capabilities Yin et al. (2023), has emerged as a game changer for GUI automation, redefining the the way agents interact with graphical user interfaces. Beginning with models like ChatGPT Wu et al. (2023c), LLMs have demonstrated extraordinary proficiency in natural language understanding, code generation, and generalization across diverse tasks Liu et al. (2024g); Shen (2024); Feng et al.; Zhao et al.

---

[1] By LLMs, we refer to the general concept of foundation models capable of accepting various input modalities (*e.g.*, visual language models (VLMs), multimodal LLMs (MLLMs)) while producing output exclusively in textual sequences contributors (2024).

(2023b). The integration of visual language models (VLMs) has further extended these capabilities, enabling these models to process visual data, such as the intricate layouts of GUIs Hong et al. (2023). This evolution bridges the gap between linguistic and visual comprehension, empowering intelligent agents to interact with GUIs in a more human-like and adaptive manner. By leveraging these advancements, LLMs and VLMs offer transformative potential, enabling agents to navigate complex digital environments, execute tasks dynamically, and revolutionize the field of GUI automation.

## 1.1   Motivation for LLM-Powered GUI agents

With a LLM serving as its cognitive core, LLM-powered GUI automation introduces a new class of intelligent agents capable of interpreting natural language instructions, analyzing GUI elements, and autonomously executing corresponding actions. These **LLM-powered GUI agents** operate without reliance on complex platform-specific scripts or hard-coded workflows, enabling a more flexible and generalizable approach to automation. We define them as:

> *Intelligent agents that operate within GUI environments, leveraging LLMs as their core inference and cognitive engine to generate, plan, and execute actions in a flexible and adaptive manner.*

This paradigm fosters dynamic, human-like interactions and real-time decision-making across diverse platforms, as illustrated in Figure 1. To summarize, the motivation behind LLM-powered GUI agents can be distilled into the following three key aspects:

**Overcoming the Limitations of Traditional Automation.**   Traditional GUI automation tools are typically constrained by rigid, rule-based logic and narrow task scopes. They struggle to generalize across diverse applications or adapt to dynamic environments. In contrast, LLM-powered GUI agents integrate natural language understanding, visual perception, and reasoning into a unified framework. This enables them to handle a broader range of tasks and react intelligently to context changes. Furthermore, unlike API-based agents—which depend on exposed or accessible interfaces—GUI agents interact with applications through their graphical frontends. This GUI-centric approach allows non-intrusive automation even in closed-source or legacy applications, significantly broadening the applicability of these agents across platforms and ecosystems.

**Democratizing Access Through Natural Interaction.**   By enabling users to control complex software systems using natural language, LLM-powered GUI agents lower the barrier to automation for non-technical users. They eliminate the need for programming skills or manual scripting, making multi-step operations more intuitive and accessible. Examples such as SeeAct Zheng et al. (2024a) for web navigation, AppAgent Zhang et al. (2023a) for mobile apps, and UFO Zhang et al. (2024a) for Windows automation demonstrate the growing breadth of use cases. Multi-platform agents like CogAgent CogAgent Team (2024) further extend these capabilities across heterogeneous environments. These systems are increasingly resembling intelligent virtual assistants Guan et al. (2023)—akin to J.A.R.V.I.S. from Iron Man—that understand user intent and perform tasks fluidly across applications.

**Emerging Real-World Adoption.**   LLM-powered GUI agents are rapidly transitioning from research to deployment. For instance, Microsoft Power Automate integrates LLMs to assist users in building automation workflows across applications using natural language[2]. Similarly, AI copilots in productivity software like Microsoft Copilot[3] are bridging the gap between user intent and application behavior. These agents also hold

---

[2] https://www.microsoft.com/en-us/power-platform/blog/power-automate/revolutionize-the-way-you-work-with-automation-and-ai/
[3] https://copilot.microsoft.com/

promise for improving accessibility—for example, enabling visually impaired users to control GUI interfaces using voice or text input Aljedaani et al. (2024).

The convergence of LLMs and GUI automation has sparked growing research interest, spanning topics such as application frameworks Zhang et al. (2024a), data collection and GUI grounding Cheng et al. (2024a), model optimization Hong et al. (2023), and evaluation methodologies Zhuge et al. (2024). This momentum reflects both the transformative potential and the technical complexity of the field. Despite recent advances, fundamental challenges remain—such as robust GUI grounding, error recovery, and generalization across unseen interfaces. However, a comprehensive survey of this rapidly evolving domain is still lacking, creating an urgent need for systematic synthesis and analysis. Our work aims to address this gap.

## 1.2 Scope of the Survey

To address this gap, this paper provides a pioneering, comprehensive survey of LLM-powered GUI agents. We cover the historical evolution of GUI agents, provide a step-by-step guide to building these agents, summarize essential and advanced techniques, review notable tools and research related to frameworks, data and models, showcase representative applications, and outline future directions. Specifically, this survey aims to answer the following research questions (RQs):

1. **RQ1:** What are the key background, enabling factors, and unique characteristics of LLM-powered GUI agents? (Section 3)

2. **RQ2:** What are the essential components and advanced technologies that form the foundation of LLM-powered GUI agents? (Section 4)

3. **RQ3:** What are the principal frameworks for LLM GUI agents across web, mobile, computer and multi-platform and what are their defining characteristics? (Section 5)

4. **RQ4:** What are the existing datasets, and how can comprehensive datasets be collected to train optimized LLMs for GUI agents? (Section 6)

5. **RQ5:** How can the collected data be used to train purpose-built Large Action Models (LAMs)[4] for GUI agents, and what are the current leading models in the field? (Section 7)

6. **RQ6:** What metrics and benchmarks are used to evaluate the capability and performance of GUI agents? (Section 8)

7. **RQ7:** What are the most significant real-world applications of LLM-powered GUI agents, and how have they been adapted for practical use? (Section 9)

8. **RQ8:** What are the major challenges, limitations, and future research directions for developing robust and intelligent GUI agents? (Section 10)

Through these questions, this survey aims to provide a comprehensive overview of the current state of the field, offer a guide for building LLM-powered GUI agents, identify key research gaps, and propose directions for future work. This survey is one of the pioneers to systematically examine the domain of LLM-powered GUI agents, integrating perspectives from LLM advancements, GUI automation, and human-computer interaction. To start, we present a structural overview in Figure 2. For ease of reference, a list of abbreviations used throughout the paper is provided in Table 1.

## 2 Related Work

The integration of LLMs with GUI agents is an emerging and rapidly growing field of research. Several related surveys and tutorials provide foundational insights and guidance. We provide a brief review of existing

---

[4]In our context, LAM denotes a fine-tuned large language or vision-language model that is specifically tailored for executing GUI agent tasks.

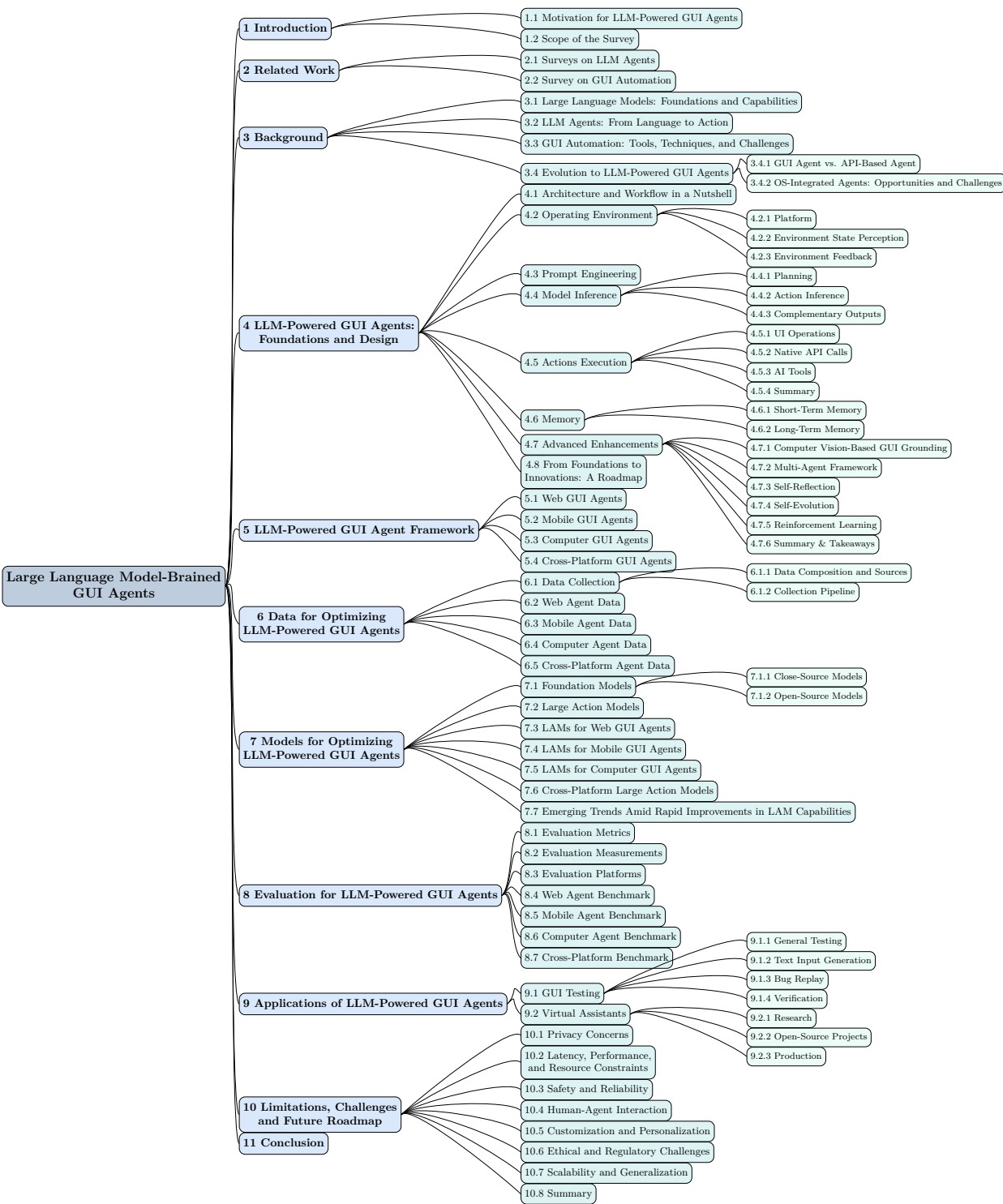

Figure 2: The structure of the survey on LLM-powered GUI agents.

overview articles on GUI automation and LLM agents, as these topics closely relate to and inform our research focus. To begin, we provide an overview of representative surveys and books on GUI automation, LLM agents, and their integration, as summarized in Table 2. These works either directly tackle one or two core areas in GUI automation and LLM-driven agents, or provide valuable insights that, while not directly addressing the topic, contribute indirectly to advancing the field.

Table 1: List of abbreviations in alphabetical order.

| Acronym | Explanation |
|---|---|
| AI | Artificial Intelligence |
| AITW | Android in the Wild |
| AITZ | Android in The Zoo |
| API | Application Programming Interface |
| CLI | Command-Line Interface |
| CLIP | Contrastive Language-Image Pre-Training |
| CoT | Chain-of-Thought |
| CSS | Cascading Style Sheets |
| CuP | Completion under Policy |
| CV | Computer Vision |
| DOM | Document Object Model |
| DPO | Direct Preference Optimization |
| GCC | General Computer Control |
| GPT | Generative Pre-trained Transformers |
| GUI | Graphical User Interface |
| HCI | Human-Computer Interaction |
| HTML | Hypertext Markup Language |
| ICL | In-Context Learning |
| IoU | Intersection over Union |
| LAM | Large Action Model |
| LLM | Large Language Model |
| LSTM | Long Short-Term Memory |
| LTM | Long-Term Memory |
| MCTS | Monte Carlo Tree Search |
| MoE | Mixture of Experts |
| MDP | Markov Decision Process |
| MLLM | Multimodal Large Language Model |
| OCR | Optical Character Recognition |
| OS | Operation System |
| RAG | Retrieval-Augmented Generation |
| ReAct | Reasoning and Acting |
| RL | Reinforcement Learning |
| RLHF | Reinforcement Learning from Human Feedback |
| RNN | Recurrent Neural Network |
| RPA | Robotic Process Automation |
| UI | User Interface |
| UX | User Experience |
| VAB | VisualAgentBench |
| VLM | Visual Language Models |
| ViT | Vision Transformer |
| VQA | Visual Question Answering |
| SAM | Segment Anything Model |
| SoM | Set-of-Mark |
| STM | Short-Term Memory |

Table 2: Summary of representative surveys and books on GUI automation and LLM agents. A ✓symbol indicates that a publication explicitly addresses a given domain, while an ◯ symbol signifies that the publication does not focus on the area but offers relevant insights. Publications covering both GUI automation and LLM agents are highlighted for emphasis.

| Survey | One Sentence Summary | Scope | | |
|---|---|---|---|---|
| | | GUI Automation | LLM Agent | LLM Agent + GUI Automation |
| Li & Wu (2006) | A book on how to develop an automated GUI testing tool. | ✓ | | |
| Rodríguez-Valdés et al. (2021) | A survey on automated GUI testing in 30 years. | ✓ | | |
| Arnatovich & Wang (2018) | A survey on automated techniques for mobile functional GUI testing. | ✓ | | |
| *et al.*, Ivančić et al. (2019) | A literature review on RPA. | ✓ | | |
| Said et al. (2020) | An overview on mobile GUI testing. | ✓ | | |
| Li (2023) | An survey on Android GUI testing. | ✓ | | |
| Oksanen (2023) | GUI testing on Windows OS. | ✓ | | |
| Deshmukh et al. (2023) | A survey on GUI testing for improving user experience. | ✓ | | |
| Bajammal et al. (2020) | A survey on the use of computer vision for software engineering. | ✓ | | |
| Yu et al. (2023) | A survey on using computer for mobile app GUI testing. | ✓ | | |
| Syed et al. (2020) | A review of contemporary themes and challenges in RPA. | ✓ | | |
| Chakraborti et al. (2020) | A review of emerging trends of intelligent process automation. | ✓ | | |
| Enríquez et al. (2020) | A scientific and industrial systematic mapping study of RPA. | ✓ | | |
| Ribeiro et al. (2021) | A review of combining AI and RPA in industry 4.0. | ✓ | | |
| Nass et al. (2021) | Discuss the chanllenges of GUI testing. | ✓ | | |
| Agostinelli et al. (2019) | Discuss the research challenges of intelligent RPA. | ✓ | | |
| Wali et al. (2023) | A review on task automation with intelligent agents. | ✓ | | |
| Zhao et al. (2023b) | A comprehensive survey of LLMs. | | ✓ | |
| Zhao et al. (2023a) | A survey of LLM-based agents. | | ✓ | |
| Cheng et al. (2024b) | An overview of LLM-based AI agent. | | ✓ | |
| Li et al. (2024i) | A survey on personal LLM agents on their capability, efficiency and security. | | ✓ | |
| Xi et al. (2023) | A comprehensive survey of LLM-based agents. | | ✓ | |
| Wang et al. (2024g) | A survey on LLM-based autonomous agents. | | ✓ | |
| Guo et al. (2024b) | A survey of mult-agent LLM frameworks. | | ✓ | |
| Han et al. (2024) | A survey on LLM multi-agent systems, with their challenges and open problems. | | ✓ | |
| Sun et al. (2024a) | A survey on LLM-based multi-agent reinforcement learning. | | ✓ | |
| Huang et al. (2024b) | A survey on planning in LLM agents. | | ✓ | |
| Aghzal et al. (2025) | A survey on automated planning in LLMs. | | ✓ | |
| Zheng et al. (2025c) | Discuss the roadmap of lifelong learning in LLM agents. | | ✓ | |
| Zhang et al. (2024q) | A survey on the memory of LLM-based agents. | | ✓ | |
| Shen (2024) | A survey of the tool usage in LLM agents. | | ✓ | |
| Chang et al. (2024) | A survey on evaluation of LLMs. | | ✓ | |
| Li et al. (2024d) | A survey on benchmarks multimodal applications. | | ✓ | |
| Li et al. (2025d) | A survey on benchmarking evaluations, applications, and challenges of visual LLMs. | | ✓ | |
| Huang & Zhang (2024) | A survey on evaluation of multimodal LLMs. | | ✓ | |
| Xie et al. (2024a) | A survey on LLM based multimodal agent. | | ✓ | ◯ |
| Durante et al. (2024) | A survey of multimodal interaction with AI agents. | | ✓ | ◯ |
| Wu et al. (2024a) | A survey of foundations and trend on multimodal mobile agents. | | ✓ | ✓ |
| Wang et al. (2024k) | A survey on the integration of foundation models with GUI agents. | | ✓ | ✓ |
| Gao et al. (2024f) | A survey on autonomous agents across digital platforms. | | ✓ | ✓ |
| Nguyen et al. (2024) | A survey on GUI agents. | | ✓ | ✓ |
| Liu et al. (2025b) | A survey on GUI agent on phone automation. | | ✓ | ✓ |
| Hu et al. (2024b) | A survey on MLLM based agents for OS. | | ✓ | ✓ |
| Shi et al. (2025) | A survey of building trustworthy GUI agents. | | ✓ | ✓ |
| Ning et al. (2025) | A survey of agents for Web automation. | | ✓ | ✓ |
| Tang et al. (2025b) | A survey of GUI agents powered by (multimodal) LLMs. | | ✓ | ✓ |
| Sager et al. (2025) | A review of AI agent for computer use. | ◯ | ✓ | ✓ |
| **Our work** | **A comprehensive survey on LLM-brained GUI agents, on their foundations, technologies, frameworks, data, models, applications, challenges and future roadmap.** | ◯ | ✓ | ✓ |

## 2.1 Survey on GUI Automation

GUI automation has a long history and wide applications in industry, especially in GUI testing Li & Wu (2006); Rodríguez-Valdés et al. (2021); Arnatovich & Wang (2018) and RPA Ivančić et al. (2019) for task automation Wali et al. (2023).

Said *et al.*, Said et al. (2020) provide an overview of GUI testing for mobile applications, covering objectives, approaches, and challenges within this domain. Focusing on Android applications, Li Li (2023) narrows the scope further, while Oksanen *et al.*, Oksanen (2023) explore automatic testing techniques for Windows GUI applications, a key platform for agent operations. Similarly, Moura *et al.*, Moura et al. (2023) review GUI testing for web applications, which involves diverse tools, inputs, and methodologies. Deshmukh *et al.*, Deshmukh et al. (2023) discuss automated GUI testing for enhancing user experience, an area where LLMs also bring new capabilities. A cornerstone of modern GUI testing is computer vision (CV), which is used to interpret UI elements and identify actionable controls Bajammal et al. (2020). Yu *et al.*, Yu et al. (2023) survey the application of CV in mobile GUI testing, highlighting both its significance and associated challenges. In LLM-powered GUI agents, application UI screenshots are equally essential, serving as key inputs for reliable task comprehension and execution.

On the other hand, RPA, which focuses on automating repetitive human tasks, also relies heavily on GUI automation for relevant processes. Syed *et al.*, Syed et al. (2020) review this field and highlight contemporary RPA themes, identifying key challenges for future research. Chakraborti *et al.*, Chakraborti et al. (2020) emphasize the importance of shifting from traditional, script-based RPA toward more intelligent, adaptive paradigms, offering a systematic overview of advancements in this direction. Given RPA's extensive industrial applications, Enriquez *et al.*, Enríquez et al. (2020) and Ribeiro *et al.*, Ribeiro et al. (2021) survey the field from an industrial perspective, underscoring its significance and providing a comprehensive overview of RPA methods, development trends, and practical challenges.

Both GUI testing Nass et al. (2021) and RPA Agostinelli et al. (2019) continue to face significant challenges in achieving greater intelligence and robustness. LLM-powered GUI agents are poised to play a transformative role in these fields, providing enhanced capabilities and adding substantial value to address these persistent issues.

## 2.2 Surveys on LLM Agents

The advent of LLMs has significantly enhanced the capabilities of intelligent agents Zhao et al. (2023a), enabling them to tackle complex tasks previously out of reach, particularly those involving natural language understanding and code generation Cheng et al. (2024b). This advancement has spurred substantial research into LLM-based agents designed for a wide array of applications Li et al. (2024i).

Both Xie *et al.*, Xi et al. (2023) and Wang *et al.*, Wang et al. (2024g) offer comprehensive surveys on LLM-powered agents, covering essential background information, detailed component breakdowns, taxonomies, and various applications. These surveys serve as valuable references for a foundational understanding of LLM-driven agents, laying the groundwork for further exploration into LLM-based GUI agents. Xie *et al.*, Xie et al. (2024a) provide an extensive overview of multimodal agents, which can process images, videos, and audio in addition to text. This multimodal capability significantly broadens the scope beyond traditional text-based agents Durante et al. (2024). Notably, most GUI agents fall under this category, as they rely on image inputs, such as screenshots, to interpret and interact with graphical interfaces effectively. Multi-agent frameworks are frequently employed in the design of GUI agents to enhance their capabilities and scalability. Surveys by Guo *et al.*, Guo et al. (2024b) and Han *et al.*, Han et al. (2024) provide comprehensive overviews of the current landscape, challenges, and future directions in this area. Sun *et al.*, Sun et al. (2024a) provide an overview of recent methods that leverage reinforcement learning to strengthen multi-agent LLM systems, opening new pathways for enhancing their capabilities and adaptability. These surveys offer valuable insights and guidance for designing effective multi-agent systems within GUI agent frameworks.

In the realm of digital environments, Wu *et al.*, Wu et al. (2024a) presents a survey on LLM agents operating in mobile environments, covering key aspects of mobile GUI agents. In a boarder scope, Wang *et al.*, Wang et al. (2024k) present a survey on the integration of foundation models with GUI agents. Another survey by Gao *et al.*, provides an overview of autonomous agents operating across various digital platforms Gao et al. (2024f), highlighting their capabilities, challenges, and applications. All these surveys highlighting emerging trends in this area.

Regarding individual components within LLM agents, several surveys provide detailed insights that are especially relevant for GUI agents. Huang *et al.*, Huang et al. (2024b) examine planning mechanisms in

LLM agents, which are essential for executing long-term tasks—a frequent requirement in GUI automation. Zhang *et al.*, Zhang et al. (2024q) explore memory mechanisms, which allow agents to store critical historical information, aiding in knowledge retention and decision-making. Additionally, Shen Shen (2024) surveys the use of tools by LLMs (such as APIs and code) to interact effectively with their environments, grounding actions in ways that produce tangible impacts. Further, Chang *et al.*, Chang et al. (2024) provide a comprehensive survey on evaluation methods for LLMs, which is crucial for ensuring the robustness and safety of GUI agents. Two additional surveys, Li et al. (2024d) and Huang & Zhang (2024), provide comprehensive overviews of benchmarks and evaluation methods specifically tailored to multimodal LLMs. The evaluation also facilitates a feedback loop, allowing agents to improve iteratively based on assessment results. Together, these surveys serve as valuable resources, offering guidance on essential components of LLM agents and forming a foundational basis for LLM-based GUI agents.

Compared to existing surveys, our work offers a significantly more comprehensive and up-to-date overview of the LLM-powered GUI agent landscape. We curate and synthesize over 500 references, covering a wide range of topics including foundation models, data sources, system frameworks, benchmarks, evaluation methodologies, and practical deployments. While prior surveys often concentrate on narrower aspects on selected platform (*e.g.*, web, mobile), our survey takes a holistic perspective that spans the full development and deployment lifecycle. Beyond narrative summaries, we also provide consolidated reference tables for each subdomain, enabling readers to quickly categorize and locate relevant works across platforms and research themes—serving as a practical handbook for both researchers and practitioners. Furthermore, we incorporate foundational background material and propose evaluation taxonomies that make the survey accessible to newcomers, addressing gaps in prior work that often assume a high degree of prior familiarity.

## 3 Background

The development of LLM-powered GUI agents is grounded in three major advancements: *(i)* large language models (LLMs) Zhao et al. (2023b), which bring advanced capabilities in natural language understanding and code generation, forming the core intelligence of these agents; *(ii)* accompanying agent architectures and tools Wang et al. (2024g) that extend LLM capabilities, bridging the gap between language models and physical environments to enable tangible impacts; and *(iii)* GUI automation Yeh et al. (2009), which has cultivated a robust set of tools, models, and methodologies essential for GUI agent functionality. Each of these components has played a critical role in the emergence of LLM-powered GUI agents. In the following subsections, we provide a brief overview of these areas to set the stage for our discussion.

### 3.1 Large Language Models: Foundations and Capabilities

The study of language models has a long and rich history Shannon (1951), beginning with early statistical language models Cavnar et al. (1994) and smaller neural network architectures Chung et al. (2014). Building on these foundational concepts, recent advancements have focused on transformer-based LLMs, such as the Generative Pre-trained Transformers (GPTs) Mann et al. (2020). These models are pretrained on extensive text corpora and feature significantly larger model sizes, validating scaling laws and demonstrating exceptional capabilities across a wide range of natural language tasks. Beyond their sheer size, these LLMs exhibit enhanced language understanding and generation abilities, as well as emergent properties that are absent in smaller-scale language models Wei et al. (2021).

Early neural language models, based on architectures like recurrent neural networks (RNNs) Medsker et al. (2001) and long short-term memory networks (LSTMs) Hochreiter (1997), were limited in both performance and generalization. The introduction of the Transformer model, built on the attention mechanism Vaswani (2017), marked a transformative milestone, establishing the foundational architecture now prevalent across almost all subsequent LLMs. This development led to variations in model structures, including encoder-only models (*e.g.*, BERT Devlin (2018), RoBERTa Liu (2019), ALBERT Lan (2019)), decoder-only models (*e.g.*, GPT-1 Radford (2018), GPT-2 Radford et al. (2019)), and encoder-decoder models (*e.g.*, T5 Raffel et al. (2020), BART Lewis (2019)). In 2022, ChatGPT Wu et al. (2023c) based on GPT-3.5 Ouyang et al. (2022) launched as a groundbreaking LLM, fundamentally shifting perceptions of what language models can achieve. Since then, numerous advanced LLMs have emerged, including GPT-4 Achiam et al. (2023), LLaMA-3 Dubey

et al. (2024), and Gemini Team et al. (2023), propelling the field into rapid growth. Today's LLMs are highly versatile, with many of them are capable of processing multimodal data and performing a range of tasks, from question answering to code generation, making them indispensable tools in various applications Hurst et al. (2024); Jiang et al. (2024); Zhang et al. (2024b); Liu et al. (2024d).

The emergence of LLMs has also introduced significant advanced properties that invigorate their applications, making previously challenging tasks, such as natural language-driven GUI agents feasible. These advancements include:

1. **Few-Shot Learning Mann et al. (2020):** Also referred to as in-context learning Dong et al. (2022), LLMs can acquire new tasks from a small set of demonstrated examples presented in the prompt during inference, eliminating the need for retraining. This capability is crucial for enabling GUI agents to generalize across different environments with minimal effort.

2. **Instruction Following Zhang et al. (2023c):** After undergoing instruction tuning, LLMs exhibit a remarkable ability to follow instructions for novel tasks, demonstrating strong generalization skills Ouyang et al. (2022). This allows LLMs to effectively comprehend user requests directed at GUI agents and to follow predefined objectives accurately.

3. **Long-Term Reasoning Huang & Chang (2022):** LLMs possess the ability to plan and solve complex tasks by breaking them down into manageable steps, often employing techniques like chain-of-thought (CoT) reasoning Wei et al. (2022); Ding et al. (2023). This capability is essential for GUI agents, as many tasks require multiple steps and a robust planning framework.

4. **Code Generation and Tool Utilization Chen et al. (2021a):** LLMs excel in generating code and utilizing various tools, such as APIs Shen (2024). This expertise is vital, as code and tools form the essential toolkit for GUI agents to interact with their environments.

5. **Multimodal Comprehension Yin et al. (2023):** Advanced LLMs can integrate additional data modalities, such as images, into their training processes, evolving into multimodal models. This ability is particularly important for GUI agents, which must interpret GUI screenshots presented as images in order to function effectively White et al. (2019).

To further enhance the specialization of LLMs for GUI agents, researchers often fine-tune these models with domain-specific data, such as user requests, GUI screenshots, and action sequences, thereby increasing their customization and effectiveness. In Section 7, we delve into these advanced, tailored models for GUI agents, discussing their unique adaptations and improved capabilities for interacting with graphical interfaces.

### 3.2 LLM Agents: From Language to Action

Traditional AI agents have often focused on enhancing specific capabilities, such as symbolic reasoning or excelling in particular tasks like Go or Chess. In contrast, the emergence of LLMs has transformed AI agents by providing them with a natural language interface, enabling human-like decision-making capabilities, and equipping them to perform a wide variety of tasks and take tangible actions in diverse environments Wang et al. (2024g); Kim et al. (2023); Liu et al. (2024g); Qiao et al. (2023). In LLM agents, if LLMs form the "brain" of a GUI agent, then its accompanying components serve as its "eyes and hands", enabling the LLM to perceive the environment's status and translate its textual output into actionable steps that generate tangible effects Xi et al. (2023). These components transform LLMs from passive information sources into interactive agents that execute tasks on behalf of users, which redefine the role of LLMs from purely text-generative models to systems capable of driving actions and achieving specific goals.

In the context of GUI agents, the agent typically perceives the GUI status through screenshots and widget trees Boshart & Kosa (2003), then performs actions to mimic user operations (*e.g.*, mouse clicks, keyboard inputs, touch gestures on phones) within the environment. Since tasks can be long-term, effective planning and task decomposition are often required, posing unique challenges. Consequently, an LLM-powered GUI agent usually possess multimodal capabilities Xie et al. (2024a), a robust planning system Huang et al.

(2024b), a memory mechanism to analyze historical interactions Zhang et al. (2024q), and a specialized toolkit to interact with its environment Li & Wu (2006). We will discuss these tailored designs for GUI agents in detail in Section 4.

### 3.3 GUI Automation: Tools, Techniques, and Challenges

GUI automation has been a critical area of research and application since the early days of GUIs in computing. Initially developed to improve software testing efficiency, GUI automation focused on simulating user actions, such as clicks, text input, and navigation, across graphical applications to validate functionality Said et al. (2020). Early GUI automation tools were designed to execute repetitive test cases on static interfaces Rodríguez-Valdés et al. (2021). These approaches streamlined quality assurance processes, ensuring consistency and reducing manual testing time. As the demand for digital solutions has grown, GUI automation has expanded beyond testing to other applications, including RPA Ivančić et al. (2019) and Human-Computer Interaction (HCI) Li & Hilliges (2021). RPA leverages GUI automation to replicate human actions in business workflows, automating routine tasks to improve operational efficiency. Similarly, HCI research employs GUI automation to simulate user behaviors, enabling usability assessments and interaction studies. In both cases, automation has significantly enhanced productivity and user experience by minimizing repetitive tasks and enabling greater system adaptability Abuaddous et al. (2022); Gao et al. (2024b).

Traditional GUI automation methods have primarily depended on scripting and rule-based frameworks Hellmann & Maurer (2011); Qian et al. (2020). Scripting-based automation utilizes languages such as Python, Java, and JavaScript to control GUI elements programmatically. These scripts simulate a user's actions on the interface, often using tools like Selenium Bruns et al. (2009) for web-based automation or AutoIt Rupp et al. (2022) and SikuliX Granda et al. (2021) for desktop applications. Rule-based approaches, meanwhile, operate based on predefined heuristics, using rules to detect and interact with specific GUI elements based on properties such as location, color, and text labels Hellmann & Maurer (2011). While effective for predictable, static workflows Xu et al. (2024c), these methods struggle to adapt to the variability of modern GUIs, where dynamic content, responsive layouts, and user-driven changes make it challenging to maintain rigid, rule-based automation Gove & Faytong (2012).

CV has become essential for interpreting the visual aspects of GUIs Yu et al. (2023); Li et al. (2021); Chang et al. (2010), enabling automation tools to recognize and interact with on-screen elements even as layouts and designs change. CV techniques allow GUI automation systems to detect and classify on-screen elements, such as buttons, icons, and text fields, by analyzing screenshots and identifying regions of interest Zou et al. (2023); Ye et al. (2021); Chen et al. (2020). Optical Character Recognition (OCR) further enhances this capability by extracting text content from images, making it possible for automation systems to interpret labels, error messages, and form instructions accurately Qian et al. (2022). Object detection models add robustness, allowing automation agents to locate GUI elements even when the visual layout shifts White et al. (2019). By incorporating CV, GUI automation systems achieve greater resilience and adaptability in dynamic environments.

Despite advances, traditional GUI automation methods fall short in handling the complexity and variability of contemporary interfaces. Today's applications often feature dynamic, adaptive elements that cannot be reliably automated through rigid scripting or rule-based methods alone Gambino et al. (2018); He et al. (2008). Modern interfaces increasingly require contextual awareness Stefanidi et al. (2022), such as processing on-screen text, interpreting user intent, and recognizing visual cues. These demands reveal the limitations of existing automation frameworks and the need for more flexible solutions capable of real-time adaptation and context-sensitive responses.

LLMs offer a promising solution to these challenges. With their capacity to comprehend natural language, interpret context, and generate adaptive scripts, LLMs can enable more intelligent, versatile GUI automation Liu et al. (2024k). Their ability to process complex instructions and learn from context allows them to bridge the gap between static, rule-based methods and the dynamic needs of contemporary GUIs Brie et al. (2023). By integrating LLMs with GUI agents, these systems gain the ability to generate scripts on-the-fly based on the current state of the interface, providing a level of adaptability and sophistication that traditional methods cannot achieve. The combination of LLMs and GUI agents paves the way for an advanced, user-centered

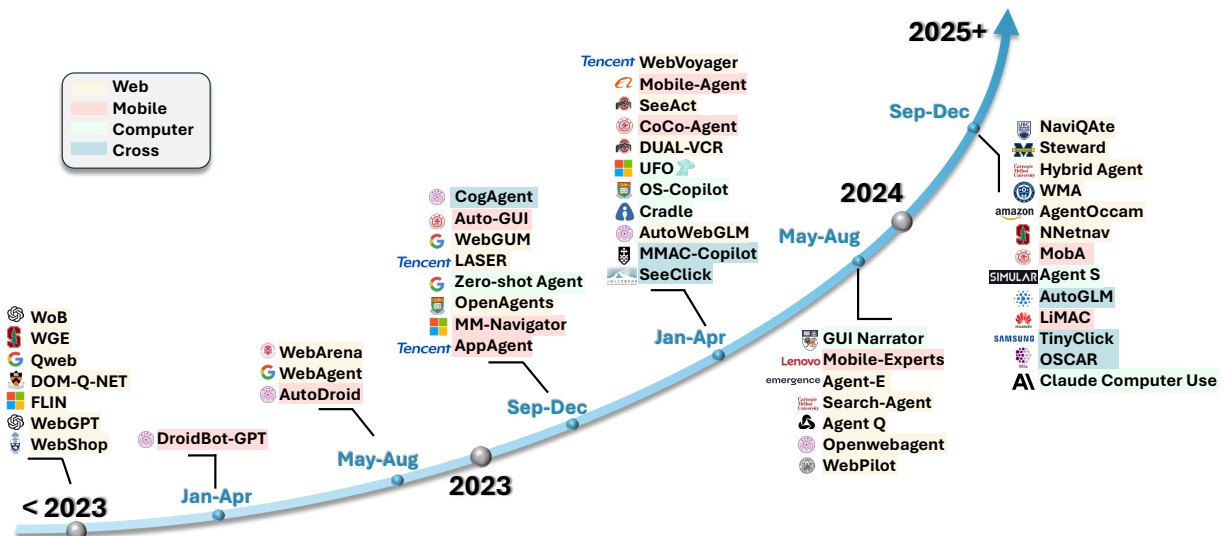

Figure 3: An overview of GUI agents evolution by year and platforms.

automation paradigm, capable of responding flexibly to user requests and interacting seamlessly with complex, evolving interfaces.

### 3.4 Evolution to LLM-Powered GUI Agents

As illustrated in Figure 3, prior to 2023 and the emergence of LLMs, work on GUI agents was limited in both scope and capability. Since then, the proliferation of LLM-based approaches has fostered numerous notable developments across platforms including web, mobile, and desktop environments. This surge is ongoing and continues to drive innovation in the field. The key takeaway is that GUI agents are not only growing in number but also rapidly advancing in terms of data, models, frameworks, and applications. This section takes you on a journey tracing the evolution of GUI agents, emphasizing key milestones that have brought the field to its present state.

### 3.4.1 GUI Agent vs. API-Based Agent

In the field of LLM-powered agents operating within digital environments, the action space can be broadly categorized into two types:

1. **GUI Agents**, which primarily rely on GUI operations (*e.g.*, clicks, keystrokes) to complete tasks.

2. **API-Based Agents**, which utilize system or application-native APIs to fulfill objectives.

We show the principle of both agent types in Figure 4. Each type has distinct advantages, and a deeper understanding of these approaches is critical for designing effective agents.

GUI operations provide a **universal control interface** that can operate across diverse applications using the same action primitives. This makes GUI agents highly generalizable, as they can interact with a wide range of software environments without requiring application-specific adaptations. However, GUI-based interactions are inherently more complex; even simple tasks may require multiple sequential steps, which can increase both the decision-making cost for the agent and the computational resources required for long-term, multi-step workflows. Another key aspect is the transparency of actions in GUI agents. Since GUI agents interact with applications in the same way a human would, by clicking, typing, and navigating through the interface, their actions are inherently more observable and interpretable to users. This transparency fosters better trust and comprehension in agent-computer interactions.

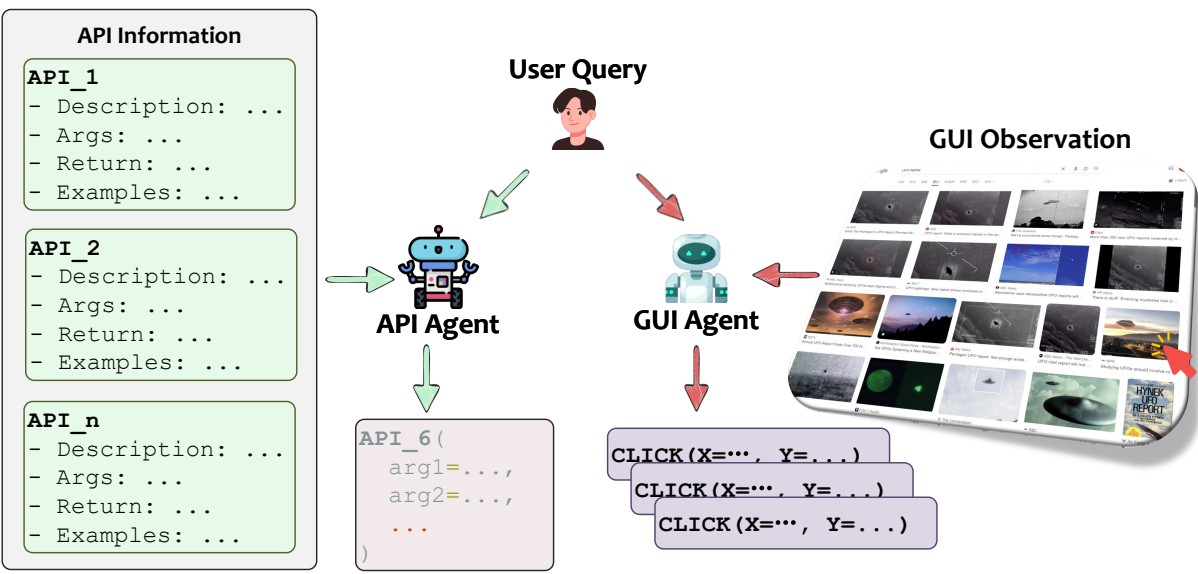

Figure 4: The comparison between API agent vs. GUI agent.

In contrast, API-based agents offer a more **efficient and direct approach** to task completion. By leveraging native APIs, tasks can often be fulfilled with a single, precise call, significantly reducing execution time and complexity. However, these native APIs are often private or restricted to specific applications, limiting accessibility and generalizability. This makes API-based agents less versatile in scenarios where API access is unavailable or insufficient. In addition, API-based agents operate behind the scenes, executing tasks through direct system calls, which, while often more efficient and reliable, can make their operations less visible and harder to debug for end users.

The most effective digital agents are likely to operate in a **hybrid manner**, combining the strengths of both approaches. Such agents can utilize GUI operations to achieve broad compatibility across software while exploiting native APIs where available to maximize efficiency and effectiveness. These hybrid agents strike a balance between generalization and task optimization, making them a **critical focus area in this survey**. For a more comprehensive comparison between GUI agents and API agents, please refer to Zhang et al. (2025a).

### 3.4.2 OS-Integrated Agents: Opportunities and Challenges

As agent capabilities continue to advance, we are witnessing the emergence of OS-integrated assistants such as *Apple Intelligence* (built into iOS/macOS) and *Microsoft Copilot* (integrated across Windows and Microsoft 365). These systems exemplify a growing trend in which foundational models are embedded directly into the operating system, offering native capabilities for task execution, summarization, and user assistance.

While these assistants represent significant progress in bringing foundation models closer to users, their functionality is largely shaped by API-based control paradigms. They are optimized for environments where fine-grained APIs are available and sanctioned, enabling streamlined and secure task execution. However, as discussed in Section 3.4.1, this model imposes inherent limitations: many third-party or legacy applications lack sufficient API exposure, and system-level privileges are often restricted, especially in closed ecosystems like iOS. This restricts the agent's ability to generalize across diverse software landscapes.

GUI-based agents offer a compelling complement in this context. By simulating human interaction with visual interfaces, GUI agents can extend automation capabilities to software that is otherwise inaccessible to OS-level API agents. This makes them especially valuable in scenarios where application internals are opaque, APIs are limited or unstable, or extensibility across software is desired. In essence, GUI agents serve as a universal fallback mechanism—enabling broader coverage and versatility without requiring privileged access

or deep integration. The growing presence of OS-level agents amplifies the importance of developing hybrid agent architectures that combine the precision and efficiency of APIs with the generality and accessibility of GUI-based control. For example, an assistant like Microsoft Copilot could selectively use GUI-based interactions for third-party or legacy apps while relying on APIs for native tools like Word or Excel. Such hybridization can enhance task coverage, robustness, and user experience across heterogeneous environments.

Looking forward, we envision increasing pressure for these integrated systems to support modular and extensible agent interfaces, allowing third-party developers and researchers to build on top of foundation agents with GUI-based capabilities. This direction not only broadens the deployment landscape for LLM-powered agents but also fosters a richer ecosystem of intelligent assistants that are adaptable to real-world variability in software availability, permissions, and user needs.

# 4 LLM-Brained GUI Agents: Foundations and Design

In essence, LLM-powered GUI agents are designed to process user instructions or requests given in natural language, interpret the current state of the GUI through screenshots or GUI element trees, and execute actions that simulate human interaction across various software interfaces Zhang et al. (2024a). These agents harness the sophisticated natural language understanding, reasoning, and generative capabilities of LLMs to accurately comprehend user intent, assess the GUI context, and autonomously engage with applications across diverse environments, thereby enabling the completion of complex, multi-step tasks. This integration allows them to seamlessly interpret and respond to user requests, bringing adaptability and intelligence to GUI automation.

As a specialized type of LLM agent, most current GUI agents adopt a similar foundational framework, integrating core components such as planning, memory, tool usage, and advanced enhancements like multi-agent collaboration, among others Wang et al. (2024g). However, each component must be tailored to meet the specific objectives of GUI agents to ensure adaptability and functionality across various application environments.

In the following sections, we provide an in-depth overview of each component, offering a practical guide and tutorial on building an LLM-powered GUI agent from the ground up. This comprehensive breakdown serves as a cookbook for creating effective and intelligent GUI automation systems that leverage the capabilities of LLMs.

## 4.1 Architecture and Workflow In a Nutshell

In Figure 5, we present the architecture of an LLM-brained GUI agent, showcasing the sequence of operations from user input to task completion. The architecture comprises several integrated components, each contributing to the agent's ability to interpret and execute tasks based on user-provided natural language instructions. Upon receiving a user request, the agent follows a systematic workflow that includes environment perception, prompt engineering, model inference, action execution, and continuous memory utilization until the task is fully completed. It is important to recognize that user–agent interaction can be bi-directional, as opposed to a passive, one-shot task execution. GUI agents may initiate communication with users to request clarification, recover from errors, or resume control when needed—mechanisms that significantly improve task completion reliability and user experience.

In general, it consists of the following components:

1. **Operating Environment:** The environment defines the operational context for the agent, encompassing platforms such as mobile devices, web browsers, and desktop operating systems like Windows. To interact meaningfully, the agent perceives the environment's current state through screenshots, widget trees, or other methods of capturing UI structure Memon et al. (2003b). It continuously monitors feedback on each action's impact, adjusting its strategy in real time to ensure effective task progression.

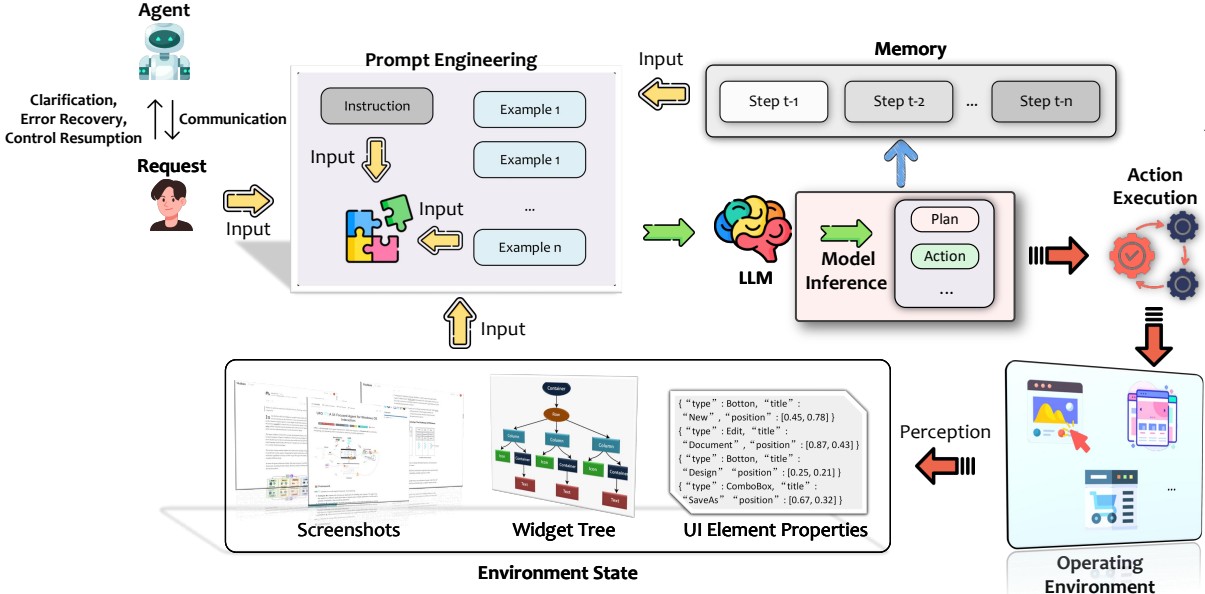

Figure 5: An overview of the architecture and workflow of a basic LLM-powered GUI agent.

2. **Prompt Engineering:** Following environment perception, the agent constructs a detailed prompt to guide the LLM's inference Wang et al. (2023b). This prompt incorporates user instructions, processed visual data (*e.g.*, screenshots), UI element layouts, properties, and any additional context relevant to the task. This structured input maximizes the LLM's ability to generate coherent, context-aware responses aligned with the current GUI state.

3. **Model Inference:** The constructed prompt is passed to a LLM, the agent's inference core, which produces a sequence of plans, actions and insights required to fulfill the user's request. This model may be a general-purpose LLM or a specialized model fine-tuned with GUI-specific data, enabling a more nuanced understanding of GUI interactions, user flows, and task requirements.

4. **Actions Execution:** Based on the model's inference results, the agent identifies specific actions (such as mouse clicks, keyboard inputs, touchscreen gestures, or API calls) required for task execution Shen (2024). An executor within the agent translates these high-level instructions into actionable commands that impact the GUI directly, effectively simulating human-like interactions across diverse applications and devices.

5. **Memory:** For multi-step tasks, the agent maintains an internal memory to track prior actions, task progress, and environment states Zhang et al. (2024q). This memory ensures coherence throughout complex workflows, as the agent can reference previous steps and adapt its actions accordingly. An external memory module may also be incorporated to enable continuous learning, access external knowledge, and enhance adaptation to new environments or requirements.

By iteratively traversing these stages and assembling the foundational components, the LLM-powered GUI agent operates intelligently, seamlessly adapting across various software interfaces and bridging the gap between language-based instruction and concrete action. Each component is critical to the agent's robustness, responsiveness, and capability to handle complex tasks in dynamic environments. In the following subsections, we detail the design and core techniques underlying each of these components, providing a comprehensive guide for constructing LLM-powered GUI agents from the ground up.

Table 3: Summary of platform-specific challenges, action spaces, and typical tasks for Web, Mobile, and Computer GUI environments.

| Platform | Typical GUI Challenges | Action Space | Representative Tasks |
|---|---|---|---|
| **Mobile** | • Constrained screen real estate
• Heavy reliance on touch and gesture recognition Mitra & Acharya (2007)
• App architectures (native vs. hybrid)
• Accessibility frameworks (e.g., Android's Accessibility API, iOS VoiceOver)
• Platform-specific constraints (permissions, security, privacy) | • Tap, swipe, pinch, and other touch gestures
• Virtual keyboard input
• In-app navigation (menus, tabs)
• Accessing hardware features (camera, GPS) | • App-based login and form filling
• Messaging, social media posting
• Location-based services and map interactions
• Handling push notifications and permission dialogs |
| **Web** | • Dynamic and responsive layouts
• Asynchronous updates (AJAX, fetch APIs)
• HTML/DOM-based structures
• Cross-browser inconsistencies | • Click, hover, scroll
• DOM-based form filling
• Link navigation and element inspection
• JavaScript event triggering | • Form completion (registrations, checkouts)
• Data extraction/web scraping
• Searching and filtering (e.g., e-commerce)
• Multi-step web navigation (redirects, pop-ups) |
| **Computer** | • Full-fledged OS-level interfaces
• Multi-window operations and system-level shortcuts
• Automation APIs (*e.g.*, Windows UI Automation Oksanen (2023))
• Frequent software updates requiring adaptation
• Complex, multi-layered software suites | • Mouse click, drag-and-drop
• Keyboard shortcuts and text input
• Menu navigation, toolbars
• Access to multiple application windows | • File management and system settings
• Productivity software usage (office suites, IDEs)
• Installing/uninstalling applications
• Coordinating multi-application workflows |

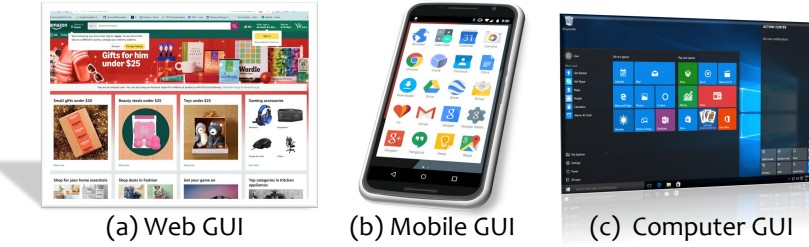

(a) Web GUI   (b) Mobile GUI   (c) Computer GUI

Figure 6: Examples of GUIs from web, mobile and computer platforms.

## 4.2 Operating Environment

The operating environment for LLM-powered GUI agents encompasses various platforms, such as mobile, web, and desktop operating systems, where these agents can interact with graphical interfaces. Each platform

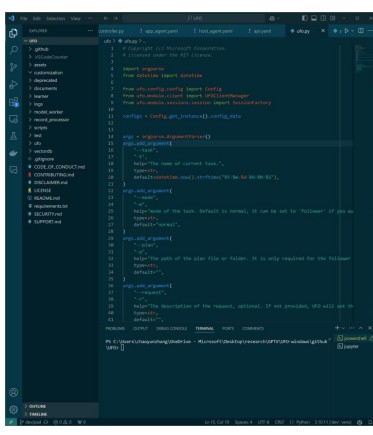 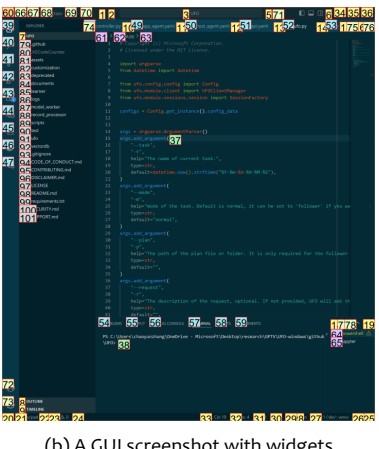 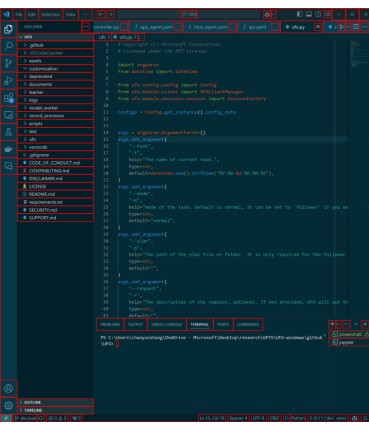

(a) A clean GUI screenshot.

(b) A GUI screenshot with widgets highlighted by SoM.

(c) A GUI screenshot with widgets highlighted by bounding boxes.

Figure 7: Examples of different variants of VS Code GUI screenshots.

has distinct characteristics that impact the way GUI agents perceive, interpret, and act within it. Examples of GUIs from each platform are shown in Figure 6. This section details the nuances of each platform, the ways agents gather environmental information, and the challenges they face in adapting to diverse operating environments.

### 4.2.1 Platform

GUI agents can interact with a wide range of platforms, including mobile devices, web applications, and computer operating systems like Windows. Each platform offers unique capabilities and constraints for GUI automation, requiring agents to adapt their perception and interaction strategies accordingly. We provide a comparative overview of the platforms, including their challenges, action spaces, and representative tasks, in Table 3.

1. **Mobile Platforms:** Mobile devices operate within constrained screen real estate, rely heavily on touch interactions Hardy & Rukzio (2008), and offer varied app architectures (*e.g.*, native vs. hybrid apps). Mobile platforms often use accessibility frameworks, such as Android's Accessibility API[5]Lee et al. (2022) and iOS's VoiceOver Accessibility Inspector[6], to expose structured information about GUI elements. However, GUI agents must handle additional complexities in mobile environments, such as gesture recognition Mitra & Acharya (2007), app navigation Jokinen (2008), and platform-specific constraints (*e.g.*, security and privacy permissions) Enck et al. (2011); Egele et al. (2011).

2. **Web Platforms:** Web applications provide a relatively standardized interface, typically accessible through Hypertext Markup Language (HTML) and Document Object Model (DOM) structures Sierkowski (2002); Fernandes et al. (2011). GUI agents can leverage HTML attributes, such as element ID, class, and tag, to identify interactive components. Web environments also present dynamic content, responsive layouts, and asynchronous updates (*e.g.*, AJAX requests) Garrett et al. (2005), requiring agents to continuously assess the DOM and adapt their actions to changing interface elements.

3. **Computer Platforms:** Computer OS platforms, such as Windows, offer full control over GUI interactions. Agents can utilize system-level automation APIs, such as Windows UI Automation[7] Oksanen (2023), to obtain comprehensive GUI element data, including type, label, position, and bounding box. These platforms often support a broader set of interaction types, mouse, keyboard,

---

[5]https://developer.android.com/reference/android/accessibilityservice/AccessibilityService

[6]https://developer.apple.com/documentation/accessibility/accessibility-inspector

[7]https://learn.microsoft.com/en-us/dotnet/framework/ui-automation/ui-automation-overview

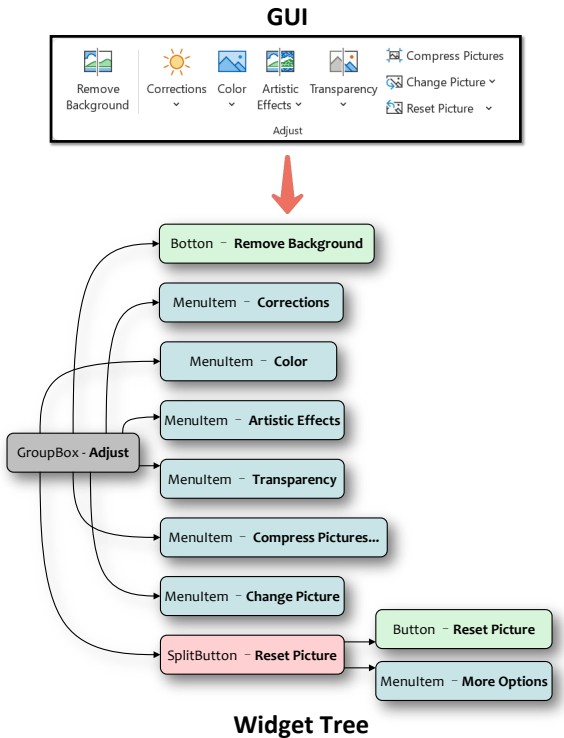

Figure 8: An example of a GUI and its widget tree.

and complex multi-window operations. These enable GUI agents to execute intricate workflows. However, these systems also require sophisticated adaptation for diverse applications, ranging from simple GUI to complex, multi-layered software suites.

In summary, the diversity of platforms, spanning mobile, web, and desktop environments, enable GUI agents to deliver broad automation capabilities, making them a generalized solution adaptable across a unified framework. However, each platform presents unique characteristics and constraints at both the system and application levels, necessitating a tailored approach for effective integration. By considering these platform-specific features, GUI agents can be optimized to address the distinctive requirements of each environment, thus enhancing their adaptability and reliability in varied automation scenarios.

### 4.2.2 Environment State Perception

Accurately perceiving the current state of the environment is essential for LLM-powered GUI agents, as it directly informs their decision-making and action-planning processes. This perception is enabled by gathering a combination of structured data, such as widget trees, and unstructured data, like screenshots, to capture a complete representation of the interface and its components. In Table 4, we outline key toolkits available for collecting GUI environment data across various platforms, and below we discuss their roles in detail:

1. **GUI Screenshots:** Screenshots provide a visual snapshot of the application, capturing the entire state of the GUI at a given moment. They offer agents a reference for layout, design, and visual content, which is crucial when structural details about GUI elements are either limited or unavailable. Visual elements like icons, images, and other graphical cues that may hold important context can be analyzed directly from screenshots. Many platforms have built-in tools to capture screenshots (*e.g.*,

Table 4: Key toolkits for collecting GUI environment data.

| Tool | Platform | Environment | Accessible Information | Highlight | Link |
|---|---|---|---|---|---|
| Selenium | Web | Browser (Cross-platform) | DOM elements, HTML structure, CSS properties | Extensive browser support and automation capabilities | https://www.selenium.dev/ |
| Puppeteer | Web | Browser (Chrome, Firefox) | DOM elements, HTML/CSS, network requests | Headless browser automation with rich API | https://pptr.dev/ |
| Playwright | Web | Browser (Cross-platform) | DOM elements, HTML/CSS, network interactions | Multi-browser support with automation and testing capabilities | https://playwright.dev/ |
| TestCafe | Web | Browser (Cross-platform) | DOM elements, HTML structure, CSS properties | Easy setup with JavaScript/-TypeScript support | https://testcafe.io/ |
| BeautifulSoup | Web | HTML Parsing | HTML content, DOM elements | Python library for parsing HTML and XML documents | https://www.crummy.com/software/BeautifulSoup/ |
| Protractor | Web | Browser (Angular) | DOM elements, Angular-specific attributes | Designed for Angular applications, integrates with Selenium | https://www.protractortest.org/ |
| WebDriverIO | Web | Browser (Cross-platform) | DOM elements, HTML/CSS, network interactions | Highly extensible with a vast plugin ecosystem | https://webdriver.io/ |
| Ghost Inspector | Web | Browser (Cross-platform) | DOM elements, screenshots, test scripts | Cloud-based automated browser testing and monitoring | https://ghostinspector.com/ |
| Cypress | Web | Browser (Cross-platform) | DOM elements, HTML/CSS, network requests | Real-time reloads and interactive debugging | https://www.cypress.io/ |
| UIAutomator | Mobile | Android | GUI hierarchy, widget properties, screen content | Native Android GUI testing framework | https://developer.android.com/training/testing/ui-automator |
| Espresso | Mobile | Android | GUI components, view hierarchy, widget properties | Google's native Android GUI testing framework | https://developer.android.com/training/testing/espresso |
| Android View Hierarchy | Mobile | Android | GUI hierarchy, widget properties, layout information | View hierarchy accessible via developer tools | https://developer.android.com/studio/debug/layout-inspector |
| iOS Accessibility Inspector | Mobile | iOS | Accessibility tree, GUI elements, properties | Tool for inspecting iOS app GUI elements | https://developer.apple.com/documentation/accessibility/accessibility-inspector |
| XCUITest | Mobile | iOS | GUI elements, accessibility properties, view hierarchy | Apple's iOS GUI testing framework | https://developer.apple.com/documentation/xctest/user_interface_tests |
| Flutter Driver | Mobile | Flutter apps | Widget tree, properties, interactions | Automation for Flutter applications | https://flutter.dev/docs/testing |
| Android's MediaProjection API | Mobile | Android | Screenshots, screen recording | Capturing device screen content programmatically | https://developer.android.com/reference/android/media/projection/MediaProjection |
| Windows GUI Automation | Computer | Windows | Control properties, widget trees, accessibility tree | Native Windows support with OS integration | https://docs.microsoft.com/windows/win32/winauto/entry-uiauto-win32 |
| Sikuli | Computer | Windows, macOS, Linux | Screenshots (image recognition), GUI elements | Image-based automation using computer vision | http://sikulix.com/ |
| AutoIt | Computer | Windows | Window titles, control properties, coordinates | Scripting language for Windows GUI automation | https://www.autoitscript.com/site/autoit/ |
| Inspect.exe | Computer | Windows | GUI elements, control properties, accessibility tree | Tool for inspecting Windows GUI elements | https://docs.microsoft.com/windows/win32/winauto/inspect-objects |
| macOS Accessibility API | Computer | macOS | Accessibility tree, GUI elements, control properties | macOS support for accessibility and GUI automation | https://developer.apple.com/accessibility/ |
| Pywinauto | Computer | Windows | Control properties, GUI hierarchy, window information | Python-based Windows GUI automation | https://pywinauto.readthedocs.io/ |
| Electron Inspector | Computer | Electron apps | DOM elements, HTML/CSS, JavaScript state | Tool for Electron applications | https://www.electronjs.org/docs/latest/tutorial/automated-testing |
| Windows Snipping Tool | Computer | Windows | Screenshots | Tool for capturing screenshots in Windows | https://www.microsoft.com/en-us/windows/tips/snipping-tool |
| macOS Screenshot Utility | Computer | macOS | Screenshots, screen recording | Tool for capturing screenshots and recording screen | https://support.apple.com/guide/mac-help/take-a-screenshot-or%2Dscreen-recording%2Dmh26782/mac |
| AccessKit | Cross-Platform | Various OS | Accessibility tree, control properties, roles | Standardized APIs across platforms | https://github.com/AccessKit/accesskit |
| Appium | Cross-Platform | Android, iOS, Windows, macOS | GUI elements, accessibility properties, gestures | Mobile automation framework | https://appium.io/ |
| Robot Framework | Cross-Platform | Web, Mobile, Desktop | GUI elements, DOM, screenshots | Extensible with various libraries | https://robotframework.org/ |
| Cucumber | Cross-Platform | Web, Mobile, Desktop | Step definitions, GUI interactions | BDD framework supporting automation tools | https://cucumber.io/ |
| TestComplete | Cross-Platform | Web, Mobile, Desktop | GUI elements, DOM, control properties | Tool with extensive feature set | https://smartbear.com/product/testcomplete/overview/ |
| Katalon Studio | Cross-Platform | Web, Mobile, Desktop | GUI elements, DOM, screenshots | All-in-one automation solution | https://www.katalon.com/ |
| Ranorex | Cross-Platform | Web, Mobile, Desktop | GUI elements, DOM, control properties | Tool with strong reporting features | https://www.ranorex.com/ |
| Applitools | Cross-Platform | Web, Mobile, Desktop | Screenshots, visual checkpoints, DOM elements | AI-powered visual testing | https://applitools.com/ |

| Widget | Widget Name | Position | Attributes |
|--------|-------------|----------|------------|
| Remove Background | Button - 'Remove Background' | L-3810, T128, R-3708, B243 | title='Remove Background'; auto_id='PictureBackgroundRemoval'; control_type='Button' |
| Corrections | MenuItem - 'Corrections' | L-3689, T128, R-3592, B243 | title='Corrections'; auto_id='PictureCorrectionsMenu'; control_type='MenuItem' |
| Color | MenuItem - 'Color' | L-3589, T128, R-3527, B243 | title='Color'; auto_id='PictureColorMenu'; control_type='MenuItem' |
| Artistic Effects | MenuItem - 'Artistic Effects' | L-3524, T128, R-3448, B243 | title='Artistic Effects'; auto_id='PictureArtisticEffectsGallery'; control_type='MenuItem' |
| Transparency | MenuItem - 'Transparency' | L-3445, T128, R-3336, B243 | title='Transparency'; auto_id='PictureTransparencyGallery'; control_type='MenuItem' |
| Compress Pictures | Button - 'Compress Pictures...' | L-3333, T128, R-3138, B164 | title='Compress Pictures...'; auto_id='PicturesCompress'; control_type='Button' |
| Change Picture | MenuItem - 'Change Picture' | L-3333, T167, R-3149, B203 | title='Change Picture'; auto_id='PictureChangeMenu'; control_type='MenuItem' |
| Reset Picture | SplitButton - 'Reset Picture' | L-3333, T206, R-3160, B242 | title='Reset Picture'; control_type='SplitButton' |

Figure 9: Examples of UI element properties in the PowerPoint application for GUI Agent interaction.

Windows Snipping Tool[8], macOS Screenshot Utility[9], and Android's MediaProjection API[10]), and screenshots can be enhanced with additional annotations, such as Set-of-Mark (SoM) highlights Yang et al. (2023) or bounding boxes Wu et al. (2023d) around key GUI components, to streamline agent decisions. Figure 7 illustrates various screenshots of the VS Code GUI, including a clean version, as well as ones with SoM and bounding boxes that highlight actionable components, helping the agent focus on the most critical areas of the interface.

2. **Widget Trees:** Widget trees present a hierarchical view of interface elements, providing structured data about the layout and relationships between components Gamma (1995). We show an example of a GUI and its widget tree in Figure 8. By accessing the widget tree, agents can identify attributes such as element type, label, role, and relationships within the interface, all of which are essential for contextual understanding. Tools like Windows GUI Automation and macOS's Accessibility API[11] provide structured views for desktop applications, while Android's Accessibility API and HTML DOM structures serve mobile and web platforms, respectively. This hierarchical data is indispensable for agents to map out logical interactions and make informed choices based on the GUI structure.

3. **GUI Element Properties:** Each GUI element in the interface contains specific properties, such as control type, label text, position, and bounding box dimensions, that help agents target the appropriate components. These properties are instrumental for agents to make decisions about spatial relationships (*e.g.*, adjacent elements) and functional purposes (*e.g.*, distinguishing between buttons and text fields). For instance, web applications reveal properties like DOM attributes (id,

---

[8]https://support.microsoft.com/en-us/windows/use-snipping-tool-to-capture%2Dscreenshots%2D00246869%2D1843%2D655f%2Df220%2D97299b865f6b

[9]https://support.apple.com/guide/mac-help/take-a-screenshot-mh26782/mac

[10]https://developer.android.com/reference/android/media/projection/MediaProjection

[11]https://developer.apple.com/library/archive/documentation/Accessibility/Conceptual/AccessibilityMacOSX/

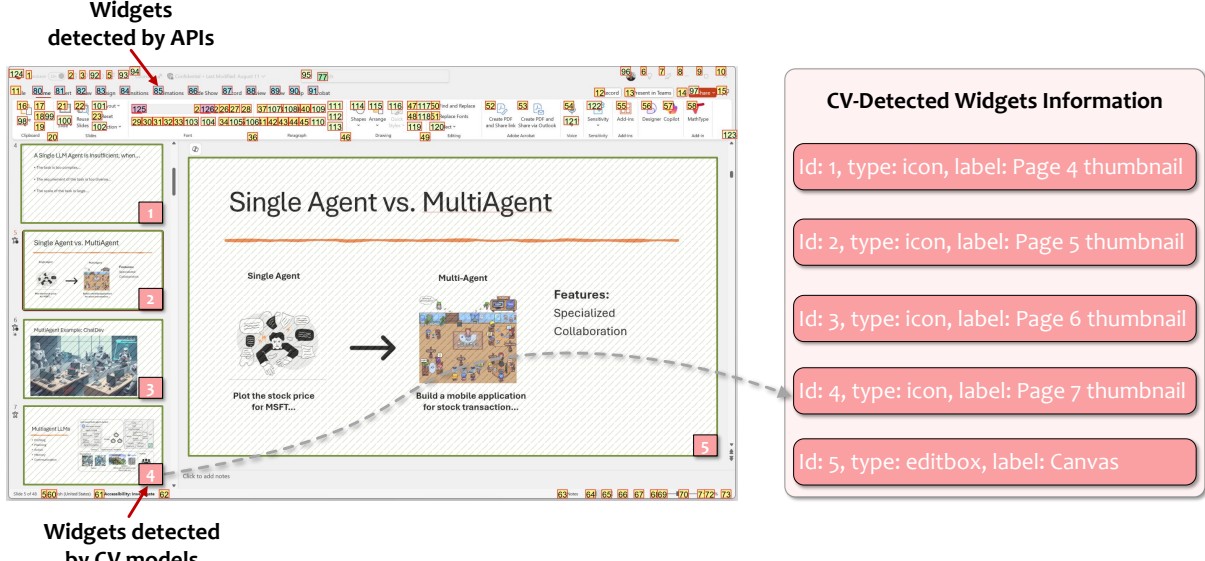

Figure 10: An example illustrating the use of a CV approach to parse a PowerPoint GUI and detect non-standard widgets, inferring their types and labels.

class, name) and CSS styles that provide context and control information. These attributes assist agents in pinpointing precise elements for interaction, enhancing their ability to navigate and operate within diverse GUI environments. Figure 9 illustrates examples of selected GUI element properties extracted by the Windows UI Automation API, which support GUI agents in decision-making.

4. **Complementary CV Approaches:** When structured information is incomplete or unavailable, computer vision techniques can provide additional insights Wang et al. (2024a). For instance, OCR allows agents to extract text content directly from screenshots, facilitating the reading of labels, error messages, and instructions Qian et al. (2022). Furthermore, advanced object detection Chen et al. (2020) models like SAM (Segment Anything Model) Kirillov et al. (2023), DINO Liu et al. (2023a) and OmniParser Lu et al. (2024d) can identify and classify GUI components in various layouts, supporting the agent in dynamic environments where GUI elements may frequently change. These vision-based methods ensure robustness, enabling agents to function effectively even in settings where standard GUI APIs are insufficient. We illustrate an example of this complementary information in Figure 10 and further detail these advanced computer vision approaches in Section 4.7.1.

Together, these elements create a comprehensive, multimodal representation of the GUI environment's current state, delivering both structured and visual data. By incorporating this information into prompt construction, agents are empowered to make well-informed, contextually aware decisions without missing critical environmental cues.

### 4.2.3 Environment Feedback

Effective feedback mechanisms are essential for GUI agents to assess the success of each action and make informed decisions for subsequent steps. Feedback can take several forms, depending on the platform and interaction type. Figure 11 presents examples of various types of feedback obtained from the environment.

1. **Screenshot Update:** By comparing before-and-after screenshots, agents can identify visual differences that signify state changes in the application. Screenshot analysis can reveal subtle variations in the interface, such as the appearance of a notification, visual cues, or confirmation messages, that may not be captured by structured data Moran et al. (2018).

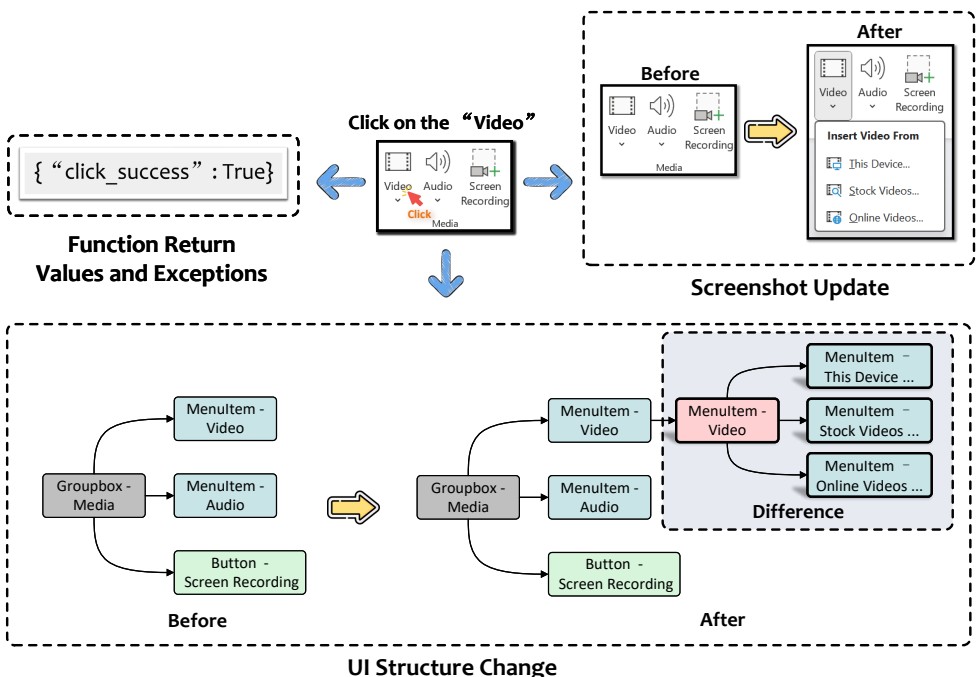

Figure 11: Examples of various types of feedback obtained from a PowerPoint application environment.

2. **GUI Structure Change:** After executing an action, agents can detect modifications in the widget tree structure, such as the appearance or disappearance of elements, updates to element properties, or hierarchical shifts Ricós et al. (2023). These changes indicate successful interactions (*e.g.*, opening a dropdown or clicking a button) and help the agent determine the next steps based on the updated environment state.

3. **Function Return Values and Exceptions:** Certain platforms offer direct feedback on action outcomes through function return values or system-generated exceptions Du et al. (2024). For example, API responses or JavaScript return values can confirm action success on web platforms, while exceptions or error codes can signal failed interactions, guiding the agent to retry or select an alternative approach.

These feedback provided by the environment is crucial for GUI agents to assess the outcomes of their previous actions. This real-time information enables agents to evaluate the effectiveness of their interventions and determine whether to adhere to their initial plans or pivot towards alternative strategies. Through this process of self-reflection, agents can adapt their decision-making, optimizing task execution and enhancing overall performance in dynamic and varied application environments.

## 4.3 Prompt Engineering

In the operation of LLM-powered GUI agents, effective prompt construction is a crucial step that encapsulates all necessary information for the agent to generate appropriate responses and execute tasks successfully Wang et al. (2023b). After gathering the relevant data from the environment, the agent formulates a comprehensive prompt that combines various components essential for inference by the LLM. Each component serves a specific purpose, and together they enable the agent to execute the user's request efficiently. Figure 12 illustrates a basic example of prompt construction in an LLM-powered GUI agent. The key elements of the prompt are summarized as follows:

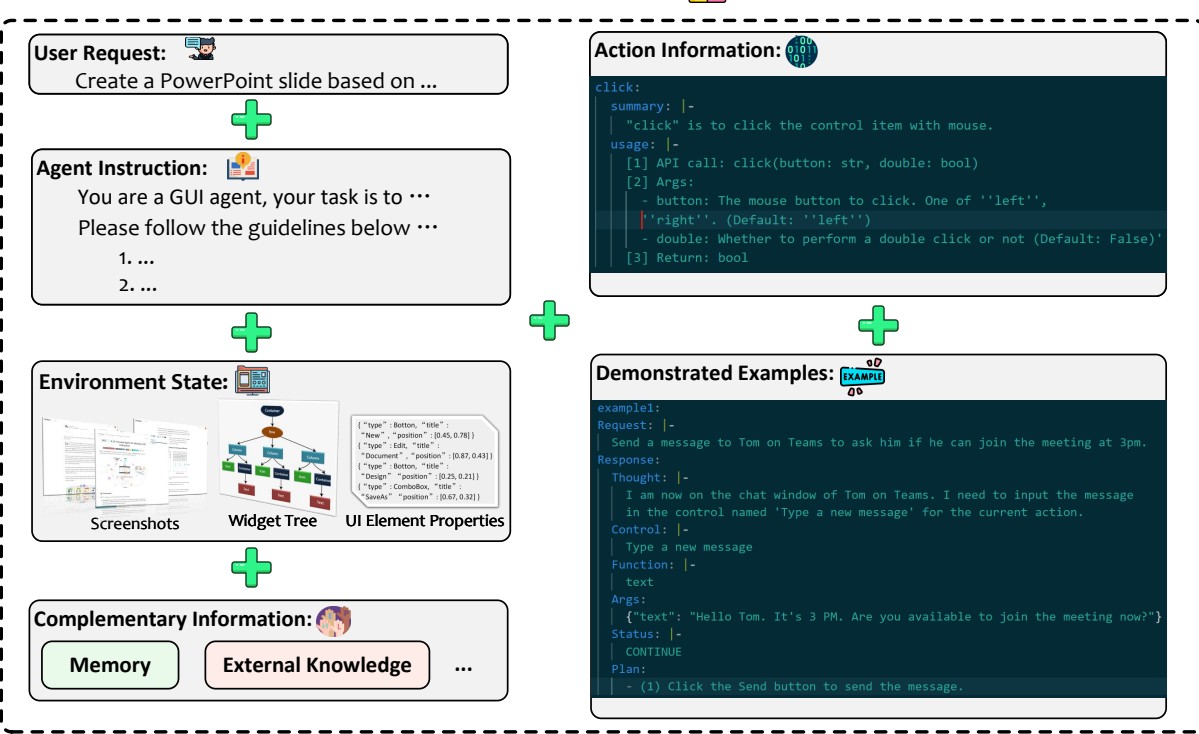

Figure 12: A basic example of prompt construction in a LLM-powered GUI agent.

1. **User Request:** This is the original task description provided by the user, outlining the objective and desired outcome. It serves as the foundation for the agent's actions and is critical for ensuring that the LLM understands the context and scope of the task.

2. **Agent Instruction:** This section provides guidance for the agent's operation, detailing its role, rules to follow, and specific objectives. Instructions clarify what inputs the agent will receive and outline the expected outputs from the LLM, establishing a framework for the inference process. The core agent instructions are usually embedded within the base system prompt of the LLM, with supplementary instructions dynamically injected or updated based on environmental feedback and contextual adaptation.

3. **Environment States:** The agent includes perceived GUI screenshots and GUI information, as introduced in Section 4.2.2. This multimodal data may consist of various versions of screenshots (*e.g.*, a clean version and a SoM annotated version) to ensure clarity and mitigate the risk of GUI controls being obscured by annotations. This comprehensive representation of the environment is vital for accurate decision-making.

4. **Action Documents:** This component outlines the available actions the agent can take, detailing relevant documentation, function or tools names, schemata, arguments, return values, and any other necessary parameters. Providing this information equips the LLM with the context needed to select and generate appropriate actions for the task at hand.

5. **Demonstrated Examples:** Including example input/output pairs is essential to activate the in-context learning Dong et al. (2022) capability of the LLM. These examples help the model comprehend and generalize the task requirements, enhancing its performance in executing the GUI agent's responsibilities.

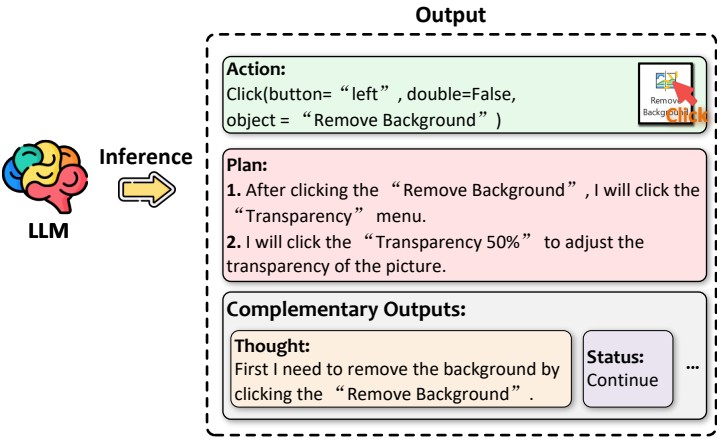

Figure 13: An example of the LLM's inference output in a GUI agent.

6. **Complementary Information:** Additional context that aids in planning and inference may also be included. This can consist of historical data retrieved from the agent's memory (as detailed in Section 4.6) and external knowledge sources, such as documents obtained through retrieval-augmented generation (RAG) methods Lewis et al. (2020); Gao et al. (2023). This supplemental information can provide valuable insights that further refine the agent's decision-making processes.

The construction of an effective prompt is foundational for the performance of LLM-powered GUI agents. By systematically incorporating aforementioned information, the agent ensures that the LLM is equipped with the necessary context and guidance to execute tasks accurately and efficiently.

## 4.4 Model Inference

The constructed prompt is submitted to the LLM for inference, where the LLM is tasked with generating both a plan and the specific actions required to execute the user's request. This inference process is critical as it dictates how effectively the GUI agent will perform in dynamic environments. It typically involves two main components: planning and action prediction, as well as the generation of complementary outputs. Figure 13 shows an example of the LLM's inference output.

### 4.4.1 Planning

Successful execution of GUI tasks often necessitates a series of sequential actions, requiring the agent to engage in effective planning Zhang et al. (2024j). Analogous to human cognitive processes, thoughtful planning is essential to organize tasks, schedule actions, and ensure successful completion Huang et al. (2024b); Cho et al. (2024). The LLM must initially conceptualize a long-term goal while simultaneously focusing on short-term actions to initiate progress toward that goal Dagan et al. (2023).

To effectively navigate the complexity of multi-step tasks, the agent should decompose the overarching task into manageable subtasks and establish a timeline for their execution Khot et al. (2022). Techniques such as CoT reasoning Wei et al. (2022) can be employed, enabling the LLM to develop a structured plan that guides the execution of actions. This plan, which can be stored for reference during future inference steps, enhances the organization and focus of the agent's activities.

The granularity of planning may vary based on the nature of the task and the role of the agent Huang et al. (2024b). For complex tasks, a hierarchical approach that combines global planning (identifying broad subgoals) with local planning (defining detailed steps for those subgoals) can significantly improve the agent's ability to manage long-term objectives effectively Chen et al. (2024j).

Table 5: Overview of actions for GUI agents.

| Action | Category | Original Executor | Examples | Platform | Environment | Toolkit |
|---|---|---|---|---|---|---|
| Mouse actions | GUI Operations | Mouse | Click, scroll, hover, drag | Computer | Windows | GUI Automation 7, Pywinauto Sweigart (2024) |
| Mouse actions | GUI Operations | Mouse | Click, scroll, hover, drag | Computer | macOS | AppleScript [10], Automator [11] |
| Mouse actions | GUI Operations | Mouse | Click, scroll, hover, drag | Web | Browser | Selenium, Puppeteer |
| Keyboard actions | GUI Operations | Keyboard | Typing, key presses, shortcuts | Computer | Windows | GUI Automation 7, Pywinauto Sweigart (2024) |
| Keyboard actions | GUI Operations | Keyboard | Typing, key presses, shortcuts | Computer | macOS | AppleScript [10], Automator [11] |
| Keyboard actions | GUI Operations | Keyboard | Typing, key presses, shortcuts | Web | Browser | Selenium, Puppeteer |
| Touch actions | GUI Operations | Touchscreen | Tap, swipe, pinch, zoom | Mobile | Android | Appium, UIAutomator |
| Touch actions | GUI Operations | Touchscreen | Tap, swipe, pinch, zoom | Mobile | iOS | Appium, XCUITest |
| Gesture actions | GUI Operations | User hand | Rotate, multi-finger gestures | Mobile | Android, iOS | Appium, GestureTools [12] |
| Voice commands | GUI Operations | User voice | Speech input, voice commands | Mobile | Android | SpeechRecognizer [13] |
| Voice commands | GUI Operations | User voice | Speech input, voice commands | Mobile | iOS | SiriKit [14] |
| Clipboard operations | GUI Operations | System clipboard | Copy, paste | Cross-platform | Cross-OS | Pyperclip [15], Clipboard.js [16] |
| Screen interactions | GUI Operations | User | Screen rotation, shake | Mobile | Android, iOS | Device sensors APIs [17] |
| Shell Commands | Native API Calls | Command Line Interface | File manipulation, system operations, script execution | Computer | Unix/Linux, macOS | Bash, Terminal |
| Application APIs | Native API Calls | Application APIs | Send email, create document, fetch data | Computer | Windows | Microsoft Office COM APIs [18] |
| Application APIs | Native API Calls | Application APIs | Access calendar, send messages | Mobile | Android | Android SDK APIs [19] |
| Application APIs | Native API Calls | Application APIs | Access calendar, send messages | Mobile | iOS | iOS SDK APIs [20] |
| System APIs | Native API Calls | System APIs | File operations, network requests | Computer | Windows | Win32 API [21] |
| System APIs | Native API Calls | System APIs | File operations, network requests | Computer | macOS | Cocoa APIs [22] |
| Web APIs | Native API Calls | Web Services | Fetch data, submit forms | Web | Browser | Fetch API [23], Axios [24] |
| AI Models | AI Tools | AI Models | Screen understanding, summarization, image generation | Cross-platform | Cross-OS | DALL·E Ramesh et al. (2021), OpenAI APIs [26] |

---

[10]https://developer.apple.com/library/archive/documentation/AppleScript/Conceptual/AppleScriptLangGuide/introduction/ASLR_intro.html

[11]https://www.macosxautomation.com/automator/

[12]https://docs.blender.org/manual/en/latest/sculpt_paint/sculpting/introduction/gesture_tools.html

[13]https://developer.android.com/reference/android/speech/SpeechRecognizer

### 4.4.2 Action Prediction

Action prediction is the core objective of the inference stage, as it translates the planning into executable tasks. The inferred actions are typically expressed as function call strings, encompassing the function name and relevant parameters. These strings can be readily converted into real-world interactions with the environment, such as clicks, keyboard inputs, mobile gestures, or API calls. A detailed discussion of these action types is presented in Section 4.5.

The input prompt must include a predefined set of actions available for the agent to select from. The agent can choose an action from this set or, if allowed, generate custom code or API calls to interact with the environment Tan et al. (2024a). This flexibility can enhance the agent's adaptability to unforeseen circumstances; however, it may introduce reliability concerns, as the generated code may be prone to errors.

### 4.4.3 Complementary Outputs

In addition to planning and action prediction, the LLM can also generate complementary outputs that enhance the agent's capabilities. These outputs may include reasoning processes that clarify the agent's decision-making (*e.g.*, CoT reasoning), messages for user interaction, or communication with other agents or systems, or the status of the task (*e.g.*, continue or finished). The design of these functionalities can be tailored to meet specific needs, thereby enriching the overall performance of the GUI agent. We note that that such complementary outputs typically capture immediate reasoning, intermediate reflections (*e.g.*, <think> in DeepSeek-r1 Guo et al. (2025)), or communications directed at users or system components, rather than representing a structured sequence of future actions as in explicit planning.

By effectively balancing planning and action prediction while incorporating complementary outputs, agents can navigate complex tasks with a higher degree of organization and adaptability.

### 4.5 Actions Execution

Following the inference process, a crucial next step is for the GUI agent to execute the actions derived from the inferred commands within the GUI environment and subsequently gather feedback. Although the term "GUI agent" might suggest a focus solely on user interface actions, the action space can be greatly expanded by incorporating various toolboxes that enhance the agent's versatility. Broadly, the actions available to GUI agents fall into three main categories: *(i)* GUI operations Li et al. (2020a), *(ii)* native API calls Gu et al. (2016), and *(iii)* AI tools Masterman et al. (2024). Each category offers unique advantages and challenges, enabling the agent to tackle a diverse range of tasks more effectively. We summarize the various actions commonly used in GUI agents, categorized into distinct types, in Table 5, and provide detailed explanations of each category below.

### 4.5.1 UI Operations

UI operations encompass the fundamental interactions that users typically perform with GUIs in software applications. These operations include various forms of input, such as mouse actions (clicks, drags, hovers), keyboard actions (key presses, combinations), touch actions (taps, swipes), and gestures (pinching, rotating).

---

[14] https://developer.apple.com/documentation/sirikit/
[15] https://pypi.org/project/pyperclip/
[16] https://clipboardjs.com/
[17] https://developer.android.com/develop/sensors-and-location/sensors/sensors_overview
[18] https://learn.microsoft.com/en-us/previous-versions/office/office-365-api/
[19] https://developer.android.com/reference
[20] https://developer.apple.com/ios/
[21] https://learn.microsoft.com/en-us/windows/win32/api/
[22] https://developer.apple.com/library/archive/documentation/Cocoa/Conceptual/CocoaFundamentals/WhatIsCocoa/WhatIsCocoa.html
[23] https://developer.mozilla.org/en-US/docs/Web/API/Fetch_API
[24] https://axios-http.com/docs/api_intro
[25] https://platform.openai.com/docs/overview

The specifics of these actions may differ across platforms and applications, necessitating a tailored approach for each environment.

While GUI operations form the foundation of agent interactions with the GUI, they can be relatively slow due to the sequential nature of these tasks. Each operation must be executed step by step, which can lead to increased latency, especially for complex workflows that involve numerous interactions. Despite this drawback, GUI operations are crucial for maintaining a broad compatibility across various applications, as they leverage standard user interface elements and interactions.

### 4.5.2 Native API Calls

In contrast to GUI operations, some applications provide native APIs that allow GUI agents to perform actions more efficiently. These APIs offer direct access to specific functionalities within the application, enabling the agent to execute complex tasks with a single command Lu et al. (2024a). For instance, calling the Outlook API allows an agent to send an email in one operation, whereas using GUI operations would require a series of steps, such as navigating through menus and filling out forms Song et al. (2024b).

While native APIs can significantly enhance the speed and reliability of action execution, their availability is limited. Not all applications or platforms expose APIs for external use, and developing these interfaces can require substantial effort and expertise. Consequently, while native APIs present a powerful means for efficient task completion, they may not be as generalized across different applications as GUI operations.

### 4.5.3 AI Tools

The integration of AI tools into GUI agents represents a transformative advancement in their capabilities. These tools can assist with a wide range of tasks, including content summarization from screenshots or text, document enhancement, image or video generation (*e.g.*, calling ChatGPT Wu et al. (2023c), DALL · E Ramesh et al. (2021)), and even invoking other agents or Copilot tools for collaborative assistance. The rapid development of generative AI technologies enables GUI agents to tackle complex challenges that were previously beyond their capabilities.

By incorporating AI tools, agents can extend their functionality and enhance their performance in diverse contexts. For example, a GUI agent could use an AI summarization tool to quickly extract key information from a lengthy document or leverage an image generation tool to create custom visuals for user presentations. This integration not only streamlines workflows but also empowers agents to deliver high-quality outcomes in a fraction of the time traditionally required.

### 4.5.4 Summary

An advanced GUI agent should adeptly leverage all three categories of actions: GUI operations for broad compatibility, native APIs for efficient execution, and AI tools for enhanced capabilities. This multifaceted approach enables the agent to operate reliably across various applications while maximizing efficiency and effectiveness. By skillfully navigating these action types, GUI agents can fulfill user requests more proficiently, ultimately leading to a more seamless and productive user experience.

## 4.6 Memory

For a GUI agent to achieve robust performance in complex, multi-step tasks, it must retain memory, enabling it to manage states in otherwise stateless environments. Memory allows the agent to track its prior actions, their outcomes, and the task's overall status, all of which are crucial for informed decision-making in subsequent steps Lee et al. (2023). By establishing continuity, memory transforms the agent from a reactive system into a proactive, stateful one, capable of self-adjustment based on accumulated knowledge. The agent's memory is generally divided into two main types: Short-Term Memory Lu et al. (2023) and Long-Term Memory Wang et al. (2024n). We show an overview of different types of memory in GUI agents in Table 6.

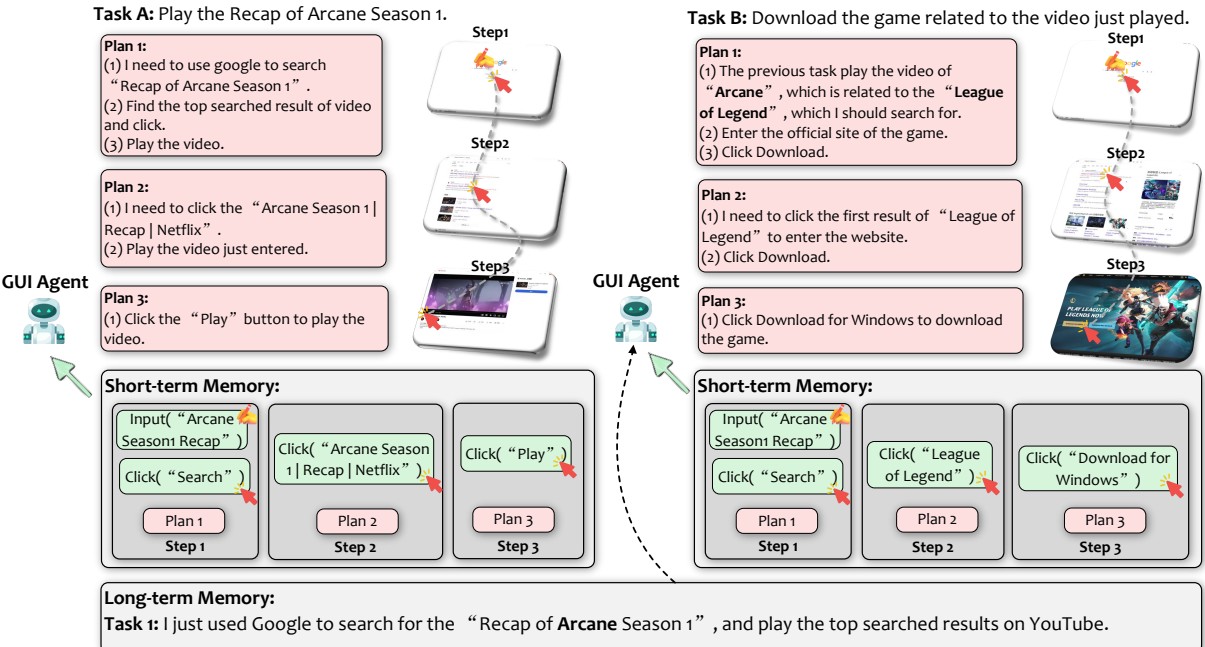

Figure 14: Illustration of short-term memory and long-term memory in an LLM-brained GUI agent.

Table 6: Summary of memory in GUI agents.

| Memory Element | Memory Type | Description | Storage Medium/Method |
|---|---|---|---|
| Action | Short-term | Historical actions trajectory taken in the environment | In-memory, Context window |
| Plan | Short-term | Plan passed from previous step | In-memory, Context window |
| Execution Results | Short-term | Return values, error traces, and other environmental feedback | In-memory, Context window |
| Environment State | Short-term | Important environment state data, e.g., UI elements | In-memory, Context window |
| Self-experience | Long-term | Task completion trajectories from historical tasks | Database, Disk |
| Self-guidance | Long-term | Guidance and rules summarized from historical trajectories | Database, Disk |
| External Knowledge | Long-term | Other external knowledge sources aiding task completion | External Knowledge Base |
| Task Success Metrics | Long-term | Metrics from task success or failure rates across sessions | Database, Disk |

### 4.6.1 Short-Term Memory

Short-Term Memory (STM) provides the primary, ephemeral context used by the LLM during runtime Tack et al. (2024). STM stores information pertinent to the current task, such as recent plans, actions, results, and environmental states, and continuously updates to reflect the task's ongoing status. This memory is particularly valuable in multi-step tasks, where each decision builds on the previous one, requiring the agent to maintain a clear understanding of the task's trajectory. As illustrated in Figure 14, during the completion of independent tasks, the task trajectory, comprising actions and plans—is stored in the STM. This allows the agent to track task progress effectively and make more informed decisions.

However, STM is constrained by the LLM's context window, limiting the amount of information it can carry forward. To manage this limitation, agents can employ selective memory management strategies, such as selectively discarding or summarizing less relevant details to prioritize the most impactful information. Despite its limited size, STM is essential for ensuring coherent, contextually aware interactions and supporting the agent's capacity to execute complex workflows with immediate, relevant feedback.

### 4.6.2 Long-Term Memory

Long-Term Memory (LTM) serves as an external storage repository for contextual information that extends beyond the immediate runtime Zhu et al. (2023a). Unlike STM, which is transient, LTM retains historical

task data, including previously completed tasks, successful action sequences, contextual tips, and learned insights. LTM can be stored on disk or in a database, enabling it to retain larger volumes of information than what is feasible within the LLM's immediate context window. In the example shown in Figure 14, when the second task requests downloading a game related to the previous task, the agent retrieves relevant information from its LTM. This enables the agent to accurately identify the correct game, facilitating efficient task completion.

LTM contributes to the agent's self-improvement over time by preserving examples of successful task trajectories, operational guidelines, and common interaction patterns. When approaching a new task, the agent can leverage RAG techniques to retrieve relevant historical data, which enhances its ability to adapt strategies based on prior success. This is similar to the lifelong learning Zheng et al. (2025c), which makes LTM instrumental in fostering an agent's capacity to "learn" from experience, enabling it to perform tasks with greater accuracy and efficiency as it accumulates insights across sessions. For instance, Zheng et al. (2024d) provides an illustrative example of using past task trajectories stored in memory to guide and enhance future decision-making, a technique that is highly adaptable for GUI agents. It also enables better personalization by retaining information about previous tasks.

### 4.7 Advanced Enhancements

While most LLM-powered GUI agents incorporate fundamental components such as perception, planning, action execution, and memory, several advanced techniques have been developed to significantly improve the reasoning and overall capabilities of these agents. Here, we outline shared advancements widely adopted in research to guide the development of more specialized and capable LLM-powered GUI agents.

#### 4.7.1 Computer Vision-Based GUI Grounding

Although various tools (Section 4) enable GUI agents to access information like widget location, captions, and properties, certain non-standard GUIs or widgets may not adhere to these tools' protocols Zhan et al. (2021), rendering their information inaccessible. Additionally, due to permission management, these tools are not always usable. Such incomplete information can present significant challenges for GUI agents, as the LLM may need to independently locate and interact with required widgets by estimating their coordinates to perform actions like clicking—a task that is inherently difficult without precise GUI data.

CV models offer a non-intrusive solution for GUI grounding directly from screenshots, enabling the detection, localization, segmentation, and even functional estimation of widgets Li et al. (2020b); White et al. (2019); Wang et al. (2021); Bai et al. (2021). This approach allows agents to interpret the visual structure and elements of the GUI without relying on system-level tools or internal metadata, which may be unavailable or incomplete. CV-based GUI parsing provides agents with valuable insights into interactive components, screen layout, and widget functionalities based solely on visual cues, enhancing their ability to recognize and act upon elements on the screen. Figure 10 provides an illustrative example of how a CV-based GUI parser works. While standard API-based detection captures predefined widgets, the CV model can identify additional elements, such as thumbnails and canvases, which may not have explicit API representations in the PowerPoint interface. This enhances widget recognition, allowing the agent to detect components beyond the scope of API detection. We show an overview of related GUI grounding models and benchmarks in Table 7, 8, 9 and 10.

A notable example is OmniParser Lu et al. (2024d), which implements a multi-stage parsing technique involving a fine-tuned model for detecting interactable icons, an OCR module for extracting text, and an icon description model that generates localized semantic descriptions for each UI element. By integrating these components, OmniParser constructs a structured representation of the GUI, enhancing an agent's understanding of interactive regions and functional elements. This comprehensive parsing strategy has shown to significantly improve GPT-4V's screen comprehension and interaction accuracy.

Such CV-based GUI grounding layers provide critical grounding information that significantly enhances an agent's ability to interact accurately and intuitively with diverse GUIs. This is particularly beneficial for handling custom or non-standard elements that deviate from typical accessibility protocols. Additionally,

Table 7: A summary of of GUI grounding models (Part I).

| Model | Platform | Foundation Model | Size | Dataset | Input | Output | Highlight | Link |
|---|---|---|---|---|---|---|---|---|
| OmniParser Lu et al. (2024d) | Mobile, Desktop, and Web | BLIP-2 Li et al. (2023c), YOLOv8 Reis et al. (2023) | / | 67,000 UI screenshots with bounding box annotations and 7,185 icon–description pairs generated using GPT-4 | UI screenshots | IDs, bounding boxes, and descriptions of interactable elements | Introduces a purely vision-based screen parsing framework for general UI understanding without external information, significantly improving action prediction accuracy for LLM-driven agents | `https://github.com/microsoft/OmniParser` |
| Iterative Narrowing Nguyen (2024) | Mobile, Web, and Desktop | Qwen2-VL and OS-Atlas-Base | / | ScreenSpot Cheng et al. (2024a) | A GUI screenshot and a natural language query | (x,y) coordinates representing the target location in the GUI | Progressively crops regions of the GUI to refine predictions, enhancing precision for GUI grounding tasks | `https://github.com/ant-8/GUI-Grounding-via-Iterative-Na` |
| Iris Ge et al. (2024) | Mobile (iOS, Android), Desktop (Windows, macOS), and Web | Qwen-VL Bai et al. (2023b) | 9.6B | 850K GUI-specific annotations and 150K vision–language instructions | High-resolution GUI screenshots with natural language instructions | Referring: Generates detailed descriptions of UI elements. Grounding: Locates UI elements on the screen. | Information-Sensitive Cropping for efficient handling of high-resolution GUI images, and Self-Refining Dual Learning to iteratively enhance GUI grounding and referring tasks without additional annotations | / |
| Attention-driven Grounding Xu et al. (2024b) | Mobile, Web, and Desktop | MiniCPM-Llama3-V 2.5 | 8.5B | Mind2Web Deng et al. (2023), ScreenSpot Cheng et al. (2024a), Visual-WebBench Liu et al. (2024f) | GUI screenshots and textual user queries | Element localization via bounding boxes, text-to-image mapping for grounding, and actionable descriptions of GUI components | Utilizes attention mechanisms in pre-trained MLLMs without fine-tuning | `https://github.com/HeimingX/TAG` |
| Aria-UI Yang et al. (2024b) | Web, Desktop, and Mobile | Aria Li et al. (2024b) | 3.9B | 3.9 million elements and 11.5 million samples | GUI screenshots, user instructions, and action histories | Pixel coordinates for GUI elements and corresponding actions | A purely vision-based approach avoiding reliance on AXTree-like inputs | `https://ariaui.github.io` |

prompting methods like iterative narrowing have shown promise in improving the widget grounding capabilities of VLMs Nguyen (2024). Together, these approaches pave the way for more adaptable and resilient GUI agents, capable of operating effectively across a broader range of screen environments and application contexts.

Several works have introduced benchmarks to evaluate the GUI grounding capabilities of models and agents. For instance, ScreenSpot Cheng et al. (2024a) serves as a pioneering benchmark designed to assess the GUI grounding performance of LLM-powered agents across diverse platforms, including iOS, Android, macOS, Windows, and web environments. It features a dataset with over 600 screenshots and 1,200 instructions, focusing on complex GUI components such as widgets and icons. This benchmark emphasizes the importance of GUI grounding in enhancing downstream tasks like web automation and mobile UI interaction. Building upon this, ScreenSpot-Pro Li et al. (2025b) extends the scope to more professional, high-resolution environments. This evolved version includes 1,581 tasks with high-quality annotations, encompassing domains such as

Table 8: A summary of of GUI grounding models (Part II).

| Model | Platform | Foundation Model | Size | Dataset | Input | Output | Highlight | Link |
|---|---|---|---|---|---|---|---|---|
| UGround Gou et al. (2024) | Web, Desktop (Windows, MacOS, Linux), Mobile (Android, iOS) | LLaVA-NeXT-7B Liu et al. (2024b) | 7B | Web-Hybrid and other existing datasets | GUI screenshots, user queries | Pixel coordinates of GUI elements | A universal GUI grounding model that relies solely on vision, eliminating the need for text-based representations | https://osu-nlp-group.github.io/UGround/ |
| GUI-Bee Fan et al. (2025) | Web | SeeClick Cheng et al. (2024a), Qwen-GUI Chen et al. (2024i), and UIX-7B Liu et al. (2024e) | 7B-13B | NovelScreenSpot | GUI screenshots, user queries, accessibility tree | GUI element grounding locations, actions and function calls, navigation steps, predicted GUI changes after interaction | Autonomously explores GUI environments, with Q-ICRL optimizing exploration efficiency and enhancing data diversity. | https://gui-bee.github.io |
| RWKV-UI Yang & Hou (2025) | Web | SIGLIP Zhai et al. (2023), DINOv2 Oquab et al. (2023), SAM Kirillov et al. (2023) | 1.6B | Websight Laurençon et al. (2024b), WebUI-7kbal Wu et al. (2023b), Web2Code Yun et al. (2024) | High-resolution webpage images | Element grounding, Action prediction, CoT reasoning | Introduces a high-resolution three-encoder architecture with visual prompt engineering and CoT reasoning. | / |
| TRISHUL Singh et al. (2025) | Web, Desktop, and Mobile platforms | / (Training-Free) | / (Training-Free) | / (Training-Free) | GUI Screenshots, user instructions/queries, hierarchical screen parsing outputs, OCR-extracted text descriptors | Action grounding, functionality descriptions of GUI elements, GUI referring, and SoMs | Utilizes hierarchical screen parsing and spatially enhanced element descriptions to enhance LVLMs without additional training. | / |

software development, creative tools, CAD, scientific applications, and office productivity. Key features of ScreenSpot-Pro include authentic high-resolution screenshots and meticulous annotations provided by domain experts.

These benchmarks provide critical evaluation criteria for assessing GUI grounding capabilities, thereby advancing the development of GUI agents for improved GUI understanding and interaction.

### 4.7.2 Multi-Agent Framework

The adage "two heads are better than one" holds particular relevance for GUI automation tasks, where a single agent, though capable, can be significantly enhanced within a multi-agent framework Li et al. (2023a); Chen et al. (2024h). Multi-agent systems leverage the collective intelligence, specialized skills, and complementary strengths of multiple agents to tackle complex tasks more effectively than any individual agent could alone. In the context of GUI agents, multi-agent systems offer advanced capabilities through two primary mechanisms: *(i)* specialization and *(ii)* inter-agent collaboration. Figure 15 illustrates an example of how an LLM-powered multi-agent collaborates to create a desk.

Table 9: A summary of of GUI grounding models (Part III).

| Model | Platform | Foundation Model | Size | Dataset | Input | Output | Highlight | Link |
|---|---|---|---|---|---|---|---|---|
| AutoGUI Li et al. (2025a) | Web, Mobile | Qwen-VL-10B Bai et al. (2023b), SliME-8B Zhang et al. (2024o) | 10B / 8B | AutoGUI-704k | GUI screenshots, User queries | Element functionalities, Element locations | Automatically labels UI elements based on interaction-induced changes, making it scalable and high-quality. | `https://autogui-project.github.io/` |
| Query Inference Wu et al. (2025d) | Mobile Android | Qwen2-VL-7B-Instruct Wang et al. (2024j) | 7B | UIBERT Wu et al. (2024f) | GUI screenshots | Action-oriented queries, Co-ordinates | Improves reasoning without requiring large-scale training data. | `https://github.com/ZrW00/GUIPivot` |
| WinClick Hui et al. (2025) | Windows OS | Phi3-Vision Abdin et al. (2024) | 4.2B | WinSpot Benchmark | GUI screenshots, Natural language instructions | Element locations | The first GUI grounding model specifically tailored for Windows. | `https://github.com/zackhuiiiiii/WinSpot` |
| FOCUS Tang et al. (2025a) | Web, mobile applications, and desktop | Qwen2-VL-2B-Instruct Wang et al. (2024j) | 2B | GUICourse Chen et al. (2024i), Aguvis-stage1 Xu et al. (2024k), Wave-UI Zheng et al. (2024c), Desktop-UI Lin et al. (2024c) | GUI screenshot + task instruction | Normalized coordinates (x, y) | A dual-system GUI grounding architecture inspired by human cognition, which dynamically switches between fast (intuitive) and slow (analytical) grounding modes based on task complexity | `https://github.com/sugarandgugu/Focus` |
| UI-E2I-Synth Liu et al. (2025d) | Web, Windows, and Android | InternVL2-4B and Qwen2-VL-7B | 4B and 7B | 1.6M screenshots, 9.9M instructions | GUI screenshot | Element coordinates | Introduces a three-stage synthetic data pipeline for GUI grounding with both explicit and implicit instruction synthesis | `https://colmon46.github.io/i2e-bench-leaderboard/` |
| RegionFocus Luo et al. (2025b) | Web-based and Desktop interfaces | UI-TARS and Qwen2.5-VL | 72B | None (test-time only) | GUI screenshots with a point of interest | Coordinate-based actions | Introduces a visual test-time scaling framework that zooms into salient UI regions and integrates an image-as-map mechanism to track history and avoid repeated mistakes—boosting grounding accuracy without model retraining | `https://github.com/tiangeluo/RegionFocus` |

1. **Specialization of Agents:** In a multi-agent framework, each agent is designed to specialize in a specific role or function, leveraging its unique capabilities to contribute to the overall task. As illustrated in the Figure 15, specialization enables distinct agents to focus on different aspects of the task pipeline. For instance, the "Document Extractor" specializes in extracting relevant content from local documents, such as PDFs, while the "Web Retriever" focuses on gathering additional information from online sources. Similarly, the "Designer" transforms the retrieved information into visually appealing slides, and the "Evaluator" provides feedback to refine and improve the output. This functional separation ensures that each agent becomes highly adept at its designated task, leading to improved efficiency and quality of results Song et al. (2024d).

Table 10: A summary of of GUI grounding benchmarks.

| Benchmark | Platform | Dataset | Input | Output | Highlight | Link |
|---|---|---|---|---|---|---|
| ScreenSpot Cheng et al. (2024a) | iOS, Android, macOS, and Windows | Over 600 screenshots and 1,200 instructions | GUI screenshots accompanied by user instructions | Bounding boxes or coordinates of actionable GUI elements | A realistic and diverse GUI grounding benchmark covering multiple platforms and a variety of elements | `https://github.com/njucckevin/SeeClick` |
| ScreenSpot-Pro Li et al. (2025b) | Windows, macOS, and Linux | 1,581 instruction–screenshot pairs covering 23 applications across 5 industries and 3 operating systems | High-resolution GUI screenshots paired with natural language instructions | Bounding boxes for locating target UI elements | Introduces a high-resolution benchmark for professional environments | `https://github.com/likaixin2000/ScreenSpot-Pro-GUI-Grounding` |
| PixelWeb Yang et al. (2025c) | Web | 100,000 webpages | Rendered webpage screenshots and DOM information | BBox, mask, contour | The first GUI dataset to provide pixel-level annotations—including mask and contour—for web UIs, enabling high-precision GUI grounding and detection tasks | `https://huggingface.co/datasets/cyberalchemist/PixelWeb` |

**Task:** Create a desk for LLM-based multi-agent system.

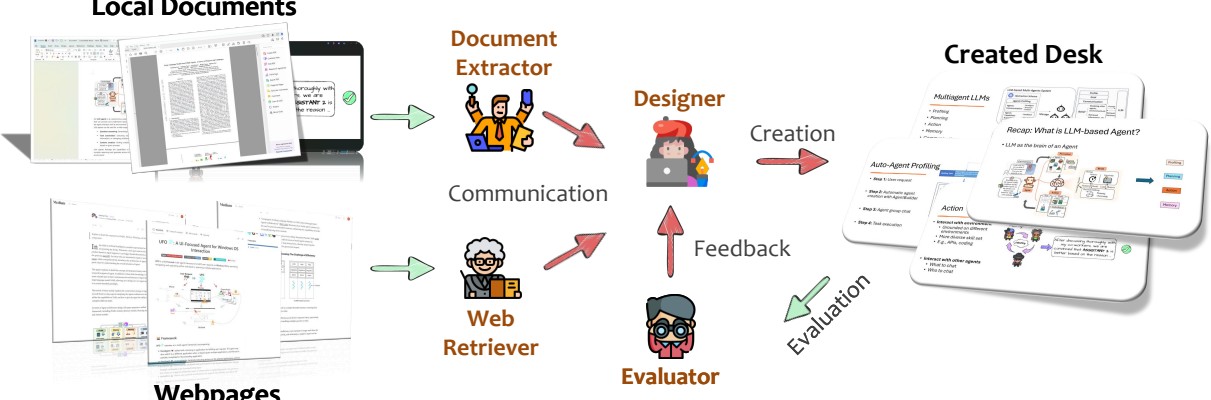

Figure 15: An example of multi-agent system collaboration in creating a desk.

2. **Collaborative Inter-Agent Dynamics:** The multi-agent system shown in the Figure 15 exemplifies how agents collaborate dynamically to handle complex tasks. The process begins with the "Document Extractor" and "Web Retriever", which work in parallel to collect information from local and online sources. The retrieved data is communicated to the "Designer", who synthesizes it into a cohesive set of slides. Once the slides are created, the "Evaluator" reviews the output, providing feedback for refinement. These agents share information, exchange context, and operate in a coordinated manner, reflecting a human-like teamwork dynamic. For example, as depicted, the agents' roles are tightly integrated—each output feeds into the next stage, creating a streamlined workflow that mirrors real-world collaborative environments Zhang et al. (2024a).

In such a system, agents can collectively engage in tasks requiring planning, discussion, and decision-making. Through these interactions, the system taps into each agent's domain expertise and latent potential for specialization, maximizing overall performance across diverse, multi-step processes.

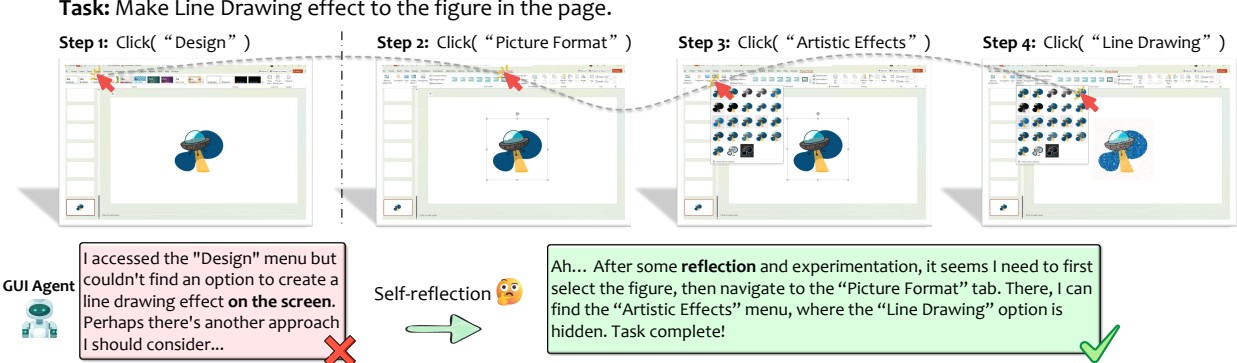

Figure 16: An example of self-reflection in task completion of an LLM-powered GUI agent.

### 4.7.3 Self-Reflection

"A fault confessed is half redressed". In the context of GUI multi-agent systems, self-reflection refers to the agents' capacity to introspectively assess their reasoning, actions, and decisions throughout the task execution process Renze & Guven (2024). This capability allows agents to detect potential mistakes, adjust strategies, and refine actions, thereby improving the quality and robustness of their decisions, especially in complex or unfamiliar GUI environments. By periodically evaluating their own performance, self-reflective agents can adapt dynamically to produce more accurate and effective results Pan et al. (2024a).

Self-reflection is particularly critical for GUI agents due to the variable nature of user interfaces and the potential for errors, even in human-operated systems. GUI agents frequently encounter situations that deviate from expectations, such as clicking the wrong button, encountering unexpected advertisements, navigating unfamiliar interfaces, receiving error messages from API calls, or even responding to user feedback on task outcomes. To ensure task success, a GUI agent must quickly reflect on its actions, assess these feedback signals, and adjust its plans to better align with the desired objectives.

As illustrated in Figure 16, when the agent initially fails to locate the "Line Drawing" option in the Design menu, self-reflection enables it to reconsider and identify its correct location under Artistic Effects" in the "Picture Format" menu, thereby successfully completing the task.

In practice, self-reflection techniques for GUI agents typically involve two main approaches: *(i)* **ReAct** Yao et al. (2022b) and *(ii)* **Reflexion** Shinn et al. (2024).

1. **ReAct (Reasoning and Acting):** ReAct integrates self-reflection into the agent's action chain by having the agent evaluate each action's outcome and reason about the next best step. In this framework, the agent doesn't simply follow a linear sequence of actions; instead, it adapts dynamically, continuously reassessing its strategy in response to feedback from each action. For example, if a GUI agent attempting to fill a form realizes it has clicked the wrong field, it can adjust by backtracking and selecting the correct element. Through ReAct, the agent achieves higher consistency and accuracy, as it learns to refine its behavior with each completed step.

2. **Reflexion:** Reflexion emphasizes language-based feedback, where agents receive and process feedback from the environment as linguistic input, referred to as self-reflective feedback. This feedback is contextualized and used as input in subsequent interactions, helping the agent to learn rapidly from prior mistakes. For instance, if a GUI agent receives an error message from an application, Reflexion enables the agent to process this message, update its understanding of the interface, and avoid similar mistakes in future interactions. Reflexion's iterative feedback loop promotes continuous improvement and is particularly valuable for GUI agents navigating complex, multi-step tasks.

Overall, self-reflection serves as an essential enhancement in GUI multi-agent systems, enabling agents to better navigate the variability and unpredictability of GUI environments. This introspective capability not

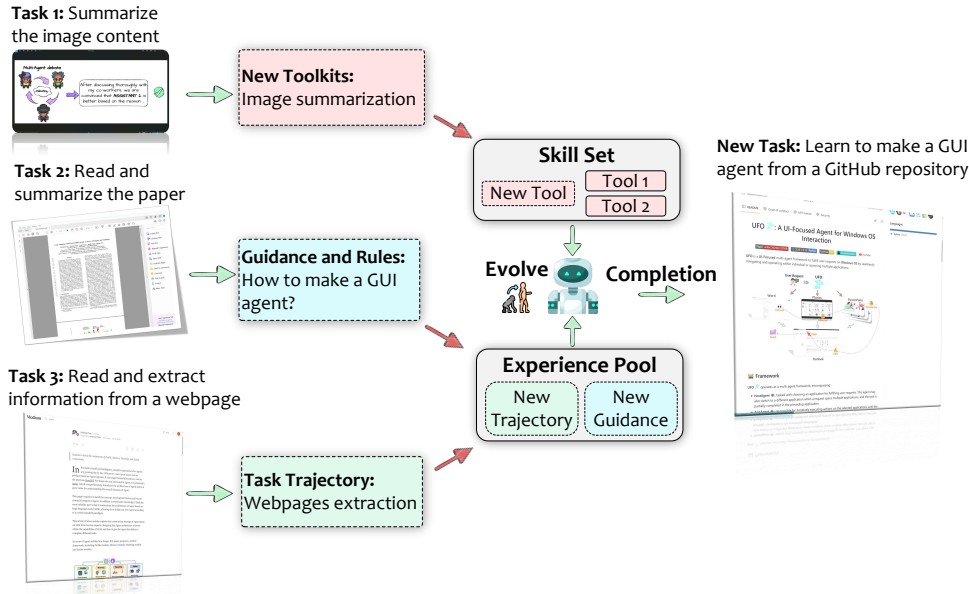

Figure 17: An example self-evolution in a LLM-powered GUI agent with task completion.

only boosts individual agent performance but also promotes resilience, adaptability, and long-term learning in a collaborative setting.

### 4.7.4 Self-Evolution

Self-evolution Tao et al. (2024) is a crucial attribute that GUI agents should possess, enabling them to enhance their performance progressively through accumulated experience. In the context of GUI multi-agent systems, self-evolution allows not only individual agents to improve but also facilitates collective learning and adaptation by sharing knowledge and strategies among agents. During task execution, GUI agents generate detailed action trajectories accompanied by complementary information such as environment states, internal reasoning processes (the agent's thought processes), and evaluation results. This rich data serves as a valuable knowledge base from which GUI agents can learn and evolve. The knowledge extracted from this experience can be categorized into three main areas:

1. **Task Trajectories**: The sequences of actions executed by agents, along with the corresponding environment states, are instrumental for learning Zhao et al. (2024a). These successful trajectories can be leveraged in two significant ways. First, they can be used to fine-tune the core LLMs that underpin the agent models with the state-action pairs Wang et al. (2024h). Fine-tuning with such domain-specific and task-relevant data enhances the model's ability to generalize and improves performance on similar tasks in the future. Second, after summarized by LLMs Tao et al. (2024), these trajectories can be utilized as demonstration examples to activate the in-context learning capabilities of LLMs during prompt engineering. By including examples of successful task executions in the prompts, agents can better understand and replicate the desired behaviors without additional model training.

   For instance, suppose an agent successfully completes a complex task that involves automating data entry across multiple applications. The recorded action trajectory—comprising the steps taken, decisions made, and contextual cues—can be shared with other agents. These agents can then use this trajectory as a guide when faced with similar tasks, reducing the learning curve and improving efficiency.

2. **Guidance and Rules:** From the accumulated experiences, agents can extract high-level rules or guidelines that encapsulate best practices, successful strategies, and lessons learned from past

Figure 18: An example of MDP modeling for task completion in a GUI agent.

mistakes Zhu et al. (2023b); Zhang et al. (2024p). Such guidance can be acquired by the LLM itself through trajectory summarization Zhu et al. (2023b), or even via search-based algorithms, such as Monte Carlo Tree Search (MCTS) Zhang et al. (2024p). This knowledge can be formalized into policies or heuristics that agents consult during decision-making processes, thereby enhancing their reasoning capabilities.

For example, if agents repeatedly encounter errors when attempting to perform certain actions without proper prerequisites (*e.g.*, trying to save a file before specifying a file path), they can formulate a rule to check for these prerequisites before executing the action. This proactive approach reduces the likelihood of errors and improves task success rates.

3. **New Toolkits:** Throughout their interactions, GUI agents may discover or develop more efficient methods, tools, or sequences of actions that streamline task execution Tan et al. (2024a). These may include optimized API calls, macros, or combinations of GUI operations that accomplish tasks more effectively than previous approaches. LLMs can be leveraged to automatically analyze execution trajectories in order to summarize, discover, and generate high-level shortcuts or frequently used fast APIs, which can then be reused for future executions Jiang et al. (2025). By incorporating these new tools into their repertoire, agents expand their capabilities and enhance overall efficiency.

   As an example, an agent might find that using a batch processing API can automate repetitive tasks more efficiently than performing individual GUI operations in a loop. This new approach can be shared among agents within the multi-agent system, allowing all agents to benefit from the improved method and apply it to relevant tasks.

Figure 17 illustrates how a GUI agent evolves through task completion. During its operations, the agent adds new capabilities to its skill set, such as an image summarization toolkit, gains insights from reading a paper on creating GUI agents, and stores task trajectories like webpage extraction in its experience pool. When assigned a new task, such as "Learn to make a GUI agent from a GitHub repository", the agent draws on its acquired skills and past experiences to adapt and perform effectively.

This dynamic evolution highlights the agent's ability to continually learn, grow, and refine its capabilities. By leveraging past experiences, incorporating new knowledge, and expanding its toolset, GUI agents can adapt to diverse challenges, improve task execution, and significantly enhance the overall performance of the system, fostering a collaborative and ever-improving environment.

### 4.7.5 Reinforcement Learning

Reinforcement Learning (RL) Kaelbling et al. (1996) has witnessed significant advancements in aligning LLMs with desired behaviors Wang et al. (2023c), and has recently been employed in the development of LLM agents Sun et al. (2024a); Zhai et al. (2024). In the context of GUI multi-agent systems, RL offers substantial potential to enhance the performance, adaptability, and collaboration of GUI agents. GUI automation tasks naturally align with the structure of a Markov Decision Process (MDP) Puterman (1990), making them particularly well-suited for solutions based on RL. In this context, the *state* corresponds to the environment perception (such as GUI screenshots, GUI element properties, and layout configurations), while *actions* map directly to GUI operations, including mouse clicks, keyboard inputs, and API calls. *Rewards* can be explicitly defined based on various performance metrics, such as task completion, efficiency, and accuracy, allowing the agent to optimize its actions for maximal effectiveness. Figure 18 illustrates an example of MDP modeling for task completion in a GUI agent, where state, action and reward are clearly defined.

By formulating GUI agent interactions as an MDP, we can leverage RL techniques to train agents that learn optimal policies for task execution through trial and error Toyama et al. (2021a). This approach enables agents to make decisions that maximize cumulative rewards over time, leading to more efficient and effective task completion. For example, an agent learning to automate form filling in a web application can use RL to discover the most efficient sequence of actions to input data and submit the form successfully, minimizing errors and redundant steps. This process helps align the agents more closely with desired behaviors in GUI automation tasks, especially in complex or ambiguous situations where predefined action sequences are insufficient.

As a representative approach, Bai *et al.*, introduce DigiRL Bai et al. (2024), a two-phase RL framework for training GUI agents in dynamic environments. DigiRL begins with an offline RL phase that uses offline data to initialize the agent model, followed by online fine-tuning, where the model interacts directly with an environment to refine its strategies through live data within an Android learning environment using an LLM evaluator that provides reliable reward signals. This adaptive setting enables the agent to learn and respond effectively to the complexities of dynamic GUIs. Wang *et al.*, propose DistRL Wang et al. (2024l), an RL fine-tuning pipeline specifically designed for on-device mobile control agents operating within Android. DistRL employs an asynchronous architecture, deploying RL fine-tuned agents across heterogeneous worker devices and environments for decentralized data collection. By leveraging off-policy RL techniques, DistRL enables centralized training with data gathered remotely from diverse environments, significantly enhancing the scalability and robustness of the model. These representative methods illustrate the potential of RL to improve GUI agents, demonstrating how both centralized and distributed RL frameworks can enable more responsive, adaptable, and effective GUI automation models in real-world applications.

### 4.7.6 Summary & Takeaways

In conclusion, the advanced techniques significantly enhance the capabilities of LLM-brained GUI agents, making them more versatile, efficient, and adaptive within multi-agent frameworks. Importantly, these techniques are not mutually exclusive—many can be integrated to create more powerful agents. For instance, incorporating self-reflection within a multi-agent framework allows agents to collaboratively improve task strategies and recover from errors. By leveraging these advancements, developers can design LLM-brained GUI agents that are not only adept at automating complex, multi-step tasks but also capable of continuously improving through self-evolution, adaptability to dynamic environments, and effective inter-agent collaboration. Future research is expected to yield even more sophisticated techniques, further extending the scope and robustness of GUI automation.

### 4.8 From Foundations to Innovations: A Roadmap

Building robust, adaptable, and effective LLM-powered GUI agents is a multifaceted process that requires careful integration of several core components. With a solid foundation in architecture, design, environment interaction, and memory, as outlined in Section 4, we now shift our focus to the critical elements required for deploying these agents in practical scenarios. This exploration begins with an in-depth review of state-of-the-art LLM-powered GUI agent frameworks in Section 5, highlighting their advancements and unique contributions

Table 11: Taxonomy of LLM-powered GUI agent frameworks by target platform.

| Platform | References |
|---|---|
| **Web** | Zheng et al. (2024a); Song et al. (2024b); Chae et al. (2024); Gur et al. (2024); Ma et al. (2024b); He et al. (2024b); Lai et al. (2024); Xie et al. (2023); Kil et al. (2024); Abuelsaad et al. (2024); Koh et al. (2024b); Zhang et al. (2024n); Yang et al. (2024a); Murty et al. (2024); Shahbandeh et al. (2024); Iong et al. (2024); Tang & Shin (2024); Putta et al. (2024); Gu et al. (2024); Verma et al. (2024); Kim et al. (2024b); Shen et al. (2024b); Zhou et al. (2024a); Liu et al. (2024c); Huang et al. (2025b); Zhang et al. (2025e); Pahuja et al. (2025); Wornow et al. (2024); Zhang et al. (2025c); Dammu (2025); Erdogan et al. (2025); Zheng et al. (2025a); Wang et al. (2025g); Zhang et al. (2025f) |
| **Mobile** | Zhang et al. (2023a); Wang et al. (2024e); Wen et al. (2024a); Zhang et al. (2024g); Song et al. (2024c); Wen et al. (2024c); Ma et al. (2024d); Zhang & Zhang (2024); Yan et al. (2023a); Li et al. (2024g); Wen et al. (2024b); Wang et al. (2024d); Zhang et al. (2024e); Christianos et al. (2024); Zhu et al. (2024b); Lee et al. (2024c); Wang et al. (2025e); Hoscilowicz et al. (2024); Wu et al. (2025c); Wang et al. (2025c); Huang et al.; Wang et al. (2025b); Liu et al. (2025g); Jiang et al. (2025); Zhou et al. (2025); Cheng et al. (2025a); Dai et al. (2025); Liu et al. (2025a); Lai et al. (2025); Wang et al. (2023a); Kahlon et al. (2025); Bishop et al. (2024) |
| **Computer** | Zhang et al. (2024a); Tan et al. (2024a); Wu et al. (2024e); Li et al. (2024g); Agashe et al. (2024); Wu et al. (2024b); Li et al. (2023d); He et al. (2024d); Liu et al. (2025c); Aggarwal & Welleck (2025); Zhao et al. (2025a); Lu et al. (2025); Zhang et al. (2025b); Yin et al. (2025) |
| **Cross-Platform** | Liu et al. (2024h); Xu et al. (2024k); Pawlowski et al. (2024); Song et al. (2024d); Su et al. (2025); He et al. (2025); Wang & Liu (2024); Jia et al. (2024); Wang et al. (2024o); Liu et al. (2025e); Agashe et al. (2025); Hu et al. (2025); Huang et al. (2025a) |

to the field. Building on this, we delve into the methodologies for optimizing LLMs for GUI agents, starting with data collection and processing strategies in Section 6, and progressing to model optimization techniques in Section 7. To ensure robust development and validation, we then examine evaluation methodologies and benchmarks in Section 8, which are essential for assessing agent performance and reliability. Finally, we explore a diverse range of practical applications in Section 9, demonstrating the transformative impact of these agents across various domains.

Note that we intentionally limit in-depth discussions in the main text (Sections 4–9) to a carefully curated set of representative works. The selection criteria include demonstrated impact (e.g., citation count, GitHub adoption), methodological novelty, and alignment with our core taxonomy. Works that are less mature or narrower in scope—regardless of publication status—are instead cataloged in summary tables (Table 16–69), which have been relocated to the appendix to improve readability and streamline the main narrative. Together, these sections provide a comprehensive roadmap for advancing LLM-powered GUI agents from foundational concepts to real-world implementation and innovation. This roadmap, spanning from foundational components to real-world deployment, encapsulates the essential pipeline required to bring an LLM-powered GUI agent concept from ideation to implementation.

# 5 LLM-Powered GUI Agent Framework

The integration of LLMs has unlocked new possibilities for constructing GUI agents, enabling them to interpret user requests, analyze GUI components, and autonomously perform actions across diverse environments. By

equipping these models with essential components and functionalities, as outlined in Section 4, researchers have created sophisticated frameworks tailored to various platforms and applications. These frameworks represent a rapidly evolving area of research, with each introducing innovative techniques and specialized capabilities that push the boundaries of what GUI agents can achieve.

The landscape of GUI agent frameworks has seen notable advancements, particularly in terms of multi-agent architectures, multimodal inputs, and enhanced action sets. These developments are laying the groundwork for more versatile and powerful agents capable of handling complex, dynamic environments. Key takeaways from recent advancements include:

1. **Multi-Agent Synergy:** Multi-agent systems, such as those in UFO Zhang et al. (2024a) and MMAC-Copilot Song et al. (2024d), represent a significant trend in GUI agent development. By assigning specialized roles to different agents within a framework, multi-agent systems can enhance task efficiency, adaptability, and overall performance. As agents take on more complex tasks across diverse platforms, the coordinated use of multiple agents is proving to be a powerful approach, enabling agents to handle intricate workflows with greater precision and speed.

2. **Multimodal Input Benefits:** While some agents still rely solely on text-based inputs (*e.g.*, DOM structures or HTML), incorporating visual inputs, such as screenshots, has shown clear performance advantages. Agents like WebVoyager He et al. (2024b) and SeeAct Zheng et al. (2024a) highlight how visual data, combined with textual inputs, provides a richer representation of the environment state, helping agents make better-informed decisions. This integration of multimodal inputs is essential for accurate interpretation in visually complex or dynamic environments where text alone may not capture all necessary context.

3. **Expanding Action Sets Beyond UI Operations:** Recent agents have expanded their action sets beyond standard UI operations to include API calls and AI-driven actions, as seen in Hybrid Agent Song et al. (2024b) and AutoWebGLM Lai et al. (2024). Incorporating diverse actions allows agents to achieve higher levels of interaction and task completion, particularly in environments where data can be directly retrieved or manipulated through API calls. This flexibility enhances agent capabilities, making them more efficient and adaptable across a wider range of applications.

4. **Emerging Techniques for Improved Decision-Making:** Novel approaches such as world models in WMA Chae et al. (2024) and search-based strategies in Search-Agent Koh et al. (2024b) represent promising directions for more advanced decision-making. World models allow agents to simulate action outcomes, reducing unnecessary interactions and improving efficiency, especially in long-horizon tasks. Similarly, search-based algorithms like best-first and MCTS help agents explore action pathways more effectively, enhancing their adaptability in complex, real-time environments.

5. **Toward Cross-Platform Generalization:** Cross-platform frameworks, such as AutoGLM Liu et al. (2024h) and OSCAR Wang & Liu (2024), underscore the value of generalizability in GUI agent design. These agents are pioneering efforts to create solutions that work seamlessly across mobile, desktop, and web platforms, moving closer to the goal of a one-stop GUI agent that can operate across multiple ecosystems. Cross-platform flexibility will be crucial for agents that aim to assist users consistently across their digital interactions.

6. **Pure Vision-Based Agent:** To enable universal GUI control, pure vision-based frameworks have emerged as a prominent solution. These agents rely solely on screenshots for decision-making, eliminating the need for access to metadata such as widget trees or element properties. Notable work like AGUVIS Xu et al. (2024k) exemplifies this approach. While pure vision-based methods offer greater generalizability and bypass system API limitations, they require strong "grounding" capabilities to accurately locate and interact with UI elements—an ability often lacking in many foundational models. Fine-tuning models specifically for visual grounding and GUI understanding, or integrating GUI parsing techniques like OmniParser Lu et al. (2024d), can address this challenge and enhance the agent's ability to perform precise interactions.

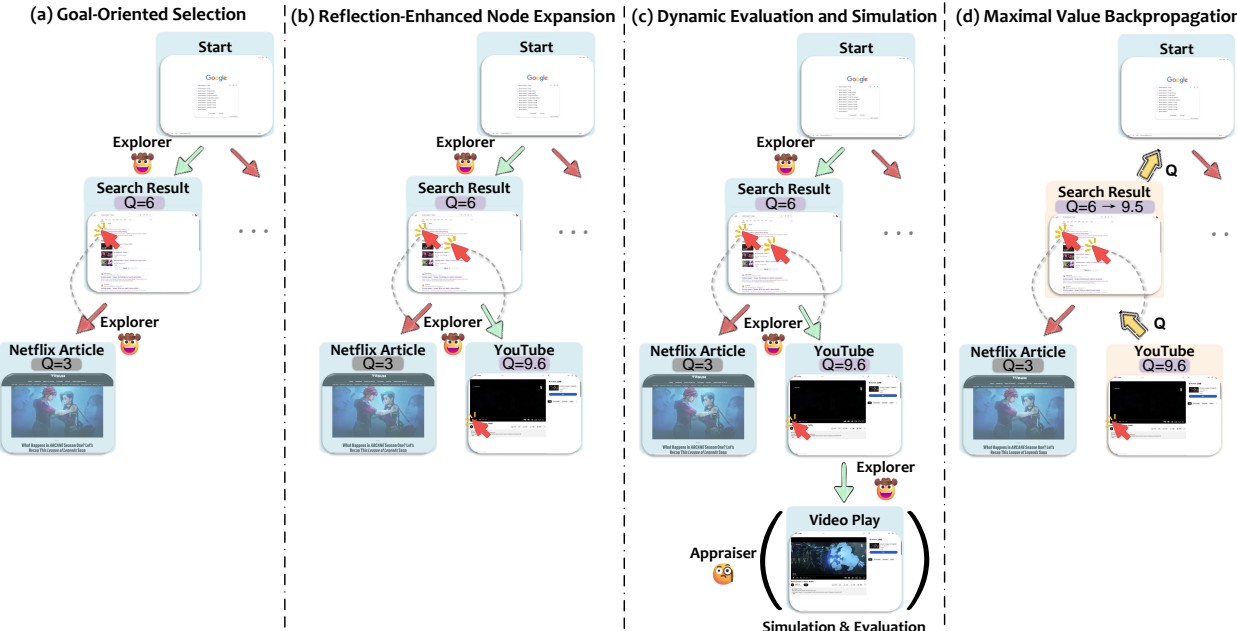

Figure 19: An illustration of the local optimization stage in WebPilot Zhang et al. (2024n) using MCTS. Figure adapted from the original paper.

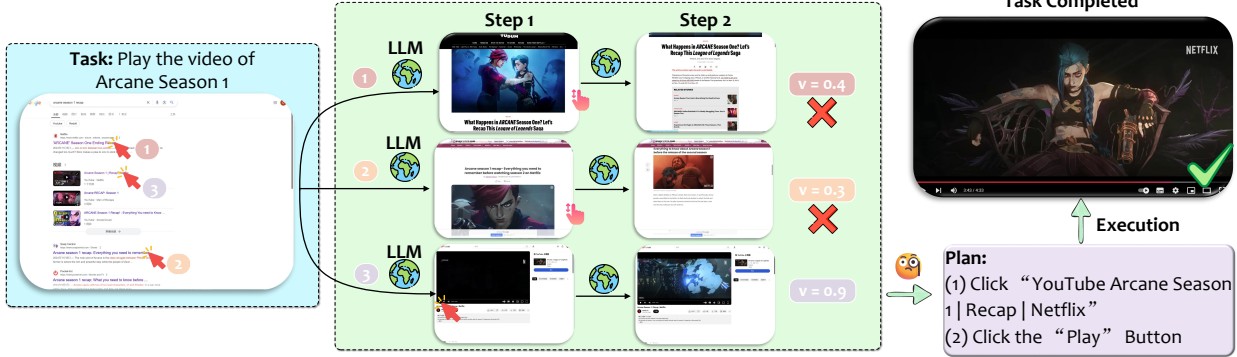

Figure 20: An example illustrating how WebDreamer Gu et al. (2024) uses an LLM to simulate the outcome of each action. Figure adapted from the original paper.

We offer a detailed discussion of each framework, examining their foundational design principles, technical advancements, and the specific challenges they address in the realm of GUI automation. By delving into these aspects, we aim to provide deeper insights into how these agents are shaping the future of human-computer interaction and task automation, and the critical role they play in advancing this transformative field. Before delving into the details, we first present a taxonomy of LLM-powered GUI agent frameworks categorized by target platform, as shown in Table 11.

## 5.1 Web GUI Agents

Advancements in web GUI agents have led to significant strides in automating complex tasks within diverse and dynamic web environments. Recent frameworks have introduced innovative approaches that leverage multimodal inputs, predictive modeling, and task-specific optimizations to enhance performance, adaptability, and efficiency. In this subsection, we first summarize key web GUI agent frameworks in Tables 16, 17, 18, 19,

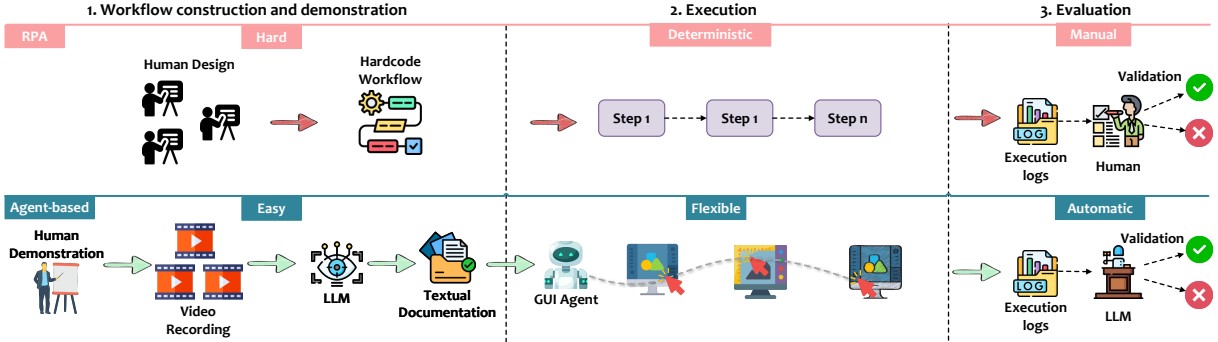

Figure 21: Comparison of RPA and agent based automation. Figure adapted from Wornow et al. (2024).

20 and 21, then delve into representative frameworks, highlighting their unique contributions and how they collectively push the boundaries of web-based GUI automation.

One prominent trend is the integration of multimodal capabilities to improve interaction with dynamic web content. For instance, **SeeAct** Zheng et al. (2024a) harnesses GPT-4V's multimodal capacities to ground actions on live websites effectively. By leveraging both visual data and HTML structure, SeeAct integrates grounding techniques using image annotations, HTML attributes, and textual choices, optimizing interactions with real-time web content. This approach allows SeeAct to achieve a task success rate of 51.1% on real-time web tasks, highlighting the importance of dynamic evaluation in developing robust web agents.

Building upon the advantages of multimodal inputs, **WebVoyager** He et al. (2024b) advances autonomous web navigation by supporting end-to-end task completion across real-world web environments. Utilizing GPT-4V for both visual (screenshots) and textual (HTML elements) inputs, WebVoyager effectively interacts with dynamic web interfaces, including those with dynamically rendered content and intricate interactive elements. This multimodal capability allows WebVoyager to manage complex interfaces with a success rate notably surpassing traditional text-only methods, setting a new benchmark in web-based task automation.

In addition to multimodal integration, some frameworks focus on parsing intricate web structures and generating executable code to navigate complex websites. **WebAgent** Gur et al. (2024) employs a two-tiered model approach by combining HTML-T5 for parsing long, complex HTML documents with Flan-U-PaLM Chung et al. (2024) for program synthesis. This modular design enables WebAgent to translate user instructions into executable Python code, autonomously handling complex, real-world websites through task-specific sub-instructions. WebAgent demonstrates a 50% improvement in success rates on real websites compared to traditional single-agent models, showcasing the advantages of integrating HTML-specific parsing with code generation for diverse and dynamic web environments.

To enhance decision-making in web navigation, several frameworks introduce state-space exploration and search algorithms. **LASER** Ma et al. (2024b) models web navigation as state-space exploration, allowing flexible backtracking and efficient decision-making without requiring extensive in-context examples. By associating actions with specific states and leveraging GPT-4's function-calling feature for state-based action selection, LASER minimizes errors and improves task success, particularly in e-commerce navigation tasks such as WebShop and Amazon. This state-based approach provides a scalable and efficient solution, advancing the efficiency of LLM agents in GUI navigation.

Similarly, **Search-Agent** Koh et al. (2024b) innovatively introduces a best-first search algorithm to enhance multi-step reasoning in interactive web environments. By exploring multiple action paths, this approach improves decision-making, achieving up to a 39% increase in success rates across benchmarks like WebArena Zhou et al.. Search-Agent's compatibility with existing multimodal LLMs demonstrates the effectiveness of search-based algorithms for complex, interactive web tasks.

Expanding on search-based strategies, **WebPilot** Zhang et al. (2024n) employs a dual optimization strategy combining global and local Monte Carlo Tree Search (MCTS) Browne et al. (2012) to improve adaptability

in complex and dynamic environments. As illustrated in Figure 19, WebPilot decomposes overarching tasks into manageable sub-tasks, with each undergoing localized optimization. This approach allows WebPilot to continuously adjust its strategies in response to real-time observations, mimicking human-like decision-making and flexibility. Extensive testing on benchmarks like WebArena Zhou et al. and MiniWoB++ Liu et al. (2018) demonstrates WebPilot's state-of-the-art performance, showcasing exceptional adaptability compared to existing methods.

Furthering the concept of predictive modeling, the **WMA** Chae et al. (2024) introduces a world model to simulate and predict the outcomes of UI interactions. By focusing on transition-based observations, WMA allows agents to simulate action results before committing, reducing unnecessary actions and increasing task efficiency. This predictive capability is particularly effective in long-horizon tasks that require high accuracy, with WMA demonstrating strong performance on benchmarks such as WebArena Zhou et al. and Mind2Web Deng et al. (2023).

Along similar lines, **WebDreamer**Gu et al. (2024) introduces an innovative use of LLMs for model-based planning in web navigation, as depicted in Figure 20. WebDreamer simulates and evaluates potential actions and their multi-step outcomes using LLMs before execution Yao et al. (2024), akin to a "dreamer" that envisions various scenarios. By preemptively assessing the potential value of different plans, WebDreamer selects and executes the plan with the highest expected value. This approach addresses critical challenges in web automation, such as safety concerns and the need for robust decision-making in complex and dynamic environments, demonstrating superiority over reactive agents in benchmarks like VisualWebArena Koh et al. (2024a) and Mind2Web-live Pan et al. (2024b).

Beyond predictive modeling, integrating API interactions into web navigation offers enhanced flexibility and efficiency. The **Hybrid Agent** Song et al. (2024b) combines web browsing and API interactions, dynamically switching between methods based on task requirements. By utilizing API calls for structured data interaction, the Hybrid Agent reduces the time and complexity involved in traditional web navigation, achieving higher accuracy and efficiency in task performance. This hybrid architecture underscores the benefits of integrating both structured API data and human-like browsing capabilities in AI agent systems.

Addressing the challenges of complex web structures and cross-domain interactions, **AutoWebGLM** Lai et al. (2024) offers an efficient solution by simplifying HTML to focus on key webpage components, thereby improving task accuracy. Using reinforcement learning and rejection sampling for fine-tuning, AutoWebGLM excels in complex navigation tasks on both English and Chinese sites. Its bilingual dataset and structured action-perception modules make it practical for cross-domain web interactions, emphasizing the importance of efficient handling in diverse web tasks.

**ECLAIR Wornow et al. (2024)** represents a pioneering application that replaces traditional RPA with a foundation model-powered GUI agent for enterprise automation. Unlike conventional RPA, which relies on manually programmed rules and rigid scripts, ECLAIR dynamically learns workflows from video demonstrations and textual SOPs (Standard Operating Procedures), significantly reducing setup time and improving adaptability. It operates on enterprise web applications, leveraging GPT-4V and CogAgent CogAgent Team (2024) to perceive GUI elements, plan actions, and execute workflows, and validate automatically. By eliminating the high maintenance costs and execution brittleness of RPA, ECLAIR introduces a more flexible and scalable approach to GUI automation. We show a comparison of such agent-based vs. RPA automation in Figure 21. This work establishes an important foundation for LLM-powered GUI automation, demonstrating how multimodal foundation models can bridge the gap between process mining, RPA, and fully autonomous enterprise workflows.

**Summary**   Recent advances in web GUI agents have significantly broadened the design space of intelligent web automation systems. Across the surveyed frameworks, we observe three dominant trends: (i) Multimodal grounding, exemplified by SeeAct and WebVoyager, boosts success rates on dynamic websites by integrating screenshots, HTML, and textual cues; (ii) Predictive planning and simulation, as employed by WMA and WebDreamer, reduces action errors and improves long-horizon reasoning by modeling environment dynamics before execution; and (iii) Search-based and hybrid control, as seen in LASER, Search-Agent, and WebPilot,

enables robust decision-making via MCTS or state-based planning, achieving up to 39% improvement in complex web benchmarks like WebArena and Amazon.

Overall, these frameworks collectively demonstrate how combining multimodal perception, planning, and hybrid action strategies leads to marked gains in success rates (up to 51.1%), adaptability, and robustness in real-world web environments. We envision future research further exploring generalization across unseen domains, long-horizon task stability, and the integration of memory and self-correction mechanisms.

## 5.2 Mobile GUI Agents

The evolution of mobile GUI agents has been marked by significant advancements, leveraging multimodal models, complex architectures, and adaptive planning to address the unique challenges of mobile environments. These agents have progressed from basic interaction capabilities to sophisticated systems capable of dynamic, context-aware operations across diverse mobile applications. We first provide an overview of mobile GUI agent frameworks in Tables 22, 23, 24, 25 and 26.

**Wang _et al._,** Wang et al. (2023a) pioneer the use of LLMs to enable conversational interaction with mobile UIs, establishing one of the earliest foundations for mobile GUI agents. Their approach involves directly prompting foundation models such as PaLM using structured representations of Android view hierarchies, which are transformed into HTML-like text to better align with the LLM's training distribution. The authors define and evaluate four core tasks, including Screen Summarization, Screen QA, Screen Question Generation, and Instruction-to-UI Mapping—demonstrating that strong performance can be achieved with as few as two prompt examples per task. Emphasizing practicality and accessibility, the work enables rapid prototyping without model fine-tuning, and stands out as a seminal effort in prompt-based evaluation of LLM-powered GUI agents for mobile applications.

Early efforts focused on enabling human-like GUI interactions without requiring backend system access. One such pioneering framework is **AppAgent** Zhang et al. (2023a), which utilizes GPT-4V's multimodal capabilities to comprehend and respond to both visual and textual information. By performing actions like tapping and swiping using real-time screenshots and structured XML data, AppAgent can interact directly with the GUI across a variety of applications, from social media to complex image editing. Its unique approach of learning through autonomous exploration and observing human demonstrations allows for rapid adaptability to new apps, highlighting the effectiveness of multimodal capabilities in mobile agents.

Building upon this foundation, **AppAgent-V2** Li et al. (2024g) advances the framework by enhancing visual recognition and incorporating structured data parsing. This enables precise, context-aware interactions and the ability to perform complex, multi-step operations across different applications. AppAgent-V2 also introduces safety checks to handle sensitive data and supports cross-app tasks by tracking and adapting to real-time interactions. This progression underscores the importance of advanced visual recognition and structured data processing in improving task precision and safety in real-time mobile environments.

Parallel to these developments, vision-centric approaches emerged to further enhance mobile task automation without relying on app-specific data. For instance, **Mobile-Agent** Wang et al. (2024e) leverages OCR, CLIP Radford et al. (2021), and Grounding DINO Liu et al. (2023a) for visual perception. By using screenshots and visual tools, Mobile-Agent performs operations ranging from app navigation to complex multitasking, following instructions iteratively and adjusting for errors through a self-reflective mechanism. This vision-based method positions Mobile-Agent as a versatile and adaptable assistant for mobile tasks.

To address challenges in long-sequence navigation and complex, multi-app scenarios, **Mobile-Agent-v2** Wang et al. (2024d) introduces a multi-agent architecture that separates planning, decision-making, and reflection. By distributing responsibilities among three agents, this framework optimizes task progress tracking, retains memory of task-relevant information, and performs corrective actions when errors occur. Integrated with advanced visual perception tools like Grounding DINO Liu et al. (2023a) and Qwen-VL-Int4 Bai et al. (2023b), Mobile-Agent-v2 showcases significant improvements in task completion rates on both Android and Harmony OS, highlighting the potential of multi-agent systems for handling complex mobile tasks.

In addition to vision-centric methods, some frameworks focus on translating GUI states into language to enable LLM-based action planning. **VisionTasker** Song et al. (2024c) combines vision-based UI interpretation

with sequential LLM task planning by processing mobile UI screenshots into structured natural language. Supported by YOLO-v8 Reis et al. (2023) and PaddleOCR[29] for widget detection, VisionTasker allows the agent to automate complex tasks across unfamiliar apps, demonstrating higher accuracy than human operators on certain tasks. This two-stage design illustrates a versatile and adaptable framework, setting a strong precedent in mobile automation.

Similarly, **DroidBot-GPT** Wen et al. (2024c) showcases an innovative approach by converting GUI states into natural language prompts, enabling LLMs to autonomously decide on action sequences. By interpreting the GUI structure and translating it into language that GPT models can understand, DroidBot-GPT generalizes across various apps without requiring app-specific modifications. This adaptability underscores the transformative role of LLMs in handling complex, multi-step tasks with minimal custom data.

To enhance action prediction and context awareness, advanced frameworks integrate perception and action systems within a multimodal LLM. **CoCo-Agent** Ma et al. (2024d) exemplifies this by processing GUI elements like icons and layouts through its Comprehensive Event Perception and Comprehensive Action Planning modules. By decomposing actions into manageable steps and leveraging high-quality data from benchmarks like Android in the Wild (AITW) Rawles et al. (2023) and META-GUI Sun et al. (2022), CoCo-Agent demonstrates its ability to automate mobile tasks reliably across varied smartphone applications.

Further advancing this integration, **CoAT** Zhang et al. (2024g) introduces a chain-of-action-thought process to enhance action prediction and context awareness. Utilizing sophisticated models such as GPT-4V and set-of-mark tagging, CoAT addresses the limitations of traditional coordinate-based action recognition. By leveraging the Android-In-The-Zoo (AITZ) dataset it builds, CoAT provides deep context awareness and improves both action prediction accuracy and task completion rates, highlighting its potential for accessibility and user convenience on Android platforms.

Addressing the need for efficient handling of multi-step tasks with lower computational costs, **AutoDroid** Wen et al. (2024a) combines LLM-based comprehension with app-specific knowledge. Using an HTML-style GUI representation and a memory-based approach, AutoDroid reduces dependency on extensive LLM queries. Its hybrid architecture of cloud and on-device models enhances responsiveness and accessibility, making AutoDroid a practical solution for diverse mobile tasks. AutoDroid-V2 Wen et al. (2024b) enhances its predecessor AutoDroid, by utilizing on-device language models to generate and execute multi-step scripts for user task automation. By transforming dynamic and complex GUI elements of mobile apps into structured app documents, it achieves efficient and accurate automation without depending on cloud-based resources. The script-based approach reduces computational overhead by minimizing query frequency, thereby improving task efficiency and addressing the limitations of stepwise agents. This advancement enables privacy-preserving and scalable task automation on mobile platforms.

**MobileGPT Lee et al. (2024c)** automates tasks on Android devices using a human-like app memory system that emulates the cognitive process of task decomposition—Explore, Select, Derive, and Recall. This approach results in highly efficient and accurate task automation. Its hierarchical memory structure supports modular, reusable, and adaptable tasks and sub-tasks across diverse contexts. MobileGPT demonstrates superior performance over state-of-the-art systems in task success rates, cost efficiency, and adaptability, highlighting its potential for advancing mobile task automation.

In a more advanced distributed setting, **FedMobileAgent Wang et al. (2025c)** employs a federated learning framework to train mobile automation agents using self-sourced data from users' phone interactions. It addresses the high cost and privacy concerns associated with traditional human-annotated datasets by introducing Auto-Annotation, which leverages vision-language models (VLMs) to infer user intentions from screenshots and actions. The system enables decentralized training through federated learning while preserving user privacy, and its adaptive aggregation method enhances model performance under non-IID data conditions. Experimental results on several mobile benchmarks demonstrate that FedMobileAgent achieves performance comparable to human-annotated models at a fraction of the cost.

**Summary**  In summary, mobile GUI agents have evolved significantly, progressing from single-agent systems to complex, multi-agent frameworks capable of dynamic, context-aware operations. These innovations

---

[29]https://github.com/PaddlePaddle/PaddleOCR

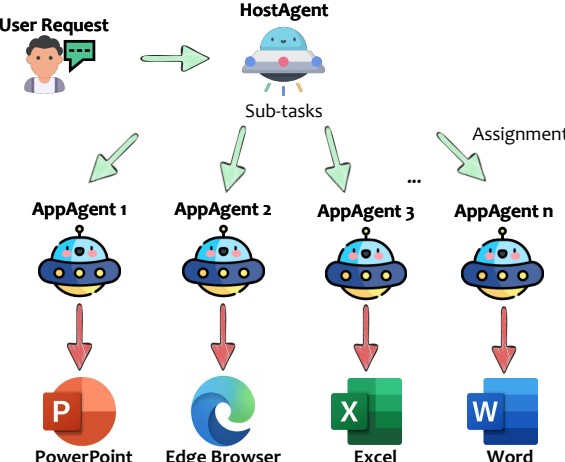

Figure 22: The multi-agent architecture employed in UFO Zhang et al. (2024a). Figure adapted from the original paper.

demonstrate that sophisticated architectures, multimodal processing, and advanced planning strategies are essential in handling the diverse challenges of mobile environments. Notably, frameworks like AppAgent-V2 and Mobile-Agent-v2 emphasize strong real-time perception and multi-agent collaboration to enable robust cross-app operations, achieving notable improvements in task completion accuracy and stability. Systems such as VisionTasker and DroidBot-GPT further highlight the effectiveness of translating GUI states into structured language to facilitate generalizable and instruction-following behavior without app-specific tuning. Meanwhile, solutions like CoCo-Agent and CoAT integrate high-quality benchmark-driven training and modular perception-action pipelines, leading to significant gains in action prediction accuracy and reliability across unseen mobile apps. On the efficiency front, AutoDroid-V2 and MobileGPT push the boundary by incorporating on-device models and reusable memory structures, demonstrating improvements not only in success rates but also in latency and query efficiency—critical for practical deployment. Collectively, these frameworks mark substantial progress in the field, revealing how combining vision-language reasoning, hierarchical planning, and efficient execution mechanisms leads to higher task generalization, reduced computational overhead, and improved real-world applicability of mobile GUI agents.

### 5.3 Computer GUI Agents

Computer GUI agents have evolved to offer complex automation capabilities across diverse operating systems, addressing challenges such as cross-application interaction, task generalization, and high-level task planning. They have led to the development of sophisticated frameworks capable of handling complex tasks across desktop environments. These agents have evolved from simple automation tools to intelligent systems that leverage multimodal inputs, advanced architectures, and adaptive learning to perform multi-application tasks with high efficiency and adaptability. We provide an overview of computer GUI agent frameworks in Table 28, 29 and 30.

One significant development in this area is the introduction of multi-agent architectures that enhance task management and execution. For instance, the UI-Focused Agent, **UFO** Zhang et al. (2024a) represents a pioneering framework specifically designed for the Windows operating system. UFO redefines UI-focused automation through its advanced dual-agent architecture, leveraging GPT-Vision to interpret GUI elements and execute actions autonomously across multiple applications. The framework comprises a HostAgent, responsible for global planning, task decomposition, and application selection, and an AppAgent, tasked with executing assigned subtasks within individual applications, as illustrated in Figure 22. This centralized structure enables UFO to manage complex, multi-application workflows such as aggregating information and generating reports. Similar architectural approach has also been adopted by other GUI agent frameworks AgentSeaf AI (2024); Zhu et al. (2024b); Zhang et al. (2024e). By incorporating safeguards and customizable

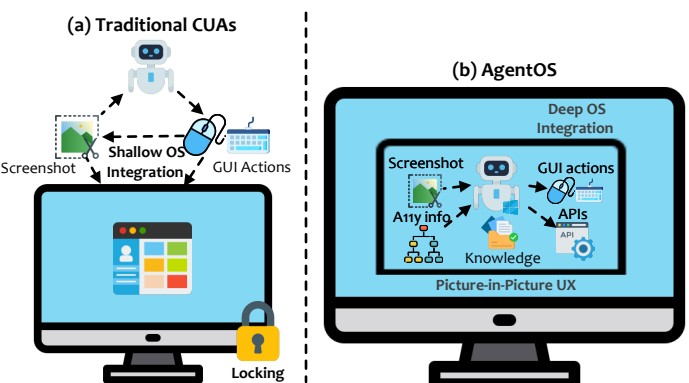

Figure 23: The comparison of traditional CUAs and the Desktop AgentOS UFO$^2$. Figure adapted from the original paper.

actions, UFO ensures efficiency and security when handling intricate commands, positioning itself as a cutting-edge assistant for Windows OS. Its architecture, exemplifies dynamic adaptability and robust task-solving capabilities across diverse applications, demonstrating the potential of multi-agent systems in desktop automation.

**UFO$^2$** Zhang et al. (2025b), the successor to UFO, elevates GUI automation from a vision-only prototype to a deeply integrated, Windows-native *AgentOS* (Figure 23). It coordinates tasks through a centralized *HostAgent*, which delegates subtasks to application-specialized *AppAgents*. A hybrid perception pipeline that fuses Windows UI Automation (UIA) metadata with OmniParser-v2 visual grounding delivers robust control identification even for custom widgets. Via a unified GUI–API action layer, AppAgents preferentially invoke high-level application APIs and fall back to pixel-level clicks only when necessary, cutting both latency and brittleness. A picture-in-picture virtual desktop cleanly isolates agent execution from the user's main session, enabling non-intrusive multitasking. Runtime performance is further boosted by retrieval-augmented help documents and execution logs, coupled with speculative multi-action planning that executes several steps per single LLM invocation. Tested on 20+ real Windows applications, **UFO$^2$** exceeds Operator OpenAI (2025b) and other CUAs by more than 10 percentage points in success rate while halving LLM calls. Because the framework is model-agnostic, swapping GPT-4o for a stronger LLM such as *o1* yields additional gains without code changes.

Building upon the theme of adaptability and generalist capabilities, **Cradle** Tan et al. (2024a) pushes the boundaries of general computer control by utilizing VLMs for interacting with various software, ranging from games to professional applications, without the need for API access. Cradle employs GPT-4o to interpret screen inputs and perform low-level actions, making it versatile across different types of software environments. Its six-module structure, covering functions such as information gathering and self-reflection, enables the agent to execute tasks, reason about actions, and utilize past interactions to inform future decisions. Cradle's capacity to function in dynamic environments, including complex software, marks it as a significant step toward creating generalist agents with broad applicability across desktop environments.

Extending the capabilities of computer GUI agents to multiple operating systems, **OS-Copilot** Wu et al. (2024e) introduces a general-purpose framework designed to operate across Linux and macOS systems. Its notable feature, **FRIDAY**, showcases the potential of self-directed learning by adapting to various applications and performing tasks without explicit training for each app. Unlike application-specific agents, FRIDAY integrates APIs, keyboard and mouse controls, and command-line operations, creating a flexible platform that can autonomously generate and refine tools as it interacts with new applications. OS-Copilot's ability to generalize across unseen applications, validated by its performance on the GAIA benchmark, provides a foundational model for OS-level agents capable of evolving in complex environments. This demonstrates promising directions for creating adaptable digital assistants that can handle diverse desktop environments and complex task requirements.

In the emerging field of LLM-powered GUI agents for desktop environments, Programming with Pixels (PwP) Aggarwal & Welleck (2025) introduces a compelling alternative to traditional tool-based software engineering agents, as illustrated in Figure X. Rather than relying on predefined API calls, PwP enables agents to interact directly with an IDE using visual perception, keyboard inputs, and mouse clicks, mimicking the way human developers operate within an IDE. This approach allows for generalization beyond predefined APIs, providing a highly expressive environment where agents can execute a wide range of software engineering tasks, including debugging, UI generation, and code editing. Evaluations conducted on PwP-Bench demonstrate that computer-use agents, despite lacking direct access to structured APIs, can match or even surpass traditional tool-based approaches in certain scenarios.

**Summary**  In summary, computer GUI agents have evolved significantly, progressing from single-task automation tools to advanced multi-agent systems capable of performing complex, multi-application tasks and learning from interactions. Frameworks such as UFO, Cradle, and OS-Copilot exemplify different design philosophies, each addressing core challenges like cross-application control, adaptability to unseen environments, and efficient long-horizon planning. UFO showcases a highly structured dual-agent architecture that achieves robust task decomposition and accurate subtask execution within the Windows ecosystem, demonstrating superior performance in cross-app workflows like document synthesis and report generation. In contrast, Cradle focuses on generality and reasoning by leveraging a six-module pipeline with self-reflection and memory to handle diverse software types, including games and creative tools—highlighting strong adaptability without relying on application-specific APIs. Meanwhile, OS-Copilot pushes the boundaries of platform generalization, achieving cross-OS compatibility (Linux/macOS) via a hybrid interaction approach combining CLI, GUI, and API-level control, with demonstrated success on the GAIA benchmark for unseen applications. Together, these systems illustrate a clear trend: the most effective computer GUI agents integrate multimodal perception, modular reasoning, and dynamic learning mechanisms to balance generality and performance, setting the foundation for future AgentOS frameworks that are scalable, OS-agnostic, and capable of long-term autonomous operation in real-world desktop environments.

## 5.4 Cross-Platform GUI Agents

Cross-platform GUI agents have emerged as versatile solutions capable of interacting with various environments, from desktop and mobile platforms to more complex systems. These frameworks prioritize adaptability and efficiency, leveraging both lightweight models and multi-agent architectures to enhance cross-platform operability. In this subsection, we first We overview cross-platform GUI agent frameworks in Table 31 and 32, then explore key frameworks that exemplify the advancements in cross-platform GUI automation.

A significant stride in this domain is represented by **AutoGLM** Liu et al. (2024h), which bridges the gap between web browsing and Android control by integrating large multimodal models for seamless GUI interactions across platforms. AutoGLM introduces an Intermediate Interface Design that separates planning and grounding tasks, improving dynamic decision-making and adaptability. By employing a self-evolving online curriculum with reinforcement learning, the agent learns incrementally from real-world feedback and can recover from errors. This adaptability and robustness make AutoGLM ideal for real-world deployment in diverse user applications, setting a new standard in cross-platform automation and offering promising directions for future research in foundation agents.

While some frameworks focus on integrating advanced models for cross-platform interactions, others emphasize efficiency and accessibility. **TinyClick** Pawlowski et al. (2024) addresses the need for lightweight solutions by focusing on single-turn interactions within GUIs. Utilizing the Florence-2-Base Vision-Language Model, TinyClick executes tasks based on user commands and screenshots with only 0.27 billion parameters. Despite its compact size, it achieves high accuracy—73% on Screenspot Cheng et al. (2024a) and 58.3% on OmniAct Kapoor et al. (2024) —outperforming larger multimodal models like GPT-4V while maintaining efficiency. Its multi-task training and MLLM-based data augmentation enable precise UI element localization, making it suitable for low-resource environments and addressing latency and resource constraints in UI grounding and action execution.

In addition to lightweight models, multi-agent architectures play a crucial role in enhancing cross-platform GUI interactions. **OSCAR** Wang & Liu (2024) exemplifies this approach by introducing a generalist GUI

agent capable of autonomously navigating and controlling both desktop and mobile applications. By utilizing a state machine architecture, OSCAR dynamically handles errors and adjusts its actions based on real-time feedback, making it suitable for automating complex workflows guided by natural language. The integration of standardized OS controls, such as keyboard and mouse inputs, allows OSCAR to interact with applications in a generalized manner, improving productivity across diverse GUI environments. Its open-source design promotes broad adoption and seamless integration, offering a versatile tool for cross-platform task automation and productivity enhancement.

Expanding on the concept of multi-agent systems, **AgentStore** Jia et al. (2024) introduces a flexible and scalable framework for integrating heterogeneous agents to automate tasks across operating systems. The key feature of AgentStore is the MetaAgent, which uses the innovative AgentToken strategy to dynamically manage a growing number of specialized agents. By enabling dynamic agent enrollment, the framework fosters adaptability and scalability, allowing both specialized and generalist capabilities to coexist. This multi-agent architecture supports diverse platforms, including desktop and mobile environments, leveraging multimodal perceptions such as GUI structures and system states. AgentStore's contributions highlight the importance of combining specialization with generalist capabilities to overcome the limitations of previous systems.

Further advancing cross-platform GUI interaction, **MMAC-Copilot** Song et al. (2024d) employs a multi-agent, multimodal approach to handle tasks across 3D gaming, office, and mobile applications without relying on APIs. By utilizing specialized agents like Planner, Viewer, and Programmer, MMAC-Copilot collaborates to adapt to the complexities of visually rich environments. Using GPT-4V for visual recognition and OCR for text analysis, it achieves high task completion rates in visually complex environments. The framework's integration with VIBench, a benchmark for non-API applications, underscores its real-world relevance and adaptability. MMAC-Copilot's robust foundation for dynamic interaction across platforms extends applications to industries like gaming, healthcare, and productivity.

**AGUVIS** Xu et al. (2024k) leverages a pure vision approach to automate GUI interactions, overcoming limitations of text-based systems like HTML or accessibility trees. Its platform-agnostic design supports web, desktop, and mobile applications while reducing inference costs. AGUVIS employs a two-stage training process: the first focuses on GUI grounding, and the second integrates planning and reasoning within a unified model. This approach delivers state-of-the-art performance in both offline and online scenarios, streamlining decision-making and execution.

**Agent S2 Agashe et al. (2025)** builds upon its predecessor, **Agent S Agashe et al. (2024)**, by introducing a hierarchical and compositional framework for GUI agents that integrates generalist models with specialized grounding modules. Departing from monolithic architectures, it employs a Mixture of Grounding (MoG) strategy to delegate fine-grained grounding tasks to expert modules, and adopts Proactive Hierarchical Planning (PHP) to dynamically revise action plans based on evolving observations. Relying solely on GUI screenshots, Agent S2 generalizes effectively across Ubuntu, Windows, and Android platforms. It demonstrates strong scalability and consistently outperforms larger monolithic models by strategically distributing cognitive responsibilities. The design of Agent S2 underscores the advantages of modular architectures for handling long-horizon, high-fidelity GUI interactions.

**Summary**  In summary, cross-platform GUI agents have demonstrated significant progress in bridging heterogeneous environments through varied design choices—ranging from lightweight efficiency to multi-agent scalability and robust multimodal perception. Frameworks such as AutoGLM and MMAC-Copilot highlight the power of large multimodal models and curriculum-based reinforcement learning to achieve high task success in diverse, visually rich environments without reliance on APIs. These systems exhibit strong adaptability, error recovery, and high task completion rates across domains such as mobile control, 3D gaming, and office automation. In contrast, models like TinyClick illustrate the feasibility of achieving competitive performance—surpassing even GPT-4V in some benchmarks—using highly compact architectures, thus addressing latency and resource constraints critical for real-time or on-device applications. Meanwhile, OSCAR and AgentStore exemplify the growing role of modular multi-agent systems, combining state tracking, error correction, and flexible enrollment of specialized agents to achieve scalability and platform coverage. These architectures enable dynamic agent composition, facilitating generalist behavior without sacrificing

domain-specific precision. Notably, AGUVIS introduces a vision-only paradigm, unifying GUI grounding, planning, and reasoning into a platform-agnostic model that generalizes across web, mobile, and desktop systems. Collectively, these frameworks reflect a clear trend toward combining scalability, multimodal grounding, and architectural efficiency to meet the demands of cross-platform GUI automation in increasingly complex, real-world settings.

# 6 Data for Optimizing LLM-Powered GUI Agents

In the previous section, we explored general frameworks for LLM-powered GUI agents, most of which rely on foundational LLMs such as GPT-4V and GPT-4o. However, to elevate these agents' performance and efficiency, optimizing their "brain", the underlying model is crucial. Achieving this often involves fine-tuning foundational models using large-scale, diverse, and high-quality contextual GUI datasets Li et al. (2024e), which are specifically curated to enable these models to excel in GUI-specific tasks. Collecting such datasets, particularly those rich in GUI screenshots, metadata, and interactions, necessitates an elaborate process of data acquisition, filtering, and preprocessing, each requiring substantial effort and resources Chen et al. (2024b).

As GUI agents continue to gain traction, researchers have focused on assembling datasets that represent a broad spectrum of platforms and capture the diverse intricacies of GUI environments. These datasets are pivotal in training models that can generalize effectively, thanks to their coverage of varied interfaces, workflows, and user interactions. To ensure comprehensive representation, innovative methodologies have been employed to collect and structure these data assets. In the sections that follow, we detail an end-to-end pipeline for data collection and processing tailored to training GUI-specific LLMs. We also examine significant datasets from various platforms, providing insights into their unique features, the methodologies used in their creation, and their potential applications in advancing the field of LLM-powered GUI agents.

## 6.1 Data Collection

Data is pivotal in training a purpose-built GUI agent, yet gathering it requires substantial time and effort due to the task's complexity and the varied environments involved.

### 6.1.1 Data Composition and Sources

The essential data components for GUI agent training closely align with the agent's perception and inference requirements discussed in Sections 4.2.2 and 4.4. At a high level, this data comprises:

1. **User Instructions:** These provide the task's overarching goal, purpose, and specific details, typically in natural language, offering a clear target for the agent to accomplish, *e.g.*, "change the font size of all text to 12".

2. **Environment Perception:** This typically includes GUI screenshots, often with various visual augmentations, as well as optional supplementary data like widget trees and UI element properties to enrich the context. This should encompass both the static assessment of environment states (Section 4.2.2) and the dynamic environment feedback that captures post-action changes (Section 4.2.3), thereby providing sufficient contextual information.

3. **Task Trajectory:** This contains the critical action sequence required to accomplish the task, along with supplementary information, such as the agent's plan. A trajectory usually involves multiple steps and actions to navigate through the task.

While user instructions and environmental perception serve as the model's input, the expected model output is the task trajectory. This trajectory's action sequence is then grounded within the environment to complete the task.

For **user instructions**, it is crucial to ensure that they are realistic and reflective of actual user scenarios. Instructions can be sourced in several ways: *(i)* directly from human designers, who can provide insights

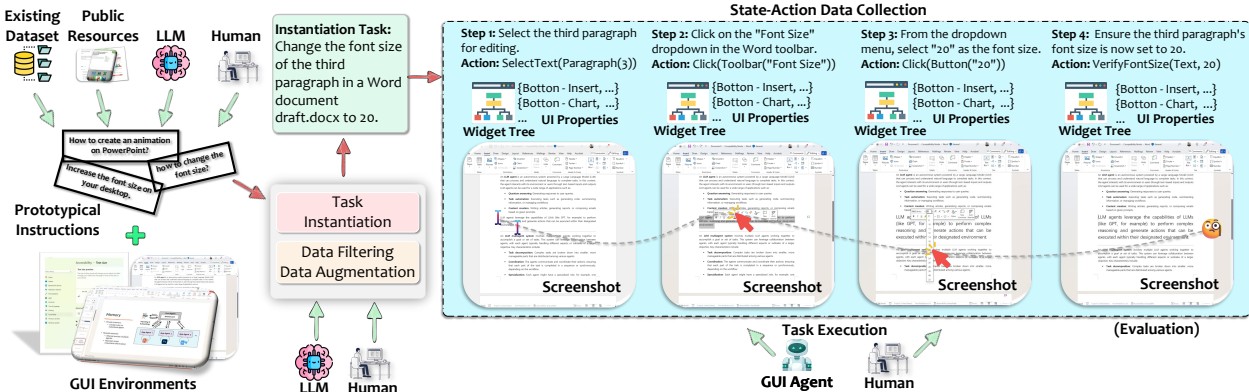

Figure 24: A complete pipeline for data collection for training a GUI agent model.

based on real-world applications; *(ii)* extracted from existing, relevant datasets if suitable data is available; *(iii)* sourcing from public materials, such as websites, application help documentation, and other publicly available resources; and *(iv)* generated by LLMs, which can simulate a broad range of user requests across different contexts. Additionally, LLMs can be employed for data augmentation Ding et al. (2024a), increasing both the quality and diversity of instructions derived from the original data.

For gathering **environment perception** data, various toolkits—such as those discussed in Section 4.2.2—can be used to capture the required GUI data. This can be done within an environment emulator (*e.g.*, Android Studio Emulator[30], Selenium WebDriver[31], Windows Sandbox[32]) or by directly interfacing with a real environment to capture the state of GUI elements, including screenshots, widget trees, and other metadata essential for the agent's operation.

Collecting **task trajectories**, which represent the agent's action sequence to complete a task, is often the most challenging aspect. Task trajectories need to be accurate, executable, and well-validated. Collection methods include *(i)* using programmatically generated scripts, which define action sequences for predefined tasks, providing a highly controlled data source; *(ii)* employing human annotators, who complete tasks in a crowdsourced manner with each step recorded, allowing for rich, authentic action data; and *(iii)* leveraging model or agent bootstrapping Tan et al. (2024b), where an existing LLM or GUI agent attempts to complete the task and logs its actions, though this method may require additional validation due to potential inaccuracies. All these methods demand considerable effort, reflecting the complexities of gathering reliable, task-accurate data for training GUI agents.

### 6.1.2 Collection Pipeline

Figure 24 presents a complete pipeline for data collection aimed at training a GUI agent model. The process begins with gathering initial user instructions, which may come from various aforementioned sources. These instructions are typically prototypical, not yet tailored or grounded to a specific environment Liu et al. (2024i). For instance, an instruction like "how to change the font size?" from a general website lacks specificity and doesn't align with the concrete requests a user might make within a particular application. To address this, an instantiation step is required Liu et al. (2024i), where instructions are contextualized within a specific environment, making them more actionable. For example, the instruction might be refined to "Change the font size of the third paragraph in a Word document of `draft.docx` to 20.", giving it a clear, environment-specific goal. This instantiation process can be conducted either manually by humans or programmatically with an LLM.

---

[30]https://developer.android.com/studio

[31]https://www.selenium.dev/

[32]https://learn.microsoft.com/en-us/windows/security/application-security/application-isolation/windows-sandbox/windows-sandbox-overview

Following instantiation, instructions may undergo a filtering step to remove low-quality data, ensuring only relevant and actionable instructions remain. Additionally, data augmentation techniques can be applied to expand and diversify the dataset, improving robustness. Both of these processes can involve human validation or leverage LLMs for efficiency.

Once instruction refinement is complete, task trajectories and environment perceptions are collected simultaneously. As actions are performed within the environment, each step is logged, providing a record of the environment's state and the specific actions taken. After a full task trajectory is recorded, an evaluation phase is necessary to identify and remove any failed or inaccurate sequences, preserving the quality of the dataset. By iterating this pipeline, a high-quality dataset of GUI agent data can be compiled, which is crucial for training optimized models.

Data collection and curation for LLM-powered GUI agents is an intensive process, often requiring substantial human involvement, particularly for generating accurate action sequences and annotations. While early datasets were limited in scale and task diversity, recent advancements have led to large-scale, multi-platform datasets that support more complex and realistic GUI interactions. Key insights from these developments include:

1. **Scale and Diversity:** High-quality, large-scale data is essential for training robust GUI agents capable of handling diverse UI states and tasks. Datasets like MobileViews Gao et al. (2024e) and ScreenAI Baechler et al. (2024) illustrate the importance of vast and varied data to accommodate the dynamic nature of mobile and desktop applications, enhancing the agent's resilience across different environments.

2. **Cross-Platform Flexibility:** Cross-platform datasets such as VisualAgentBench Liu et al. (2024i) and GUI-World Chen et al. (2024c) underscore the value of generalizability, enabling agents to perform consistently across mobile, web, and desktop environments. This cross-platform adaptability is a crucial step towards creating one-stop solutions where a single GUI agent can operate seamlessly across multiple platforms.

3. **Automated Data Collection:** AI-driven data collection tools, as exemplified by OmniParser Lu et al. (2024d) and MobileViews Gao et al. (2024e), showcase the potential to significantly reduce manual efforts and accelerate scalable dataset creation. By automating the annotation process, these tools pave the way for more efficient data pipelines, moving towards a future where AI supports AI by expediting data gathering and labeling for complex GUI interactions.

4. **Unified Data Formats and Protocols:** xLAM's unified data format is an essential innovation that improves compatibility across diverse platforms Zhang et al. (2024d), addressing a significant bottleneck in cross-platform GUI agent development. Establishing standardized protocols or action spaces for data collection, particularly given the varied data formats, action spaces, and environment representations across platforms, will be vital in furthering agent generalization and consistency.

In the following sections, we review existing GUI agent datasets across various platforms, offering insights into current practices and potential areas for improvement. Before delving into the details, we first present a taxonomy of LLM-powered GUI agent datasets, categorized by target platform, as shown in Table 12.

## 6.2   Web Agent Data

Web-based GUI agents demand datasets that capture the intricate complexity and diversity of real-world web interactions. These datasets often encompass varied website structures, including DOM trees and HTML content, as well as multi-step task annotations that reflect realistic user navigation and interaction patterns. Developing agents that can generalize across different websites and perform complex tasks requires comprehensive datasets that provide rich contextual information. We provide an overview of web-based GUI agents dataset in Table 33 and 34.

Building upon this need, several significant datasets have been developed to advance web-based GUI agents. Unlike traditional datasets focusing on narrow, predefined tasks, **Mind2Web** Deng et al. (2023) represents a

Table 12: Taxonomy of LLM-powered GUI agent datasets by target platform.

| Platform | References |
|---|---|
| **Web** | Deng et al. (2023); Chen et al. (2024e); Lu et al. (2024c); Pan et al. (2024b); Xu et al. (2024j); Trabucco et al. (2025) |
| **Mobile** | Li et al. (2020a); Zhang et al. (2024g); You et al. (2025); Wu et al. (2024c); Meng et al. (2024); Deka et al. (2017); Burns et al. (2022); Sun et al. (2022); Rawles et al. (2023); Lu et al. (2024b); Chai et al. (2024); Chen et al. (2024f); Wang et al. (2024f); Xu et al. (2024i); Gao et al. (2024e); Sun et al. (2025b) |
| **Computer** | Niu et al. (2024); Wang et al. (2024h); Xu et al. (2025) |
| **Cross-Platform** | Lu et al. (2024d); Chen et al. (2024i); Gou et al. (2024); Sun et al. (2024b); Chawla et al. (2024); Chen et al. (2024c); Zhang et al. (2024d); Shen et al. (2024a); Liu et al. (2024i); Baechler et al. (2024); Chaimalas et al. (2025); Cheng et al. (2025b) |

significant step forward by emphasizing open-ended task descriptions, pushing agents to interpret high-level goals independently. It offers over 2,350 human-annotated tasks across 137 diverse websites, capturing complex interaction patterns and sequences typical in web navigation. This setup aids in evaluating agents' generalization across unseen domains and serves as a benchmark for language grounding in web-based GUIs, enhancing adaptability for real-world applications.

Similarly, **WebVLN** Chen et al. (2024e) expands on web GUI tasks by combining navigation with question-answering. It provides agents with text-based queries that guide them to locate relevant web pages and extract information. By leveraging both HTML and visual content from websites, WebVLN aligns with real-world challenges of web browsing. This dataset is particularly valuable for researchers aiming to develop agents capable of complex, human-like interactions in GUI-driven web spaces.

Moreover, **WebLINX** Lu et al. (2024c) focuses on conversational GUI agents, particularly emphasizing real-world web navigation through multi-turn dialogue. Featuring over 2,300 expert demonstrations across 155 real-world websites, WebLINX creates a rich environment with DOM trees and screenshots for training and evaluating agents capable of dynamic, user-guided navigation tasks. This dataset promotes agent generalization across new sites and tasks, with comprehensive action and dialogue data that provide insights into enhancing agent responsiveness in realistic web-based scenarios.

**MultiUI** Liu et al. (2024e) is a large-scale dataset designed to enhance GUI agents' text-rich visual understanding. It comprises 7.3 million multimodal instruction samples collected from 1 million websites, covering key web UI tasks such as element grounding, action prediction, and interaction modeling. Unlike traditional datasets that rely on raw HTML, MultiUI utilizes structured accessibility trees to generate high-quality multimodal instructions. Models trained on MultiUI demonstrate substantial performance improvements, achieving a 48% gain on VisualWebBench Liu et al. (2024f) and a 19.1% increase in element accuracy on Mind2Web Deng et al. (2023).

**InSTA Trabucco et al. (2025)** is an Internet-scale dataset for training GUI-based web agents, generated entirely through an automated LLM pipeline without human annotations. It covers 150k diverse websites sourced from Common Crawl and includes rich web navigation tasks, trajectories in Playwright API calls, and evaluations using LLM-based judges. The dataset highlights strong generalization capabilities and data efficiency, significantly outperforming human-collected datasets like Mind2Web Deng et al. (2023) and WebLINX Lu et al. (2024c) in zero-shot and low-resource settings. InSTA represents a key advancement in scalable data curation for LLM-powered GUI agents, offering unprecedented coverage across real-world web interfaces.

Collectively, these datasets represent essential resources that enable advancements in web agent capabilities, supporting the development of adaptable and intelligent agents for diverse web applications.

### 6.3 Mobile Agent Data

Mobile platforms are critical for GUI agents due to the diverse range of apps and unique user interactions they involve. To develop agents that can effectively navigate and interact with mobile interfaces, datasets must offer a mix of single and multi-step tasks, focusing on natural language instructions, UI layouts, and user interactions. We first overview mobile GUI agents dataset in Tables 35, 36 and 37.

An early and foundational contribution in this domain is the **Rico** dataset Deka et al. (2017), which provides over 72,000 unique UI screens and 10,811 user interaction traces from more than 9,700 Android apps. Rico has been instrumental for tasks such as UI layout similarity, interaction modeling, and perceptual modeling, laying the groundwork for mobile interface understanding and GUI agent development.

Building upon the need for grounding natural language instructions to mobile UI actions, **PIXELHELP** Li et al. (2020a) introduces a dataset specifically designed for this purpose. It includes multi-step instructions, screenshots, and structured UI element data, enabling detailed analysis of how verbal instructions can be converted into mobile actions. This dataset has significant applications in accessibility and task automation, supporting agents that autonomously execute tasks based on verbal cues.

Further expanding the scope, the **Android in the Wild (AITW)** dataset Rawles et al. (2023) offers one of the most extensive collections of natural device interactions. Covering a broad spectrum of Android applications and diverse UI states, AITW captures multi-step tasks emulating real-world device usage. Collected through interactions with Android emulators, it includes both screenshots and action sequences, making it ideal for developing GUI agents that navigate app interfaces without relying on app-specific APIs. Due to its scale and diversity, AITW has become a widely used standard in the field.

In addition, **META-GUI** Sun et al. (2022) provides a unique dataset for mobile task-oriented dialogue systems by enabling direct interaction with mobile GUIs, bypassing the need for API-based controls. This approach allows agents to interact across various mobile applications using multi-turn dialogues and GUI traces, broadening their capabilities in real-world applications without custom API dependencies. The dataset's support for complex interactions and multi-turn dialogue scenarios makes it valuable for building robust conversational agents.

Recently, **MobileViews** Gao et al. (2024e) emerged as the largest mobile screen dataset to date, offering over 600,000 screenshot–view hierarchy pairs from 20,000 Android apps. Collected with an LLM-enhanced app traversal tool, it provides a high-fidelity resource for mobile GUI agents in tasks such as screen summarization, tappability prediction, and UI component identification. Its scale and comprehensive coverage of screen states make MobileViews a key resource for advancing mobile GUI agent capabilities.

Collectively, mobile platforms currently boast the richest set of datasets due to their versatile tools, emulator support, and diverse use cases, reflecting the demand for high-quality, adaptive GUI agents in mobile applications.

### 6.4 Computer Agent Data

In contrast to mobile and web platforms, the desktop domain for GUI agents has relatively fewer dedicated datasets, despite its critical importance for applications like productivity tools and enterprise software. However, notable efforts have been made to support the development and evaluation of agents designed for complex, multi-step desktop tasks. We show related dataset for computer GUI agents in Table 38.

A significant contribution in this area is **ScreenAgent** Niu et al. (2024), a dedicated dataset and model designed to facilitate GUI control in Linux and Windows desktop environments. ScreenAgent provides a comprehensive pipeline that enables agents to perform multi-step task execution autonomously, encompassing planning, action, and reflection phases. By leveraging annotated screenshots and detailed action sequences, it allows for high precision in UI element positioning and task completion, surpassing previous models in accuracy. This dataset is invaluable for researchers aiming to advance GUI agent capabilities in the desktop domain, enhancing agents' decision-making accuracy and user interface interactions.

The **LAM** Wang et al. (2024h) is specifically designed to train and evaluate Large Action Models (LAMs) for GUI environments, bridging natural language task understanding and action execution. It comprises two core components: Task-Plan data, detailing user tasks with step-by-step plans, and Task-Action data, translating these plans into executable GUI actions. Sourced from application documentation, WikiHow articles, and Bing search queries, the dataset is enriched and structured using GPT-4. Targeting the Windows OS, with a focus on automating tasks in Microsoft Word, it includes 76,672 task-plan pairs and 2,688 task-action trajectories, making it one of the largest collections for GUI-based action learning. Data quality is ensured through a robust validation pipeline that combines LLM-based instantiation, GUI interaction testing, and manual review. Each entry is complemented with GUI screenshots and metadata, enabling models to learn both high-level task planning and low-level execution. The dataset's modular design supports fine-tuning for specific GUI tasks and serves as a replicable framework for building datasets in other environments, marking a significant contribution to advancing GUI-based automation.

Although the desktop domain has fewer datasets compared to mobile and web, efforts like ScreenAgent and LAMs highlight the growing interest and potential for developing sophisticated GUI agents for computer systems.

## 6.5 Cross-Platform Agent Data

Cross-platform datasets play a pivotal role in developing versatile GUI agents that can operate seamlessly across mobile, computer, and web environments. Such datasets support generalizability and adaptability, enabling agents to handle varied interfaces and tasks in real-world applications. We provide an overview of related dataset for cross-platform GUI agents in Table 39 and Table 40.

One significant contribution is **ScreenAI** Baechler et al. (2024), which extends the scope of data collection to include both mobile and desktop interfaces. Covering tasks such as screen annotation, question-answering, and navigation, ScreenAI offers hundreds of millions of annotated samples. Its comprehensive scale and mixed-platform coverage make it a robust foundation for GUI agents that need to manage complex layouts and interactions across diverse interfaces. By emphasizing element recognition and screen summarization, ScreenAI advances the development of multi-platform GUI agents capable of handling varied visual structures.

Building upon the need for evaluating visual foundation models across environments, **VisualAgentBench** Liu et al. (2024i) is a groundbreaking cross-platform benchmark designed to assess GUI agents in both mobile and web settings. It emphasizes interaction-focused tasks, using environments like Android Virtual Device and WebArena-Lite Zhou et al. to evaluate and improve agent responses to GUI layouts and user interface actions. The dataset's innovative collection method, which combines program-based solvers and large multimodal model bootstrapping, facilitates robust training trajectories that enhance adaptability and error recovery in GUI agent tasks.

Furthermore, **GUI-World** Chen et al. (2024c) spans multiple platforms, including desktop, mobile, and XR environments, with over 12,000 annotated videos. Designed to address the challenges of dynamic and sequential GUI tasks, GUI-World allows researchers to benchmark GUI agent capabilities across diverse interfaces. By providing detailed action sequences and QA pairs, it sets a high standard for evaluating agents in complex, real-world scenarios.

Additionally, **xLAM**Zhang et al. (2024d) contributes significantly to actionable agent development by providing a unified dataset format designed to support multi-turn interactions, reasoning, and function-calling tasks. Sourced from datasets like WebShopYao et al. (2023), ToolBench Guo et al. (2024c), and AgentBoard Ma et al. (2024a), xLAM standardizes data formats across diverse environments, addressing the common issue of inconsistent data structures that hinder agent training and cross-environment compatibility. By offering a consistent structure, xLAM enhances the adaptability and error detection capabilities of GUI agents, allowing for more seamless integration and performance across different applications.

OS-Genesis Sun et al. (2024b) adopts a reverse task synthesis approach for the Android and web platforms. It leverages GPT-4o to interactively explore the environment and generate instructions in a reverse manner. This process constructs high-quality, diverse GUI trajectories without relying on human annotations or

predefined tasks. By eliminating these dependencies, OS-Genesis achieves scalable and efficient training for GUI agents while significantly enhancing the diversity and quality of the generated data.

Collectively, these cross-platform datasets contribute to building multi-platform GUI agents, paving the way for agents that can seamlessly navigate and perform tasks across different interfaces, fostering more generalized and adaptable systems.

## 7 Models for Optimizing LLM-Powered GUI Agents

LLMs act as the "brain" of GUI agents, empowering them to interpret user intents, comprehend GUI screens, and execute actions that directly impact their environments. While several existing foundation models are robust enough to serve as this core, they can be further fine-tuned and optimized to evolve into Large Action Models (LAMs)—specialized models tailored to improve the performance and efficiency of GUI agents. These LAMs bridge the gap between general-purpose capabilities and the specific demands of GUI-based interactions.

The exploration of LAMs for GUI agents has revealed several key insights that are shaping the future of intelligent interaction with graphical user interfaces:

1. **Smaller Models for On-Device Inference:** Many of the optimized LAMs are built from smaller foundational models, often ranging from 1 billion to 7 billion parameters. This reduction in model size enhances computational efficiency, making it feasible to deploy these models on resource-constrained devices such as mobile phones and edge devices. The ability to perform on-device inference without relying on cloud services addresses privacy concerns and reduces latency, leading to a more responsive user experience.

2. **Enhanced GUI Comprehension Reduces Reliance on Structured Data:** Models like VGA Meng et al. (2024) and OmniParser Lu et al. (2024d) emphasize the importance of visual grounding and image-centric fine-tuning to reduce dependency on structured UI metadata. By improving GUI comprehension directly from visual inputs, agents become more adaptable to different software environments, including those where structured data may be inaccessible or inconsistent.

3. **Reinforcement Learning Bridges Static and Dynamic Environments:** The application of reinforcement learning in models like DigiRL Bai et al. (2024) demonstrates the effectiveness of bridging static training data with dynamic real-world environments. This approach allows agents to learn from interactions, recover from errors, and adapt to changes, enhancing their robustness and reliability in practical applications.

4. **Unified Function-Calling Enhances Interoperability:** Efforts to standardize data formats and function-calling mechanisms, as seen in models like xLAM Zhang et al. (2024d), facilitate multi-turn interactions and reasoning across different platforms. This unification addresses compatibility issues and enhances the agent's ability to perform complex tasks involving multiple APIs and services.

5. **Inference-Time Computing and Reasoning Models:** Recent work highlights the importance of inference-time computing, where models plan, reason, and decompose tasks on the fly without architectural changes. Techniques such as extended context windows and chain-of-thought prompting (*e.g.*, "o1-style" reasoning) enable more robust, long-horizon decision-making. UI-R1 Lu et al. (2025b), GUI-R1 Xia & Luo (2025) and InfiGUI-R1 Liu et al. (2025f) are pioneering efforts in this direction. There is also growing interest in rule-based rewards and cost functions that guide inference-time behavior, integrating explicit heuristics to improve the stability, interpretability, and generalization of GUI agents.

In this section, we first introduce the foundation models that currently form the backbone of GUI agents, highlighting their strengths and limitations. We then delve into the concept of LAMs, discussing how these models are fine-tuned with GUI-specific datasets to enhance their adaptability, accuracy, and action-orientation in GUI environments. Through this exploration, we illustrate the progression from general-purpose LLMs

Table 13: Taxonomy of LLM-powered GUI agent models by target platform.

| Platform | References |
|---|---|
| **Foundation Models** | Wang et al. (2024j); Abdin et al. (2024); OpenAI (2025a); Li et al. (2023c); Bai et al. (2023b); Team et al. (2023); Hurst et al. (2024); Anthropic (2024); OpenAI (2023); Chen et al. (2024l;k); Wang et al. (2024m); You et al. (2023); Liu et al. (2024b;a); Huang et al. (2025c) |
| **Web** | Furuta et al. (2023); Putta et al. (2024); Fereidouni & Siddique (2024); Thil et al. (2024); He et al. (2024c); Qi et al. (2024); Zhang et al. (2025d) |
| **Mobile** | Chen et al. (2024f); Bai et al. (2024); You et al. (2025); Wu et al. (2024c); Meng et al. (2024); Qian et al. (2024); Chen et al. (2024g); Chen & Li (2024a;b;c); Nong et al. (2024); Li et al. (2024f); Wu et al. (2025b); Papoudakis et al. (2025); Bai et al. (2025a); Zheng et al. (2025b); Wang et al. (2025f); Lu et al. (2025b); Zhang et al. (2025d); Luo et al. (2025a) |
| **Computer** | Niu et al. (2024); Yang et al. (2025b); Wang et al. (2024h); Jin et al. (2025) |
| **Cross-Platform** | Hong et al. (2023); Cheng et al. (2024a); Shen et al. (2024a); Zhang et al. (2023e); Lu et al. (2024d); Baechler et al. (2024); Zhang et al. (2024d); Li et al. (2024j); Lin et al. (2024c); Wu et al. (2024f); Qin et al. (2025); Rahman et al. (2024); Yang et al. (2025a); Huang et al. (2025c); Xia & Luo (2025); Liu et al. (2025f) |

to purpose-built LAMs, laying the foundation for advanced, intelligent GUI agents. Before delving into the details, we first present a taxonomy of LLM-powered GUI agent models, categorized by target platform, as shown in Table 13.

## 7.1 Foundation Models

Foundation models serve as the core of LLM-powered GUI agents, providing the essential capabilities for understanding and interacting with graphical user interfaces. Recent advancements in both close-source and open-source MLLMs have significantly enhanced the potential of GUI agents, offering improvements in efficiency, scalability, and multimodal reasoning. This subsection explores these foundation models, highlighting their innovations, contributions, and suitability for GUI agent applications. For a quick reference, Table 41 and 42 present an overview of the key models and their characteristics.

### 7.1.1 Close-Source Models

While proprietary models are not openly available for customization, they offer powerful capabilities that can be directly utilized as the "brain" of GUI agents.

Among these, GPT-4V OpenAI (2023) and GPT-4o Hurst et al. (2024) are most commonly used in existing GUI agent frameworks due to their strong abilities, as discussed in Section 5. GPT-4V represents a significant advancement in multimodal AI, combining text and image analysis to expand the functionality of traditional LLMs. Its ability to understand and generate responses based on both textual and visual inputs makes it well-suited for GUI agent tasks that require deep multimodal reasoning. Although its deployment is limited due to safety and ethical considerations, GPT-4V underscores the potential of foundation models to revolutionize GUI agent development with enhanced efficiency and flexibility.

Similarly, GPT-4o offers a unified multimodal autoregressive architecture capable of processing text, audio, images, and video. This model excels in generating diverse outputs efficiently, achieving faster response times at lower costs compared to its predecessors. Its rigorous safety and alignment practices make it reliable for sensitive tasks, positioning it as a robust tool for intelligent GUI agents that require comprehensive multimodal comprehension.

The **Gemini** model family Team et al. (2023) advances multimodal AI modeling by offering versions tailored for high-complexity tasks, scalable performance, and on-device efficiency. Notably, the Nano models demonstrate significant capability in reasoning and coding tasks despite their small size, making them suitable for resource-constrained devices. Gemini's versatility and efficiency make it a compelling choice for powering GUI agents that require both performance and adaptability.

Emphasizing industry investment in GUI automation, **Claude 3.5 Sonnet (Computer Use)** introduces a pioneering approach by utilizing a vision-only paradigm for desktop task automation Anthropic (2024); Hu et al. (2024a). It leverages real-time screenshots to observe the GUI state and generate actions, eliminating the need for metadata or underlying GUI structure. This model effectively automates GUI tasks by interpreting the screen, moving the cursor, clicking buttons, and typing text. Its unique architecture integrates a ReAct-based Yao et al. (2022b) reasoning paradigm with selective observation, reducing computational overhead by observing the environment only when necessary. Additionally, Claude 3.5 maintains a history of GUI screenshots, enhancing task adaptability and enabling dynamic interaction with software environments in a human-like manner. Despite challenges in handling dynamic interfaces and error recovery, this model represents a significant step forward in creating general-purpose GUI agents. Its development highlights substantial industry investment in this area, indicating a growing focus on leveraging LLMs for advanced GUI automation.

The Operator model OpenAI (2025a;b), developed by OpenAI, represents a new frontier in Computer-Using Agents (CUA), akin to Claude 3.5 Sonnet (Computer Use). Designed to interact with GUI environments through LLM-powered reasoning and vision capabilities, Operator builds upon GPT-4o, integrating reinforcement learning to navigate and execute tasks across digital interfaces such as browsers, forms, and applications. By perceiving screenshots, interpreting UI elements, and performing actions via a virtual cursor and keyboard, Operator enables the automation of complex GUI-based workflows, including online purchases, email management, and document editing. Notably, Operator excels in understanding and manipulating digital environments, establishing itself as a powerful tool for human-computer interaction automation. Its exceptional performance on various benchmarks underscores its leading capabilities in GUI-based task automation.

### 7.1.2 Open-Source Models

Open-source models provide flexibility for customization and optimization, allowing developers to tailor GUI agents with contextual data and deploy them on devices with limited resources.

The **Qwen-VL** series Bai et al. (2023b) is notable for its fine-grained visual understanding and multimodal capabilities. With a Vision Transformer-based visual encoder and the Qwen-7B language model Bai et al. (2023a), it achieves state-of-the-art results on vision-language benchmarks while supporting multilingual interactions. Its efficiency and open-source availability, along with quantized versions for resource efficiency, make it suitable for developing GUI agents that require precise visual comprehension.

Building upon this, **Qwen2-VL** Wang et al. (2024j) introduces innovations like Naive Dynamic Resolution and Multimodal Rotary Position Embedding, enabling efficient processing of diverse modalities including extended-length videos. The scalable versions of Qwen2-VL balance computational efficiency and performance, making them adaptable for both on-device applications and complex multimodal tasks in GUI environments.

**InternVL-2** Chen et al. (2024l;k) combines a Vision Transformer with a Large Language Model to handle text, images, video, and medical data inputs. Its progressive alignment strategy and availability in various sizes allow for flexibility in deployment. By achieving state-of-the-art performance in complex multimodal tasks, InternVL-2 demonstrates powerful capabilities that are valuable for GUI agents requiring comprehensive multimodal understanding.

Advancing efficient integration of visual and linguistic information, **CogVLM** Wang et al. (2024m) excels in cross-modal tasks with a relatively small number of trainable parameters. Its ability to deeply integrate visual and language features while preserving the full capabilities of large language models makes it a cornerstone for GUI agent development, especially in applications where resource efficiency is critical.

Enhancing spatial understanding and grounding, **Ferret** You et al. (2023) offers an innovative approach tailored for GUI agents. By unifying referring and grounding tasks within a single framework and employing a hybrid region representation, it provides precise interaction with graphical interfaces. Its robustness against object hallucinations and efficient architecture make it ideal for on-device deployment in real-time GUI applications.

The **LLaVA** model Liu et al. (2024b) integrates a visual encoder with a language decoder, facilitating efficient alignment between modalities. Its lightweight projection layer and modular design enable quick experimentation and adaptation, making it suitable for GUI agents that require fast development cycles and strong multimodal reasoning abilities. Building on this, **LLaVA-1.5** Liu et al. (2024a) introduces a novel MLP-based cross-modal connector and scales to high-resolution image inputs, achieving impressive performance with minimal training data. Its data efficiency and open-source availability pave the way for widespread use in GUI applications requiring detailed visual reasoning.

**BLIP-2** Li et al. (2023c) employs a compute-efficient strategy by leveraging frozen pre-trained models and introducing a lightweight Querying Transformer. This design allows for state-of-the-art performance on vision-language tasks with fewer trainable parameters. BLIP-2's modularity and efficiency make it suitable for resource-constrained environments, highlighting its potential for on-device GUI agents.

Finally, **Phi-3.5-Vision** Abdin et al. (2024) achieves competitive performance in multimodal reasoning within a compact model size. Its innovative training methodology and efficient integration of image and text understanding make it a robust candidate for GUI agents that require multimodal reasoning and on-device inference without the computational overhead of larger models.

In summary, both close-source and open-source foundation models have significantly advanced the capabilities of LLM-powered GUI agents. While proprietary models offer powerful out-of-the-box performance, open-source models provide flexibility for customization and optimization, enabling tailored solutions for diverse GUI agent applications. The innovations in multimodal reasoning, efficiency, and scalability across these models highlight the evolving landscape of foundation models, paving the way for more intelligent and accessible GUI agents.

## 7.2 Large Action Models

While general-purpose foundation LLMs excel in capabilities like multimodal understanding, task planning, and tool utilization, they often lack the specialized optimizations required for GUI-oriented tasks. To address this, researchers have introduced *Large Action Models* (LAMs)—foundation LLMs fine-tuned with contextual, GUI-specific datasets (as outlined in Section 6) to enhance their action-driven capabilities. These models represent a significant step forward in refining the "brain" of GUI agents for superior performance.

In the realm of GUI agents, LAMs provide several transformative advantages:

1. **Enhanced Action Orientation:** By specializing in action-oriented tasks, LAMs enable accurate interpretation of user intentions and generation of precise action sequences. This fine-tuning ensures that LAMs can seamlessly align their outputs with GUI operations, delivering actionable steps tailored to user requests.

2. **Specialized Planning for Long, Complex Tasks:** LAMs excel in devising and executing intricate, multi-step workflows. Whether the tasks span multiple applications or involve interdependent operations, LAMs leverage their training on extensive action sequence datasets to create coherent, long-term plans. This makes them ideal for productivity-focused tasks requiring sophisticated planning across various tools.

3. **Improved GUI Comprehension and Visual Grounding:** Training on datasets that incorporate GUI screenshots allows LAMs to advance their abilities in detecting, localizing, and interpreting UI components such as buttons, menus, and forms. By utilizing visual cues instead of relying solely on structured UI metadata, LAMs become highly adaptable, performing effectively across diverse software environments.

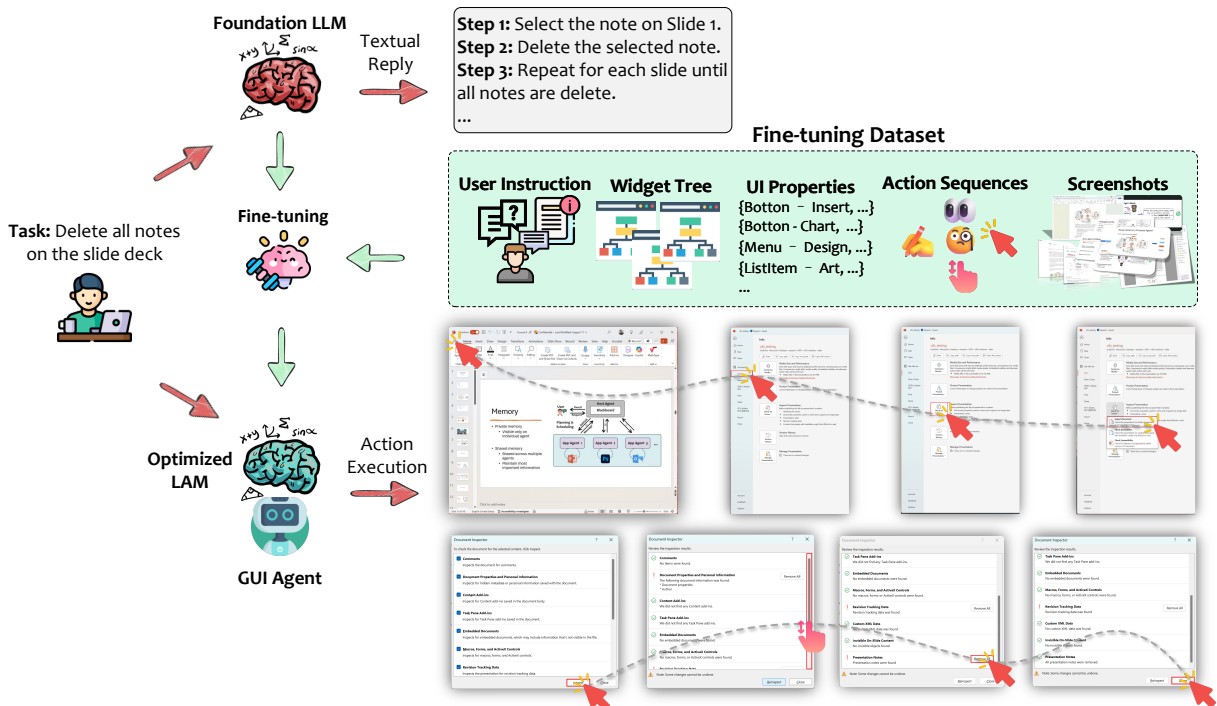

Figure 25: The evolution from foundation LLMs to GUI agent-optimized LAM with fine-tuning.

4. **Efficiency through Model Size Reduction:** Many LAMs are built on smaller foundational models—typically around 7 billion parameters—that are optimized for GUI-specific tasks. This compact, purpose-driven design reduces computational overhead, enabling efficient operation even in resource-constrained environments, such as on-device inference.

As illustrated in Figure 25, the process of developing a purpose-built LAM for GUI agents begins with a robust, general-purpose foundation model, ideally with VLM capabilities. Fine-tuning these models on comprehensive, specialized GUI datasets—including user instructions, widget trees, UI properties, action sequences, and annotated screenshots—transforms them into optimized LAMs, effectively equipping them to serve as the "brain" of GUI agents.

This optimization bridges the gap between planning and execution. A general-purpose LLM might provide only textual plans or abstract instructions in response to user queries, which may lack precision. In contrast, a LAM-empowered GUI agent moves beyond planning to actively and intelligently execute tasks on GUIs. By interacting directly with application interfaces, these agents perform tasks with remarkable precision and adaptability. This paradigm shift marks the evolution of GUI agents from passive task planners to active, intelligent executors.

## 7.3   LAMs for Web GUI Agents

In the domain of web-based GUI agents, researchers have developed specialized LAMs that enhance interaction and navigation within web environments. These models are tailored to understand the complexities of web GUIs, including dynamic content and diverse interaction patterns. We present an analysis of LAMs tailored for web GUI agents in Table 43.

Building upon the need for multimodal understanding, **WebGUM** Furuta et al. (2023) integrates HTML understanding with visual perception through temporal and local tokens. It leverages Flan-T5 Chung et al. (2024) for instruction fine-tuning and ViT Dosovitskiy et al. (2021) for visual inputs, enabling it to process both textual and visual information efficiently. This multimodal grounding allows WebGUM to generalize

tasks effectively, significantly outperforming prior models on benchmarks like MiniWoB++ Liu et al. (2018) and WebShop Yao et al. (2023). With its data-efficient design and capacity for multi-step reasoning, WebGUM underscores the importance of combining multimodal inputs in enhancing GUI agent performance.

Addressing the challenge of multi-step reasoning and planning in GUI environments, researchers have introduced frameworks that incorporate advanced search and learning mechanisms. For instance, **Agent Q** Putta et al. (2024) employs MCTS combined with self-critique mechanisms and Direct Preference Optimization (DPO) Rafailov et al. (2024) to improve success rates in complex tasks such as product search and reservation booking. By fine-tuning the LLaMA-3 70B model Dubey et al. (2024) to process HTML DOM representations and generate structured action plans, thoughts, and environment-specific commands, this framework showcases the power of integrating reasoning, search, and iterative fine-tuning for autonomous agent development.

Leveraging smaller models for efficient web interaction, **GLAINTEL** Fereidouni & Siddique (2024) demonstrates that high performance can be achieved without large computational resources. Utilizing the Flan-T5 Chung et al. (2024) model with 780M parameters, it focuses on dynamic web environments like simulated e-commerce platforms. The model incorporates RL to optimize actions such as query formulation and navigation, effectively integrating human demonstrations and unsupervised learning. Achieving results comparable to GPT-4-based methods at a fraction of the computational cost, GLAINTEL underscores the potential of reinforcement learning in enhancing web-based GUI agents for task-specific optimization.

To enable continuous improvement and generalization across varied web domains, **OpenWebVoyager** He et al. (2024c) combines imitation learning with an iterative exploration-feedback-optimization cycle. Leveraging large multimodal models like Idefics2-8B Laurençon et al. (2024a), it performs autonomous web navigation tasks. By training on diverse datasets and fine-tuning using trajectories validated by GPT-4 feedback, the agent addresses real-world complexities without relying on synthetic environments. This approach significantly advances GUI agent frameworks by demonstrating the capability to generalize across varied web domains and tasks.

Moreover, tackling challenges such as sparse training data and policy distribution drift, **WebRL** Qi et al. (2024) introduces a self-evolving curriculum and robust reward mechanisms for training LLMs as proficient web agents. By dynamically generating tasks based on the agent's performance, WebRL fine-tunes models like Llama-3.1 Dubey et al. (2024) and GLM-4 GLM et al. (2024), achieving significant success rates in web-based tasks within the WebArena environment. This framework outperforms both proprietary APIs and other open-source models, highlighting the effectiveness of adaptive task generation and sustained learning improvements in developing advanced GUI agents.

These advancements in LAMs for web GUI agents illustrate the importance of integrating multimodal inputs, efficient model designs, and innovative training frameworks to enhance agent capabilities in complex web environments.

### 7.4 LAMs for Mobile GUI Agents

Mobile platforms present unique challenges for GUI agents, including diverse screen sizes, touch interactions, and resource constraints. Researchers have developed specialized LAMs to address these challenges, enhancing interaction and navigation within mobile environments. We present an overview of LAMs tailored for mobile GUI agents in Table 44, 45 and 46.

Focusing on detailed UI understanding, **MobileVLM** Wu et al. (2024c) introduces an advanced vision-language model designed specifically for mobile UI manipulation tasks. Built on Qwen-VL-Chat Bai et al. (2023b), it incorporates mobile-specific pretraining tasks for intra- and inter-UI comprehension. By leveraging the Mobile3M dataset—a comprehensive corpus of 3 million UI pages and interaction traces organized into directed graphs—the model excels in action prediction and navigation tasks. MobileVLM's novel two-stage pretraining framework significantly enhances its adaptability to mobile UIs, outperforming existing VLMs in benchmarks like ScreenQA Hsiao et al. (2024) and Auto-UI Zhang & Zhang (2024). This work highlights the effectiveness of tailored pretraining in improving mobile GUI agent performance.

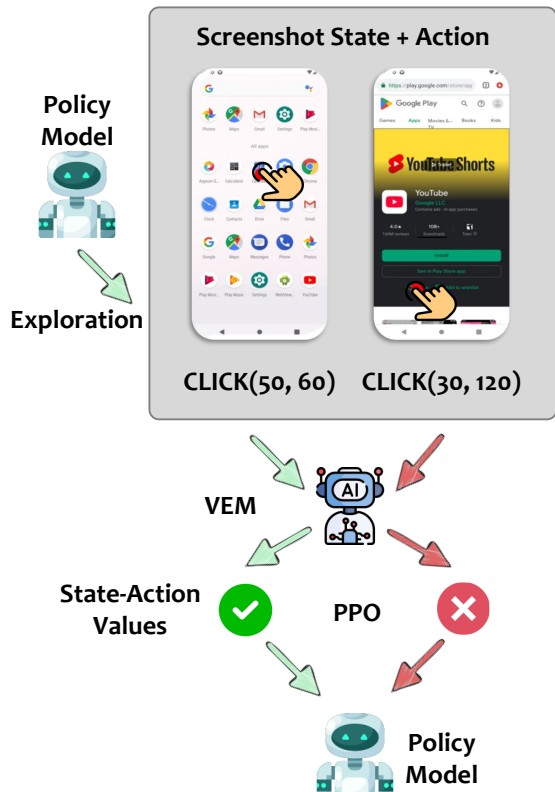

Figure 26: The PPO training process of VEM Zheng et al. (2025b). Figure adapted from the original paper.

Addressing the need for robust interaction in dynamic environments, **DigiRL** Bai et al. (2024) presents a reinforcement learning-based framework tailored for training GUI agents in Android environments. By leveraging offline-to-online RL, DigiRL adapts to real-world stochasticity, making it suitable for diverse, multi-step tasks. Unlike prior models reliant on imitation learning, DigiRL autonomously learns from interaction data, refining itself to recover from errors and adapt to new scenarios. The use of a pre-trained Vision-Language Model with 1.3 billion parameters enables efficient processing of GUI screenshots and navigation commands. Its performance on the AITW dataset demonstrates a significant improvement over baseline methods, positioning DigiRL as a benchmark in the development of intelligent agents optimized for complex GUI interactions.

Both **Digi-Q** Bai et al. (2025a) and **VEM** Zheng et al. (2025b) investigate the use of offline RL to enhance the performance of GUI agents without requiring direct interaction with the environment. Digi-Q employs temporal-difference learning to train a Q-function offline and derives policies through a Best-of-N selection strategy based on the predicted Q-values. Similarly, VEM introduces an environment-free RL framework tailored for training LLM-powered GUI agents using PPO. It directly estimates state-action values from offline data by fine-tuning with annotated value data from GPT-4o, thereby enabling policy training without real-time execution in a GUI environment. At inference time, only the policy model is utilized. Figure 26 illustrates the overall architecture of VEM. The study further demonstrates that offline RL with structured credit assignment can achieve performance comparable to interactive RL models. Overall, VEM offers a scalable and layout-agnostic approach for training GUI agents while minimizing interaction costs. Both works underscore the potential of offline RL for GUI agent training.

To enhance GUI comprehension and reduce reliance on textual data, **VGA** Meng et al. (2024) employs fine-tuned vision-language models that prioritize image-based cues such as shapes, colors, and positions. Utilizing the RICO Deka et al. (2017) dataset for training, VGA is tailored for Android GUIs and employs a two-stage fine-tuning process to align responses with both visual data and human intent. The model

excels in understanding GUI layouts, predicting design intents, and facilitating precise user interactions. By outperforming existing models like GPT-4V in GUI comprehension benchmarks, VGA sets a new standard for accuracy and efficiency in mobile GUI agents.

In the context of lightweight and efficient models, **UINav** Li et al. (2024f) demonstrates a practical system for training neural agents to automate UI tasks on mobile devices. It balances accuracy, generalizability, and computational efficiency through macro actions and an error-driven demonstration collection process. UINav uses a compact encoder-decoder architecture and SmallBERT Turc et al. (2019) for text and screen element encoding, making it suitable for on-device inference. A key innovation is its ability to generalize across diverse tasks and apps with minimal demonstrations, addressing key challenges in UI automation with a versatile framework.

**UI-R1** Lu et al. (2025b) introduces a RL–based training paradigm aimed at enhancing GUI action prediction for multimodal large language models (MLLMs). The resulting model, UI-R1-3B, fine-tunes Qwen2.5-VL-3B using a novel rule-based reward function that jointly evaluates action type correctness and click coordinate accuracy, while also enabling o1-style Jaech et al. (2024) chain-of-thought (CoT) reasoning through structured `<think>` tags. UI-R1 relies on only 136 high-quality samples selected via a three-stage filtering strategy. Despite this limited supervision, UI-R1-3B achieves significant improvements on both in-domain and out-of-domain benchmarks. By leveraging Group Relative Policy Optimization (GRPO) Shao et al. (2024), the framework aligns policy optimization with the goals of GUI grounding and task execution. UI-R1 establishes a scalable and data-efficient approach for training GUI agents via RL and paves the way for lightweight yet effective agent design. Its methodology has also been successfully extended to cross-platform agents Xia & Luo (2025); Liu et al. (2025f), demonstrating strong generalization capabilities.

In addition to action models, **ViMo** Luo et al. (2025a) introduces a novel generative visual world model for GUI agents, aimed at improving App agent decision-making by predicting the next GUI state as an image rather than a textual description. A key innovation of ViMo is the Symbolic Text Representation (STR), which replaces GUI text regions with structured placeholders to facilitate accurate and legible text synthesis. This decoupled design allows the system to handle GUI graphics generation using a fine-tuned diffusion model, and text generation through an LLM, thereby achieving high visual fidelity and semantic precision. ViMo significantly boosts both GUI prediction quality and downstream agent performance, with a reported 29.14% relative improvement in GUI generation metrics and enhanced planning accuracy for long-horizon tasks. As a forward simulator, ViMo represents a crucial advancement toward reliable world models for mobile GUI agents, supporting more effective decision evaluation and trajectory planning in visual environments.

These models collectively advance the field of mobile GUI agents by addressing platform-specific challenges through innovative training methods, efficient model architectures, and specialized datasets.

### 7.5 LAMs for Computer GUI Agents

For desktop and laptop environments, GUI agents must handle complex applications, multitasking, and varied interaction modalities. Specialized LAMs for computer GUI agents enhance capabilities in these settings, enabling more sophisticated task execution. We overview of LAMs for computer GUI agents across in Table 47.

Integrating planning, acting, and reflecting phases, **ScreenAgent** Niu et al. (2024) is designed for autonomous interaction with computer screens. Based on CogAgent Hong et al. (2023), it is fine-tuned using the ScreenAgent Dataset, providing comprehensive GUI interaction data across diverse tasks. With inputs as screenshots and outputs formatted in JSON for mouse and keyboard actions, ScreenAgent achieves precise UI element localization and handles continuous multi-step tasks. Its capability to process real-time GUI interactions using a foundation model sets a new benchmark for LLM-powered GUI agents, making it an ideal reference for future research in building more generalized intelligent agents.

Bridging high-level planning with real-world manipulation, **Octopus** Yang et al. (2025b) represents a pioneering step in embodied vision-language programming. Leveraging the MPT-7B MosaicML (2023) and CLIP ViT-L/14 Radford et al. (2021), Octopus integrates egocentric and bird's-eye views for visual comprehension, generating executable action code. Trained using the OctoVerse suite, its datasets encompass

richly annotated environments like OmniGibson, Minecraft, and GTA-V, covering routine and reasoning-intensive tasks. Notably, Octopus innovates through Reinforcement Learning with Environmental Feedback, ensuring adaptive planning and execution. Its vision-dependent functionality offers seamless task generalization in unseen scenarios, underscoring its capability as a unified model for embodied agents operating in complex GUI environments.

Wang *et al.*, Wang et al. (2024h) present a comprehensive overview of **LAMs**, a new paradigm in AI designed to perform tangible actions in GUI environments, using UFO Zhang et al. (2024a) at Windows OS as a case study platform. Built on the Mistral-7B Jiang et al. (2023) foundation, LAMs advance beyond traditional LLMs by integrating task planning with actionable outputs. Leveraging structured inputs from tools like the UI Automation (UIA) API, LAMs generate executable steps for dynamic planning and adaptive responses. A multi-phase training strategy—encompassing task-plan pretraining, imitation learning, self-boosting exploration, and reinforcement learning—ensures robustness and accuracy. Evaluations on real-world GUI tasks highlight LAMs' superior task success rates compared to standard models. This innovation establishes a foundation for intelligent GUI agents capable of transforming user requests into real-world actions, driving significant progress in productivity and automation.

These developments in computer GUI agents highlight the integration of advanced visual comprehension, planning, and action execution, paving the way for more sophisticated and capable desktop agents.

## 7.6 Cross-Platform Large Action Models

To achieve versatility across various platforms, cross-platform LAMs have been developed, enabling GUI agents to operate seamlessly in multiple environments such as mobile devices, desktops, and web interfaces. We provide an analysis of LAMs tailored for cross-platform GUI agents in Tables 48, 49 and 50.

**CogAgent** Hong et al. (2023) stands out as an advanced visual language model specializing in GUI understanding and navigation across PC, web, and Android platforms. Built on CogVLM Wang et al. (2024m), it incorporates a novel high-resolution cross-module to process GUI screenshots efficiently, enabling detailed comprehension of GUI elements and their spatial relationships. Excelling in tasks requiring OCR and GUI grounding, CogAgent achieves state-of-the-art performance on benchmarks like Mind2Web Deng et al. (2023) and AITW Rawles et al. (2023). Its ability to generate accurate action plans and interface with GUIs positions it as a pivotal step in developing intelligent agents optimized for GUI environments. CogAgent has further evolved into its beta version, GLM-PC CogAgent Team (2024), offering enhanced control capabilities.

Focusing on universal GUI understanding, **Ferret-UI 2** Li et al. (2024j) from Apple is a state-of-the-art multimodal large language model designed to master UI comprehension across diverse platforms, including iPhones, Android devices, iPads, web, and AppleTV. By employing dynamic high-resolution image encoding, adaptive gridding, and high-quality multimodal training data generated through GPT-4, it outperforms its predecessor and other competing models in UI referring, grounding, and interaction tasks. Ferret-UI 2's advanced datasets and innovative training techniques ensure high accuracy in spatial understanding and user-centered interactions, setting a new benchmark for cross-platform UI adaptability and performance.

Advancing GUI automation, **ShowUI** Lin et al. (2024c) introduces a pioneering Vision-Language-Action model that integrates high-resolution visual inputs with textual understanding to perform grounding, navigation, and task planning. Optimized for web, desktop, and mobile environments, ShowUI leverages the Phi-3.5-vision-instruct backbone and comprehensive datasets to achieve robust results across benchmarks like ScreenSpot Cheng et al. (2024a) and GUI-Odyssey Lu et al. (2024b). Its ability to process multi-frame and dynamic visual inputs alongside JSON-structured output actions highlights its versatility. With innovations in interleaved image-text processing and function-calling capabilities, ShowUI sets a new standard for LLM-powered GUI agents.

Addressing the need for a unified action space, **OS-ATLAS** Wu et al. (2024f) introduces a foundational action model specifically designed for GUI agents across platforms like Windows, macOS, Linux, Android, and the web. By leveraging a massive multi-platform dataset and implementing a unified action space, OS-ATLAS achieves state-of-the-art performance in GUI grounding and out-of-distribution generalization tasks. Its scalable configurations adapt to varying computational needs while maintaining versatility in handling natural

language instructions and GUI elements. As a powerful open-source alternative to commercial solutions, OS-ATLAS marks a significant step toward democratizing access to advanced GUI agents.

Magma Yang et al. (2025a) is a foundation model for multimodal AI agents that integrates LLMs with vision and action understanding to complete UI navigation and robotic manipulation tasks. Unlike previous models optimized for either UI automation or robotics, Magma jointly trains on a heterogeneous dataset (about 39M samples) spanning UI screenshots, web navigation, robot trajectories, and instructional videos. It employs SoM and Trace-of-Mark techniques, which enhance action grounding and prediction by labeling actionable elements in GUI environments and tracking motion traces in robotic tasks.

UI-TARS Qin et al. (2025) is an advanced, vision-based Large Action Model (LAM) optimized for multi-platform GUI agents. Unlike traditional approaches, it relies solely on GUI screenshots for perception, eliminating the need for structured representations. By incorporating a unified action space, UI-TARS enables seamless execution across Web, Windows, macOS, and Android environments. Built on Qwen-2-VL, it is trained on 6 million GUI tutorials, large-scale screenshot datasets, and multiple open-source benchmarks. A key innovation of UI-TARS is its System-2 reasoning capability, which allows it to generate explicit reasoning steps before executing actions, enhancing decision-making in dynamic environments. Additionally, it employs an iterative self-improvement framework, refining its performance through reflection-based learning. Experimental results demonstrate that UI-TARS outperforms existing models, including GPT-4o and Claude, in task execution benchmarks.

These cross-platform LAMs demonstrate the potential of unified models that can adapt to diverse environments, enhancing the scalability and applicability of GUI agents in various contexts.

### 7.7 Emerging Trends Amid Rapid Improvements in LAM Capabilities

Recent breakthroughs in LAM have driven considerable innovation in GUI agent research. While these large models excel at general reasoning, open-domain conversation, and multimodal interpretation, certain longstanding challenges in GUI-based automation remain only partially addressed. In this subsection, we outline three interrelated trends that are poised to remain central—even as baseline LAM capabilities continue to rise—and discuss how these trends may benefit from, or be constrained by, ongoing advancements in foundation models.

**Live Environments and Data Coverage.** A recurring challenge in GUI automation is acquiring rich, diverse data that truly reflect the complexities of real user interfaces. Many LAM systems are trained on static snapshots or curated datasets, which may not capture the dynamic, event-driven nature of GUI environments. As these agents become more capable, the need for live environment interaction grows in importance. Operating in a live environment allows agents to iteratively collect data and refine their understanding of various apps, platforms, or usage contexts—thereby improving coverage of niche scenarios and rare interface states. Nevertheless, live data collection can be both time-consuming and costly, raising questions about how to balance on-device exploration with robust offline datasets. Expanding open-source benchmarks and real-time simulators can help address coverage gaps without incurring prohibitive data-collection expenses, ensuring that agent training encompasses both realistic and representative GUI states.

**Reinforcement Learning for GUI Agents.** A second trend persists in using Reinforcement Learning (RL) to optimize agent behavior over extended interactions. With the advent of more powerful LAMs, RL can integrate dynamic feedback signals—such as rewards or constraints derived from user feedback, success rates, or time-to-completion. In online RL, agents interact with live systems to continually refine their policies, though the exploration cost can be substantial if an agent must repeatedly trial-and-error in production environments. Offline RL is a compelling alternative, enabling large-scale pretraining on recorded trajectories or user logs to reduce reliance on potentially expensive or risky online exploration. These methods can leverage improved baseline model capabilities to interpret and navigate complex UIs more effectively, while learning from failures or suboptimal states without exposing end users to frequent disruptions or errors.

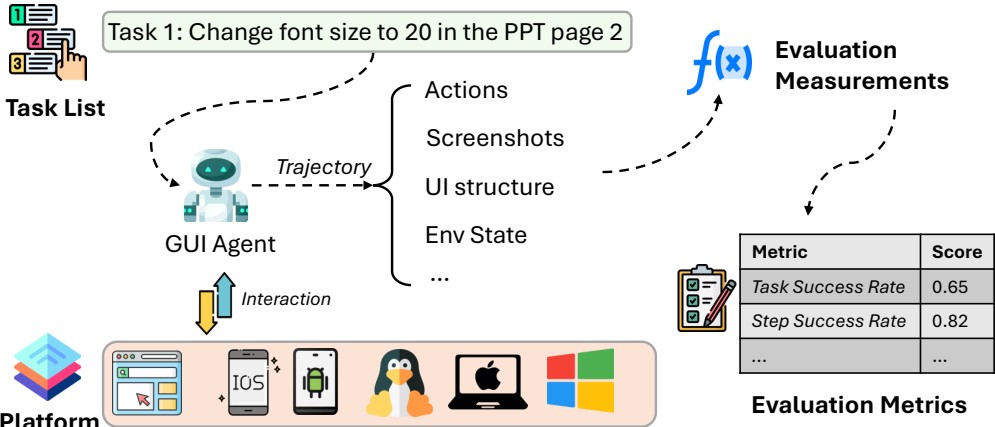

Figure 27: An illustrative example of evaluation of task completion by a GUI agent.

**Inference-Time Computing and Reasoning Models.** As LAM research accelerates, there is a growing emphasis on inference-time computing—where models dynamically plan, reason, and decompose tasks without extensive architectural modifications. Novel approaches (*e.g.*, "R1-like" extended context windows or chain-of-thought prompting) can deliver longer-horizon reasoning, thereby improving the accuracy and stability of GUI interactions. This can be especially beneficial for long or multi-step tasks that require the agent to keep track of evolving interface states and dependencies. In tandem, we see increased interest in rule-based rewards or cost functions that guide inference-time decisions—for example, penalizing unnecessary clicks or prioritizing user-facing safety checks. By integrating explicit heuristics with advanced model reasoning, GUI agents can achieve more stable and interpretable behavior, despite the inherent open-endedness of LLM-driven interactions.

# 8 Evaluation for LLM-Powered GUI Agents

In the domain of GUI agents, evaluation is crucial for enhancing both functionality and user experience Li et al. (2024d); Huang & Zhang (2024) and should be conducted across multiple aspects. By systematically assessing these agents' effectiveness across various tasks, evaluation not only gauges their performance in different dimensions but also provides a framework for their continuous improvement Liu et al. (2023b). Furthermore, it encourages innovation by identifying areas for potential development, ensuring that GUI agents evolve in tandem with advancements in LLMs and align with user expectations.

As illustrated in Figure 27, when a GUI agent completes a task, it produces an action sequence, captures screenshots, extracts UI structures, and logs the resulting environment states. These outputs serve as the foundation for evaluating the agent's performance through various metrics and measurements across diverse platforms. In the subsequent sections, we delve into these evaluation methodologies, discussing the metrics and measurements used to assess GUI agents comprehensively. We also provide an overview of existing benchmarks tailored for GUI agents across different platforms, highlighting their key features and the challenges they address.

## 8.1 Evaluation Metrics

Evaluating GUI agents requires robust and multidimensional metrics to assess their performance across various dimensions, including accuracy, efficiency, and compliance (*e.g.*, safety). In a typical benchmarking setup, the GUI agent is provided with a natural language instruction as input and is expected to autonomously execute actions until the task is completed. During this process, various assets can be collected, such as the sequence of actions taken by the agent, step-wise observations (*e.g.*, DOM or HTML structures), screenshots,

runtime logs, final states, and execution time. These assets enable evaluators to determine whether the task has been completed successfully and to analyze the agent's performance. In this section, we summarize the key evaluation metrics commonly used for benchmarking GUI agents. Note that different research works may use different names for these metrics, but with similar calculations. We align their names in this section.

1. **Step Success Rate:** Completing a task may require multiple steps. This metric measures the ratio of the number of steps that are successful over the total steps within a task. A high step success rate indicates precise and accurate execution of granular steps, which is essential for the reliable performance of tasks involving multiple steps Deng et al. (2023); Pan et al. (2024b); Rawles et al. (2023).

2. **Turn Success Rate:** A *turn* indicates a single interaction between the user and the agent. A turn may consist of multiple steps, and completing a task may consist of multiple turns. This metric measures the ratio of turns that successfully address the request in that interaction over all turns. It focuses on the agent's ability to understand and fulfill user expectations during interactive or dialog-based tasks, ensuring the agent's responsiveness and reliability across iterative interactions, particularly in tasks requiring dynamic user-agent communication Lu et al. (2024c); Deng et al. (2024b).

3. **Task Success Rate:** Task success rate measures the successful task completion over all tasks set in the benchmark. It evaluates whether the final task completion state is achieved while ignoring the intermediate steps. This metric provides an overall measure of end-to-end task completion, reflecting the agent's ability to handle complex workflows holistically Yao et al. (2023); Zhang et al. (2024c); Xie et al. (2024b).

4. **Efficiency Score:** Efficiency score evaluates how effectively the agent completes tasks while considering resource consumption, execution time, or total steps the agent might take. This metric can be broken down into the following sub-metrics:

   - **Time Cost:** Measures the time taken to complete tasks.
   - **Resource Cost:** Measures the memory/CPU/GPU usage to complete tasks.
   - **LLM Cost:** Evaluates the computational or monetary cost of LLM calls used during task execution.
   - **Step Cost:** Measures the total steps required to complete tasks.

   Depending on the specific metrics used, the efficiency score can be interpreted differently in different papers Chen et al. (2024d); Deng et al. (2024a).

5. **Completion under Policy:** This metric measures the rate at which tasks are completed successfully while adhering to policy constraints. It ensures that the agent complies with user-defined or organizational rules, such as security, ethical, safety, privacy, or business guidelines, during task execution. This metric is particularly relevant for applications where compliance is as critical as task success Levy et al. (2024).

6. **Risk Ratio:** Similar to the previous metric, the risk ratio evaluates the potential risk associated with the agent's actions during task execution. It identifies vulnerabilities, errors, or security concerns that could arise during task handling. A lower risk ratio indicates higher trustworthiness and reliability, while a higher ratio may suggest areas needing improvement to minimize risks and enhance robustness Levy et al. (2024).

The implementation of metrics in each GUI agent benchmark might vary depending on the platform and the task formulation. In all tables in this section, we mapped the original metrics used in the benchmarks, which may possess different names, to the categories that we defined above.

## 8.2 Evaluation Measurements

To effectively evaluate GUI agents, various measurement techniques are employed to assess their accuracy and alignment with expected outputs. These measurements validate different aspects of agent performance, ranging from textual and visual correctness to interaction accuracy and system state awareness, using code, models, and even agents as evaluators Zhuge et al. (2024). Below, we summarize key measurement approaches used in benchmarking GUI agents. Based on these measurements, the evaluation metrics defined beforehand can be calculated accordingly.

1. **Text Match:** This measurement evaluates whether the text-based outputs of the agent match the expected results. For example, whether a target product name is reached when the agent is browsing an e-commerce website. It can involve different levels of strictness, including:

   - **Exact Match:** Ensures the output perfectly matches the expected result.
   - **Partial or Fuzzy Match:** Allows for approximate matches, which are useful for handling minor variations such as typos or synonyms.
   - **Semantic Similarity:** Measures deeper alignment in semantic meaning using techniques like cosine similarity of text embeddings or other semantic similarity measures.

   Text Match is widely applied in tasks involving textual selections, data entry, or natural language responses.

2. **Image Match:** Image Match focuses on validating whether the agent acts or stops on the expected page (*e.g.*, webpage, app UI), or selects the right image. It involves comparing screenshots, selected graphical elements, or visual outcomes against ground truth images using image similarity metrics or visual question answering (VQA) methods. This measurement is particularly crucial for tasks requiring precise visual identification.

3. **Element Match:** This measurement checks whether specific widget elements (*e.g.*, those in HTML, DOM, or application UI hierarchies) interacted with by the agent align with the expected elements. These may include:

   - **HTML Tags and Attributes:** Ensuring the agent identifies and interacts with the correct structural elements.
   - **URLs and Links:** Validating navigation-related elements.
   - **DOM Hierarchies:** Confirming alignment with expected DOM structures in dynamic or complex web interfaces.
   - **UI Controls and Widgets:** Verifying interactions with platform-specific controls such as buttons, sliders, checkboxes, dropdown menus, or other GUI components in desktop and mobile applications.
   - **Accessibility Identifiers:** Utilizing accessibility identifiers or resource IDs in mobile platforms like Android and iOS to ensure correct element selection.
   - **View Hierarchies:** Assessing alignment with expected view hierarchies in mobile applications, similar to DOM hierarchies in web applications.
   - **System Controls and APIs:** Ensuring correct interaction with operating system controls or APIs, such as file dialogs, system menus, or notifications in desktop environments.

   Element Match ensures robust interaction with user interface components across different platforms during task execution.

4. **Action Match:** This measurement assesses the accuracy of the agent's actions, such as clicks, scrolls, or keystrokes, by comparing them against an expected sequence. It involves:

   - **Action Accuracy:** Validates that each action (including action type and its arguments) is performed correctly (*e.g.*, clicking the correct button, typing the right input).
   - **Action Sequence Alignment:** Ensures actions occur in the correct order to meet task requirements.

- **Location Prediction:** Checks that spatial actions, such as mouse clicks or touch gestures, target the intended regions of the interface.

  Action Match is vital for evaluating step-wise correctness in task completion.

5. **State Information:** State Information captures runtime data related to the system's environment during task execution. It provides insights into contextual factors that may influence the agent's behavior, such as:

   - **Application State:** Information about the state of the application being interacted with (*e.g.*, open files, active windows, saved files in given locations).
   - **System Logs:** Detailed logs recording the agent's decisions and interactions.
   - **Environment Variables:** Contextual data about the operating system or runtime environment.

   This measurement is valuable for debugging, performance analysis, and ensuring reliability under diverse conditions.

Each of these measurement techniques contributes to a comprehensive evaluation framework, ensuring that the agent not only completes tasks but does so with precision, efficiency, and adaptability. Together, they help build trust in the agent's ability to perform reliably in real-world scenarios while maintaining compliance with policy constraints.

### 8.3 Evaluation Platforms

Evaluating GUI agents requires diverse platforms to capture the varying environments in which these agents operate. The platforms span web, mobile, and desktop environments, each with unique characteristics, challenges, and tools for evaluation. This section summarizes the key aspects of these platforms and their role in benchmarking GUI agents.

1. **Web:** Web platforms are among the most common environments for GUI agents, reflecting their prevalence in everyday tasks such as browsing, form filling, and data scraping. Key characteristics of web platforms for evaluation include:

   - **Dynamic Content:** Web applications often involve dynamic elements generated through JavaScript, AJAX, or similar technologies, requiring agents to handle asynchronous updates effectively.
   - **Diverse Frameworks:** The variety of web technologies (*e.g.*, HTML, CSS, JavaScript frameworks) demands robust agents capable of interacting with a range of interface designs and structures.
   - **Tools and Libraries:** Evaluation often uses tools such as Selenium, Puppeteer, or Playwright to emulate browser interactions, collect runtime information, and compare outcomes against expected results.
   - **Accessibility Compliance:** Metrics like WCAG (Web Content Accessibility Guidelines) adherence can also be evaluated to ensure inclusivity.

2. **Mobile:** Mobile platforms, particularly Android and iOS, pose unique challenges for GUI agents due to their constrained interfaces and touch-based interactions. Evaluating agents on mobile platforms involves:

   - **Screen Size Constraints:** Agents must adapt to limited screen real estate, ensuring interactions remain accurate and efficient.
   - **Touch Gestures:** Evaluating the agent's ability to simulate gestures such as taps, swipes, and pinches is essential.
   - **Platform Diversity:** Android devices vary significantly in terms of screen sizes, resolutions, and system versions, while iOS offers more standardized conditions.

Table 14: Taxonomy of LLM-powered GUI agent benchmarks by target platform.

| Platform | References |
|---|---|
| **Web** | Liu et al. (2024f); Shahbandeh et al. (2024); Lai et al. (2024); Xu et al. (2021); Shi et al. (2017); Liu et al. (2018); Deng et al. (2023); Chen et al. (2024e); Lu et al. (2024c); Zhou et al.; Koh et al. (2024a); Deng et al. (2024c); Zhang et al. (2024s); Pan et al. (2024b); Levy et al. (2024); Furuta et al. (2024); Xu et al. (2024e); Xie et al. (2024b); Drouin et al. (2024); Jang et al. (2024); Ma et al. (2024c); Wornow et al.; Zheng et al. (2024b); Yao et al. (2023); Chezelles et al. (2024); Wu et al. (2025a); Thomas et al. (2025); Tur et al. (2025); Kara et al.; Xue et al. (2025); Zharmagambetov et al. (2025); Lù et al. (2025); Ye et al. (2025); Garg et al. (2025); Song et al. (2025); Evtimov et al. (2025) |
| **Mobile** | Rawles et al. (2023); Xu et al. (2024i); Wen et al. (2024a); Li et al. (2020a); Toyama et al. (2021a); Wang et al. (2024e); Zhang et al. (2024c); Lee et al. (2024b); Rawles et al. (2024); Xing et al. (2024); Deng et al. (2024a); Lee et al. (2024a); Chen et al. (2024d); Zhang et al. (2024h); Wang et al. (2024i); Zhao et al. (2024b); Chai et al. (2025); Ran et al. (2025); Chen et al. (2025c); Sun et al. (2025a); Wang et al. (2025d) |
| **Computer** | Gao et al. (2024b); Bonatti et al. (2024); Xie et al. (2024b); Cao et al. (2024); Wang et al. (2024p); Zheng et al. (2024c); Zhao et al. (2025c); Nayak et al. (2025); Wang et al. (2025a) |
| **Cross-Platform** | Wu et al. (2024b); Liu et al. (2024i); Kapoor et al. (2024); Lin et al. (2024b); Xu et al. (2024g); Cheng et al. (2024a); Fan et al. (2024) |

- **Evaluation Tools:** Tools like Appium and Espresso (for Android) or XCTest (for iOS) and emulators are commonly used for testing and evaluation.

3. **Desktop:** Desktop platforms provide a richer and more complex environment for GUI agents, spanning multiple operating systems such as Windows, macOS, and Linux. Evaluations on desktop platforms often emphasize:

- **Application Diversity:** Agents must handle a wide range of desktop applications, including productivity tools, web browsers, and custom enterprise software.
- **Interaction Complexity:** Desktop interfaces often include advanced features such as keyboard shortcuts, drag-and-drop, and context menus, which agents must handle correctly.
- **Cross-Platform Compatibility:** Evaluations may involve ensuring agents can operate across multiple operating systems and versions.
- **Automation Frameworks:** Tools such as Windows UI Automation, macOS Accessibility APIs, and Linux's AT-SPI are used to automate and monitor agent interactions.
- **Resource Usage:** Memory and CPU usage are significant metrics, particularly for long-running tasks or resource-intensive applications.

Each platform presents distinct challenges and opportunities for evaluating GUI agents. Web platforms emphasize scalability and dynamic interactions, mobile platforms focus on touch interfaces and performance, and desktop platforms require handling complex workflows and cross-application tasks. Some benchmarks are cross-platform, requiring agents to be robust, adaptable, and capable of generalizing across different environments. All the metrics, measurements, and platforms discussed are essential for a comprehensive evaluation of GUI agents across multiple aspects. Most existing benchmarks rely on them for evaluation.

The evolution of GUI agent benchmarks reflects a broader shift towards more realistic, interactive, and comprehensive evaluation environments:

1. **Towards More Interactive and Realistic Environments:** Recent advancements in GUI agent benchmarking emphasize the transition from synthetic scenarios to more interactive and realistic environments. This shift is evident in the use of simulators, Docker containers, and real-world applications to create "live" environments that better mimic genuine user interactions. Such environments not only provide a more accurate assessment of agent capabilities but also pose new challenges in terms of performance and robustness.

2. **Cross-Platform Benchmarks:** The emergence of cross-platform benchmarks that encompass mobile, web, and desktop environments represents a significant step towards evaluating the generalizability of GUI agents. However, these benchmarks introduce fundamental challenges unique to each platform. A unified interface for accessing platform-specific information, such as HTML and DOM structures, could substantially streamline the benchmarking process and reduce implementation efforts. Future work should focus on standardizing these interfaces to facilitate seamless agent evaluation across diverse environments.

3. **Increased Human Interaction and Realism:** There is a growing trend towards incorporating more human-like interactions in benchmarks, as seen in multi-turn and conversational scenarios. These setups mirror real-world use cases more closely, thereby providing a rigorous test of an agent's ability to handle dynamic, iterative interactions. As GUI agents become more sophisticated, benchmarks must continue to evolve to include these nuanced interaction patterns, ensuring agents can operate effectively in complex, human-centric environments.

4. **Scalability and Automation Challenges:** Scalability remains a significant concern in benchmarking GUI agents. The creation of realistic tasks and the development of evaluation methods for individual cases often require substantial human effort. Automation of these processes could alleviate some of the scalability issues, enabling more extensive and efficient benchmarking. Future research should explore automated task generation and evaluation techniques to enhance scalability.

5. **Emphasis on Safety, Privacy, and Compliance:** There is a notable trend towards evaluating GUI agents on safety, privacy, and compliance metrics. These considerations are increasingly important as agents are integrated into sensitive and regulated domains. Encouraging this trend will help ensure that agents not only perform tasks effectively but also adhere to necessary legal and ethical standards. Future benchmarks should continue to expand on these dimensions, incorporating evaluations that reflect real-world compliance and data security requirements.

In what follows, we first present a taxonomy of LLM-powered GUI agent benchmarks, categorized by target platform, as shown in Table 14, and detail these benchmarks for GUI agents selectively.

## 8.4 Web Agent Benchmarks

Evaluating GUI agents in web environments necessitates benchmarks that capture the complexities and nuances of web-based tasks. Over the years, several benchmarks have been developed, each contributing unique perspectives and challenges to advance the field. We first provide an overview of these benchmarks in Tables 51, 52, 53, 54, 55 and 56.

One of the pioneering efforts in this domain is **MiniWoB++** Shi et al. (2017); Liu et al. (2018), focusing on assessing reinforcement learning agents on web-based GUI tasks. It introduces realistic interaction scenarios, including clicking, typing, and navigating web elements, and leverages workflow-guided exploration (WGE) to improve efficiency in environments with sparse rewards. Agents are evaluated based on success rates, determined by their ability to achieve final goal states, highlighting adaptability and robustness across various complexities.

Building upon the need for more realistic environments, **Mind2Web**Deng et al. (2023) represents a significant advancement by enabling agents to handle real-world HTML environments rather than simplified simulations. Established after the advent of LLMsYan et al. (2023b), it offers a large dataset of over 2,000 tasks spanning multiple domains, presenting challenges from basic actions to complex multi-page workflows. The benchmark

emphasizes end-to-end task performance through metrics like Element Accuracy and Task Success Rate, encouraging rigorous evaluation of agents.

Extending Mind2Web's capabilities, **MT-Mind2Web** Deng et al. (2024c) introduces conversational web navigation, requiring sophisticated interactions that span multiple turns with both users and the environment. This advanced benchmark includes 720 web navigation conversation sessions with 3,525 instruction and action sequence pairs, averaging five user-agent interactions per session, thereby testing agents' conversational abilities and adaptability.

To further enhance realism, **WebArena** Zhou et al. sets a new standard with its realistic web environment that mimics genuine human interactions. Featuring 812 tasks across multiple domains, it requires agents to perform complex, long-horizon interactions over multi-tab web interfaces. By focusing on functional correctness rather than surface-level matches, WebArena promotes thorough assessment of agents' practical abilities.

Recognizing the importance of multimodal capabilities, **VisualWebArena**, an extension of WebArena Zhou et al., was designed to assess agents on realistic visually grounded web tasks. Comprising 910 diverse tasks in domains like Classifieds, Shopping, and Reddit, it adds new visual functions for measuring open-ended tasks such as visual question answering and fuzzy image matching, thereby challenging agents in multimodal understanding.

Similarly, **VideoWebArena** Jang et al. (2024) focuses on evaluating agents' abilities to comprehend and interact with video content on the web. It presents 74 videos across 2,021 tasks, challenging agents in video-based information retrieval, contextual reasoning, and skill application. This benchmark highlights critical deficiencies in current models, emphasizing the need for advancements in agentic reasoning and video comprehension.

Complementing this, **VisualWebBench** Liu et al. (2024f) offers a multimodal benchmark that assesses understanding, OCR, grounding, and reasoning across website, element, and action levels. Spanning 1.5K samples from real-world websites, it identifies challenges such as poor grounding and subpar OCR with low-resolution inputs, providing a crucial evaluation perspective distinct from general multimodal benchmarks.

Beyond the challenges of multimodality, understanding agents' resilience to environmental distractions is crucial. **EnvDistraction** Ma et al. (2024c) introduces a benchmark that evaluates the faithfulness of multimodal GUI agents under non-malicious distractions, such as pop-ups and recommendations. The study demonstrates that even advanced agents are prone to such distractions, revealing vulnerabilities that necessitate robust multimodal perception for reliable automation.

Focusing on safety and trustworthiness, **ST-WebAgentBench** Levy et al. (2024) takes a unique approach by emphasizing the management of unsafe behaviors in enterprise settings. It features a human-in-the-loop system and a policy-driven hierarchy, introducing the Completion under Policy (CuP) metric to evaluate agents' compliance with organizational, user, and task-specific policies. This benchmark operates in web environments using BrowserGym Chezelles et al. (2024) and includes 235 tasks with policies addressing various safety dimensions, providing a comprehensive framework for evaluating agents in enterprise scenarios.

Addressing the automation of enterprise software tasks, **WorkArena** Drouin et al. (2024) offers a benchmark emphasizing tasks commonly performed within the ServiceNow platform. With 19,912 unique instances across 33 tasks, it highlights the significant performance gap between current state-of-the-art agents and human capabilities in enterprise UI automation, setting a trajectory for future innovation.

BrowserGym Chezelles et al. (2024) builds ecosystem designed for web agent research. It unifies various benchmarks like MiniWoB(++) Liu et al. (2018), WebArena Zhou et al., and WorkArena Drouin et al. (2024) under a single framework, addressing the issue of fragmentation in web agent evaluation. By leveraging standardized observation and action spaces, it enables consistent and reproducible experiments. BrowserGym's extensible architecture make it a vital tool for developing and testing GUI-driven agents powered by LLMs, significantly accelerating innovation in web automation research.

In the realm of interacting with live websites, **WebOlympus** Zheng et al. (2024b) introduces an open platform that enables web agents to interact with live websites through a Chrome extension-based interface.

Supporting diverse tasks and integrating a safety monitor to prevent harmful actions, it promotes safer automation of web-based tasks and provides a critical tool for evaluating agent performance in realistic scenarios.

Collectively, these benchmarks have significantly contributed to advancing the evaluation of web-based GUI agents, each addressing different aspects such as realism, multimodality, safety, and enterprise applicability. Their developments reflect the evolving challenges and requirements in creating sophisticated agents capable of complex web interactions.

### 8.5  Mobile Agent Benchmarks

Evaluating GUI agents on mobile platforms presents unique challenges due to the diversity of interactions and the complexity of mobile applications. Several benchmarks have been developed to address these challenges, each contributing to the advancement of mobile agent evaluation. We first provide an analysis for these mobile benchmarks in Tables 57, 58, 59 and 60.

An early effort in this domain is **PIXELHELP** Li et al. (2020a), which focuses on grounding natural language instructions to actions on mobile user interfaces. Addressing the significant challenge of interpreting and executing complex, multi-step tasks, PIXELHELP provides a comprehensive dataset pairing English instructions with human-performed actions on a mobile UI emulator. It comprises 187 multi-step instructions across four task categories, offering a robust resource for evaluating models on task accuracy through metrics like Complete Match and Partial Match.

Building upon the need for systematic evaluation, **ANDROIDLAB** Xu et al. (2024i) establishes a comprehensive framework for Android-based autonomous agents. It introduces both an action space and operational modes that support consistent evaluations for text-only and multimodal models. By providing XML and SoM operation modes, ANDROIDLAB allows LLMs and LMMs to simulate real-world interactions in equivalent environments. The benchmark includes 138 tasks across nine apps, encompassing typical Android functionalities, and evaluates agents using metrics such as Success Rate and Reversed Redundancy.

To further challenge agents in handling both API and UI operations, **Mobile-Bench** Deng et al. (2024a) offers an innovative approach by combining these elements within a realistic Android environment. Its multi-app setup and three distinct task categories test agents' capabilities in handling simple and complex mobile interactions, pushing beyond traditional single-app scenarios. The evaluation leverages CheckPoint metrics, assessing agents at each key action step, providing insights into planning and decision-making skills.

Emphasizing safety in mobile device control, **MobileSafetyBench** Lee et al. (2024a) provides a structured evaluation framework that prioritizes both helpfulness and safety. It rigorously tests agents across common mobile tasks within an Android emulator, focusing on layered risk assessment, including legal compliance and privacy. A distinctive feature is its indirect prompt injection test to probe agent robustness. The evaluation ensures agents are scored on practical success while managing risks, advancing research in LLM reliability and secure autonomous device control.

Expanding the scope to multiple languages and application scenarios, **SPA-BENCH** Chen et al. (2024d) introduces an extensive benchmark for smartphone agents. It assesses both single-app and cross-app tasks in a plug-and-play framework that supports seamless agent integration. With a diverse task collection across Android apps, including system and third-party apps, SPA-BENCH offers a realistic testing environment measuring agent capabilities in understanding UIs and handling app navigation through metrics like success rate, efficiency, and resource usage.

Focusing on efficient and user-friendly evaluation, **MobileAgentBench** Wang et al. (2024i) presents a benchmark tailored for agents on Android devices. It offers a fully autonomous testing process, leveraging final UI state matching and real-time app event tracking. With 100 tasks across 10 open-source Android applications categorized by difficulty, it accommodates multiple paths to success, enhancing reliability and applicability. Comprehensive metrics, including task success rate, efficiency, latency, and token cost, provide insights into agent performance.

Complementing these efforts, **LlamaTouch** Zhang et al. (2024h) introduces a benchmark and testbed for mobile UI task automation in real-world Android environments. Emphasizing essential state annotation, it enables precise evaluation of tasks regardless of execution path variability or dynamic UI elements. With 496 tasks spanning 57 unique applications, LlamaTouch demonstrates scalability and fidelity through advanced matching techniques, integrating pixel-level screenshots and textual screen hierarchies, reducing false negatives and supporting diverse task complexities.

Zhao *et al.*, introduce GTArena Zhao et al. (2024b), a formalized framework and benchmark designed to advance autonomous GUI testing agents. GTArena provides a standardized evaluation environment tailored for multimodal large language models. Central to its design is the novel Transition Tuple data structure, which systematically captures and analyzes GUI defects. The benchmark assesses three core tasks—test intention generation, task execution, and defect detection—using a diverse dataset comprising real-world, artificially injected, and synthetic defects, establishing GTArena as a pioneering benchmark for GUI testing agents.

Collectively, these benchmarks have significantly advanced the evaluation of mobile-based GUI agents, addressing challenges in task complexity, safety, efficiency, and scalability. Their contributions are instrumental in developing more capable and reliable agents for mobile platforms.

### 8.6 Computer Agent Benchmarks

Evaluating GUI agents on desktop computers involves diverse applications and complex workflows. Several benchmarks have been developed to assess agents' capabilities in these environments, each addressing specific challenges and advancing the field. We overview benchmarks for computer GUI agents in Tables 61 and 62.

An early benchmark in this domain is **Act2Cap** Wu et al. (2024b), which focuses on capturing and narrating GUI actions in video formats using a cursor as a pivotal visual guide. Act2Cap emphasizes the detailed nuances of GUI interactions, particularly cursor-based actions like clicks and drags, essential for advancing automation capabilities in GUI-intensive tasks. It includes a substantial dataset of 4,189 samples across various Windows GUI environments, employing metrics based on element-wise Intersection over Union to evaluate semantic accuracy and temporal and spatial precision.

To provide a scalable and genuine computer environment for multimodal agents, **OSWorld** Xie et al. (2024b) introduces a pioneering framework that supports task setup, execution-based evaluation, and interactive learning across multiple operating systems, including Ubuntu, Windows, and macOS. OSWorld serves as a unified environment that mirrors the complexity and diversity of real-world computer use, accommodating arbitrary applications and open-ended computer tasks. It includes a comprehensive suite of 369 tasks on Ubuntu and 43 tasks on Windows, utilizing execution-based evaluation metrics like success rate for rigorous assessment.

Building on OSWorld, **WindowsArena** Bonatti et al. (2024) adapts the framework to create over 150 diverse tasks specifically for the Windows operating system. Focusing on multi-modal, multi-step tasks, it requires agents to demonstrate abilities in planning, screen understanding, and tool usage within a real Windows environment. Addressing the challenge of slow evaluation times, WindowsArena enables parallelized deployment in the Azure cloud, drastically reducing evaluation time and allowing for comprehensive testing across various applications and web domains.

Focusing on office automation tasks, **OFFICEBENCH** Wang et al. (2024p) introduces a groundbreaking framework for benchmarking LLM agents in realistic office workflows. Simulating intricate workflows across multiple office applications like Word, Excel, and Email within a Linux Docker environment, it evaluates agents' proficiency in cross-application automation. The benchmark challenges agents with complex tasks at varying difficulty levels, demanding adaptability to different complexities and use cases. Customized metrics assess operation accuracy and decision-making, providing critical insights into agents' capabilities in managing multi-application office scenarios.

Addressing the automation of data science and engineering workflows, **Spider2-V** Cao et al. (2024) offers a distinctive benchmark. It features 494 real-world tasks across 20 enterprise-level applications, spanning the entire data science workflow from data warehousing to visualization. Assessing agents' abilities to handle

both code generation and complex GUI interactions within authentic enterprise software environments on Ubuntu, it employs a multifaceted evaluation method that includes information-based validation, file-based comparison, and execution-based verification.

In the realm of productivity software, **AssistGUI** Gao et al. (2024b) provides a pioneering framework for evaluating agents' capabilities. It introduces an Actor-Critic Embodied Agent framework capable of complex hierarchical task planning, GUI parsing, and action generation. The dataset includes diverse tasks across design, office work, and system settings, supported by project files for reproducibility. By emphasizing outcome-driven evaluation with pixel-level precision and procedural adherence, AssistGUI highlights the potential and limitations of current LLM-based agents in managing intricate desktop software workflows.

**WorldGUI Zhao et al. (2025c)** is a benchmark designed to evaluate GUI agents under dynamic conditions on the Windows platform. Unlike previous static benchmarks, it introduces varied initial states to simulate real-world interactions across both desktop and web applications. Rather than always starting from a fixed default state, agents must adapt to changing UI layouts, user interactions, system settings, and pre-existing conditions, requiring robust adaptability to perform effectively. The benchmark comprises 315 tasks spanning 10 popular software applications and incorporates instructional videos, project files, and multiple pre-action scenarios, providing a comprehensive and realistic evaluation framework for assessing an agent's ability to handle complex task execution.

**Computer Agent Arena Wang et al. (2025a)** presents a new paradigm for benchmarking LLM-based GUI agents through live, user-configured desktop environments. Unlike traditional static datasets, it provides an interactive cloud-based infrastructure where agents are evaluated on tasks spanning web browsing, programming, and productivity using real applications like Google Docs, VSCode, and Slack. Its innovation lies in using head-to-head agent comparisons, human judgment, and Elo-based ranking to evaluate general-purpose digital agents in realistic settings. The benchmark supports Windows and Ubuntu, with MacOS support planned, and allows customized task scenarios with diverse software and website setups. By enabling crowdsourced evaluations and planning open-source releases, it fosters community-driven improvements and robust comparisons.

Collectively, these benchmarks provide comprehensive evaluation frameworks for GUI agents on desktop platforms, addressing challenges in task complexity, cross-application automation, scalability, and fidelity. Their contributions are instrumental in advancing the development of sophisticated agents capable of complex interactions in desktop environments.

## 8.7 Cross-Platform Agent Benchmarks

To develop GUI agents capable of operating across multiple platforms, cross-platform benchmarks are essential. These benchmarks challenge agents to adapt to different environments and interfaces, evaluating their versatility and robustness. We provide an overview of benchmarks for cross-platform GUI agents in Tables 63.

Addressing this need, **VisualAgentBench** (VAB) Liu et al. (2024i) represents a pioneering benchmark for evaluating GUI and multimodal agents across a broad spectrum of realistic, interactive tasks. Encompassing platforms such as Web (WebArena-Lite Zhou et al.), Android (VAB-Mobile Xu et al. (2024i)), and game environments, VAB focuses on vision-based interaction and high-level decision-making tasks. The benchmark employs a multi-level data collection strategy involving human demonstrations, program-based solvers, and model bootstrapping. Evaluation metrics concentrate on success rates, ensuring comprehensive performance assessments in tasks like navigation and content modification, thereby filling a significant gap in benchmarking standards for GUI-based LLM agents.

Complementing this, **CRAB** Xu et al. (2024g) introduces an innovative benchmark by evaluating multimodal language model agents in cross-environment interactions. It uniquely supports seamless multi-device task execution, evaluating agents in scenarios where tasks span both Ubuntu Linux and Android environments. By introducing a graph-based evaluation method that breaks down tasks into sub-goals and accommodates multiple correct paths to completion, CRAB provides a nuanced assessment of planning, decision-making,

and adaptability. Metrics such as Completion Ratio, Execution Efficiency, Cost Efficiency, and Success Rate offer comprehensive insights into agent performance.

Focusing on GUI grounding for cross-platform visual agents, **ScreenSpot** Cheng et al. (2024a) offers a comprehensive benchmark emphasizing tasks that rely on interpreting screenshots rather than structured data. ScreenSpot includes over 600 screenshots and 1,200 diverse instructions spanning mobile (iOS, Android), desktop (macOS, Windows), and web platforms. It evaluates click accuracy and localization precision by measuring how effectively agents can identify and interact with GUI elements through visual cues alone. By challenging models with a wide variety of UI elements, ScreenSpot addresses real-world complexities, making it an essential resource for evaluating visual GUI agents across varied environments.

These cross-platform benchmarks collectively advance the development of GUI agents capable of operating seamlessly across multiple platforms. By providing comprehensive evaluation frameworks, they are instrumental in assessing and enhancing the versatility and adaptability of agents in diverse environments.

## 9 Applications of LLM-Powered GUI Agents

As LLM-powered GUI agents continue to mature, a growing number of applications leverage this concept to create more intelligent, user-friendly, and natural language-driven interfaces. These advancements are reflected in research papers, open-source projects, and industry solutions. Typical applications encompass *(i)* **GUI testing**, which has transitioned from traditional script-based approaches to more intuitive, natural language-based interactions, and *(ii)* **virtual assistants**, which automate users' daily tasks in a more adaptive and responsive manner through natural language interfaces.

The application of LLM-powered GUI agents has ushered in new capabilities and interfaces for tasks such as GUI testing and virtual assistance, introducing natural language interactions, enhanced automation, and improved accessibility across platforms. These agents are transforming the way users interact with software applications by simplifying complex tasks and making technology more accessible. However, despite these advancements, LLM-powered GUI agents are still in their infancy, and several challenges need to be addressed for them to reach maturity. Key insights from recent developments include:

1. **Natural Language-Driven Interactions:** LLM-powered GUI agents have enabled users to interact with applications using natural language, significantly lowering the barrier to entry for non-expert users. In GUI testing, tools like GPTDroid Liu et al. (2024k) and AUITestAgent Hu et al. (2024c) allow testers to specify test cases and requirements in plain language, automating the execution and verification processes. Similarly, virtual assistants like LLMPA Guan et al. (2024a) and ProAgent Ye et al. (2023) interpret user commands to perform complex tasks, showcasing the potential of natural language interfaces in simplifying user interactions across platforms.

2. **Enhanced Automation of Complex Tasks:** These agents have demonstrated the ability to automate multi-step and intricate workflows without the need for manual scripting. Projects like AutoTask Pan et al. (2023b) and GPTVoiceTasker Vu et al. (2024) autonomously explore and interact with GUI environments, executing tasks based on high-level goals or voice commands. In GUI testing, agents have improved coverage and efficiency by automating the generation of test inputs and reproducing bugs from textual descriptions, as seen in CrashTranslator Huang et al. (2024c) and AdbGPT Feng & Chen (2024).

3. **Multimodal Perception and Interaction:** Integrating visual and textual inputs has enhanced the agents' understanding of GUI contexts, leading to better decision-making and interaction accuracy. Agents like VizAbility Gorniak et al. (2024) and OpenAdapt OpenAdapt AI (2024) utilize screenshots, UI trees, and OCR to perceive the environment more comprehensively. This multimodal approach is crucial for applications that require precise identification and manipulation of GUI elements, especially in dynamic or visually complex interfaces.

4. **Improved Accessibility and User Experience:** LLM-powered GUI agents have contributed to making technology more accessible to users with disabilities or limited technical proficiency. Tools like

Table 15: Taxonomy of application scenarios for GUI agents.

| Application Area | References |
|---|---|
| **GUI Testing** | Zimmermann & Koziolek (2023b); Liu et al. (2024k); Yoon et al. (2024); Hu et al. (2024c); Liu et al. (2024l); Taeb et al. (2024); Cui et al. (2024); Liu et al. (2023c); Huang et al. (2024c); Feng & Chen (2024); Ding et al. (2024b); Beyzaei et al. (2024); Lu et al. (2025a); Ran et al. (2024); Li et al. (2024h); Demissie et al. (2025); Yapağcı et al. (2025); Li et al. (2025c); Chevrot et al. (2025); Rosenbach et al. (2025); Kong et al.; Feng et al. (2025) |
| **Virtual Assistants** | Microsoft (2024); Gorniak et al. (2024); Ye et al. (2023); Guan et al. (2024a); Vu et al. (2024); Pan et al. (2023b); Gao et al. (2024a); Huang et al. (2024a); Gao et al. (2024g); OpenAdapt AI (2024); AgentSeaf AI (2024); Interpreter (2024); MultiOn AI (2024); HONOR (2024); Srinivasan & Patapati (2025) |

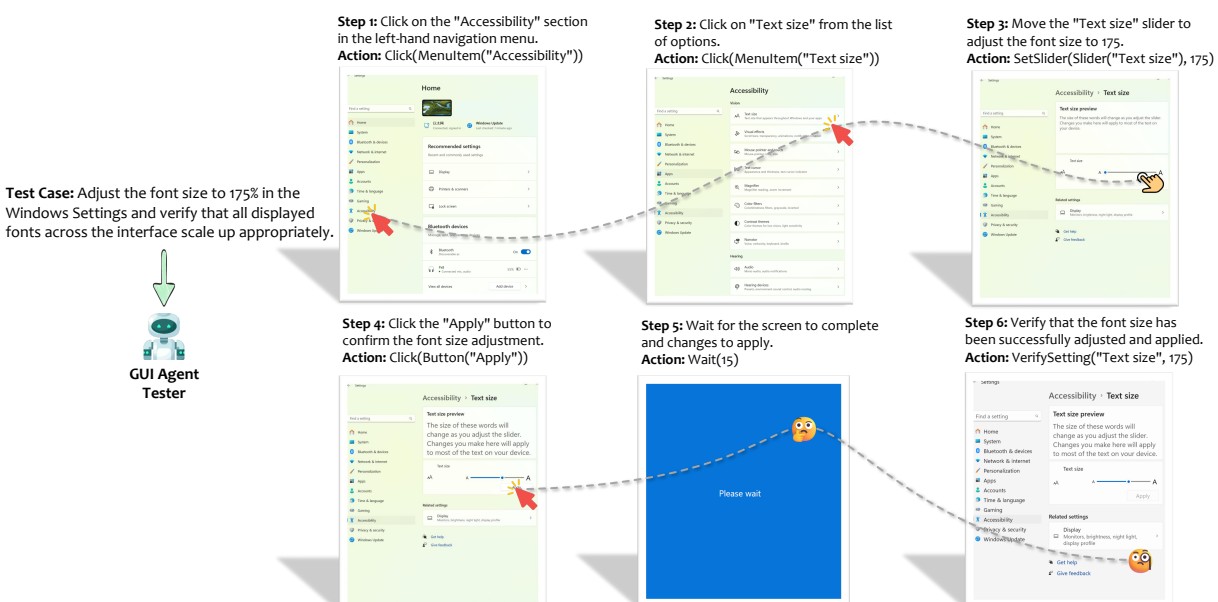

Figure 28: An example of testing font size adjustment using an LLM-powered GUI agent.

VizAbility Gorniak et al. (2024) aid blind and low-vision users in understanding data visualizations, while EasyAsk Gao et al. (2024g) assists older adults in navigating smartphone functions. By tailoring interactions to the needs of diverse user groups, these agents enhance inclusivity and user experience.

Next, we present a taxonomy of application scenarios for GUI agents, categorized by application area, as shown in Table 15, followed by a detailed discussion of representative works.

## 9.1 GUI Testing

GUI testing evaluates a software application's graphical user interface to ensure compliance with specified requirements, functionality, and user experience standards. It verifies interface elements like buttons, menus, and windows, as well as their responses to user interactions. Initially conducted manually, GUI testing evolved with the advent of automation tools such as Selenium and Appium, enabling testers to automate repetitive tasks, increase coverage, and reduce testing time Wang et al. (2024c); Yu et al. (2023). However, LLM-powered GUI agents have introduced a paradigm shift, allowing non-experts to test GUIs intuitively

through natural language interfaces. These agents cover diverse scenarios, including general testing, input generation, and bug reproduction, without the need for traditional scripting Wang et al. (2024c).

Figure 28 and illustrates the use of an LLM-powered GUI agent to test font size adjustment on Windows OS. With only a natural language test case description, the agent autonomously performs the testing by executing UI operations, navigating through the settings menu, and leveraging its screen understanding capabilities to verify the final outcome of font size adjustment. This approach dramatically reduces the effort required for human or script-based testing. Next, we detail the GUI testing works powered by GUI agents, and first provide an overview Tables 64, 65 , 66 and 67.

### 9.1.1 General Testing

Early explorations demonstrated how LLMs like GPT-3 could automate GUI testing by interpreting natural language test cases and programmatically executing them. For example, one approach integrates GUI states with GPT-3 prompts, leveraging tools like Selenium and OpenCV to reduce manual scripting and enable black-box testing Zimmermann & Koziolek (2023a). Building on this, a subsequent study employed GPT-4 and Selenium WebDriver for web application testing, achieving superior branch coverage compared to traditional methods like monkey testing Zimmermann & Koziolek (2023b). These advances highlight how LLMs simplify GUI testing workflows while significantly enhancing coverage and efficiency.

Further pushing boundaries, **GPTDroid** reframed GUI testing as an interactive Q&A task. By extracting structured semantic information from GUI pages and leveraging memory mechanisms for long-term exploration, it increased activity coverage by 32%, uncovering critical bugs with remarkable precision Liu et al. (2024k). This approach underscores the potential of integrating conversational interfaces with memory for comprehensive app testing. For Android environments, **DROIDAGENT** introduced an intent-driven testing framework. It automates task generation and execution by perceiving GUI states in JSON format and using LLMs for realistic task planning. Its ability to set high-level goals and achieve superior feature coverage demonstrates how intent-based testing can transform functional verification in GUI applications Yoon et al. (2024).

ProphetAgent Kong et al. introduces a novel approach to LLM-powered GUI testing by automatically synthesizing Android application test scripts from natural language descriptions. Departing from previous methods that directly apply LLMs to GUI screenshots or app behaviors, ProphetAgent builds a Clustered UI Transition Graph (CUTG) enriched with semantic annotations. This structured representation enables more accurate mapping between natural language test steps and GUI operations, leading to significant improvements in completion rate (78.1%) and action accuracy (83.3%). The system employs a dual-agent architecture: SemanticAgent handles semantic annotation, while GenerationAgent generates executable scripts. ProphetAgent demonstrates strong scalability and real-world applicability—reducing tester workload by over 70% at ByteDance. Its performance underscores the effectiveness of combining LLMs with explicit semantic knowledge graphs in GUI-based environments.

**AUITestAgent** extended the capabilities of LLM-powered GUI testing by bridging natural language-driven requirements and GUI functionality Hu et al. (2024c). Employing multi-modal analysis and dynamic agent organization, it efficiently executes both simple and complex testing instructions. This framework highlights the value of combining multi-source data extraction with robust language models to automate functional testing in commercial apps. Incorporating vision-based methods, **VisionDroid** redefined GUI testing by aligning screenshots with textual contexts to detect non-crash bugs Liu et al. (2024l). This innovation ensures application reliability by identifying logical inconsistencies and exploring app functionalities that conventional methods often overlook.

Accessibility testing has also benefited from LLM-powered agents. **AXNav** addresses challenges in iOS accessibility workflows, automating tests for features like VoiceOver and Dynamic Type using natural language instructions and pixel-based models. Its ability to generate annotated videos for interactive review positions AXNav as a scalable and user-friendly solution for accessibility testing Taeb et al. (2024).

### 9.1.2 Text Input generation

In the realm of text input generation, Cui *et al.*, demonstrated how GPT-3.5 and GPT-4 could enhance Android app testing by generating context-aware text inputs for UI fields Cui et al. (2024). By systematically evaluating these inputs across multiple apps, they revealed the potential of LLMs in improving test coverage and detecting unique bugs with minimal manual intervention. Similarly, **QTypist** formulated text input generation as a fill-in-the-blank task, leveraging LLMs to improve activity and page coverage by up to 52% Liu et al. (2023c).

### 9.1.3 Bug Replay

For bug reproduction, **CrashTranslator** automated the reproduction of crashes from stack traces by integrating reinforcement learning with LLMs. Its iterative navigation and crash prediction steps significantly reduced debugging time and outperformed state-of-the-art methods Huang et al. (2024c). Meanwhile, **AdbGPT** demonstrated how few-shot learning and chain-of-thought reasoning could transform textual bug reports into actionable GUI operations. By dynamically inferring GUI actions, AdbGPT provided an efficient and lightweight solution for bug replay Feng & Chen (2024).

**BugCraft** Yapağcı et al. (2025) leverages LLM-powered GUI agents to automate bug reproduction in games, specifically targeting the open-ended and complex environment of Minecraft. It employs GPT-4o as the inference engine, integrating textual bug reports, visual GUI understanding through OmniParser Lu et al. (2024d), and external knowledge from the Minecraft Wiki to generate and execute structured reproduction steps. Actions are carried out via a custom Macro API, enabling robust interaction with both the game's GUI and environment. BugCraft's ability to translate unstructured bug descriptions into executable in-game behaviors highlights the strong potential of vision-enhanced LLM agents for advancing software testing and debugging.

### 9.1.4 Verification

Finally, as a novel application in testing, **MagicWand** showcased the potential of LLMs in automating "How-to" verifications. By extracting, executing, and refining instructions from search engines, it addressed critical challenges in user-centric task automation, improving the reliability of GUI-driven workflows Ding et al. (2024b).

In summary, LLM-powered GUI agents have revolutionized GUI testing by introducing natural language-driven methods, vision-based alignment, and automated crash reproduction. These innovations have enhanced test coverage, efficiency, and accessibility, setting new benchmarks for intelligent GUI testing frameworks.

## 9.2 Virtual Assistants

Virtual assistants, such as Siri[33], are AI-driven applications that help users by performing tasks, answering questions, and executing commands across various platforms, including web browsers, mobile phones, and computers. Initially, these assistants were limited to handling simple commands via voice or text input, delivering rule-based responses or running fixed workflows similar to RPA. They focused on basic tasks, such as setting alarms or checking the weather.

With advancements in LLMs and agents, virtual assistants have evolved significantly. They now support more complex, context-aware interactions on device GUIs through textual or voice commands and provide personalized responses, catering to diverse applications and user needs on various platforms. This progression has transformed virtual assistants from basic utilities into intelligent, adaptive tools capable of managing intricate workflows and enhancing user productivity across platforms. Figure 29 presents a conceptual example of a GUI agent-powered virtual assistant on a smartphone[34]. In this scenario, the agent enables users to interact through chat, handling tasks such as setting up a screenshot shortcut on their behalf. This

---

[33]https://www.apple.com/siri/

[34]The application and scenario depicted in the figure are conceptual and fabricated. They do not reflect the actual functionality of any specific smartphone. Readers should consult the phone manual or official guidance for accurate information on AI assistant capabilities.

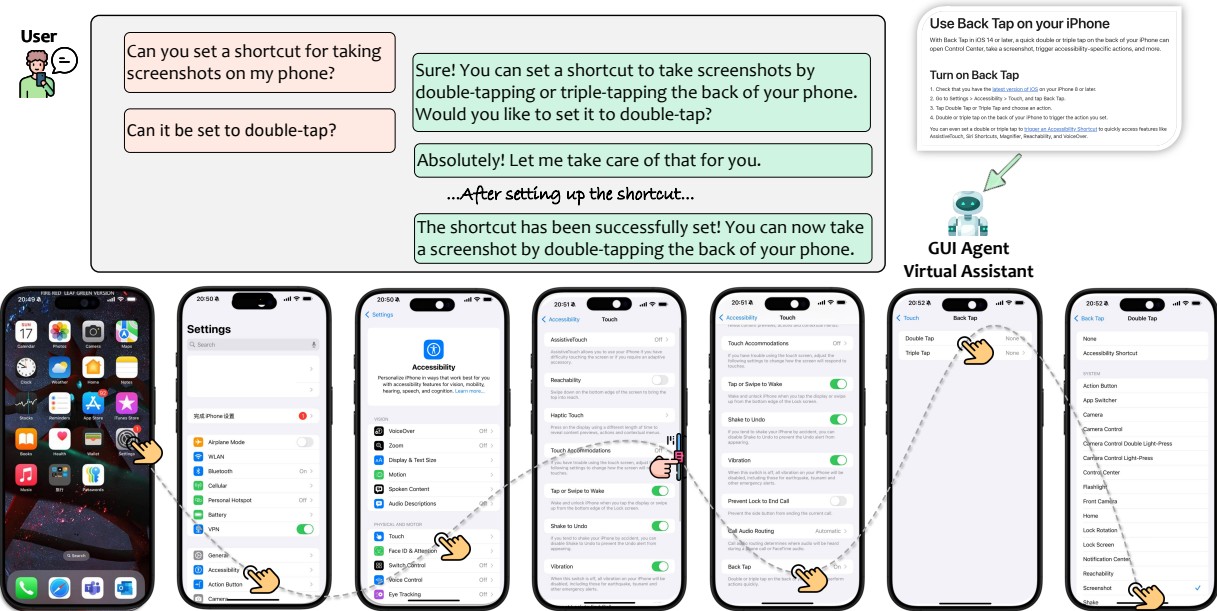

Figure 29: A conceptual example of a GUI agent-powered virtual assistant on a smartphone.

feature is particularly beneficial for users unfamiliar with the phone's functionalities, simplifying complex tasks into conversational commands.

To explore more real-world applications of virtual assistants powered by GUI agents, we provide an overview of advancements across research, open-source initiatives, and production-level applications, as summarized in Table 68 69 and 70.

### 9.2.1 Research

Recent research efforts have significantly advanced the capabilities of virtual assistants by integrating LLM-powered GUI agents, enabling more intelligent and adaptable interactions within various applications.

Firstly, the integration of LLMs into GUI-based automation has been explored to enhance business process automation. For instance, Ye et al. (2023) introduces Agentic Process Automation through the development of **ProAgent**, which automates both the creation and execution of workflows in GUI environments. By utilizing agents like ControlAgent and DataAgent, it supports complex actions such as dynamic branching and report generation in applications like Slack and Google Sheets. This approach transcends traditional RPA by enabling flexible, intelligent workflows, significantly reducing the need for manual intervention and highlighting the transformative potential of LLM-powered agents in virtual assistants.

Building upon the idea of integrating LLMs with GUI environments, researchers have focused on mobile platforms to automate complex tasks. **LLMPA** Guan et al. (2024a) is a pioneering framework that leverages LLMs to automate multi-step tasks within mobile applications like Alipay. It interacts directly with app GUIs, mimicking human actions such as clicks and typing, and employs UI tree parsing and object detection for precise environment understanding. A unique controllable calibration module ensures logical action execution, demonstrating the potential of LLM-powered virtual assistants to handle intricate workflows and real-world impact in assisting users with diverse tasks.

Similarly, the automation of smartphone tasks through natural language prompts has been addressed by **PromptRPA** Huang et al. (2024a). Utilizing a multi-agent framework, it automates tasks within smartphone GUI environments, tackling challenges like interface updates and user input variability. Advanced perception methods, including OCR and hierarchical GUI analysis, are employed to understand and interact with

mobile interfaces. By supporting real-time feedback and iterative improvements, PromptRPA underscores the importance of user-centered design in LLM-driven virtual assistants.

In the realm of accessibility, LLM-powered GUI agents have been instrumental in enhancing user experience for individuals with disabilities. For example, **VizAbility** Gorniak et al. (2024) enhances the accessibility of data visualizations for blind and low-vision users. By combining structured chart navigation with LLM-based conversational interactions, users can ask natural language queries and receive insights on chart content and trends. Leveraging frameworks like Olli[35] and chart specifications such as Vega-Lite[36], VizAbility allows exploration of visual data without direct visual perception, addressing real-world accessibility challenges in GUIs.

Furthermore, addressing the needs of older adults, **EasyAsk** Gao et al. (2024g) serves as a context-aware in-app assistant that enhances usability for non-technical users. By integrating multi-modal inputs, combining natural voice queries and touch interactions with GUI elements, it generates accurate and contextual tutorial searches. EasyAsk demonstrates how GUI agents can enhance accessibility by integrating contextual information and interactive tutorials, empowering users to navigate smartphone functions effectively.

Voice interaction has also been a focus area, with tools like **GPTVoiceTasker** Vu et al. (2024) facilitating hands-free interaction with Android GUIs through natural language commands. It bridges the gap between voice commands and GUI-based actions using real-time semantic extraction and a hierarchical representation of UI elements. By automating multi-step tasks and learning from user behavior, it enhances task efficiency and reduces cognitive load, highlighting the transformative potential of LLMs in improving accessibility and user experience in mobile environments.

Expanding on voice-powered interactions, **AutoTask** Pan et al. (2023b) enables virtual assistants to execute multi-step tasks in GUI environments without predefined scripts. It autonomously explores and learns from mobile GUIs, effectively combining voice command interfaces with dynamic action engines to interact with GUI elements. Utilizing trial-and-error and experience-driven learning, AutoTask adapts to unknown tasks and environments, showcasing its potential in enhancing voice-driven virtual assistants for hands-free interactions.

Finally, in the domain of creative workflows, **AssistEditor** Gao et al. (2024a) exemplifies a multi-agent framework for automating video editing tasks. By interacting with GUI environments, it autonomously performs complex workflows using dialogue systems and video understanding models to bridge user intent with professional editing tasks. The innovative use of specialized agents ensures efficient task distribution and execution, demonstrating the practical application of LLM-powered GUI agents in real-world scenarios and expanding automation into creative domains.

These research endeavors collectively showcase significant advancements in LLM-powered GUI agents, highlighting their potential to transform virtual assistants into intelligent, adaptable tools capable of handling complex tasks across various platforms and user needs.

### 9.2.2 Open-Source Projects

In addition to research prototypes, open-source projects have contributed substantially to the development and accessibility of LLM-brained GUI agents, enabling wider adoption and customization.

One such project is **OpenAdapt** OpenAdapt AI (2024), an open-source framework that utilizes large multimodal models to automate tasks by observing and replicating user interactions within GUI environments. It captures screenshots and records user inputs, employing computer vision techniques to understand and execute standard UI operations. Designed to streamline workflows across various industries, OpenAdapt learns from user demonstrations, thereby reducing the need for manual scripting and showcasing adaptability in GUI-based task automation.

Similarly, **AgentSea** AgentSeaf AI (2024) offers a comprehensive and modular toolkit for creating intelligent agents that can navigate and interact with various GUI environments across multiple platforms. Its flexibility

---

[35]https://mitvis.github.io/olli/
[36]https://vega.github.io/

is particularly beneficial for developing virtual assistants capable of automating complex tasks within applications, enhancing user productivity. By adhering to the UNIX philosophy, AgentSea ensures that each tool is specialized, promoting ease of use and extensibility. Its open-source nature fosters community collaboration and innovation in AI-driven GUI automation.

**Open Interpreter** Interpreter (2024) further exemplifies the potential of open-source contributions by leveraging large language models to execute code locally. Users can interact with their computer's GUI through natural language commands, supporting multiple programming languages and operating across various platforms. By facilitating tasks such as data analysis, web automation, and system management, Open Interpreter provides unrestricted access to system resources and libraries, enhancing flexibility and control. Its customization capabilities make it a valuable asset for users aiming to streamline operations through AI-powered virtual assistance.

These open-source projects not only advance the state of LLM-powered GUI agents but also democratize access to intelligent virtual assistants, enabling developers and users to tailor solutions to specific needs and applications.

### 9.2.3 Production

The integration of LLM-brained GUI agents into production environments demonstrates their practical viability and impact on enhancing user experiences in commercial applications.

**Power Automate** Microsoft (2024) exemplifies an AI-powered GUI agent that enhances user interaction with desktop applications. By allowing users to describe tasks in natural language while recording actions, it translates these descriptions into automated workflows, effectively bridging the gap between user intent and execution. Its ability to record and replicate user actions within the GUI streamlines the automation of repetitive tasks, making it a valuable tool for increasing efficiency and highlighting advancements in user-friendly automation solutions.

In the realm of web interactions, **MultiOn** MultiOn AI (2024) serves as a personal AI agent that autonomously interacts with web-based GUIs to execute user-defined tasks. Leveraging large language models, it interprets natural language commands and translates them into precise web actions, effectively automating complex or repetitive tasks. MultiOn's approach to perceiving and manipulating web elements enables seamless functioning across various web platforms, enhancing user productivity and streamlining web interactions.

On mobile platforms, the **YOYO Agent** in *MagicOS* HONOR (2024) exemplifies an LLM-powered GUI agent operating within the MagicOS 9.0 interface. Utilizing Honor's MagicLM, it comprehends and executes user commands across various applications, learning from user behavior to offer personalized assistance. This integration demonstrates how large language models can enhance virtual assistants, enabling them to perform complex tasks within GUI environments and improving user experience and productivity on mobile devices.

Eko AI (2025) serves as a prime example of a versatile and efficient tool for developing intelligent agents capable of interacting with GUIs across various platforms. Its integration with multiple LLMs and the innovative Visual-Interactive Element Perception (VIEP) technology highlight its capability to perform complex tasks through natural language instructions. Eko's comprehensive tool support make it a valuable resource for developers aiming to create customizable and production-ready agent-based workflows. By facilitating seamless interaction within GUI environments, Eko exemplifies the advancements in virtual assistants powered by LLMs.

These production-level implementations highlight the practical applications and benefits of LLM-brained GUI agents in enhancing automation, productivity, and user engagement across different platforms and industries.

## 10 Limitations, Challenges and Future Roadmap

Despite significant advancements in the development of LLM-brained GUI agents, it is important to acknowledge that this field is still in its infancy. Several technical challenges and limitations hinder their widespread adoption in real-world applications. Addressing these issues is crucial to enhance the agents' effectiveness,

safety, and user acceptance. In this section, we outline key limitations and propose future research directions to overcome these challenges, providing concrete examples to illustrate each point.

## 10.1 Privacy Concerns

Privacy is a critical concern uniquely intensified in the context of LLM-powered GUI agents. These agents often require access to sensitive user data—such as screenshots, interaction histories, personal credentials, and confidential documents—to effectively perceive and interact with the GUI environment. In many cases, this data must be transmitted to remote servers for model inference, especially when relying on cloud-based LLMs Liao et al. (2024); He et al. (2024a); Gan et al. (2024). Such deployments raise significant privacy risks, including data breaches, unauthorized access, and misuse of personal information. These concerns are further amplified when sensitive inputs are routed through third-party APIs or processed off-device, creating compliance and security vulnerabilities that can deter real-world adoption.

For instance, a GUI agent tasked with managing a user's email inbox may need to read, classify, and respond to messages containing highly personal or confidential content. Offloading this processing to the cloud introduces risks of exposure, prompting hesitation among users and organizations due to potential privacy violations Zharmagambetov et al. (2025); Yang et al. (2024c); Zhang et al. (2024l). Compared to traditional LLM applications, GUI agents operate at a finer granularity of user activity and often require broader system access, making privacy-preserving deployment strategies a critical and domain-specific challenge.

**Potential Solutions:** To mitigate privacy concerns, future research should focus on enabling *on-device inference*, where the language model operates directly on the user's device without uploading personal data Xu et al. (2024d); Qu et al. (2024). Achieving this requires advancements in model compression techniques Lin et al. (2024a), on-device optimization Liu et al. (2024j), and efficient inference algorithms Zhou et al. (2024b) to accommodate the computational limitations of user devices. In addition, frameworks must incorporate data redaction, secure communication channels, and explicit scoping of data usage within the agent's context. Furthermore, integration with system-level privacy controls and user consent mechanisms (*e.g.*, runtime permission dialogs or sandboxed execution) is essential for deployment in regulated domains.

From the technical perspective, implementing privacy-preserving techniques like federated learning Kuang et al. (2024), differential privacy Mai et al. (2023), and homomorphic encryption de Castro et al. can enhance data security while allowing the model to learn from user data. Furthermore, developers of GUI agents should collaborate with privacy policymakers to ensure that user data and privacy are appropriately protected Wolff et al. (2024). They should make the data handling processes transparent to users, clearly informing them about what data are being transmitted and how they are used, and obtain explicit user consent Zhang et al. (2024r).

## 10.2 Latency, Performance, and Resource Constraints

One challenge that is particularly salient for GUI agents—distinct from general LLM applications is the issue of latency in interactive, multi-step execution environments. Since GUI agents rely on large language models to plan and issue actions, their computational demands can lead to high latency and slow response times, which directly impact user experience Li et al. (2024a). This is especially critical in time-sensitive or interactive scenarios, where delays in action execution can cause user frustration or even trigger unintended system behavior. Unlike single-shot LLM tasks, GUI agents typically operate over extended sequences of steps, making latency cumulative and more disruptive over time. The problem is further amplified in on-device deployments, where computational resources are limited. For example, running an LLM-powered agent within a mobile app may result in sluggish performance or rapid battery depletion, significantly undermining usability on resource-constrained platforms Xu et al. (2024f); Chen et al. (2024a); Krupp et al. (2025). These concerns are uniquely pronounced in GUI agents due to their need for real-time perception, decision-making, and UI control in dynamic environments Chen et al. (2024a).

**Potential Solutions:** Future work should aim to reduce inference latency by optimizing model architectures for speed and efficiency Wan et al. (2023). Techniques such as model distillation can create smaller, faster models without substantially compromising performance Xu et al. (2024h). Leveraging hardware accelerators

like GPUs, TPUs, or specialized AI chips, and exploring parallel processing methods can enhance computational efficiency Kachris (2024). Implementing incremental inference and caching mechanisms may also improve responsiveness by reusing computations where applicable Lee et al. (2024d). Additionally, research into model optimization and compression techniques, such as pruning Wang et al. (2019) and quantization Lin et al. (2024a) can produce lightweight models suitable for deployment on resource-constrained devices. Exploring edge computing Qu et al. (2024) and distributed inference Wu et al. (2023a) can help distribute the computational load effectively.

Moreover, GUI agents should collaborate with application developers to encourage them to expose high-level native APIs for different functionalities Song et al. (2024b); Lu et al. (2024a), which combine several UI operations into single API calls. By integrating these APIs into the GUI agent, tasks can be completed with fewer steps, making the process much faster and reducing cumulative latency.

## 10.3 Safety and Reliability

The real-world actuation capabilities of GUI agents introduce unique and significant safety and reliability risks beyond those faced by general-purpose LLMs. Because GUI agents can directly manipulate user interfaces—clicking buttons, deleting files, submitting forms, or initiating system-level operations—errors in action generation can have irreversible consequences Anwar et al. (2024); Gan et al. (2024). These may include data corruption, accidental message dispatches, application crashes, or unauthorized access to sensitive system components Zhong & Wang (2023); Yuan et al. (2024). Such risks are compounded by the inherent uncertainty and non-determinism in LLM outputs: agents may hallucinate actions, misinterpret UI contexts, or behave inconsistently across sessions Zhang et al. (2024i); Zhao et al. (2025b); Zhang et al. (2023d); Chiang et al. (2025); Chen et al. (2025a). For example, an agent automating financial transactions could mistakenly execute the wrong transfer, leading to material losses. Furthermore, GUI agents expose a broader attack surface than traditional LLM applications—they are susceptible to black-box adversarial attacks that could manipulate their inputs or exploit their decision policies Xu et al. (2024a).

Unlike passive language models, GUI agents operate within dynamic software ecosystems where incorrect actions can propagate across applications or escalate into system-wide disruptions. Integration challenges also arise, including compatibility with evolving UI frameworks, user permission boundaries, and software-specific safety constraints, and malicious attacks Yang et al. (2025d); Aichberger et al.. These concerns, coupled with the lack of interpretability and formal guarantees, contribute to skepticism and reluctance from users and developers alike. Addressing safety and reliability in GUI agents thus requires not only robust model behavior but also runtime safeguards Lee et al. (2025), rollback mechanisms, and interface-aware verification techniques tailored specifically to this interaction paradigm.

**Potential Solutions:** Ensuring safety and reliability necessitates robust error detection and handling mechanisms Pan et al. (2023a). Future research should focus on integrating validation steps that verify the correctness of inferred actions before execution Huang et al. (2023). Developing formal verification methods Jha et al. (2023), implementing exception handling routines Zhang et al. (2023b), and establishing rollback procedures Koo & Toueg (1986) are essential for preventing and mitigating the impact of errors. Additionally, incorporating permission management Luo et al. (2017); Hao et al. (2013); Felt et al. (2011); Lutaaya (2018) to limit the agent's access rights can prevent unauthorized or harmful operations.

Furthermore, creating standardized interaction protocols can facilitate smoother and safer integration with various applications and systems Xiang et al. (2024). Ensuring that agents comply with security best practices, such as secure authentication and authorization protocols Berkovits et al. (1998), is essential.

## 10.4 Human-Agent Interaction

Human-agent interaction introduces distinct challenges in the context of GUI agents, where the agent and user operate within the same dynamic interface. Any user intervention—such as moving the mouse, altering window states, or modifying inputs—can inadvertently interfere with the agent's ongoing execution, potentially causing conflicts, unintended actions, or breakdowns in task flow Gao et al. (2024d); Bradshaw et al. (2017).

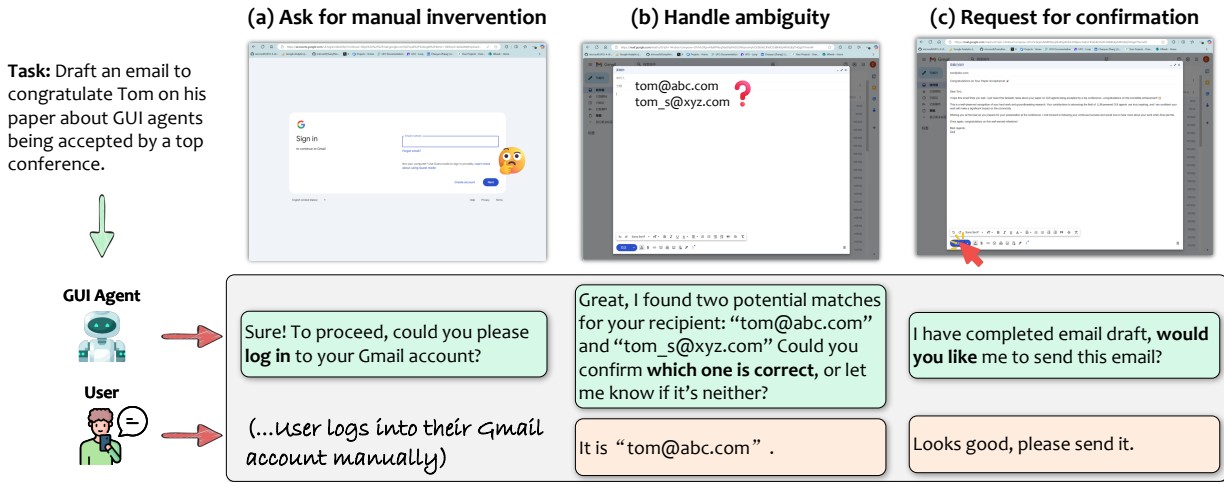

Figure 30: An illustrative example of human-agent interaction for completing an email sending request.

Designing robust collaboration protocols that govern when the agent should yield control, pause execution, or defer to the user is a non-trivial problem specific to GUI-based automation.

Further complicating this interaction is the ambiguity of user instructions. Natural language commands may be vague, under-specified, or context-dependent, leading to misinterpretations or incomplete task plans. GUI agents may also encounter runtime uncertainties—such as unexpected popups, missing inputs, or conflicting UI states—that require them to seek user clarification or feedback Zhang et al. (2024a); Feng et al. (2024). Determining when and how an agent should request user input—whether for disambiguation, permission, or verification—is critical for ensuring both reliability and user trust Amayuelas et al. (2023); Shi et al. (2025).

This challenge is exemplified in the fabricated scenario shown in Figure 30, where a GUI agent is instructed to send an email to "Tom." The agent must first prompt the user to log in securely, protecting credentials by avoiding automated input. It then encounters ambiguity when multiple contacts named "Tom" are found, and resolves it by prompting the user to select the intended recipient. Finally, before dispatching the email, the agent requests explicit confirmation, recognizing that email-sending is a non-reversible action with privacy implications Zhang et al. (2024a). Although the task appears simple, it reflects the complexity of real-world human-GUI agent collaboration, involving privacy preservation, ambiguity resolution, and intentionality confirmation Kim et al. (2024a). These are not generic LLM issues, but domain-specific challenges rooted in shared interaction with software interfaces—underscoring the need for new design paradigms around shared control, interruption handling, and proactive clarification in GUI agent systems.

**Potential Solutions:** Emphasizing *user-centered design* Lu et al. (2024e) principles can address user needs and concerns, providing options for customization and control over the agent's behavior Feng et al. (2024). Equipping agents with the ability to engage in *clarification dialogues* when user instructions are unclear can enhance task accuracy Wester et al. (2024). Natural language understanding components can detect ambiguity and prompt users for additional information. For instance, the agent could ask, "There are two contacts named John. Do you mean John Smith or John Doe?" Incorporating *human-in-the-loop* systems allows for human intervention during task execution, enabling users to guide or correct the agent's decisions when necessary Wang et al. (2024b). Developing adaptive interaction models that facilitate seamless collaboration between humans and agents is essential. Additionally, providing transparency and explainability in the agent's reasoning processes can build user trust and improve cooperation Cambria et al. (2024); Wu et al. (2024d); Shi et al. (2025); Chen et al. (2025b).

Lastly, developing a virtual desktop environment for the agent to operate in—one that connects to the user's main desktop session without disrupting their workflow, can significantly enhance the user experience (UX) in human-agent interaction. The picture-in-picture mode implemented in UFO$^2$ Zhang et al. (2025b)

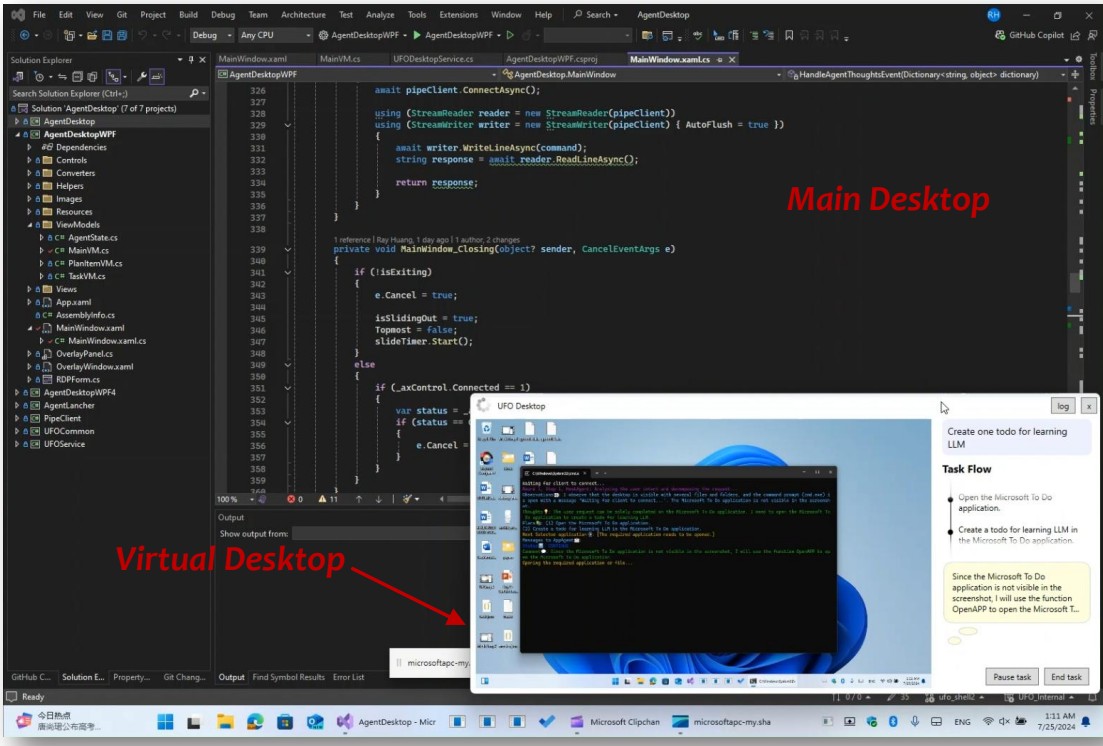

Figure 31: The Picture-in-Picture interface in UFO$^2$: a virtual desktop window enabling non-disruptive automation. Figure adapted from Zhang et al. (2025b).

demonstrates this concept in practice, as illustrated in Figure 31. By allowing the agent to run within a resizable and movable virtualized desktop, users can easily minimize or reposition the agent window as needed. This flexibility improves both the usability and the overall UX of interacting with GUI-based agents.

## 10.5 Customization and Personalization

Effective GUI agents must go beyond generic task completion and provide experiences that are personalized to individual users, adapting to their unique workflows, preferences, and behavioral patterns Li et al. (2024i); Cai et al. (2024a). Unlike general LLM applications that operate in isolated prompts or conversations, GUI agents work across software environments where user interaction styles can vary significantly. A one-size-fits-all agent may fail to align with how a particular user edits documents, navigates interfaces, or organizes tasks—resulting in friction, inefficiency, or user frustration Li et al. (2024c).

For instance, a GUI agent assisting with document editing must learn the user's preferred tone, formatting conventions, and vocabulary. Without this contextual understanding, the agent may offer irrelevant suggestions or enforce formatting inconsistent with the user's intent. Personalization in GUI agents thus requires longitudinal learning, where the agent continually adapts based on prior interactions, fine-tunes its behavior to match user expectations, and preserves consistency across sessions Li et al. (2023b).

However, this introduces new challenges. The high variability in user preferences—especially in free-form GUI environments—makes it difficult to define universal personalization strategies. Moreover, collecting and leveraging user-specific data must be done responsibly, raising critical concerns around privacy, data retention, and on-device learning. Striking a balance between effective customization and user trust is particularly

important for GUI agents, which often operate over sensitive documents, personal applications, or system-level interfaces.

**Potential Solutions:** Future research should focus on developing mechanisms for *user modeling* Tan & Jiang (2023) and *preference learning* Gao et al. (2024c), enabling agents to tailor their actions to individual users. Techniques such as reinforcement learning from user feedback Kaufmann et al. (2023), collaborative filtering Kim et al. (2024c), and context-aware computing Talukdar & Biswas (2024) can help agents learn user preferences over time. Ensuring that personalization is achieved without compromising privacy is essential Xiao & Tao (2006), potentially through on-device learning and anonymized data processing. In a more futuristic, cyberpunk-inspired scenario, agents may inversely generate GUIs tailored to users' needs, enabling greater customization and personalization Hojo et al. (2025).

### 10.6 Ethical and Regulatory Challenges

LLM-powered GUI agents raise distinct ethical and regulatory concerns due to their ability to perform real-world actions across software interfaces. Unlike traditional LLMs, these agents can autonomously trigger operations, manipulate data, and interact with sensitive applications—amplifying risks around accountability, fairness, and user consent Gan et al. (2024); Sarker (2024); Biswas & Talukdar (2023); Li et al. (2023e); Zhang et al. (2025g).

A key concern is bias inherited from training data, which can lead to unfair behavior in sensitive workflows. For example, a GUI agent assisting in hiring may unknowingly exhibit gender or racial bias Ferrara (2023); Yu et al. (2024). These risks are harder to audit at the GUI level due to limited traceability across multi-application actions. Regulatory compliance adds further complexity. GUI agents often operate across domains with strict data protection laws, but lack standardized mechanisms for logging actions or securing user consent. This makes it challenging to meet legal and ethical standards, especially when agents act in opaque or background contexts. Addressing these issues requires tailored solutions for GUI agents, including permission controls, runtime confirmations, and transparent activity logs—ensuring safe, fair, and compliant deployment across diverse environments.

**Potential Solutions:** Addressing these concerns requires establishing clear ethical guidelines and regulatory frameworks for the development and use of GUI agents Piñeiro-Martín et al. (2023). Future work should focus on creating mechanisms for auditing and monitoring agent behavior Zheng et al. to ensure compliance with ethical standards and legal requirements Chan et al. (2024). Incorporating bias detection and mitigation strategies in language models can help prevent discriminatory or unfair actions Lin et al. (2024d). Providing users with control over data usage and clear information about the agent's capabilities can enhance transparency and trust.

### 10.7 Scalability and Generalization

GUI agents often struggle to scale beyond specific applications or environments, limiting their generalization. Each software interface features unique layouts, styles, and interaction patterns—even common UI elements like pop-up windows can vary widely Zhang et al. (2024m). These variations make it difficult to design agents that operate robustly across platforms without retraining or fine-tuning.

A further challenge is the dynamic nature of real-world GUIs. Frequent changes due to software updates, A/B testing, or interface redesigns—such as repositioned buttons or modified widget hierarchies—can easily break previously functional agents. For example, an agent trained on one version of a word processor may fail when the layout changes, or when deployed on a different program with similar functionality but a different interface structure. Even when GUIs share visual similarities, agents often fail to generalize without additional exploration or adaptation Shekkizhar & Cosentino (2025). This lack of robustness restricts deployment in practical settings and increases the cost of maintenance, requiring frequent updates or retraining to stay aligned with evolving environments Grosse et al. (2023); Zhang et al. (2024k); Li & Waldo (2024). Overcoming this challenge remains critical for developing truly scalable and adaptable GUI agents.

**Potential Solutions:** To enhance scalability and generalization, one solution from the dataset perspective is to create comprehensive GUI agent datasets that cover a wide range of environments, user requests, GUI

designs, platforms, and interaction patterns. By exposing the LLM to diverse data sources during training, the model can learn common patterns and develop a more generalized understanding, enabling it to adapt to infer the functionality of new interfaces based on learned similarities Song et al. (2024a).

To further enhance adaptability, research can focus on techniques such as *transfer learning* Weiss et al. (2016) and *meta-learning* Chen et al. (2021b). *Transfer learning* involves pre-training a model on a large, diverse dataset and then fine-tuning it on a smaller, task-specific dataset. In the context of GUI agents, this means training the LLM on a wide array of GUI interactions before customizing it for a particular application or domain. *Meta-learning*, enables the model to rapidly adapt to new tasks with minimal data by identifying underlying structures and patterns across different tasks. These approaches enable agents to generalize from limited data and adapt to new environments with minimal retraining.

However, even with these measures, the agent may still encounter difficulties in unfamiliar environments. To address this, we advocate for developers to provide helpful knowledge bases, such as guidance documents, application documentation, searchable FAQs, and even human demonstrations on how to use the application Zhu et al. (2024a); Guan et al. (2024b); Hsieh et al. (2023). Techniques like RAG Gao et al. (2023) can be employed, where the agent retrieves relevant information from a knowledge base at runtime to inform its decisions Kagaya et al. (2024). For instance, if the agent encounters an unknown interface element, it can query the documentation to understand its purpose and how to interact with it. This approach enhances the agent's capabilities without requiring extensive retraining. Implementing these solutions requires collaborative efforts not only from agent developers but also from application or environment providers.

## 10.8 Summary

LLM-brained GUI agents hold significant promise for automating complex tasks and enhancing user productivity across various applications. However, realizing this potential requires addressing the outlined limitations through dedicated research and development efforts. By addressing these challenges, the community can develop more robust and widely adopted GUI agents.

Collaboration among researchers, industry practitioners, policymakers, and users is essential to navigate these challenges successfully. Establishing interdisciplinary teams can foster innovation and ensure that GUI agents are developed responsibly, with a clear understanding of technical, ethical, and societal implications. As the field progresses, continuous evaluation and adaptation will be crucial to align technological advancements with user needs and expectations, ultimately leading to more intelligent, safe, and user-friendly GUI agents.

## 11 Conclusion

The combination of LLMs and GUI automation marks a transformative moment in human-computer interaction. LLMs provide the "brain" for natural language processing, comprehension, and GUI understanding, while GUI automation tools serve as the "hands", translating the agent's cognitive abilities into actionable commands within software environments. Together, they form LLM-powered GUI agents that introduce a new paradigm in user interaction, allowing users to control applications through straightforward natural language commands instead of complex, platform-specific UI operations. This synergy has shown remarkable potential, with applications flourishing in both research and industry.

In this survey, we provide a comprehensive, systematic, and timely overview of the field of LLM-powered GUI agents. Our work introduces the core components and advanced techniques that underpin these agents, while also examining critical elements such as data collection, model development, frameworks, evaluation methodologies, and real-world applications. Additionally, we explore the current limitations and challenges faced by these agents and outline a roadmap for future research directions. We hope this survey serves as a valuable handbook for those learning about LLM-powered GUI agents and as a reference point for researchers aiming to stay at the forefront of developments in this field.

As we look to the future, the concept of LLM-brained GUI agents promises to become increasingly tangible, fundamentally enhancing productivity and accessibility in daily life. With ongoing research and development,

this technology stands poised to reshape how we interact with digital systems, transforming complex workflows into seamless, natural interactions.

# A    Evolution and Progression of LLM-Powered GUI Agents

"Rome wasn't built in a day." The development of LLM-powered GUI agents has been a gradual journey, grounded in decades of research and technical progress. Beginning with simple GUI testing scripts and rule-based automation frameworks, the field has evolved significantly through the integration of machine learning techniques, creating more intelligent and adaptive systems. The introduction of LLMs, especially multimodal models, has transformed GUI automation by enabling natural language interactions and fundamentally reshaping how users interact with software applications.

## A.1    Early Automation Systems

In the initial stages of GUI automation, researchers relied on random-based, rule-based, and script-based strategies. While foundational, these methods had notable limitations in terms of flexibility and adaptability.

### A.1.1    Random-Based Automation

Random-based automation uses random sequences of actions within the GUI without relying on specific algorithms or structured models using monkey test Wetzlmaier et al. (2016). This approach was widely used in GUI testing to uncover potential issues by exploring unpredictable input sequences Zeng et al. (2016). While effective at identifying edge cases and bugs, random-based methods were often inefficient due to a high number of redundant or irrelevant trials.

### A.1.2    Rule-Based Automation

Rule-based automation applies predefined rules and logic to automate tasks. In 2001, Memon *et al.*, Memon et al. (2001) introduced a planning approach that generated GUI test cases by transforming initial states to goal states through a series of predefined operators. Hellmann *et al.*, Hellmann & Maurer (2011) (2011) demonstrated the potential of rule-based approaches in exploratory testing, enhancing bug detection. In the RPA domain, SmartRPA Agostinelli et al. (2020) (2020) used rule-based processing to automate routine tasks, illustrating the utility of rules for streamlining structured processes.

### A.1.3    Script-Based Automation

Script-based automation relies on detailed scripts to manage GUI interactions. Tools like jRapture Steven et al. (2000) (2000) record and replay Java-based GUI sequences using Java binaries and the JVM, enabling consistent execution by precisely reproducing input sequences. Similarly, DART Memon et al. (2003a) (2003) automated the GUI testing lifecycle, from structural analysis to test case generation and execution, offering a comprehensive framework for regression testing.

### A.1.4    Tools and Software

A range of software tools were developed for GUI testing and business process automation during this period. Microsoft Power Automate Microsoft (2024) (2019) provides a low-code/no-code environment for creating automated workflows within Microsoft applications. Selenium selenium (2024) (2004) supports cross-browser web testing, while Appium appium (2024) (2012) facilitates mobile UI automation. Commercial tools like TestComplete smartbear (2024) (1999), Katalon Studio katalon (2024) (2015), and Ranorex ranorex (2024) (2007) allow users to create automated tests with cross-platform capabilities.

Although these early systems were effective for automating specific, predefined workflows, they lacked flexibility and required manual scripting or rule-based logic. Nonetheless, they established the foundations of GUI automation, upon which more intelligent systems were built.

## A.2 The Shift Towards Intelligent Agents

The incorporation of machine learning marked a major shift towards more adaptable and capable GUI agents. Early milestones in this phase included advancements in machine learning, natural language processing, computer vision, and reinforcement learning applied to GUI tasks.

### A.2.1 Machine Learning and Computer Vision

RoScript Qian et al. (2020) (2020) was a pioneering system that introduced a non-intrusive robotic testing system for touchscreen applications, expanding GUI automation to diverse platforms. AppFlow Hu et al. (2018) (2018) used machine learning to recognize common screens and UI components, enabling modular testing for broad categories of applications. Progress in computer vision also enabled significant advances in GUI testing, with frameworks Chang et al. (2010) (2010) automating visual interaction tasks. Humanoid Li et al. (2019) (2019) uses a deep neural network model trained on human interaction traces within the Android system to learn how users select actions based on an app's GUI. This model is then utilized to guide test input generation, resulting in improved coverage and more human-like interaction patterns during testing. Similarly, Deep GUI YazdaniBanafsheDaragh & Malek (2021) (2021) applies deep learning techniques to filter out irrelevant parts of the screen, thereby enhancing black-box testing effectiveness in GUI testing by focusing only on significant elements. These approaches demonstrate the potential of deep learning to make GUI testing more efficient and intuitive by aligning it closely with actual user behavior.

Widget detection, as demonstrated by White *et al.*, White et al. (2019) (2019), leverages computer vision to accurately identify UI elements, serving as a supporting technique that enables more intelligent and responsive UI automation. By detecting and categorizing interface components, this approach enhances the agent's ability to interact effectively with complex and dynamic GUIs Xie et al. (2020).

### A.2.2 Natural Language Processing

Natural language processing capabilities introduced a new dimension to GUI automation. Systems like RUSS Xu et al. (2021) (2021) and FLIN Mazumder & Riva (2020) (2020) allowed users to control GUIs through natural language commands, bridging human language and machine actions. Datasets, such as those in Li et al. (2020a) (2020), further advanced the field by mapping natural language instructions to mobile UI actions, opening up broader applications in GUI control. However, these approaches are limited to handling simple natural commands and are not equipped to manage long-term tasks.

### A.2.3 Reinforcement Learning

The development of environments like World of Bits (WoB) Shi et al. (2017) (2017) enabled the training of web-based agents using reinforcement learning (RL). Workflow-guided exploration Liu et al. (2018) (2018) improved RL efficiency and task performance. DQT Lan et al. (2024) (2024) applied deep reinforcement learning to automate Android GUI testing by preserving widget structures and semantics, while AndroidEnv Toyama et al. (2021b) (2021) offered realistic simulations for agent training on Android. WebShop Yao et al. (2022a) (2022) illustrated the potential for large-scale web interaction, underscoring the growing sophistication of RL-driven GUI automation.

While these machine learning-based approaches were more adaptable than earlier rule-based systems Zhang et al. (2019); Martins et al. (2020), they still struggled to generalize across diverse, unforeseen tasks. Their dependence on predefined workflows and limited adaptability required retraining or customization for new environments, and natural language control was still limited.

## A.3 The Advent of LLM-Powered GUI Agents

The introduction of LLMs, particularly multimodal models like GPT-4o Hurst et al. (2024) (2023), has radically transformed GUI automation by allowing intuitive interactions through natural language. Unlike previous approaches that required integration of separate modules, LLMs provide an end-to-end solution for

GUI automation, offering advanced capabilities in natural language understanding, visual recognition, and reasoning.

LLMs present several unique advantages for GUI agents, including natural language understanding, multimodal processing, planning, and generalization. These features make LLMs and GUI agents a powerful combination. While there were earlier explorations, 2023 marked a pivotal year for LLM-powered GUI agents, with significant developments across various platforms such as web, mobile, and desktop applications.

### A.3.1   Web Domain

The initial application of LLMs in GUI automation was within the web domain, with early studies establishing benchmark datasets and environments Yao et al. (2022a); Shi et al. (2017). A key milestone was WebAgent Gur et al. (2023) (2023), which, alongside WebGUM Furuta et al. (2023) (2023), pioneered real-world web navigation using LLMs. These advancements paved the way for further developments Ma et al. (2023); Zheng et al. (2024a); Deng et al. (2024b), utilizing more specialized LLMs to enhance web-based interactions.

### A.3.2   Mobile Devices

The integration of LLMs into mobile devices began with AutoDroid Wen et al. (2024a) (2023), which combined LLMs with domain-specific knowledge for smartphone automation. Additional contributions like MM-Navigator Yan et al. (2023b) (2023), AppAgent Zhang et al. (2023a) (2023), and Mobile-Agent Wang et al. (2024e) (2023) enabled refined control over smartphone applications. Research has continued to improve accuracy for mobile GUI automation through model fine-tuning Nong et al. (2024); Zhang et al. (2024f) (2024).

### A.3.3   Computer Systems

For desktop applications, UFO Zhang et al. (2024a) (2024) was one of the first systems to leverage GPT-4 with visual capabilities to fulfill user commands in Windows environments. Cradle Tan et al. (2024a) (2024) extended these capabilities to software applications and games, while Wu *et al.*, Wu et al. (2024e) (2024) provided interaction across diverse desktop applications, including web browsers, code terminals, and multimedia tools.

### A.3.4   Industry Models

In industry, the `Claude 3.5 Sonnet` model Anthropic (2024) (2024) introduced a "computer use" feature capable of interacting with desktop environments through UI operations Hu et al. (2024a). This signifies the growing recognition of LLM-powered GUI agents as a valuable application in industry, with stakeholders increasingly investing in this technology.

Undoubtedly, LLMs have introduced new paradigms and increased the intelligence of GUI agents in ways that were previously unattainable. As the field continues to evolve, we anticipate a wave of commercialization, leading to transformative changes in user interaction with GUI applications.

## B   Additional Summary in Tabular Form

While the major works have been discussed in detail in Sections XX–XX, we provide additional summary tables (Table 16–70) for quick reference and ease of lookup. This table is intended to help readers conveniently locate key references and insights associated with each section.

Table 16: Overview of LLM-brained GUI agent frameworks on web platforms (Part I).

| Agent | Platform | Perception | Action | Model | Architecture | Highlight | Link |
|---|---|---|---|---|---|---|---|
| WMA Chae et al. (2024) | Web | Accessibility tree from DOM | UI operations, *e.g.*, clock, type, and hover | Llama-3.1-8B-Instruct Dubey et al. (2024) for predicting observations and GPT-4 for policy modeling | Single-agent with simulation-based observation | Uses a world model to predict state changes before committing actions, improving task success rates and minimizing unnecessary interactions with the environment | `https://github.com/kyle8581/WMA-Agents` |
| WebAgent Gur et al. (2024) | Web | HTML structure | UI interactions | HTML-T5 for task planning and summarization and Flan-U-PaLM Chung et al. (2024) for code generation | Two-stage architecture for planning and program synthesis | Leverages specialized LLMs to achieve HTML-based task planning and programmatic action execution | / |
| LASER Ma et al. (2024b) | Web | GUI structure of the web environment, with defined states | Defined per state, such as searching, selecting items, navigating pages, and finalizing a purchase | GPT-4 | Single-agent | Uses a state-space exploration approach, allowing it to handle novel situations with flexible backtracking | `https://github.com/Mayer123/LASER` |
| WebVoyager He et al. (2024b) | Web | Screenshots with numerical labels on interactive elements | Standard UI operations | GPT-4V | Single-agent | Integrates visual and textual cues within real-world, rendered web pages, enhancing its ability to navigate complex web structures | `https://github.com/MinorJerry/WebVoyager` |
| AutoWeb-GLM Lai et al. (2024) | Web | Simplified HTML and OCR for text recognition | UI operations such as clicking, typing, scrolling, and selecting, and advanced APIs like jumping to specific URLs | ChatGLM3-6B GLM et al. (2024) | Single-agent | Its HTML simplification method for efficient webpage comprehension and its bilingual benchmark | `https://github.com/THUDM/AutoWebGLM` |

Table 17: Overview of LLM-brained GUI agent frameworks on web platforms (Part II).

| Agent | Platform | Perception | Action | Model | Architecture | Highlight | Link |
|---|---|---|---|---|---|---|---|
| OpenAgents Xie et al. (2023) | Web | DOM elements | Standard UI operations, browser-based actions controlled, API calls for tool execution, and structured data manipulation | GPT-4 and Claude Anthropic (2024) | Multi-agent architecture, with distinct agents (Data Agent, Plugins Agent, and Web Agent) | Democratizes access to language agents by providing an open-source, multi-agent framework optimized for real-world tasks | https://github.com/xlang-ai/OpenAgents |
| SeeAct Zheng et al. (2024a) | Web | Screenshot images and HTML structure | Standard UI operations | GPT-4V | Single-agent | Its use of GPT-4V's multimodal capabilities to integrate both visual and HTML information, allowing for more accurate task performance on dynamic web content | https://github.com/OSU-NLP-Group/SeeAct |
| DUAL-VCR Kil et al. (2024) | Web | HTML elements and screenshots | Standard UI operations | Flan-T5-base Chung et al. (2024) | Two-stage single-agent architecture | Dual-view contextualization | / |
| Agent-E Abuelsaad et al. (2024) | Web | DOM structure and change observation | Standard UI operations | GPT-4 Turbo | Hierarchical multi-agent architecture, composed of a planner agent and a browser navigation agent | Hierarchical architecture and adaptive DOM perception | https://github.com/EmergenceAI/Agent-E |
| Search-Agent Koh et al. (2024b) | Web | Screenshot and text descriptions | Standard UI operations | GPT-4 | Single-agent with search | Novel inference-time search algorithm that enhances the agent's ability to perform multi-step planning and decision-making | https://jykoh.com/search-agents |
| R2D2 Huang et al. (2025b) | Web | DOM | Standard UI operations | GPT-4o | Single-agent | Dynamically constructs an internal web environment representation for more robust decision-making. The integration of a replay buffer and error analysis reduces navigation errors and improves task completion rates. | https://github.com/AmenRa/retriv |

Table 18: Overview of LLM-brained GUI agent frameworks on web platforms (Part III).

| Agent | Platform | Perception | Action | Model | Architecture | Highlight | Link |
|---|---|---|---|---|---|---|---|
| ScribeAgent Shen et al. (2024b) | Web | HTML-DOM | Standard UI operations | Single-agent architecture | Specialized fine-tuning approach using production-scale workflow data to outperform general-purpose LLMs like GPT-4 in web navigation tasks | `https://github.com/colonylabs/ScribeAgent` | |
| PAE Zhou et al. (2024a) | Web | Screenshots | Standard UI Operations | Claude 3 Sonnet Anthropic (2024), Qwen2VL-7B Wang et al. (2024j), and LLaVa-1.6 Liu et al. (2024b) | A multi-agent architecture involving a task proposer to suggest tasks, an agent policy to perform tasks, and an autonomous evaluator to assess success and provide feedback using RL. | Autonomous skill discovery in real-world environments using task proposers and reward-based evaluation | `https://yanqval.github.io/PAE/` |
| WebPilot Zhang et al. (2024n) | Web | Accessibility trees and dynamic observations | Standard UI operations | GPT-4 | Multi-agent architecture, with Global Optimization and Local Optimization | Dual optimization strategy (Global and Local) with Monte Carlo Tree Search (MCTS) Browne et al. (2012), allowing dynamic adaptation to complex, real-world web environments | `https://yaoz720.github.io/WebPilot/` |
| Hybrid Agent Song et al. (2024b) | Web | Accessibility trees and screenshots | Standard UI operations, API calls, and generating code | GPT-4 | Multi-agent system, combining both API and browsing capabilities | Hybrid Agent seamlessly integrates web browsing and API calls | `https://github.com/yueqis/API-Based-Agent` |
| AgentOccam Yang et al. (2024a) | Web | HTML | Standard UI operations | GPT-4 | Single-agent | Simple design that optimizes the observation and action spaces | / |
| NNetnav Murty et al. (2024) | Web | DOM | Standard UI operations | GPT-4 | Single-agent | Trains web agents using synthetic demonstrations, eliminating the need for expensive human input | `https://github.com/MurtyShikhar/Nnetnav` |

Table 19: Overview of LLM-brained GUI agent frameworks on web platforms (Part IV).

| Agent | Platform | Perception | Action | Model | Architecture | Highlight | Link |
|---|---|---|---|---|---|---|---|
| NaviQAte Shahbandeh et al. (2024) | Web | Screenshots | Standard UI operations | GPT-4 | Single-agent system | Frames web navigation as a question-and-answer task | / |
| OpenWeb-Agent Iong et al. (2024) | Web | HTML and screenshots | UI operations, Web APIs, and self-generated code | GPT-4 and AutoWe-bGLM Lai et al. (2024) | Modular single-agent | Modular design that allows developers to seamlessly integrate various models to automate web tasks | `https://github.com/THUDM/OpenWebAgent/` |
| Steward Tang & Shin (2024) | Web | HTML and screenshots | Standard UI operations | GPT-4 | Single-agent | Ability to automate web interactions using natural language instructions | / |
| WebDreamer Gu et al. (2024) | Web | Screenshots combined with SoM, and HTML | Standard UI operations and navigation actions | GPT-4o | Model-based single-agent architecture | Pioneers the use of LLMs as world models for planning in complex web environments | `https://github.com/OSU-NLP-Group/WebDreamer` |
| Agent Q Putta et al. (2024) | Web | DOM for textual input, screenshots for visual feedback | UI interactions, querying the user for help | LLaMA-3 70B Dubey et al. (2024) for policy learning and execution, GPT-V for visual feedback | Single-agent with MCTS and RL | Combination of MCTS-guided search and self-critique mechanisms enables iterative improvement in reasoning and task execution | `https://github.com/sentient-engineering/agent-q` |
| Auto-Intent Kim et al. (2024b) | Web | HTML structure | Standard UI Operations | GPT-3.5, GPT-4, Llama-3 Dubey et al. (2024) for action inference; Mistral-7B Jiang et al. (2023) and Flan-T5XL Chung et al. (2024) for intent prediction | Single-agent with self-exploration | Introduces a unique self-exploration strategy to generate semantically diverse intent hints | / |

Table 20: Overview of LLM-brained GUI agent frameworks on web platforms (Part V).

| Agent | Platform | Perception | Action | Model | Architecture | Highlight | Link |
|---|---|---|---|---|---|---|---|
| AdaptAgent Verma et al. (2024) | Web | GUI screenshots with HTML/-DOM structures | Standard UI Operations and Playwright scripts | GPT-4o and CogAgent Hong et al. (2023) | Single-agent | Adapts to unseen tasks with just 1–2 multimodal human demonstrations | / |
| WEPO Liu et al. (2024c) | Web | HTML and DOM | Standard UI Operations | Llama3-8B Dubey et al. (2024), Mistral-7B Jiang et al. (2023), and Gemma-2B Team et al. (2024) | Single-agent architecture. | Incorporates a distance-based sampling mechanism tailored to the DOM tree structure, enhancing preference learning by distinguishing between salient and non-salient web elements with DPO Rafailov et al. (2024). | / |
| Agent-Symbiotic Zhang et al. (2025e) | Web | Accessible tree structure of web elements | Standard UI operations | Large LLMs: GPT-4o, Claude-3.5. Small LLMs: LLaMA-3 Dubey et al. (2024), DeepSeek-R1 Guo et al. (2025) | Multi-agent iterative architecture | Introduces an iterative, symbiotic learning process between large and small LLMs for web automation. Enhances both data synthesis and task performance through speculative data synthesis, multi-task learning, and privacy-preserving hybrid modes. | / |
| LiteWebAgent Zhang et al. (2025c) | Web | DOM, Screenshots | Standard UI operations, Playwright script | Any LLM and MLLM | Single-agent | First open-source, production-ready web agent integrating tree search for multi-step task execution. | https://github.com/PathOnAI/LiteWebAgent |
| ECLAIR Wornow et al. (2024) | Web | Screenshots | Standard UI operations | GPT-4V, GPT-4o, CogAgent | Single-agent architecture | Eliminates the high setup costs, brittle execution, and burdensome maintenance associated with traditional RPA by learning from video and text documentation. | https://github.com/HazyResearch/eclair-agents |
| Dammu et al., Dammu (2025) | Web | DOM elements, Webpage accessibility attributes | Standard UI operations | Not specified | Single-agent architecture | User-aligned task execution, where the agent adapts to individual user preferences in an ethical manner. | / |

Table 21: Overview of LLM-brained GUI agent frameworks on web platforms (Part VI).

| Agent | Platform | Perception | Action | Model | Architecture | Highlight | Link |
|---|---|---|---|---|---|---|---|
| Plan-and-Act Erdogan et al. (2025) | Web | HTML | Standard UI operations | LLaMA-3.3-70B-Instruct Dubey et al. (2024) | Two-stage modular architecture: PLANNER + EXECUTOR | Decouples planning from execution in LLM-based GUI agents and introduces a scalable synthetic data generation pipeline to fine-tune each component | / |
| SkillWeaver Zheng et al. (2025a) | Web | GUI screenshots and Accessibility Tree | Standard UI operations and high-level skill APIs | GPT-4o | Single-agent | Introduces a self-improvement framework for web agents that autonomously discover, synthesize, and refine reusable skill APIs through exploration | `https://github.com/OSU-NLP-Group/SkillWeaver` |
| ASI Wang et al. (2025g) | Web | Webpage Accessibility Tree | Standard GUI actions | Claude-3.5-Sonnet | Single-agent | Introduces programmatic skills that are verified through execution to ensure quality and are used as callable actions to improve efficiency | `https://github.com/zorazrw/agent-skill-induction` |
| Rollback Agent Zhang et al. (2025f) | Web | Accessibility trees | Standard GUI actions | Multi-agent architecture | Multi-module, ReAct-inspired agent architecture | Introduces a modular rollback mechanism that enables multi-step rollback to avoid dead-end states | / |

Table 22: Overview of LLM-brained GUI agent frameworks on mobile platforms (Part I).

| Agent | Platform | Perception | Action | Model | Architecture | Highlight | Link |
|---|---|---|---|---|---|---|---|
| Wang *et al.*, Wang et al. (2023a) | Android Mobile | Android view hierarchy structure | (1) Screen Question Generation, (2) Screen Summarization, (3) Screen Question Answering, and (4) Mapping Instruction to UI Action | PaLM Anil et al. (2023) | Single-agent | The first paper to study Screen Question Generation and Screen QA using LLMs | `https://github.com/google-research/google-research/tree/master/llm4mobile` |
| VisionTasker Song et al. (2024c) | Android mobile devices | UI screenshots with widget detection and text extraction | UI operations such as tapping, swiping, and entering text | ERNIE Bot Baidu Research (2024) | Single-agent with vision-based UI understanding and sequential task planning | Vision-based UI understanding approach, which allows it to interpret UI semantics directly from screenshots without view hierarchy dependencies | `https://github.com/AkimotoAyako/VisionTasker` |
| DroidBot-GPT Wen et al. (2024c) | Android mobile devices | Translates the GUI state information of Android applications into natural language prompts | UI operations, including actions like click, scroll, check, and edit | GPT | Single-agent | Automates Android applications without modifications to either the app or the model | `https://github.com/MobileLLM/DroidBot-GPT` |
| CoCo-Agent Ma et al. (2024d) | Android mobile devices | GUI screenshots, OCR layouts, and historical actions | GUI actions, such as clicking, scrolling, and typing | CLIP Radford et al. (2021) for vision encoding and LLaMA-2-chat-7B for language processing | Single-agent | Its dual approach of Comprehensive Environment Perception and Conditional Action Prediction | `https://github.com/xbmxb/CoCo-Agent` |
| Auto-GUI Zhang & Zhang (2024) | Android mobile devices | GUI screenshots | GUI operations | BLIP-2 vision encoder Li et al. (2023c) with a FLAN-Alpaca Wei et al. (2021) | Single-agent with chain-of-action | Its direct interaction with GUI elements. Its chain-of-action mechanism enables it to leverage both past and planned actions | `https://github.com/cooelf/Auto-GUI` |

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

Table 23: Overview of LLM-brained GUI agent frameworks on mobile platforms (Part II).

| Agent | Platform | Perception | Action | Model | Architecture | Highlight | Link |
|---|---|---|---|---|---|---|---|
| MobileGPT Lee et al. (2024c) | Android mobile devices | Simplified HTML representation | Standard UI operations and navigation actions | GPT-4-turbo for screen understanding and reasoning, GPT-3.5-turbo for slot-filling sub-task parameters | Single-agent architecture augmented by a hierarchical memory structure | Introduces a human-like app memory that allows for task decomposition into modular sub-tasks | `https:// mobile-gpt. github.io` |
| MM-Navigator Yan et al. (2023a) | Mobile iOS and Android | Smartphone screenshots with associated set-of-mark tags | Clickable UI operations | GPT-4V | Single-agent | Using set-of-mark prompting with GPT-4V for precise GUI navigation on smartphones | `https: //github. com/zzxslp/ MM-Navigator` |
| Prompt2Task Huang et al. | Android mobile devices | GUI structure and layout hierarchy, full-page textual descriptions, OCR-based text extraction | Standard UI operations | GPT-4 | Multi-agent architecture | Enables UI automation through free-form textual prompts, eliminating the need for users to script automation tasks. | `https:// github.com/ PromptRPA/ Prompt2TaskDataset` |
| AppAgent Zhang et al. (2023a) | Android mobile devices | Real-time screenshots and XML files detailing the interactive elements | User-like actions, like Tap, Long press, Swipe, Text input, Back and Exit | GPT-4V | Single-agent | Its ability to perform tasks on any smartphone app using a human-like interaction method | `https:// appagent-official. github.io/` |
| AppAgent-V2 Li et al. (2024g) | Android mobile devices | GUI screenshots with annotated elements, OCR for detecting text and icons, Structured XML metadata | Standard UI Operations: Tap, text input, long press, swipe, back, and stop | GPT-4 | Multi-phase architecture with Exploration Phase and Deployment Phase | Enhances adaptability and precision in mobile environments by combining structured data parsing with visual features | / |

Table 24: Overview of LLM-brained GUI agent frameworks on mobile platforms (Part III).

| Agent | Platform | Perception | Action | Model | Architecture | Highlight | Link |
|---|---|---|---|---|---|---|---|
| AutoDroid Wen et al. (2024a) | Android mobile devices | Simplified HTML-style representation | Standard UI operations | GPT-3.5, GPT-4, and Vicuna-7B Chiang et al. (2023) | Single-agent architecture | Its use of app-specific knowledge and a multi-granularity query optimization module to reduce the computational cost | `https://autodroid-sys.github.io/` |
| AutoDroid-V2 Wen et al. (2024b) | Android Mobile Devices | Structured GUI Representations | Multi-step scripts of standard UI operations and API calls | Llama-3.1-8B Dubey et al. (2024) | Script-based architecture. | Converts GUI task automation into a script generation problem, enhancing efficiency and task success rates. | / |
| CoAT Zhang et al. (2024g) | Android mobile devices | Screenshot-based context and semantic information | Standard UI operations | GPT-4V | Single-agent architecture | The integration of a chain-of-action-thought process, which explicitly maps each action to screen descriptions, reasoning steps, and anticipated outcomes | `https://github.com/ZhangL-HKU/CoAT` |
| Mobile-Agent-E Wang et al. (2025e) | Mobile Android | GUI screenshots, OCR for detecting text and icons | Standard UI operations and APIs | GPT-4o, Claude-3.5-Sonnet, Gemini-1.5-Pro | Hierarchical Multi-Agent System | Hierarchical multi-agent framework that separates planning from execution for improved long-term reasoning and self-evolution, enabling the system to learn reusable tips and shortcuts | `https://x-plug.github.io/MobileAgent` |
| FedMobile-Agent Wang et al. (2025c) | Android mobile devices | GUI Screenshots | Standard UI operations | Qwen2-VL-Instruct-7B Wang et al. (2024j) | Multi-agent federated learning | Introduces privacy-preserving federated learning for mobile automation, enabling large-scale training without centralized human annotation. | / |

Table 25: Overview of LLM-brained GUI agent frameworks on mobile platforms (Part IV).

| Agent | Platform | Perception | Action | Model | Architecture | Highlight | Link |
|---|---|---|---|---|---|---|---|
| ClickAgent Hoscilowicz et al. (2024) | Android Mobile Devices | Screenshots | Standard UI operations | InternVL-2.0 Chen et al. (2024l), TinyClick Pawlowski et al. (2024), SeeClick Cheng et al. (2024a) | Single-agent | Combines MLLM reasoning with a dedicated UI location model to enhance UI interaction accuracy | https://github.com/Samsung/ClickAgent |
| Mobile-Agent Wang et al. (2024e) | Mobile Android | Screenshots with icon detection | Standard UI operations | GPT-4V with Grounding DINO Liu et al. (2023a) and CLIP Radford et al. (2021) for icon detection | Single-agent | Vision-centric approach that eliminates dependency on system-specific data | https://github.com/X-PLUG/MobileAgent |
| Mobile-Agent-v2 Wang et al. (2024d) | Mobile Android OS and Harmony OS | Screenshots with text, icon recognition, and description | Standard UI operations on mobile phones | GPT-4V with Grounding DINO Liu et al. (2023a) and Qwen-VL-Int4 Bai et al. (2023a) | Multi-agent architecture with Planning Agent, Decision Agent, and Reflection Agent | Multi-agent architecture enhances task navigation for long-sequence operations | https://github.com/X-PLUG/MobileAgent |
| MobA Zhu et al. (2024b) | Mobile Android | GUI structures, screenshots with annotation | Standard UI operations and API function calls | GPT-4 | Two-level agent: a Global Agent and a Local Agent | Two-level agent system that separates task planning and execution into two specialized agents | https://github.com/OpenDFM/MobA |
| Mobile-Experts Zhang et al. (2024e) | Mobile Android | Interface memory and procedural memory | Standard UI operations and code-combined tool formulation | VLMs | Multi-agent framework with double-layer planning | Code-combined tool formulation method and double-layer planning mechanism for collaborative task execution | / |
| LiMAC Christianos et al. (2024) | Mobile Android | Screenshots and corresponding widget trees | Standard UI operations | Lightweight transformer and fine-tuned VLMs | Single-agent | Balances computational efficiency and natural language understanding | / |

Table 26: Overview of LLM-brained GUI agent frameworks on mobile platforms (Part V).

| Agent | Platform | Perception | Action | Model | Architecture | Highlight | Link |
|-------|----------|-----------|--------|-------|--------------|-----------|------|
| OS-Kairos Cheng et al. (2025a) | Mobile Android | GUI screenshots | Standard UI operations | OS-Atlas-Pro-7B and GPT-4o | Single-agent with critic-in-the-loop design | Introduces an adaptive interaction framework where each GUI action is paired with a confidence score, dynamically deciding between autonomous execution and human intervention | https://github.com/Wuzheng02/OS-Kairos |
| V-Droid Dai et al. (2025) | Mobile Android | Android Accessibility Tree | Standard UI operations | LLaMA-3.1-8B-Instruct Dubey et al. (2024) | Verifier-Driven Single-Agent Architecture | Introduces a novel verifier-driven architecture where the LLM does not generate actions directly but instead scores and selects from a finite set of extracted actions, improving task success rates and significantly reducing latency | / |
| ReachAgent Wu et al. (2025c) | Android mobile devices | GUI Screenshots, XML document | Standard UI operations | MobileVLM Wu et al. (2024c) | Single-agent, two-stage training | Divides tasks into subtasks: "Page Reaching" (navigating to the correct screen) and "Page Operation" (performing actions on the screen), using RL with preference-based training to improve long-term task success. | / |
| Mobile-Agent-V Wang et al. (2025b) | Mobile Android | Video guidance, XML hierarchy | Standard UI operations | GPT-4o | Multi-agent system | Introduces video-guided learning, allowing the agent to acquire operational knowledge efficiently. | https://github.com/X-PLUG/MobileAgent |
| MobileSteward Liu et al. (2025g) | Mobile Android | XML layouts, Screenshots | Standard UI interactions, Code execution | GPT-4V, GPT-4o | App-oriented multi-agent framework | Introduces an app-oriented multi-agent framework with self-evolution, overcoming the complexity of cross-app interactions by dynamically recruiting specialized agents. | https://github.com/XiaoMi/MobileSteward |
| AppAgentX Jiang et al. (2025) | Mobile Android | Screenshots | Standard UI operations | GPT-4o | Single-agent architecture | Introduces an evolutionary mechanism that enables dynamic learning from past interactions and replaces inefficient low-level operations with high-level actions. | https://appagentx.github.io/ |

Table 27: Overview of LLM-brained GUI agent frameworks on mobile platforms (Part VI).

| Agent | Platform | Perception | Action | Model | Architecture | Highlight | Link |
|---|---|---|---|---|---|---|---|
| CHOP Zhou et al. (2025) | Mobile Android | Screenshots | Standard UI operations | GPT-4o | Multi-agent architecture | Introduces a basis subtask framework, where subtasks are predefined based on human task decomposition patterns, ensuring better executability and efficiency. | https://github.com/Yuqi-Zhou/CHOP |
| LearnAct Liu et al. (2025a) | Mobile Android | GUI screenshots, UI trees, and demonstration trajectories | Standard GUI actions | Gemini-1.5-Pro, UI-TARS-7B-SFT, Qwen2-VL-7B | Multi-agent | Introduces a structured, demonstration-based learning pipeline for mobile GUI agents. It addresses long-tail generalization via few-shot demonstrations, achieving substantial performance gains on complex real-world mobile tasks | https://lgy0404.github.io/LearnAct |
| AndroidGen Lai et al. (2025) | Mobile Android | XML UI structure | Standard GUI actions | GLM-4-9B GLM et al. (2024) / LLaMA-3-70B Dubey et al. (2024) | Multi-module single-agent | Innovatively addresses data scarcity for Android agents through a self-improving architecture, a zero human-annotation training pipeline, and effective generalization from easy to hard tasks | https://github.com/THUDM/AndroidGen |
| Agent-Initiated Interaction Kahlon et al. (2025) | Android Mobile | Accessibility tree and screenshots | Standard GUI operations | Gemini 1.5 | Single-agent architecture | Pioneers agent-initiated interaction in mobile UI automation | https://github.com/google-research/google-research/tree/master/android_interaction |
| Latent State Estimation Bishop et al. (2024) | Android Mobile | Accessibility tree | Standard GUI operations | PaLM 2 | Two-module design with Reasoner and Grounder | First to formalize the estimation of latent UI states using LLMs to support UI automation | / |

Table 28: Overview of LLM-brained GUI agent frameworks on computer platforms (Part I).

| Agent | Platform | Perception | Action | Model | Architecture | Highlight | Link |
|---|---|---|---|---|---|---|---|
| UFO Zhang et al. (2024a) | Windows computer | Screenshots with annotated controls, and widget properties | Standard UI operations with additional customized operations | GPT-Vision | Dual-agent architecture, consisting of a HostAgent (for application selection and global planning) and an AppAgent (for specific task execution within applications) | Its dual-agent system that seamlessly navigates and interacts with multiple applications to fulfill complex user requests in natural language on Windows OS | https://github.com/microsoft/UFO |
| UFO$^2$ Zhang et al. (2025b) | Windows desktops | GUI screenshots and textual control properties list | Unified GUI–API action layer | GPT-4o (and GPT-4V, o1, Gemini-Flash); Vision grounding via OmniParser-v2 | Centralized HostAgent with application-specialized AppAgents | Transforms a conventional CUA into an OS-native, pluggable AgentOS with deep Windows integration, hybrid GUI–API actions, vision + UIA perception, speculative multi-action planning, retrieval-augmented knowledge, and a non-intrusive PiP virtual desktop | https://github.com/microsoft/UFO/ |
| ScreenAgent Niu et al. (2024) | Linux and Windows desktop | Screenshots | Standard UI operations | ScreenAgent model | Single-agent | Integrated planning-acting-reflecting pipeline that simulates a continuous thought process | https://github.com/niuzaisheng/ScreenAgent |
| OS-Copilot Wu et al. (2024e) | Linux and MacOS computer | Unified interface that includes mouse and keyboard control, API calls, and Bash or Python interpreters | Standard UI operations, Bash and Python commands, as well as API calls | GPT-4 | Multi-component architecture involving a planner, configurator, actor, and critic modules | Self-directed learning capability, allowing it to adapt to new applications by autonomously generating and refining tools | https://os-copilot.github.io/ |

Table 29: Overview of LLM-brained GUI agent frameworks on computer platforms (Part II).

| Agent | Platform | Perception | Action | Model | Architecture | Highlight | Link |
|---|---|---|---|---|---|---|---|
| Cradle Tan et al. (2024a) | Windows computer | Complete screen videos with Grounding DINO Liu et al. (2023a) and SAM Kirillov et al. (2023) for object detection and localization | Keyboard and mouse actions | GPT-4 | Modular single-agent architecture | Its generalizability across various digital environments, allowing it to operate without relying on internal APIs | `https://baai-agents.github.io/Cradle/` |
| Agent S Agashe et al. (2024) | Ubuntu and Windows computer | Screenshots and accessibility tree | Standard UI operations and system-level controls | GPT-4 and Claude-3.5 Sonnet Anthropic (2024) | Multi-agent architecture comprising a Manager and Worker structure | Experience-augmented hierarchical planning | `https://github.com/simular-ai/Agent-S` |
| GUI Narrator Wu et al. (2024b) | Windows computer | High-resolution screenshots | Standard UI operations | GPT-4 and QwenVL-7B Bai et al. (2023a) | Two-stage architecture, detecting the cursor location and selecting keyframes, then generating action captions | Uses the cursor as a focal point to improve understanding of high-resolution GUI actions | `https://showlab.github.io/GUI-Narrator` |
| PC Agent He et al. (2024d) | Windows Computer | Screenshots and event-based tracking | Standard UI Operations | Qwen2-VL-72B-Instruct Wang et al. (2024j) and Molmo Deitke et al. (2024) | A planning agent for decision-making combined with a grounding agent for executing actions. | Human cognition transfer framework, which transforms raw interaction data into cognitive trajectories to enable complex computer tasks. | `https://gair-nlp.github.io/PC-Agent/` |
| Zero-shot Agent Li et al. (2023d) | Computer | HTML code and DOM | Standard UI operations | PaLM-2 Anil et al. (2023) | Single-agent | Zero-shot capability in performing computer control tasks | `https://github.com/google-research/google-research/tree/master/zero_shot_structured_reflection` |
| TaskMind Yin et al. (2025) | Windows Computer | Standard GUI actions | GPT-3.5 / GPT-4 | Single-agent architecture | Introduces a novel task graph representation with cognitive dependencies, enabling LLMs to better generalize demonstrated GUI tasks | `https://github.com/Evennaire/TaskMind` | |

Saaket Agashe, Kyle Wong, Vincent Tu, Jiachen Yang, Ang Li, and Xin Eric Wang. Agent s2: A compositional generalist-specialist framework for computer use agents. *arXiv preprint arXiv:2504.00906*, 2025.

AgentSeaf AI. Introduction to agentsea platform, 2024. URL `https://www.agentsea.ai/`. Accessed: 2024-10-26.

Pranjal Aggarwal and Sean Welleck. Programming with pixels: Computer-use meets software engineering. *arXiv preprint arXiv:2502.18525*, 2025.

Mohamed Aghzal, Erion Plaku, Gregory J Stein, and Ziyu Yao. A survey on large language models for automated planning. *arXiv preprint arXiv:2502.12435*, 2025.

Table 30: Overview of LLM-brained GUI agent frameworks on computer platforms (Part III).

| Agent | Platform | Perception | Action | Model | Architecture | Highlight | Link |
|---|---|---|---|---|---|---|---|
| PC-Agent Liu et al. (2025c) | Windows computers | UI tree, Screenshots | Standard UI operations | GPT-4o | Hierarchical Multi-Agent | PC-Agent's hierarchical multi-agent design enables efficient decomposition of complex PC tasks. Its Active Perception Module enhances fine-grained GUI understanding by combining accessibility structures, OCR, and intention grounding. | `https://github.com/X-PLUG/MobileAgent/tree/main/PC-Agent` |
| PwP Aggarwal & Welleck (2025) | VSCode-based IDE in Computers | Screenshots, File system access, Terminal outputs | Standard UI interactions, File operations, Bash commands, Tools in VSCode | GPT-4o, Claude-3.5 Sonnet, Gemini-1.5 | Single-agent architecture | Shifts software engineering agents from API-based tool interactions to direct GUI-based computer use, allowing agents to interact with an IDE as a human developer would. | `https://programmingwithpixels.com` |
| COLA Zhao et al. (2025a) | Windows computers | GUI structure, properties and screenshots | Standard UI operations and system APIs | GPT-4o | Hierarchical Multi-Agent | A dynamic task scheduling mechanism with a plug-and-play agent pool, enabling adaptive handling of GUI tasks | `https://github.com/Alokia/COLA-demo` |
| STEVE Lu et al. (2025) | Windows Desktop | GUI screenshots and A11y Tree | Standard UI operations | Qwen2-VL Wang et al. (2024j) and GPT-4o | Single-agent | Introduces a scalable step verification pipeline using GPT-4o to generate binary labels for agent actions, and applies KTO optimization to incorporate both positive and negative actions into agent learning | `https://github.com/FanbinLu/STEVE` |

Simone Agostinelli, Andrea Marrella, and Massimo Mecella. Research challenges for intelligent robotic process automation. In *Business Process Management Workshops: BPM 2019 International Workshops, Vienna, Austria, September 1–6, 2019, Revised Selected Papers 17*, pp. 12–18. Springer, 2019.

Simone Agostinelli, Marco Lupia, Andrea Marrella, and Massimo Mecella. Automated generation of executable rpa scripts from user interface logs. In *Business Process Management: Blockchain and Robotic Process Automation Forum: BPM 2020 Blockchain and RPA Forum, Seville, Spain, September 13–18, 2020, Proceedings 18*, pp. 116–131. Springer, 2020.

Fellou AI. Eko - build production-ready agentic workflow with natural language. `https://eko.fellou.ai/`, 2025. Accessed: 2025-01-15.

Lukas Aichberger, Alasdair Paren, Yarin Gal, Philip Torr, and Adel Bibi. Attacking multimodal os agents with malicious image patches. In *ICLR 2025 Workshop on Foundation Models in the Wild*.

Wajdi Aljedaani, Abdulrahman Habib, Ahmed Aljohani, Marcelo Medeiros Eler, and Yunhe Feng. Does chatgpt generate accessible code? investigating accessibility challenges in llm-generated source code. In *International Cross-Disciplinary Conference on Web Accessibility*, 2024. URL `https://api.semanticscholar.org/CorpusID:273550267`.

Alfonso Amayuelas, Liangming Pan, Wenhu Chen, and William Wang. Knowledge of knowledge: Exploring known-unknowns uncertainty with large language models. *arXiv preprint arXiv:2305.13712*, 2023.

Jacob Andreas, John Bufe, David Burkett, Charles Chen, Josh Clausman, Jean Crawford, Kate Crim, Jordan DeLoach, Leah Dorner, Jason Eisner, et al. Task-oriented dialogue as dataflow synthesis. *Transactions of the Association for Computational Linguistics*, 8:556–571, 2020.

Table 31: Overview of LLM-brained cross-platform GUI agent frameworks (Part I).

| Agent | Platform | Perception | Action | Model | Architecture | Highlight | Link |
|---|---|---|---|---|---|---|---|
| AutoGLM Liu et al. (2024h) | Web and Mobile Android | Screenshots with SoM annotation and OCR | Standard UI operations, Native API interactions, and AI-driven actions | ChatGLM GLM et al. (2024) | Single-agent architecture | Self-evolving online curriculum RL framework, which enables continuous improvement by interacting with real-world environments | `https://xiao9905.github.io/AutoGLM/` |
| TinyClick Pawlowski et al. (2024) | Web, Mobile, and Windows platforms | GUI screenshots | Standard UI operations, Native API interactions, and AI-driven actions | Florence-2-Base VLM Xiao et al. (2023) | Single-agent, with single-turn tasks | Compact size (0.27B parameters) with high performance | `https://huggingface.co/Samsung/TinyClick` |
| OSCAR Wang & Liu (2024) | Desktop and Mobile | Screenshots | Standard UI operations | GPT-4 | Single-agent architecture | Ability to adapt to real-time feedback and dynamically adjust its actions | / |
| AgentStore Jia et al. (2024) | Desktop and mobile environments | GUI structures and properties, accessibility trees, screenshots and terminal output *etc* | Standard UI operations, API calls | GPT-4o and InternVL2-8B Chen et al. (2024l) | Multi-agent architecture | Dynamically integrate a wide variety of heterogeneous agents, enabling both specialized and generalist capabilities | `https://chengyou-jia.github.io/AgentStore-Home/` |
| MMAC-Copilot Song et al. (2024d) | Windows OS Desktop, mobile applications, and game environments | Screenshots | Standard UI operations, Native APIs, and Collaborative multi-agent actions | GPT-4V, SeeClick Cheng et al. (2024a) and Genimi Vision for different agents | Multi-agent architecture with Planner, Programmer, Viewer, Mentor, Video Analyst, and Librarian | Collaborative multi-agent architecture where agents specialize in specific tasks | / |
| AGUVIS Xu et al. (2024k) | Web, desktop, and mobile | Image-based observations | Standard UI operations | Fine-tuned Qwen2-VL Wang et al. (2024j) | Single-agent architecture | Pure vision-based approach for GUI interaction, bypassing textual UI representations and enabling robust cross-platform generalization | `https://aguvis-project.github.io` |
| Ponder & Press Wang et al. (2024o) | Web, Android, iOS Mobile, Windows, and macOS | Purely visual inputs | Standard UI operations | GPT-4o and Claude 3.5 Sonnet for high-level task decomposition, a fine-tuned Qwen2-VL-Instruct Wang et al. (2024j) for GUI element grounding | Divide-and-conquer architecture | Purely vision-based GUI agent that does not require non-visual inputs | `https://invinciblewyq.github.io/ponder-press-page/` |

Rohan Anil, Andrew M. Dai, Orhan Firat, Melvin Johnson, Dmitry Lepikhin, Alexandre Passos, Siamak Shakeri, Emanuel Taropa, Paige Bailey, Zhifeng Chen, Eric Chu, Jonathan H. Clark, Laurent El Shafey, Yanping Huang, Kathy Meier-Hellstern, Gaurav Mishra, Erica Moreira, Mark Omernick, Kevin Robinson, Sebastian Ruder, Yi Tay, Kefan Xiao, Yuanzhong Xu, Yujing Zhang, Gustavo Hernandez Abrego, Junwhan Ahn, Jacob Austin, Paul Barham, Jan Botha, James Bradbury, Siddhartha Brahma, Kevin Brooks, Michele Catasta, Yong Cheng, Colin Cherry, Christopher A. Choquette-Choo, Aakanksha Chowdhery, Clément Crepy, Shachi Dave, Mostafa Dehghani, Sunipa Dev, Jacob Devlin, Mark Díaz, Nan Du, Ethan Dyer, Vlad Feinberg, Fangxiaoyu Feng, Vlad Fienber, Markus Freitag, Xavier Garcia, Sebastian Gehrmann, Lucas Gonzalez, Guy Gur-Ari, Steven Hand, Hadi Hashemi, Le Hou, Joshua Howland, Andrea Hu, Jeffrey Hui, Jeremy Hurwitz, Michael Isard, Abe Ittycheriah, Matthew Jagielski, Wenhao Jia, Kathleen Kenealy, Maxim Krikun, Sneha Kudugunta, Chang Lan, Katherine Lee, Benjamin Lee, Eric Li, Music Li, Wei

Table 32: Overview of LLM-brained cross-platform GUI agent frameworks (Part II).

| Agent | Platform | Perception | Action | Model | Architecture | Highlight | Link |
|---|---|---|---|---|---|---|---|
| InfiGUIAgent Liu et al. (2025e) | Mobile, Web, Desktop | Raw screenshots | Standard UI operations. | Qwen2-VL-2B Wang et al. (2024j) | Single-agent architecture enhanced by hierarchical reasoning. | Introduces native reasoning skills, such as hierarchical and expectation-reflection reasoning, enabling advanced and adaptive task handling. | https://github.com/Reallm-Labs/InfiGUIAgent |
| Learn-by-Interact Su et al. (2025) | Web, code development, and desktops | GUI screenshots with SoM and accessibility tree | Standard UI interactions and code execution | Claude-3.5-Sonnet, Gemini-1.5-Pro Team et al. (2023), CodeGemma-7B, CodeStral-22B | Multi-agent | Introduces a fully autonomous data synthesis process, eliminating the need for human-labeled agentic data | / |
| CollabUIAgents He et al. (2025) | Mobile Android, Web | Screenshots, UI trees | Standard UI operations | Qwen2-7B Wang et al. (2024j), GPT-4 | Multi-agent system | A multi-agent reinforcement learning framework that introduces a Credit Re-Assignment (CR) strategy, using LLMs instead of environment-specific rewards to enhance performance and generalization. | https://github.com/THUNLP-MT/CollabUIAgents |
| Agent S2 Agashe et al. (2025) | Ubuntu, Windows, Android | GUI screenshot | Standard UI operations and system APIs | Claude-3.7-Sonnet, Claude-3.5-Sonnet, GPT-4o (for Manager and Worker roles), UI-TARS-72B-DPO, Tesseract OCR, and UNO (for grounding experts) | Compositional multi-agent architecture with a Manager for planning, a Worker for execution, and a Mixture of Grounding experts | Features a Mixture of Grounding technique and Proactive Hierarchical Planning, enabling more accurate grounding and adaptive replanning in long-horizon tasks | https://github.com/simular-ai/Agent-S |
| GuidNav Hu et al. (2025) | Android and Web | GUI screenshots | Standard UI operations and system APIs | GPT-4o, Gemini 2.0 Flash, Qwen-VL-Plus | Single-agent | Introduces a novel process reward model that provides fine-grained, step-level feedback to enhance GUI task accuracy and success | / |
| ScaleTrack Huang et al. (2025a) | Web, Android Mobile, and Desktop Computers | GUI screenshots | Standard GUI operations | Qwen2-VL-7B | Single-agent | First GUI agent framework to introduce backtracking—learning not only the next action but also historical action sequences | / |

Li, YaGuang Li, Jian Li, Hyeontaek Lim, Hanzhao Lin, Zhongtao Liu, Frederick Liu, Marcello Maggioni, Aroma Mahendru, Joshua Maynez, Vedant Misra, Maysam Moussalem, Zachary Nado, John Nham, Eric Ni, Andrew Nystrom, Alicia Parrish, Marie Pellat, Martin Polacek, Alex Polozov, Reiner Pope, Siyuan Qiao, Emily Reif, Bryan Richter, Parker Riley, Alex Castro Ros, Aurko Roy, Brennan Saeta, Rajkumar Samuel, Renee Shelby, Ambrose Slone, Daniel Smilkov, David R. So, Daniel Sohn, Simon Tokumine, Dasha Valter, Vijay Vasudevan, Kiran Vodrahalli, Xuezhi Wang, Pidong Wang, Zirui Wang, Tao Wang, John Wieting, Yuhuai Wu, Kelvin Xu, Yunhan Xu, Linting Xue, Pengcheng Yin, Jiahui Yu, Qiao Zhang, Steven Zheng, Ce Zheng, Weikang Zhou, Denny Zhou, Slav Petrov, and Yonghui Wu. Palm 2 technical report, 2023. URL https://arxiv.org/abs/2305.10403.

Table 33: Overview of datasets for optimizing LLMs tailored for web GUI agents (Part I).

| Dataset | Platform | Source | Content | Scale | Collection Method | Highlight | Link |
|---|---|---|---|---|---|---|---|
| Mind2Web Deng et al. (2023) | Web | Crowdsourced | Task descriptions, action sequences, webpage snapshots | 2,350 tasks from 137 websites | Human demonstrations | Develops generalist web agents with diverse user interactions on real-world websites | `https://osu-nlp-group.github.io/Mind2Web/` |
| Mind2Web-Live Pan et al. (2024b) | Web | Sampled and re-annotated from the Mind2Web Deng et al. (2023) | Textual task descriptions, intermediate evaluation states, action sequences, and metadata, GUI screenshots | 542 tasks, with 4,550 detailed annotation steps. | Annotated by human experts. | Emphasis on dynamic evaluation using "key nodes", which represent critical intermediate states in web tasks. | `https://huggingface.co/datasets/iMeanAI/Mind2Web-Live` |
| WebVLN Chen et al. (2024e) | Web | Human-designed, LLM-generated | Text instructions, plans, GUI screenshots, HTML content | 8,990 navigation paths, 14,825 QA pairs | WebVLN simulator, LLM-generated QA pairs | Vision-and-language navigation for human-like web browsing | `https://github.com/WebVLN/WebVLN` |
| WebLINX Lu et al. (2024c) | Web | From human experts | Conversational interactions, action sequences, DOM and screenshots | 2,337 demonstrations with over 100,000 interactions | Annotated by human experts | The first large-scale dataset designed to evaluate agents in real-world conversational web navigation | `https://mcgill-nlp.github.io/weblinx/` |
| AgentTrek Xu et al. (2024j) | Web | Web tutorials | Task metadata, step-by-step instructions, action sequences, visual observations, reproducible native traces | 4,902 trajectories | VLM agent guided by tutorials, with Playwright capturing the traces | Synthesizes high-quality trajectory data by leveraging web tutorials | / |

Anthropic. Introducing computer use, a new claude 3.5 sonnet, and claude 3.5 haiku, 2024. URL `https://www.anthropic.com/news/3-5-models-and-computer-use`. Accessed: 2024-10-26.

Usman Anwar, Abulhair Saparov, Javier Rando, Daniel Paleka, Miles Turpin, Peter Hase, Ekdeep Singh Lubana, Erik Jenner, Stephen Casper, Oliver Sourbut, et al. Foundational challenges in assuring alignment and safety of large language models. *arXiv preprint arXiv:2404.09932*, 2024.

appium. Appium: Cross-platform automation framework for all kinds of apps, 2024. URL `https://appium.io/docs/en/latest/`. Accessed: 2024-11-05.

Yauhen Leanidavich Arnatovich and Lipo Wang. A systematic literature review of automated techniques for functional gui testing of mobile applications. *arXiv preprint arXiv:1812.11470*, 2018.

Gilles Baechler, Srinivas Sunkara, Maria Wang, Fedir Zubach, Hassan Mansoor, Vincent Etter, Victor Cărbune, Jason Lin, Jindong Chen, and Abhanshu Sharma. Screenai: A vision-language model for ui and infographics understanding, 2024. URL `https://arxiv.org/abs/2402.04615`.

Chongyang Bai, Xiaoxue Zang, Ying Xu, Srinivas Sunkara, Abhinav Rastogi, Jindong Chen, et al. Uibert: Learning generic multimodal representations for ui understanding. *arXiv preprint arXiv:2107.13731*, 2021.

Hao Bai, Yifei Zhou, Mert Cemri, Jiayi Pan, Alane Suhr, Sergey Levine, and Aviral Kumar. Digirl: Training in-the-wild device-control agents with autonomous reinforcement learning, 2024. URL `https://arxiv.org/abs/2406.11896`.

Table 34: Overview of datasets for optimizing LLMs tailored for web GUI agents (Part II).

| Dataset | Platform | Source | Content | Scale | Collection Method | Highlight | Link |
|---|---|---|---|---|---|---|---|
| MultiUI Liu et al. (2024e) | Web | Combination of human-designed instructions and automated extraction from web structures | Textual task descriptions, plans, action sequences, GUI screenshots, accessibility trees, bounding box annotations | 7.3 million instruction samples from 1 million websites | LLMs and Playwright | Supports a broad range of UI-related tasks, including GUI understanding, action prediction, and element grounding. | `https://neulab.github.io/MultiUI/` |
| Explorer Pahuja et al. (2025) | Web | Popular URLs with systematic web exploration by LLMs | Textual task descriptions, Action sequences, GUI screenshots, Accessibility trees, HTML content | 94K successful web trajectories, 49K unique URLs, 720K screenshots | Generated by a multi-agent LLM pipeline | Largest-scale web trajectory dataset to date; dynamically explores web pages to create contextually relevant tasks | / |
| InSTA Trabucco et al. (2025) | Web | Automatically generated by LLMs across 1M websites from Common Crawl | Web navigation tasks in natural language, task plans and action sequences, HTML-based observations converted to markdown, and evaluations from LLM-based judges | 150,000 tasks across 150,000 websites | Generated by LLMs using the Playwright API and filtered by LLM-based judges | Presents a fully automated three-stage data generation pipeline—task generation, action execution, and evaluation—using only language models without any human annotations | `https://data-for-agents.github.io` |

Hao Bai, Yifei Zhou, Li Erran Li, Sergey Levine, and Aviral Kumar. Digi-q: Learning q-value functions for training device-control agents. *arXiv preprint arXiv:2502.15760*, 2025a.

Jinze Bai, Shuai Bai, Yunfei Chu, Zeyu Cui, Kai Dang, Xiaodong Deng, Yang Fan, Wenbin Ge, Yu Han, Fei Huang, Binyuan Hui, Luo Ji, Mei Li, Junyang Lin, Runji Lin, Dayiheng Liu, Gao Liu, Chengqiang Lu, Keming Lu, Jianxin Ma, Rui Men, Xingzhang Ren, Xuancheng Ren, Chuanqi Tan, Sinan Tan, Jianhong Tu, Peng Wang, Shijie Wang, Wei Wang, Shengguang Wu, Benfeng Xu, Jin Xu, An Yang, Hao Yang, Jian Yang, Shusheng Yang, Yang Yao, Bowen Yu, Hongyi Yuan, Zheng Yuan, Jianwei Zhang, Xingxuan Zhang, Yichang Zhang, Zhenru Zhang, Chang Zhou, Jingren Zhou, Xiaohuan Zhou, and Tianhang Zhu. Qwen technical report, 2023a. URL `https://arxiv.org/abs/2309.16609`.

Jinze Bai, Shuai Bai, Shusheng Yang, Shijie Wang, Sinan Tan, Peng Wang, Junyang Lin, Chang Zhou, and Jingren Zhou. Qwen-vl: A frontier large vision-language model with versatile abilities. *arXiv preprint arXiv:2308.12966*, 2023b.

Shuai Bai, Keqin Chen, Xuejing Liu, Jialin Wang, Wenbin Ge, Sibo Song, Kai Dang, Peng Wang, Shijie Wang, Jun Tang, et al. Qwen2. 5-vl technical report. *arXiv preprint arXiv:2502.13923*, 2025b.

Baidu Research. ERNIE Bot: Baidu's Knowledge-Enhanced Large Language Model Built on Full AI Stack Technology, 2024. URL `https://research.baidu.com/Blog/index-view?id=183`. [Online; accessed 9-November-2024].

Mohammad Bajammal, Andrea Stocco, Davood Mazinanian, and Ali Mesbah. A survey on the use of computer vision to improve software engineering tasks. *IEEE Transactions on Software Engineering*, 48(5): 1722–1742, 2020.

Table 35: Overview of datasets for optimizing LLMs tailored for mobile GUI agents (Part I).

| Dataset | Platform | Source | Content | Scale | Collection Method | Highlight | Link |
|---|---|---|---|---|---|---|---|
| VGA Meng et al. (2024) | Android Mobile | Rico Deka et al. (2017) | GUI screenshots, task descriptions, action sequences, bounds, layout, and functions of GUI elements | 63.8k instances, 22.3k instruction-following data pairs, 41.4k conversation data pairs | Generated by GPT-4 models | Prioritizes visual content to reduce inaccuracies | `https://github.com/Linziyang1999/Vision%2DGUI%2Dassistant` |
| Rico Deka et al. (2017) | Android Mobile | Gathered from real Android apps on Google Play Store | Textual data, screenshots, action sequences, UI structure, annotated UI representations | 72,219 unique UI screens, 10,811 user interaction traces | Crowdsourcing, automated exploration | Comprehensive dataset for mobile UI design, interaction modeling, layout generation | `http://www.interactionmining.org/` |
| PixelHelp Li et al. (2020a) | Android Mobile | Human, web "How-to", Rico UI corpus synthetic | Natural language instructions, action sequences, GUI screenshots, structured UI data | 187 multi-step instructions, 295,476 synthetic single-step commands | Human annotation and synthetic generation | Pioneering method for grounding natural language instructions to executable mobile UI actions | `https://github.com/google-research/google-research/tree/master/seq2act` |
| MoTIF Burns et al. (2022) | Android Mobile | Human-written | Natural language instructions, action sequences, GUI screenshots, structured UI data | 6,100 tasks across 125 Android apps | Human annotation | Task feasibility prediction for interactive GUI in mobile apps | `https://github.com/aburns4/MoTIF` |
| META-GUI Sun et al. (2022) | Android Mobile | SMCalFlow Andreas et al. (2020) | Dialogues, action sequences, screenshots, Android view hierarchies | 1,125 dialogues and 4,684 turns | Human annotation | Task-oriented dialogue system for mobile GUI without relying on back-end APIs | `https://x-lance.github.io/META-GU` |
| AITW Rawles et al. (2023) | Android Mobile | Human-generated instructions, LLM-generated prompts | Natural language instructions, UI screenshots, observation-action pairs | 715,142 episodes and 30,378 unique instructions | Human raters using Android emulators | Large-scale dataset for device control research with extensive app and UI diversity | `https://github.com/google-research/google-research/tree/master/android_in_the_wild` |

Shimshon Berkovits, Joshua D. Guttman, and Vipin Swarup. Authentication for mobile agents. In *Mobile Agents and Security*, 1998. URL `https://api.semanticscholar.org/CorpusID:13987376`.

Lucas Beyer, Andreas Steiner, André Susano Pinto, Alexander Kolesnikov, Xiao Wang, Daniel Salz, Maxim Neumann, Ibrahim Alabdulmohsin, Michael Tschannen, Emanuele Bugliarello, et al. Paligemma: A versatile 3b vlm for transfer. *arXiv preprint arXiv:2407.07726*, 2024.

Benyamin Beyzaei, Saghar Talebipour, Ghazal Rafiei, Nenad Medvidovic, and Sam Malek. Automated test transfer across android apps using large language models. *arXiv preprint arXiv:2411.17933*, 2024.

William E Bishop, Alice Li, Christopher Rawles, and Oriana Riva. Latent state estimation helps ui agents to reason. *arXiv preprint arXiv:2405.11120*, 2024.

Anjanava Biswas and Wrick Talukdar. Guardrails for trust, safety, and ethical development and deployment of large language models (llm). *Journal of Science & Technology*, 4(6):55–82, 2023.

Table 36: Overview of datasets for optimizing LLMs tailored for mobile GUI agents (Part II).

| Dataset | Platform | Source | Content | Scale | Collection Method | Highlight | Link |
|---|---|---|---|---|---|---|---|
| GUI Odyssey Lu et al. (2024b) | Android Mobile | Human designers, GPT-4 | Textual tasks, plans, action sequences, GUI screenshots | 7,735 episodes across 201 apps | Human demonstrations | Focuses on cross-app navigation tasks on mobile devices | `https://github.com/OpenGVLab/GUI-Odyssey` |
| Amex Chai et al. (2024) | Android Mobile | Human-designed, ChatGPT-generated | Text tasks, action sequences, high-res screenshots with multi-level annotations | 104,000 screenshots, 1.6 million interactive elements, 2,946 instructions | Human annotations, autonomous scripts | Multi-level, large-scale annotations supporting complex mobile GUI tasks | `https://yuxiangchai.github.io/AMEX/` |
| Ferret-UI You et al. (2025) | iOS, Android Mobile | Spotlight dataset, GPT-4 | Text tasks, action plans, GUI element annotations, bounding boxes | 40,000 elementary tasks, 10,000 advanced tasks | GPT-generated | Benchmark for UI-centric tasks with adjustable screen aspect ratios | `https://github.com/apple/ml-ferret` |
| AITZ Zhang et al. (2024g) | Android Mobile | AITW Rawles et al. (2023) | Screen-action pairs, action descriptions | 18,643 screen-action pairs across 70+ apps, 2,504 episodes | GPT-4V, icon detection models | Structured "Chain-of-Action-Thought" enhancing GUI navigation | `https://github.com/IMNearth/CoAT` |
| Octo-planner Chen et al. (2024f) | Android Mobile | GPT-4 generated | Text tasks, decomposed plans, action sequences | 1,000 data points | GPT-4 generated | Optimized for task planning with GUI actions | `https://huggingface.co/NexaAIDev/octopus-planning` |
| E-ANT Wang et al. (2024f) | Android tiny-apps | Human behaviors | Task descriptions, screenshots, action sequences, page element data | 40,000+ traces, 10,000 action intents | Human annotation | First large-scale Chinese dataset for GUI navigation with real human interactions | / |

Table 37: Overview of datasets for optimizing LLMs tailored for mobile GUI agents (Part III).

| Dataset | Platform | Source | Content | Scale | Collection Method | Highlight | Link |
|---|---|---|---|---|---|---|---|
| Mobile3M Wu et al. (2024c) | Android Mobile | Real-world interactions, simulations | UI screenshots, XML documents, action sequences | 3,098,786 UI pages, 20,138,332 actions | Simulation algorithm | Large-scale Chinese mobile GUI dataset with unique navigation graph | `https://github.com/Meituan-AutoML/MobileVLM` |
| AndroidLab Xu et al. (2024i) | Android Mobile | Human design, LLM self-exploration, academic datasets | Text instructions, action sequences, XML data, screenshots | 10.5k traces, 94.3k steps | Human annotation, LLM self-exploration | XML-based interaction data with unified action space | `https://github.com/THUDM/Android-Lab` |
| MobileViews Gao et al. (2024e) | Android Mobile | LLM-enhanced app traversal tool | Screenshot-view hierarchy pairs | 600,000 screenshots, VH pairs from 20,000+ apps | LLM-enhanced crawler | Largest open-source mobile screen dataset | `https://huggingface.co/datasets/mllmTeam/MobileViews` |
| FedMABench Wang et al. (2025d) | Android Mobile | AndroidControl Li et al. (2024e), AITW Rawles et al. (2023) | Textual task descriptions, action sequences, and GUI screenshots | 6 dataset series with over 30 subsets | Inferred from existing Android datasets | The first dataset designed to benchmark federated mobile GUI agents | `https://github.com/wwh0411/FedMABench` |
| GUI-Xplore Sun et al. (2025b) | Mobile Android | Combination of automated exploration and manual design | Exploration videos, textual tasks, QA pairs, view hierarchies, GUI screenshots, action sequences, and GUI transition graphs | 312 apps, 115 hours of video, 32,569 QA pairs, 41,293 actions, about 200 pages per app | Automated and human exploration | Introduces an exploration-based pretraining paradigm that provides rich app-specific priors through video data | `https://github.com/921112343/GUI-Xplore` |

Table 38: Overview of datasets for optimizing LLMs tailored for computer GUI agents.

| Dataset | Platform | Source | Content | Scale | Collection Method | Highlight | Link |
|---------|----------|--------|---------|-------|-------------------|-----------|------|
| ScreenAgent Niu et al. (2024) | Linux, Windows OS | Human-designed | GUI screenshots, action sequences | 273 task sessions, 3,005 training screenshots, 898 test screenshots | Human annotation | VLM-based agent across multiple desktop environments | `https://github.com/niuzaisheng/ScreenAgent` |
| LAM Wang et al. (2024h) | Windows OS | Application documentation, Wiki-How articles, Bing search queries | Task descriptions in natural language, step-by-step plans, action sequences, GUI screenshots | 76,672 task-plan pairs, 2,192 task-action trajectories | Instantiated using GPT-4, with actions tested and validated in the Windows environment using UFO Zhang et al. (2024a) | Provides a structured pipeline for collecting, validating, and augmenting data, enabling high-quality training for action-oriented AI models. | `https://github.com/microsoft/UFO/tree/main/dataflow` |
| DeskVision Xu et al. (2025) | Windows, macOS, and Linux desktops | Internet | GUI screenshots with annotated bounding boxes for UI elements and detailed region captions | 54,855 screenshots with 303,622 UI element annotations | UI elements detected using OmniParser and PaddleOCR | The first large-scale, open-source dataset focusing on real-world desktop GUI scenarios across operating systems | / |

Rogerio Bonatti, Dan Zhao, Francesco Bonacci, Dillon Dupont, Sara Abdali, Yinheng Li, Yadong Lu, Justin Wagle, Kazuhito Koishida, Arthur Bucker, Lawrence Jang, and Zack Hui. Windows agent arena: Evaluating multi-modal os agents at scale, 2024. URL `https://arxiv.org/abs/2409.08264`.

Mark A Boshart and Martha J Kosa. Growing a gui from an xml tree. *ACM SIGCSE Bulletin*, 35(3):223–223, 2003.

Jeffrey M Bradshaw, Paul J Feltovich, and Matthew Johnson. Human–agent interaction. In *The handbook of human-machine interaction*, pp. 283–300. CRC Press, 2017.

Paul Brie, Nicolas Burny, Arthur Sluÿters, and Jean Vanderdonckt. Evaluating a large language model on searching for gui layouts. *Proceedings of the ACM on Human-Computer Interaction*, 7(EICS):1–37, 2023.

Cameron B Browne, Edward Powley, Daniel Whitehouse, Simon M Lucas, Peter I Cowling, Philipp Rohlfshagen, Stephen Tavener, Diego Perez, Spyridon Samothrakis, and Simon Colton. A survey of monte carlo tree search methods. *IEEE Transactions on Computational Intelligence and AI in games*, 4(1):1–43, 2012.

Andreas Bruns, Andreas Kornstadt, and Dennis Wichmann. Web application tests with selenium. *IEEE software*, 26(5):88–91, 2009.

Andrea Burns, Deniz Arsan, Sanjna Agrawal, Ranjitha Kumar, Kate Saenko, and Bryan A. Plummer. A dataset for interactive vision-language navigation with unknown command feasibility, 2022. URL `https://arxiv.org/abs/2202.02312`.

Hongru Cai, Yongqi Li, Wenjie Wang, Fengbin Zhu, Xiaoyu Shen, Wenjie Li, and Tat-Seng Chua. Large language models empowered personalized web agents. *arXiv preprint arXiv:2410.17236*, 2024a.

Weilin Cai, Juyong Jiang, Fan Wang, Jing Tang, Sunghun Kim, and Jiayi Huang. A survey on mixture of experts. *arXiv preprint arXiv:2407.06204*, 2024b.

Erik Cambria, Lorenzo Malandri, Fabio Mercorio, Navid Nobani, and Andrea Seveso. XAI meets llms: A survey of the relation between explainable ai and large language models, 2024. URL `https://arxiv.org/abs/2407.15248`.

Table 39: Overview of datasets for optimizing LLMs tailored for cross-platform GUI agents (Part I).

| Dataset | Platform | Source | Content | Scale | Collection Method | Highlight | Link |
|---|---|---|---|---|---|---|---|
| Visual-AgentBench Liu et al. (2024i) | Android Mobile, Web | VAB-Mobile: Android Virtual Device, VAB-WebArena-Lite: WebArena Koh et al. (2024a) | Task instructions, action sequences, screen observations | VAB-Mobile: 1,213 trajectories, 10,175 steps; VAB-WebArena-Lite: 1,186 trajectories, 9,522 steps | Program-based solvers, agent bootstrapping, human demonstrations | Systematic evaluation of VLM as a visual foundation agent across multiple scenarios | `https://github.com/THUDM/VisualAgentBench` |
| GUICourse Chen et al. (2024i) | Android Mobile, Web | Web scraping, simulation, manual design | GUI screenshots, action sequences, OCR tasks, QA pairs | 10 million website page-annotation pairs, 67,000 action instructions | LLM-based auto-annotation, crowd-sourcing | Dataset suite for enhancing VLM GUI navigation on web and mobile platforms | `https://github.com/yiye3/GUICourse` |
| GUI-World Chen et al. (2024c) | OS, Mobile, Web, XR | Student workers, YouTube instructional videos | GUI videos, human-annotated keyframes, captions, QA data, action sequences | 12,000 videos, 83,176 frames | Human annotation | Designed for dynamic, sequential GUI tasks with video data | `https://gui-world.github.io/` |
| ScreenAI Baechler et al. (2024) | Android, iOS, Desktop/Web | Crawling apps and webpages, synthetic QA | Screen annotation, screen QA, navigation, summarization | Annotation: hundreds of millions; QA: tens of millions; Navigation: millions | Model, human annotation | Comprehensive pretraining and fine-tuning for GUI tasks across platforms | `https://github.com/google%2Dresearch%2Ddatasets/screen_annotation` |
| Web-Hybrid Gou et al. (2024) | Web, Android Mobile | Web-synthetic data | Screenshots, text-based referring expressions, coordinates on GUIs | 10 million GUI elements, 1.3 million screenshots | Rule-based synthesis, LLMs for referring expressions | Largest dataset for GUI visual grounding | `https://osu-nlp-group.github.io/UGround/` |
| GUIDE Chawla et al. (2024) | Computer and Web | Direct submissions from businesses and survey responses | Task descriptions, GUI screenshots, action sequences, CoT reasoning, spatial grounding | N/A | Collected through NEX-TAG, an automated annotation tool | Integrates images, action sequences, task descriptions, and spatial grounding into a unified dataset | `https://github.com/superagi/GUIDE` |

Ruisheng Cao, Fangyu Lei, Haoyuan Wu, Jixuan Chen, Yeqiao Fu, Hongcheng Gao, Xinzhuang Xiong, Hanchong Zhang, Yuchen Mao, Wenjing Hu, Tianbao Xie, Hongshen Xu, Danyang Zhang, Sida Wang, Ruoxi Sun, Pengcheng Yin, Caiming Xiong, Ansong Ni, Qian Liu, Victor Zhong, Lu Chen, Kai Yu, and Tao Yu. Spider2-v: How far are multimodal agents from automating data science and engineering workflows?, 2024. URL `https://arxiv.org/abs/2407.10956`.

William B Cavnar, John M Trenkle, et al. N-gram-based text categorization. In *Proceedings of SDAIR-94, 3rd annual symposium on document analysis and information retrieval*, volume 161175, pp. 14. Ann Arbor, Michigan, 1994.

Hyungjoo Chae, Namyoung Kim, Kai Tzu iunn Ong, Minju Gwak, Gwanwoo Song, Jihoon Kim, Sunghwan Kim, Dongha Lee, and Jinyoung Yeo. Web agents with world models: Learning and leveraging environment dynamics in web navigation, 2024. URL `https://arxiv.org/abs/2410.13232`.

Yuxiang Chai, Siyuan Huang, Yazhe Niu, Han Xiao, Liang Liu, Dingyu Zhang, Peng Gao, Shuai Ren, and Hongsheng Li. Amex: Android multi-annotation expo dataset for mobile gui agents, 2024. URL `https://arxiv.org/abs/2407.17490`.

Table 40: Overview of datasets for optimizing LLMs tailored for cross-platform GUI agents (Part II).

| Dataset | Platform | Source | Content | Scale | Collection Method | Highlight | Link |
|---------|----------|--------|---------|-------|-------------------|-----------|------|
| xLAM Zhang et al. (2024d) | Web and tools used | Synthesized data, and existing dataset | Textual tasks, action sequences, function-calling data | 60,000 data points | Collected using AI models with human verification steps | Provides a *unified format* across diverse environments, enhancing generalizability and error detection for GUI agents | https://github.com/SalesforceAIResearch/xLAM |
| Insight-UI Shen et al. (2024a) | iOS, Android, Windows, Linux, Web | Common Crawl corpus | Textual tasks, plans, action sequences, GUI screenshots | 434,000 episodes, 1,456,000 images | Automatic simulations performed by a browser API | Instruction-free paradigm and entirely auto-generated | / |
| OS-Genesis Sun et al. (2024b) | Web and Android | Reverse task synthesis, where the GUI environment is explored interactively without predefined tasks or human annotations. | High-level instructions, low-level instructions, action sequences, and environment states. | 1,000 synthesized trajectories. | Model-based interaction-driven approach with GPT-4o. | Reverses the conventional task-driven data collection process by enabling exploration-first trajectory synthesis. | https://qiushisun.github.io/OS-Genesis-Home/ |
| Navi-plus Cheng et al. (2025b) | Web and Android | AndroidControl Li et al. (2024e) and Mind2Web Deng et al. (2023) | Task descriptions, GUI action trajectories, low-level step instructions, screenshots, and follow-up ASK/SAY interaction pairs | / | LLM-automated with human validation | Introduces a Self-Correction GUI Navigation task featuring the novel ASK action for recovering missing information | / |
| Explorer Chaimalas et al. (2025) | Web and Android | Automated traversal of real websites and Android apps | UI screenshots, bounding boxes of interactable elements, screen similarity labels, and user actions | KhanAcademy (Web): 2,841 interactables, 378 screen similarity samples; Spotify (Android): 1,207 interactables, 451 screen similarity samples | Automated tools, HTML parsing, Accessibility Tree | Platform-independent, supports auto-labeling, and enables trace recording and voice-controlled GUI navigation | https://github.com/varnelis/Explorer |

Yuxiang Chai, Hanhao Li, Jiayu Zhang, Liang Liu, Guozhi Wang, Shuai Ren, Siyuan Huang, and Hongsheng Li. A3: Android agent arena for mobile gui agents, 2025. URL https://arxiv.org/abs/2501.01149.

Iason Chaimalas, Arnas VyĹĄniauskas, and Gabriel Brostow. Explorer: Robust collection of interactable gui elements. *arXiv preprint arXiv:2504.09352*, 2025.

Tathagata Chakraborti, Vatche Isahagian, Rania Khalaf, Yasaman Khazaeni, Vinod Muthusamy, Yara Rizk, and Merve Unuvar. From robotic process automation to intelligent process automation: –emerging trends–. In *Business Process Management: Blockchain and Robotic Process Automation Forum: BPM 2020 Blockchain and RPA Forum, Seville, Spain, September 13–18, 2020, Proceedings 18*, pp. 215–228. Springer, 2020.

Chi-Min Chan, Jianxuan Yu, Weize Chen, Chunyang Jiang, Xinyu Liu, Weijie Shi, Zhiyuan Liu, Wei Xue, and Yike Guo. Agentmonitor: A plug-and-play framework for predictive and secure multi-agent systems.

Table 41: Overview of foundation models for LLM-brained GUI agents (Part I).

| Model | Modality | Model Size | Architecture | Training Methods | Highlights | Open-Source | Link |
|---|---|---|---|---|---|---|---|
| GPT-4o Hurst et al. (2024) | Text, audio, image, and video | - | Multimodal autoregressive architecture | Pre-trained on a mix of public data, further trained for alignment with human preferences and safety considerations | Unified multimodal architecture that seamlessly processes and generates outputs across text, audio, image, and video, offering faster and more cost-effective operation than its predecessors | No | / |
| GPT-4V OpenAI (2023) | Text and image | - | - | Pre-trained on a large dataset of text and image data, followed by fine-tuning with reinforcement learning from human feedback (RLHF) | Notable for its multimodal capabilities, allowing it to analyze and understand images alongside text | No | / |
| Gemini Team et al. (2023) | Text, image, audio, and video | Nano versions: 1.8B/3.25B | Enhanced Transformer decoders | Large-scale pre-training on multimodal data, followed by supervised fine-tuning, reward modeling, and RLHF | Achieves state-of-the-art performance across multimodal tasks, including a groundbreaking 90% on the MMLU benchmark, and demonstrates capacity for on-device deployment with small model sizes | No | / |
| Claude 3.5 Sonnet (Computer Use) Anthropic (2024); Hu et al. (2024a) | Text and image | - | ReAct-based reasoning | - | Pioneering role in GUI automation as the first public beta model to utilize a vision-only paradigm for desktop task automation | No | / |
| Operator OpenAI (2025a;b) | Text and Image | - | Built on GPT-4o | Supervised learning and reinforcement learning | Trained to use a computer like a human, achieving remarkable performance on benchmarks | No | / |
| Qwen-VL Bai et al. (2023b) | Text and image | 9.6B | A Vision Transformer (ViT) Dosovitskiy et al. (2021) as the visual encoder, with a large language model based on the Qwen-7B architecture | Two stages of pre-training and a final stage of instruction fine-tuning | Achieves state-of-the-art performance on vision-language benchmarks and supports fine-grained visual understanding | Yes | httpss://github.com/QwenLM/Qwen-VL |
| Qwen2-VL Wang et al. (2024j) | Text, image, and video | 2B/7B/72B | ViT Dosovitskiy et al. (2021) as the vision encoder, paired with the Qwen2 series of language models | The ViT is trained with image-text pairs; all parameters are unfrozen for broader multimodal learning with various datasets; fine-tuning the LLM on instruction datasets | Introduces Naive Dynamic Resolution for variable resolution image processing and Multimodal Rotary Position Embedding for enhanced multimodal integration | Yes | httpss://github.com/QwenLM/Qwen2-VL |

*arXiv preprint arXiv:2408.14972*, 2024.

Tsung-Hsiang Chang, Tom Yeh, and Robert C Miller. Gui testing using computer vision. In *Proceedings of the SIGCHI Conference on Human Factors in Computing Systems*, pp. 1535–1544, 2010.

Yupeng Chang, Xu Wang, Jindong Wang, Yuan Wu, Linyi Yang, Kaijie Zhu, Hao Chen, Xiaoyuan Yi, Cunxiang Wang, Yidong Wang, et al. A survey on evaluation of large language models. *ACM Transactions on Intelligent Systems and Technology*, 15(3):1–45, 2024.

Rajat Chawla, Adarsh Jha, Muskaan Kumar, Mukunda NS, and Ishaan Bhola. Guide: Graphical user interface data for execution. *arXiv preprint arXiv:2404.16048*, 2024.

Chaoran Chen, Zhiping Zhang, Bingcan Guo, Shang Ma, Ibrahim Khalilov, Simret A Gebreegziabher, Yanfang Ye, Ziang Xiao, Yaxing Yao, Tianshi Li, et al. The obvious invisible threat: Llm-powered gui agents' vulnerability to fine-print injections. *arXiv preprint arXiv:2504.11281*, 2025a.

Table 42: Overview of foundation models for LLM-brained GUI agents (Part II).

| Model | Modality | Model Size | Architecture | Training Methods | Highlights | Open-Source | Link |
|---|---|---|---|---|---|---|---|
| InternVL-2 Chen et al. (2024l;k) | Text, image, video, and medical data | 1B/2B/4B/8B/26B/40B | ViT as the vision encoder and a LLM as the language component | Progressive alignment strategy, starting with coarse data and moving to fine data | Demonstrates powerful capabilities in handling complex multimodal tasks with various model sizes | Yes | httpss://internvl.github.io/blog/2024-07-02-InternVL-2.0/ |
| CogVLM Wang et al. (2024m) | Text and image | 17B | A ViT encoder, a two-layer MLP adapter, a pre-trained large language model, and a visual expert module | Stage 1 focuses on image captioning; Stage 2 combines image captioning and referring expression comprehension tasks | Achieves deep integration of visual and language features while preserving the full capabilities of large language models | Yes | httpss://github.com/THUDM/CogVLM |
| Ferret You et al. (2023) | Text and image | 7B/13B | Decoder-only architecture based on the Vicuna model, combined with a visual encoder | A combination of supervised training and additional instruction tuning | Ability to handle free-form region inputs via its hybrid region representation, enabling versatile spatial understanding and grounding | Yes | httpss://github.com/apple/ml-ferret |
| LLaVA Liu et al. (2024b) | Text and image | 7B/13B | A vision encoder (CLIP ViT-L/14), a language decoder (Vicuna) | Pre-training using filtered image-text pairs, fine-tuning with a multimodal instruction-following dataset | Its lightweight architecture enables quick experimentation, demonstrating capabilities close to GPT-4 in multimodal reasoning | Yes | httpss://llava-vl.github.io |
| LLaVA-1.5 Liu et al. (2024a) | Text and image | 7B/13B | A vision encoder (CLIP-ViT) and an encoder-decoder LLM architecture (*e.g.*, Vicuna or LLaMA) | Pre-training on vision-language alignment with image-text pairs; visual instruction tuning with specific task-oriented data | Notable for its data efficiency and scaling to high-resolution image inputs | Yes | httpss://llava-vl.github.io |
| BLIP-2 Li et al. (2023c) | Text and image | 3.4B/12.1B | A frozen image encoder, a lightweight Querying Transformer to bridge the modality gap, and a frozen large language model | Vision-language representation learning: trains the Q-Former with a frozen image encoder; Vision-to-language generative learning: connects the Q-Former to a frozen LLM to enable image-to-text generation | Achieves state-of-the-art performance on various vision-language tasks with a compute-efficient strategy by leveraging frozen pre-trained models | Yes | httpss://github.com/salesforce/LAVIS/tree/main/projects/blip2 |
| Phi-3.5-Vision Abdin et al. (2024) | Text and image | 4.2B | Image encoder: CLIP ViT-L/14 to process visual inputs, and transformer decoder based on the Phi-3.5-mini model for textual outputs | Pre-training on a combination of interleaved image-text datasets, synthetic OCR data, chart/table comprehension data, and text-only data; supervised fine-tuning using large-scale multimodal and text datasets; Direct Preference Optimization (DPO) to improve alignment, safety, and multimodal task performance | Excels in reasoning over visual and textual inputs, demonstrating competitive performance on single-image and multi-image tasks while being compact | Yes | httpss://github.com/microsoft/Phi-3CookBook/tree/main |

Chaoran Chen, Zhiping Zhang, Ibrahim Khalilov, Bingcan Guo, Simret A Gebreegziabher, Yanfang Ye, Ziang Xiao, Yaxing Yao, Tianshi Li, and Toby Jia-Jun Li. Toward a human-centered evaluation framework for trustworthy llm-powered gui agents. *arXiv preprint arXiv:2504.17934*, 2025b.

Daihang Chen, Yonghui Liu, Mingyi Zhou, Yanjie Zhao, Haoyu Wang, Shuai Wang, Xiao Chen, Tegawendé F Bissyandé, Jacques Klein, and Li Li. Llm for mobile: An initial roadmap. *arXiv preprint arXiv:2407.06573*, 2024a.

Daoyuan Chen, Yilun Huang, Zhijian Ma, Hesen Chen, Xuchen Pan, Ce Ge, Dawei Gao, Yuexiang Xie, Zhaoyang Liu, Jinyang Gao, et al. Data-juicer: A one-stop data processing system for large language models. In *Companion of the 2024 International Conference on Management of Data*, pp. 120–134, 2024b.

Dongping Chen, Yue Huang, Siyuan Wu, Jingyu Tang, Liuyi Chen, Yilin Bai, Zhigang He, Chenlong Wang, Huichi Zhou, Yiqiang Li, Tianshuo Zhou, Yue Yu, Chujie Gao, Qihui Zhang, Yi Gui, Zhen Li, Yao Wan,

Table 43: An overview of GUI-optimized models on web platforms.

| Model | Platform | Foundation Model | Size | Input | Output | Dataset | Highlights | Link |
|---|---|---|---|---|---|---|---|---|
| Agent Q Putta et al. (2024) | Web | LLaMA-3 70B Dubey et al. (2024) | 70B | HTML DOM representations | Plans, thoughts, actions, and action explanations | WebShop benchmark and OpenTable dataset | Combines Monte Carlo Tree Search (MCTS) with self-critique mechanisms, leveraging reinforcement learning to achieve exceptional performance | https://github.com/sentient%2Dengineering/agent-q |
| GLAINTEL Fereidouni & Siddique (2024) | Web | Flan-T5 Chung et al. (2024) | 780M | User instructions and observations of webpage state | GUI actions | 1.18M real-world products, 12,087 crowd-sourced natural language intents, 1,010 human demonstrations | Efficient use of smaller LLMs, and integration of RL and human demonstrations for robust performance | / |
| WebN-T5 Thil et al. (2024) | Web | T5 Raffel et al. (2020) | - | HTML and DOM with screenshots | Hierarchical navigation plans and GUI interactions | MiniWoB++, 13,000 human-made demonstrations | Combines supervised learning and reinforcement learning to address limitations of previous models in memorization and generalization | / |
| OpenWeb-Voyager He et al. (2024c) | Web | Idefics2-8b-instruct Laurençon et al. (2024a) | 8B | GUI screenshots, accessibility trees | Actions on GUI, planning and thought, answers to queries | Mind2Web and WebVoyager datasets, and generated queries for real-world web navigation | Combining imitation learning with a feedback loop for continuous improvement | https://github.com/MinorJerry/OpenWebVoyager |
| WebRL Qi et al. (2024) | Web | Llama-3.1 Dubey et al. (2024) and GLM-4 Du et al. (2021) | 8B/9B/70B | Task instructions, action history, HTML content | Actions, element identifiers, explanations or notes | WebArena-Lite | Introduces a self-evolving online curriculum reinforcement learning framework, which dynamically generates tasks based on past failures and adapts to the agent's skill level | https://github.com/THUDM/WebRL |
| WebGUM Furuta et al. (2023) | Web | Flan-T5 Chung et al. (2024) and Vision Transformer (ViT) Dosovitskiy et al. (2021) | 3B | HTML, screenshots, interaction history, instructions | Web navigation actions and free-form text | MiniWoB++ and WebShop benchmarks | Integrates temporal and local multimodal perception, combining HTML and visual tokens, and uses an instruction-finetuned language model for enhanced reasoning and task generalization | https://console.cloud.google.com/storage/browser/gresearch/webllm |

Pan Zhou, Jianfeng Gao, and Lichao Sun. Gui-world: A dataset for gui-oriented multimodal llm-based agents, 2024c. URL https://arxiv.org/abs/2406.10819.

Jieshan Chen, Mulong Xie, Zhenchang Xing, Chunyang Chen, Xiwei Xu, Liming Zhu, and Guoqiang Li. Object detection for graphical user interface: Old fashioned or deep learning or a combination? In *proceedings of the 28th ACM joint meeting on European Software Engineering Conference and Symposium on the Foundations of Software Engineering*, pp. 1202–1214, 2020.

Jingxuan Chen, Derek Yuen, Bin Xie, Yuhao Yang, Gongwei Chen, Zhihao Wu, Li Yixing, Xurui Zhou, Weiwen Liu, Shuai Wang, et al. Spa-bench: A comprehensive benchmark for smartphone agent evaluation. In *NeurIPS 2024 Workshop on Open-World Agents*, 2024d.

Mark Chen, Jerry Tworek, Heewoo Jun, Qiming Yuan, Henrique Ponde De Oliveira Pinto, Jared Kaplan, Harri Edwards, Yuri Burda, Nicholas Joseph, Greg Brockman, et al. Evaluating large language models trained on code. *arXiv preprint arXiv:2107.03374*, 2021a.

Qi Chen, Dileepa Pitawela, Chongyang Zhao, Gengze Zhou, Hsiang-Ting Chen, and Qi Wu. Webvln: Vision-and-language navigation on websites. *Proceedings of the AAAI Conference on Artificial Intelligence*, 38(2): 1165–1173, Mar. 2024e. doi: 10.1609/aaai.v38i2.27878. URL https://ojs.aaai.org/index.php/AAAI/article/view/27878.

Wei Chen and Zhiyuan Li. Octopus v2: On-device language model for super agent, 2024a. URL https://arxiv.org/abs/2404.01744.

Table 44: An overview of GUI-optimized models on mobile platforms (Part I).

| Model | Platform | Foundation Model | Size | Input | Output | Dataset | Highlights | Link |
|---|---|---|---|---|---|---|---|---|
| Mobile-VLM Wu et al. (2024c) | Mobile Android | Qwen-VL-Chat Bai et al. (2023b) | 9.8B | Screenshots and structured XML documents | Action predictions, navigation steps, and element locations | Mobile3M, includes 3 million UI pages, 20+ million actions, and XML data structured as directed graphs | Mobile-specific pretraining tasks that enhance intra- and inter-UI understanding, with a uniquely large and graph-structured Chinese UI dataset (Mobile3M) | `https://github.com/XiaoMi/mobilevlm` |
| Octo-planner Chen et al. (2024f) | Mobile devices | Phi-3 Mini Abdin et al. (2024) | 3.8B | User queries and available function descriptions | Execution steps | 1,000 data samples generated using GPT-4 | Optimized for resource-constrained devices to ensure low latency, privacy, and offline functionality | `https://huggingface.co/NexaAIDev/octopus-planning` |
| DigiRL Bai et al. (2024) | Mobile Android | AutoUI Zhang & Zhang (2024) | 1.3B | Screenshots | GUI actions | AiTW | Offline-to-online reinforcement learning, bridging gaps in static and dynamic environments | `https://github.com/DigiRL-agent/digirl` |
| LVG Qian et al. (2024) | Mobile Android | Swin Transformer Liu et al. (2021) and BERT Devlin (2018) | - | UI screenshots and free-form language expressions | Bounding box coordinates | UIBert dataset and synthetic dataset | Unifies detection and grounding tasks through layout-guided contrastive learning | / |
| Ferret-UI You et al. (2025) | Android and iPhone platforms | Ferret You et al. (2023) | 7B/13B | Raw screen pixels, sub-images divided for finer resolution, bounding boxes and regional annotations | Widget bounding boxes, text from OCR tasks, descriptions of UI elements or overall screen functionality, UI interaction actions | Generated from RICO (for Android) and AMP (for iPhone) | Multi-platform support with high-resolution adaptive image encoding | `https://github.com/apple/ml-ferret/tree/main/ferretui` |
| AppVLM Papoudakis et al. (2025) | Android mobile devices | Paligemma-3B-896 Beyer et al. (2024) | 3B | Annotated screenshots with bounding boxes and UI labels | GUI actions | AndroidControl Li et al. (2024e), AndroidWorld Rawles et al. (2024) | A lightweight model that achieves near-GPT-4o performance in Android control tasks while being 10× faster and more resource-efficient. | / |
| Digi-Q Bai et al. (2025a) | Mobile Android | LLaVA-1.5 Liu et al. (2024a) | 7B | GUI screenshots | GUI actions, Q-values | AitW Rawles et al. (2023) | Introduces a VLM-based Q-function for GUI agent training, enabling reinforcement learning without online interactions. | `https://github.com/DigiRL-agent/digiq` |
| VEM Zheng et al. (2025b) | Mobile Android | Qwen2VL Wang et al. (2024j) | 7B | GUI screenshots | GUI actions, Q-values | AitW Rawles et al. (2023) | Unlike traditional RL methods that require environment interactions, VEM enables training purely on offline data with a Value Environment Model. | `https://github.com/microsoft/GUI-Agent-RL` |

Wei Chen and Zhiyuan Li. Octopus v3: Technical report for on-device sub-billion multimodal ai agent, 2024b. URL `https://arxiv.org/abs/2404.11459`.

Wei Chen and Zhiyuan Li. Octopus v4: Graph of language models, 2024c. URL `https://arxiv.org/abs/2404.19296`.

Wei Chen, Zhiyuan Li, Zhen Guo, and Yikang Shen. Octo-planner: On-device language model for planner-action agents, 2024f. URL `https://arxiv.org/abs/2406.18082`.

Wei Chen, Zhiyuan Li, and Mingyuan Ma. Octopus: On-device language model for function calling of software apis, 2024g. URL `https://arxiv.org/abs/2404.01549`.

Weize Chen, Ziming You, Ran Li, Yitong Guan, Chen Qian, Chenyang Zhao, Cheng Yang, Ruobing Xie, Zhiyuan Liu, and Maosong Sun. Internet of agents: Weaving a web of heterogeneous agents for collaborative intelligence, 2024h. URL `https://arxiv.org/abs/2407.07061`.

Table 45: An overview of GUI-optimized models on mobile platforms (Part II).

| Model | Platform | Foundation Model | Size | Input | Output | Dataset | Highlights | Link |
|---|---|---|---|---|---|---|---|---|
| Octopus Chen et al. (2024g) | Mobile devices | CodeLlama-7B Rozière et al. (2024), Google Gemma 2B Team et al. (2024) | 7B, 2B | API documentation examples | Function names with arguments for API calls | RapidAPI Hub | Use of conditional masking to enforce correct output formatting | / |
| Octopus v2 Chen & Li (2024a) | Edge devices | Gemma-2B Team et al. (2024) | 2B | User queries and descriptions of available functions | Function calls with precise parameters | 20 Android APIs, with up to 1,000 data points generated for training | Functional tokenization strategy, which assigns unique tokens to function calls, significantly reducing the context length required for accurate prediction | / |
| Octopus v3 Chen & Li (2024b) | Edge devices | CLIP-based model and a causal language model | Less than 1 billion parameters | Queries and commands, images and functional tokens | Functional tokens for actions | Leveraged from Octopus v2 Chen & Li (2024a) | Introduction of functional tokens for multimodal applications enables the representation of any function as a token, enhancing the model's flexibility | / |
| Octopus v4 Chen & Li (2024c) | Serverless cloud-based platforms and edge devices | 17 models | Varies | User queries | Domain-specific answers, actions | Synthetic datasets similar to Octopus v2 | Graph-based framework integrating multiple specialized models for optimized performance | https://github.com/NexaAI/octopus-v4 |
| VGA Meng et al. (2024) | Mobile Android | LLaVA-v1.6-mistral-7B Liu et al. (2024b) | 7B | GUI screenshots with positional, visual, and hierarchical data | Actions and function calls, descriptions of GUI components, navigation and task planning | 63.8k-image dataset constructed from the RICO | Minimizes hallucinations in GUI comprehension by employing an image-centric fine-tuning approach, ensuring balanced attention between text and visual content | https://github.com/Linziyang1999/VGA%2Dvisual%2DGUI%2Dassistant |
| MobileFlow Nong et al. (2024) | Mobile phones | Qwen-VL-Chat Bai et al. (2023b) | 21B | GUI screenshots with OCR textual information and bounding boxes | GUI actions and question answering | 70k manually labeled business-specific data spanning 10 business sectors, and datasets like RefCOCO, ScreenQA, Flickr30K | Hybrid visual encoder capable of variable-resolution input and Mixture of Experts (MoE) Cai et al. (2024b) for enhanced performance and efficiency | / |

Wentong Chen, Junbo Cui, Jinyi Hu, Yujia Qin, Junjie Fang, Yue Zhao, Chongyi Wang, Jun Liu, Guirong Chen, Yupeng Huo, Yuan Yao, Yankai Lin, Zhiyuan Liu, and Maosong Sun. Guicourse: From general vision language models to versatile gui agents, 2024i. URL https://arxiv.org/abs/2406.11317.

Xi Chen, Xiao Wang, Lucas Beyer, Alexander Kolesnikov, Jialin Wu, Paul Voigtlaender, Basil Mustafa, Sebastian Goodman, Ibrahim Alabdulmohsin, Piotr Padlewski, et al. Pali-3 vision language models: Smaller, faster, stronger. *arXiv preprint arXiv:2310.09199*, 2023.

Yanan Chen, Ali Pesaranghader, Tanmana Sadhu, and Dong Hoon Yi. Can we rely on llm agents to draft long-horizon plans? let's take travelplanner as an example. *arXiv preprint arXiv:2408.06318*, 2024j.

Yanda Chen, Ruiqi Zhong, Sheng Zha, George Karypis, and He He. Meta-learning via language model in-context tuning. *arXiv preprint arXiv:2110.07814*, 2021b.

Yurun Chen, Xueyu Hu, Keting Yin, Juncheng Li, and Shengyu Zhang. Aeia-mn: Evaluating the robustness of multimodal llm-powered mobile agents against active environmental injection attacks. *arXiv preprint arXiv:2502.13053*, 2025c.

Zhe Chen, Weiyun Wang, Hao Tian, Shenglong Ye, Zhangwei Gao, Erfei Cui, Wenwen Tong, Kongzhi Hu, Jiapeng Luo, Zheng Ma, et al. How far are we to gpt-4v? closing the gap to commercial multimodal models with open-source suites. *arXiv preprint arXiv:2404.16821*, 2024k.

Table 46: An overview of GUI-optimized models on mobile platforms (Part III).

| Model | Platform | Foundation Model | Size | Input | Output | Dataset | Highlights | Link |
|-------|----------|------------------|------|-------|--------|---------|------------|------|
| UINav Li et al. (2024f) | Mobile Android | SmallBERT Turc et al. (2019) | Agent model: 320k, Referee model: 430k, Small-BERT model: 17.6MB | UI elements, utterance, screen representation | Predicted actions and element to act upon | 43 tasks across 128 Android apps and websites, collecting 3,661 demonstrations | Introduces a macro action framework and an error-driven demonstration collection process, significantly reducing training effort while enabling robust task performance with small, efficient models suitable for mobile devices | / |
| UI-R1 Lu et al. (2025b) | Mobile Android | Qwen2.5-VL-3B | 3B | GUI screenshots | Reasoning text and GUI actions | ScreenSpo and AndroidControl | Introduces a rule-based reinforcement learning approach using GRPO to enhance reasoning and action prediction in GUI tasks with only 136 examples | https://github.com/lll6gg/UI-R1 |
| ViMo Luo et al. (2025a) | Mobile Android | Pre-trained Stable Diffusion model Rombach et al. (2022) | / | Current GUI image, user action (in natural language), GUI text representation | GUI text representation of the next state and reconstructed full GUI image (visual prediction of the next screen) | Android Control and AITW | First GUI world model that predicts future visual GUI states | https://ai-agents-2030.github.io/ViMo/ |

Table 47: An overview of GUI-optimized models on computer platforms.

| Model | Platform | Foundation Model | Size | Input | Output | Dataset | Highlights | Link |
|-------|----------|------------------|------|-------|--------|---------|------------|------|
| Screen-Agent Niu et al. (2024) | Linux and Windows desktop | CogAgent Hong et al. (2023) | 18B | GUI screenshots | Mouse and keyboard actions | 273 task sessions | Comprehensive pipeline of planning, acting, and reflecting to handle real computer screen operations autonomously | https://github.com/niuzaisheng/ScreenAgent |
| Octopus Yang et al. (2025b) | Desktop | MPT-7B MosaicML (2023) and CLIP ViT-L/14 Radford et al. (2021) | 7B | Visual images, scene graphs containing objects and relations, environment messages | Executable action code and plans | OctoGibson: 476 tasks with structured initial and goal states; OctoMC: 40 tasks across biomes; OctoGTA: 25 crafted tasks spanning different game settings | Incorporates reinforcement learning with environmental feedback | https://choiszt.github.io/Octopus/ |
| LAM Wang et al. (2024h) | Windows OS | Mistral-7B Jiang et al. (2023) | 7B | Task requests in natural language, application environmental data | Plans, actions | 76,672 task-plan pairs, 2,192 task-action trajectories | The LAM model bridges the gap between planning and action execution in GUI environments. It introduces a multi-phase training pipeline combining task planning, imitation learning, self-boosting exploration, and reward-based optimization for robust action-oriented performance. | https://github.com/microsoft/UFO/tree/main/dataflow |

Zhe Chen, Jiannan Wu, Wenhai Wang, Weijie Su, Guo Chen, Sen Xing, Muyan Zhong, Qinglong Zhang, Xizhou Zhu, Lewei Lu, et al. Internvl: Scaling up vision foundation models and aligning for generic visual-linguistic tasks. In *Proceedings of the IEEE/CVF Conference on Computer Vision and Pattern Recognition*, pp. 24185–24198, 2024l.

Kanzhi Cheng, Qiushi Sun, Yougang Chu, Fangzhi Xu, Yantao Li, Jianbing Zhang, and Zhiyong Wu. Seeclick: Harnessing gui grounding for advanced visual gui agents, 2024a. URL https://arxiv.org/abs/2401.

Table 48: An overview of GUI-optimized models on cross-platform agents (Part I).

| Model | Platform | Foundation Model | Size | Input | Output | Dataset | Highlights | Link |
|---|---|---|---|---|---|---|---|---|
| RUIG Zhang et al. (2023e) | Mobile and desktop | Swin Transformer Liu et al. (2021) and BART Lewis (2019) | 4 decoder layers | UI screenshots and text instructions | Bounding box predictions in linguistic form | MoTIF dataset and RicoSCA dataset for mobile UI data and Common Crawl for desktop UI data | Innovatively uses policy gradients to improve the spatial decoding in the pixel-to-sequence paradigm | / |
| CogAgent Hong et al. (2023) | PC, web, and Android platforms | CogVLM-17B Wang et al. (2024m) | 18B | GUI screenshots combined with OCR-derived text | Task plans, action sequences, and textual descriptions | CCS400K, text recognition datasets: 80M synthetic text images, visual grounding datasets and GUI dataset Mind2Web and AiTW | High-resolution cross-module to balance computational efficiency and high-resolution input processing | https://github.com/THUDM/CogVLM |
| SeeClick Cheng et al. (2024a) | iOS, Android, macOS, Windows, and web | Qwen-VL Bai et al. (2023b) | 9.6B | GUI screenshots and textual instructions | GUI actions and element locations for interaction | 300k webpages with text and icons, RICO, and data from LLaVA | Ability to perform GUI tasks purely from screenshots and its novel GUI grounding pre-training approach | https://github.com/njuckevin/SeeClick |
| ScreenAI Baechler et al. (2024) | Mobile, desktop, and tablet UIs | PaLI-3 Chen et al. (2023) | 5B | GUI screenshots with OCR text, image captions, and other visual elements | Text-based answers for questions, screen annotations with bounding box coordinates and labels, navigation instructions, summaries of screen content | 262M mobile web screenshots and 54M mobile app screenshots | Unified representation of UIs and infographics, combining visual and textual elements | https://github.com/kyegomez/ScreenAI |
| Ferret-UI 2 Li et al. (2024j) | iPhone, Android, iPad, Web, AppleTV | Vicuna-13B Chiang et al. (2023), Gemma-2B Team et al. (2024), Llama3-8B Dubey et al. (2024) | 2B/8B/13B | UI screenshots, annotated bounding boxes and labels for UI widgets, OCR-detected text and bounding boxes for text elements, source HTML hierarchy trees for web data | Descriptions of UI elements, widget classification, OCR, tapability, and text/widget location, interaction instructions and multi-round interaction-based QA | Core-set, GroundUI-18k, GUIDE, Spotlight | Multi-platform support with high-resolution adaptive image encoding | / |

10935.

Pengzhou Cheng, Zheng Wu, Zongru Wu, Aston Zhang, Zhuosheng Zhang, and Gongshen Liu. Os-kairos: Adaptive interaction for mllm-powered gui agents. *arXiv preprint arXiv:2503.16465*, 2025a.

Yuheng Cheng, Ceyao Zhang, Zhengwen Zhang, Xiangrui Meng, Sirui Hong, Wenhao Li, Zihao Wang, Zekai Wang, Feng Yin, Junhua Zhao, et al. Exploring large language model based intelligent agents: Definitions, methods, and prospects. *arXiv preprint arXiv:2401.03428*, 2024b.

Ziming Cheng, Zhiyuan Huang, Junting Pan, Zhaohui Hou, and Mingjie Zhan. Navi-plus: Managing ambiguous gui navigation tasks with follow-up. *arXiv preprint arXiv:2503.24180*, 2025b.

Antoine Chevrot, Alexandre Vernotte, Jean-Rémy Falleri, Xavier Blanc, and Bruno Legeard. Are autonomous web agents good testers? *arXiv preprint arXiv:2504.01495*, 2025.

Table 49: An overview of GUI-optimized models on cross-platform agents (Part II).

| Model | Platform | Foundation Model | Size | Input | Output | Dataset | Highlights | Link |
|-------|----------|------------------|------|-------|--------|---------|------------|------|
| V-Zen Rahman et al. (2024) | Computers and Web | Vicuna-7B Chiang et al. (2023), DINO Liu et al. (2023a), EVA-2-CLIP Sun et al. (2023) | 7B | Text, GUI Images | Action Prediction, GUI Bounding Box | GUIDE Chawla et al. (2024) | Dual-resolution visual encoding for precise GUI grounding and task execution | https://github.com/abdur75648/V-Zen |
| ShowUI Lin et al. (2024c) | Websites, desktops, and mobile phones | Phi-3.5-Vision Abdin et al. (2024) | 4.2B | GUI screenshots with OCR for text-based UI elements and visual grounding for icons and widgets | GUI actions, navigation, element location | ScreenSpot, RICO, GUIEnv, GUIAct, AiTW, AiTZ, GUI-World | Interleaved Vision-Language-Action approach, allowing seamless navigation, grounding, and understanding of GUI environments | https://github.com/showlab/ShowUI |
| OS-ATLAS Wu et al. (2024f) | Windows, macOS, Linux, Android, and the web | InternVL-2 Chen et al. (2024l) and Qwen2-VL Bai et al. (2023b) | 4B/7B | GUI screenshots | GUI actions | AndroidControl, SeeClick, and others annotated with GPT-4, over 13 million GUI elements and 2.3 million screenshots | The first foundation action model designed for generalist GUI agents, supporting cross-platform GUI tasks, and introducing a unified action space | https://osatlas.github.io/ |
| xLAM Zhang et al. (2024d) | Diverse environments | Mistral-7B Jiang et al. (2023) and DeepSeek-Coder-7B Guo et al. (2024a) | Range from 1B to 8×22B | Unified function-calling data formats | Function calls, thought processes | Synthetic and augmented data, including over 60,000 high-quality samples generated using APIGen from 3,673 APIs across 21 categories | Excels in function-calling tasks by leveraging unified and scalable data pipelines | https://github.com/SalesforceAIResearch/xLAM |

De Chezelles, Thibault Le Sellier, Maxime Gasse, Alexandre Lacoste, Alexandre Drouin, Massimo Caccia, Léo Boisvert, Megh Thakkar, Tom Marty, Rim Assouel, et al. The browsergym ecosystem for web agent research. *arXiv preprint arXiv:2412.05467*, 2024.

Jeffrey Yang Fan Chiang, Seungjae Lee, Jia-Bin Huang, Furong Huang, and Yizheng Chen. Why are web ai agents more vulnerable than standalone llms? a security analysis. *arXiv preprint arXiv:2502.20383*, 2025.

Wei-Lin Chiang, Zhuohan Li, Zi Lin, Ying Sheng, Zhanghao Wu, Hao Zhang, Lianmin Zheng, Siyuan Zhuang, Yonghao Zhuang, Joseph E. Gonzalez, Ion Stoica, and Eric P. Xing. Vicuna: An open-source chatbot impressing gpt-4 with 90%* chatgpt quality, March 2023. URL https://lmsys.org/blog/2023-03-30-vicuna/.

Junhee Cho, Jihoon Kim, Daseul Bae, Jinho Choo, Youngjune Gwon, and Yeong-Dae Kwon. Caap: Context-aware action planning prompting to solve computer tasks with front-end ui only. *arXiv preprint arXiv:2406.06947*, 2024.

Filippos Christianos, Georgios Papoudakis, Thomas Coste, Jianye Hao, Jun Wang, and Kun Shao. Lightweight neural app control, 2024. URL https://arxiv.org/abs/2410.17883.

Hyung Won Chung, Le Hou, Shayne Longpre, Barret Zoph, Yi Tay, William Fedus, Yunxuan Li, Xuezhi Wang, Mostafa Dehghani, Siddhartha Brahma, et al. Scaling instruction-finetuned language models. *Journal of Machine Learning Research*, 25(70):1–53, 2024.

Junyoung Chung, Caglar Gulcehre, KyungHyun Cho, and Yoshua Bengio. Empirical evaluation of gated recurrent neural networks on sequence modeling. *arXiv preprint arXiv:1412.3555*, 2014.

CogAgent Team. Cogagent: Cognitive ai agent platform. https://cogagent.aminer.cn/home, 2024. Accessed: 2024-12-17.

Wikipedia contributors. Large language model — wikipedia, the free encyclopedia, 2024. URL https://en.wikipedia.org/wiki/Large_language_model. Accessed: 2024-11-25.

Table 50: An overview of GUI-optimized models on cross-platform agents (Part III).

| Model | Platform | Foundation Model | Size | Input | Output | Dataset | Highlights | Link |
|---|---|---|---|---|---|---|---|---|
| Falcon-UI Shen et al. (2024a) | iOS, Android, Windows, Linux, Web | Qwen2-VL-7B | 7B | Screenshots of GUI with node information and OCR annotations for visible elements | GUI actions and coordinates or bounding boxes for interaction elements | Insight-UI dataset, further fine-tuned on datasets such as AITW, AITZ, Android Control, and Mind2Web | Decouples GUI context comprehension from instruction-following tasks, leveraging an instruction-free pretraining approach. | / |
| UI-TARS Qin et al. (2025) | Web, Desktop (Windows, macOS), and Mobile (Android) | Qwen-2-VL 7B and 72B Wang et al. (2024j) | 7B / 72B | GUI screenshots | GUI actions | GUI screenshots and metadata collected from websites, apps, and operating systems; action trace datasets from various GUI agent benchmarks; 6M GUI tutorials for reasoning enhancement; multiple open-source datasets | Pure vision-based perception with standardized GUI actions across platforms (Web, Mobile, Desktop). | https://github.com/bytedance/UI-TARS |
| Magma Yang et al. (2025a) | Web, Mobile, Robotics | LLaMA-3-8B Dubey et al. (2024), ConvNeXt-Xxlarge Liu et al. (2022) | 8.6B | GUI screenshots, textual task descriptions | GUI actions, robotic manipulation | UI, robotics data, human instructional videos | Jointly trains on heterogeneous datasets, enabling generalization across digital and physical tasks | https://microsoft.github.io/Magma/ |
| SpiritSight Huang et al. (2025c) | Web, Android, Windows Desktop | InternVL Chen et al. (2024l) | 2B, 8B, and 26B | GUI screenshots | GUI actions | AitW Rawles et al. (2023), CommonCrawl websites, and custom annotations | Introduces a Universal Block Parsing (UBP) method to resolve positional ambiguity in high-resolution visual inputs. | https://hzhiyuan.github.io/SpiritSight-Agent |
| GUI-R1 Xia & Luo (2025) | Windows, Linux, MacOS, Android, and Web | QwenVL2.5 Bai et al. (2025b) | 3B and 7B | GUI screenshots | Reasoning text and GUI actions | Mixture of 3K high-quality samples | First framework to apply rule-based reinforcement learning (RFT) to high-level GUI tasks across platforms. | https://github.com/ritzz-ai/GUI-R1.git |
| InfiGUI-R1 Liu et al. (2025f) | Web, Desktop, and Android | Qwen2.5-VL-3B-Instruct | 3B | GUI screenshots, Accessibility Tree | Reasoning text and GUI actions | Diverse dataset mixture | Two-stage training framework Actor2Reasoner: (1) Reasoning Injection via Spatial Reasoning Distillation, and (2) Deliberation Enhancement via Reinforcement Learning with Sub-goal Guidance and Error Recovery Scenario Construction | https://github.com/Reallm-Labs/InfiGUI-R1 |
| Task Generalization Zhang et al. (2025d) | Web and Android (Mobile) | Qwen2-VL-7B-Instruct Wang et al. (2024j) | 7B | GUI screenshots | Thoughts and grounded coordinate-based actions | 11 domain datasets with 56K GUI trajectory samples | Introduces mid-training on diverse non-GUI reasoning tasks (particularly math and code) to substantially enhance GUI agent planning capabilities | https://github.com/hkust-nlp/GUIMid |

Chenhui Cui, Tao Li, Junjie Wang, Chunyang Chen, Dave Towey, and Rubing Huang. Large language models for mobile gui text input generation: An empirical study. *arXiv preprint arXiv:2404.08948*, 2024.

Gautier Dagan, Frank Keller, and Alex Lascarides. Dynamic planning with a llm. *arXiv preprint arXiv:2308.06391*, 2023.

Gaole Dai, Shiqi Jiang, Ting Cao, Yuanchun Li, Yuqing Yang, Rui Tan, Mo Li, and Lili Qiu. Advancing mobile gui agents: A verifier-driven approach to practical deployment. *arXiv preprint arXiv:2503.15937*, 2025.

Preetam Prabhu Srikar Dammu. Towards ethical and personalized web navigation agents: A framework for user-aligned task execution. In *Proceedings of the Eighteenth ACM International Conference on Web Search and Data Mining*, pp. 1074–1076, 2025.

Table 51: Overview of web GUI agent benchmarks (Part I).

| Benchmark | Year | Live | Highlight | Data Size | Metric | Measurement | Link |
|---|---|---|---|---|---|---|---|
| MiniWoB++ Shi et al. (2017); Liu et al. (2018) | 2017 | Yes | Evaluates agents on basic web interactions like clicking, typing, and form navigation. | 100 web interaction tasks | Task Success Rate | Element Match | `https://github.com/Farama%2DFoundation/miniwob%2Dplusplus` |
| RUSS Xu et al. (2021) | 2021 | No | Uses ThingTalk for mapping natural language to web actions, enabling precise web-based task execution in real HTML environments. | 741 instructions | Task Success Rate | Text Match, Element Match | `https://github.com/xnancy/russ` |
| WebShop Yao et al. (2023) | 2022 | Yes | Simulates e-commerce navigation with real-world products, challenging agents with instruction comprehension, multi-page navigation, and strategic exploration. | 12,087 instructions | Task Success Rate, Step Success Rate | Text Match | `https://webshop-pnlp.github.io/` |
| Mind2Web Deng et al. (2023) | 2023 | No | Tests adaptability on real-world, dynamic websites across domains. | 2,000 tasks | Step Success Rate, Task Success Rate | Element Match, Action Match | `https://github.com/OSU-NLP-Group/Mind2Web` |
| Mind2Web-Live Pan et al. (2024b) | 2024 | Yes | Provides intermediate action tracking for realistic task assessment, along with an updated Mind2Web-Live dataset and tools for annotation. | 542 tasks | Step Success Rate, Task Success Rate, Efficiency Score | Element Match, Text Match, trajectory length | `https://huggingface.co/datasets/iMeanAI/Mind2Web-Live` |
| Mind2Web-Live-Abstracted Shahbandeh et al. (2024) | 2024 | Yes | Abstract the descriptions by omitting task-specific details and user input information in Mind2Web-Live, which are more streamlined and less time-consuming to compose. | 104 samples | Task Success Rate, Efficiency Score | Text Match, Image Match, Element Match, Path Length | `https://anonymous.4open.science/r/naviqate` |
| WebArena Zhou et al. | 2023 | Yes | Simulates realistic, multi-tab browsing on Docker-hosted websites, focusing on complex, long-horizon tasks that mirror real online interactions. | 812 long-horizon tasks | Step Success Rate | Text Match | `https://webarena.dev/` |

Leo de Castro, Antigoni Polychroniadou, and Daniel Escudero. Privacy-preserving large language model inference via gpu-accelerated fully homomorphic encryption. In *Neurips Safe Generative AI Workshop 2024*.

Matt Deitke, Christopher Clark, Sangho Lee, Rohun Tripathi, Yue Yang, Jae Sung Park, Mohammadreza Salehi, Niklas Muennighoff, Kyle Lo, Luca Soldaini, Jiasen Lu, Taira Anderson, Erin Bransom, Kiana Ehsani, Huong Ngo, YenSung Chen, Ajay Patel, Mark Yatskar, Chris Callison-Burch, Andrew Head,

Table 52: Overview of web GUI agent benchmarks (Part II).

| Benchmark | Year | Live | Highlight | Data Size | Metric | Measurement | Link |
|---|---|---|---|---|---|---|---|
| VisualWebArena Koh et al. (2024a) | 2024 | Yes | Assesses multimodal agents on visually grounded tasks, requiring both visual and textual interaction capabilities in web environments. | 910 tasks | Step Success Rate | Text Match, Image Match | `https://jykoh.com/vwa` |
| MT-Mind2Web Deng et al. (2024c) | 2024 | No | Introduces conversational web navigation with multi-turn interactions, supported by a specialized multi-turn web dataset. | 720 sessions/3525 instructions | Step Success Rate, Turn Success Rate | Element Match, Action Match | `https://github.com/magicgh/self-map` |
| MMInA Zhang et al. (2024s) | 2024 | Yes | Tests multihop, multimodal tasks on real-world websites, requiring agents to handle cross-page information extraction and reasoning for complex tasks. | 1,050 tasks | Step Success Rate, Task Success Rate | Text Match, Element Match | `https://mmina.cliangyu.com/` |
| AutoWebBench Lai et al. (2024) | 2024 | No | Bilingual web browsing benchmark with 10,000 browsing traces, supporting evaluation across language-specific environments. | 10,000 traces | Step Success Rate, Efficiency Score | Element Match, Action Match, Time | `https://github.com/THUDM/AutoWebGLM` |
| WorkArena Drouin et al. (2024) | 2024 | Yes | Focuses on real-world enterprise software interactions, targeting tasks frequently performed by knowledge workers | 19,912 unique task instances | Task Success Rate, Efficiency Score, Completion under Policy, Turn Success Rate | Element Match, Text Match, Execution-based Validation | `https://github.com/ServiceNow/WorkArena` |
| VideoWebArena Jang et al. (2024) | 2024 | Yes | Focuses on long-context multimodal agents using video tutorials for task completion | 74 videos (approx. 4 hours), 2,021 tasks | Task Success Rate, Intermediate Intent Success Rate, Efficiency Scores | Element Match, State Information, Exact and Fuzzy Text Matches | `https://github.com/ljang0/videowebarena` |
| EnvDistraction Ma et al. (2024c) | 2024 | No | Evaluates the "faithfulness" of multimodal GUI agents by assessing their susceptibility to environmental distractions like pop-ups, fake results, or misleading elements | 1,198 tasks | Task Success Rate | Text Match, Element Match, State Information | `https://github.com/xbmxb/EnvDistraction` |

Rose Hendrix, Favyen Bastani, Eli VanderBilt, Nathan Lambert, Yvonne Chou, Arnavi Chheda, Jenna Sparks, Sam Skjonsberg, Michael Schmitz, Aaron Sarnat, Byron Bischoff, Pete Walsh, Chris Newell, Piper Wolters, Tanmay Gupta, Kuo-Hao Zeng, Jon Borchardt, Dirk Groeneveld, Crystal Nam, Sophie Lebrecht, Caitlin Wittlif, Carissa Schoenick, Oscar Michel, Ranjay Krishna, Luca Weihs, Noah A. Smith, Hannaneh Hajishirzi, Ross Girshick, Ali Farhadi, and Aniruddha Kembhavi. Molmo and pixmo: Open weights and open data for state-of-the-art vision-language models, 2024. URL `https://arxiv.org/abs/2409.17146`.

Table 53: Overview of web GUI agent benchmarks (Part III).

| Benchmark | Year | Live | Highlight | Data Size | Metric | Measurement | Link |
|---|---|---|---|---|---|---|---|
| WebVLN-v1 Chen et al. (2024e) | 2024 | No | Combines navigation and question-answering on shopping sites, integrating visual and textual content for unified web interaction evaluation. | 8,990 paths and 14,825 QA pairs | Task Success Rate, Efficiency Score | Element Match, Path Length, Trajectory Length | `https://github.com/WebVLN/WebVLN` |
| WEBLINX Lu et al. (2024c) | 2024 | No | Focuses on conversational navigation, requiring agents to follow multi-turn user instructions in realistic, dialogue-based web tasks. | 100k interactions | Turn Success Rate | Element Match, Text Match, Action Match | `https://mcgill-nlp.github.io/weblinx/` |
| ST-WebAgentBench Levy et al. (2024) | 2024 | Yes | Evaluates policy-driven safety in web agents, using the Completion under Policy metric to ensure compliance in enterprise-like environments. | 235 tasks | Task Success Rate, Completion under Policy (CuP), Risk Ratio | Element Match, Action Match, Text Match | `https://sites.google.com/view/st-webagentbench/home` |
| CompWoB Furuta et al. (2024) | 2023 | No | Tests agents on sequential, compositional tasks that require state management across multiple steps, simulating real-world automation scenarios. | 50 compositional tasks | Task Success Rate | Element Match | `https://github.com/google-research/google-research/tree/master/compositional_rl/compwob` |
| TURKING BENCH Xu et al. (2024e) | 2024 | Yes | Uses natural HTML tasks from crowdsourcing to assess interaction skills with real-world web layouts and elements. | 32.2K instances | Task Success Rate | Text Match, Element Match, Image Match | `https://turkingbench.github.io` |

Biplab Deka, Zifeng Huang, Chad Franzen, Joshua Hibschman, Daniel Afergan, Yang Li, Jeffrey Nichols, and Ranjitha Kumar. Rico: A mobile app dataset for building data-driven design applications. In *Proceedings of the 30th Annual ACM Symposium on User Interface Software and Technology*, UIST '17, pp. 845–854, New York, NY, USA, 2017. Association for Computing Machinery. ISBN 9781450349819. doi: 10.1145/3126594.3126651. URL `https://doi.org/10.1145/3126594.3126651`.

Biniam Fisseha Demissie, Yan Naing Tun, Lwin Khin Shar, and Mariano Ceccato. Vlm-fuzz: Vision language model assisted recursive depth-first search exploration for effective ui testing of android apps. *arXiv preprint arXiv:2504.11675*, 2025.

Shihan Deng, Weikai Xu, Hongda Sun, Wei Liu, Tao Tan, Jianfeng Liu, Ang Li, Jian Luan, Bin Wang, Rui Yan, et al. Mobile-bench: An evaluation benchmark for llm-based mobile agents. *arXiv preprint arXiv:2407.00993*, 2024a.

Xiang Deng, Yu Gu, Boyuan Zheng, Shijie Chen, Sam Stevens, Boshi Wang, Huan Sun, and Yu Su. Mind2web: Towards a generalist agent for the web. *Advances in Neural Information Processing Systems*, 36:28091–28114, 2023.

Yang Deng, Xuan Zhang, Wenxuan Zhang, Yifei Yuan, See-Kiong Ng, and Tat-Seng Chua. On the multi-turn instruction following for conversational web agents. *arXiv preprint arXiv:2402.15057*, 2024b.

Table 54: Overview of web GUI agent benchmarks (Part IV).

| Benchmark | Year | Live | Highlight | Data Size | Metric | Measurement | Link |
|---|---|---|---|---|---|---|---|
| VisualWebBench Liu et al. (2024f) | 2024 | No | Provides a fine-grained assessment of multimodal large language models (MLLMs) on web-specific tasks | 1,534 instances from 139 real websites across 87 sub-domains | Task Success Rate, Turn Success Rate, Efficiency Metrics | Text Match, Image Match, Element Match, Action Match | `https://visualwebbench.github.io/` |
| WONDER-BREAD Wornow et al. | 2024 | No | Focuses on business process management (BPM) tasks like documentation, knowledge transfer, and process improvement | 2,928 human demonstrations across 598 distinct workflows | Task Success Rate, Step Success Rate, Efficiency Score, Completion under Policy | Text Match, Action Match, State Information | `https://github.com/HazyResearch/wonderbread` |
| WebOlympus Zheng et al. (2024b) | 2024 | Yes | An open platform for web agents that simplifies running demos, evaluations, and data collection for web agents on live websites | 50 tasks | Task Success Rate, Step Success Rate | Action Match | / |
| BrowserGym Chezelles et al. (2024) | 2024 | Yes | Provides a unified, extensible, and open-source environment for evaluating web agents with standardized APIs and observations | Benchmarks include Mini-WoB(++) with 125 tasks, WebArena with 812 tasks, and WorkArena with up to 341 tasks per level | Task Success Rate, Step Success Rate, Turn Success Rate, Efficiency Metrics | Text-based matching and element match | `https://github.com/ServiceNow/BrowserGym` |
| WebWalkerQA Wu et al. (2025a) | 2025 | Yes | Benchmarks the capacity of LLMs to handle deep, structured, and realistic web-based navigation and reasoning tasks | 680 high-quality QA pairs | Task Success Rate, Efficiency Score | Text Match, Action Match | `https://github.com/Alibaba-NLP/WebWalker` |

Yang Deng, Xuan Zhang, Wenxuan Zhang, Yifei Yuan, See-Kiong Ng, and Tat-Seng Chua. On the multi-turn instruction following for conversational web agents, 2024c. URL `https://arxiv.org/abs/2402.15057`.

Parth S Deshmukh, Saroj S Date, Parikshit N Mahalle, and Janki Barot. Automated gui testing for enhancing user experience (ux): A survey of the state of the art. In *International Conference on ICT for Sustainable Development*, pp. 619–628. Springer, 2023.

Jacob Devlin. Bert: Pre-training of deep bidirectional transformers for language understanding. *arXiv preprint arXiv:1810.04805*, 2018.

Bosheng Ding, Chengwei Qin, Ruochen Zhao, Tianze Luo, Xinze Li, Guizhen Chen, Wenhan Xia, Junjie Hu, Luu Anh Tuan, and Shafiq Joty. Data augmentation using llms: Data perspectives, learning paradigms and challenges. In *Findings of the Association for Computational Linguistics ACL 2024*, pp. 1679–1705, 2024a.

Lei Ding, Jeshwanth Bheemanpally, and Yi Zhang. Improving technical" how-to" query accuracy with automated search results verification and reranking. *arXiv preprint arXiv:2404.08860*, 2024b.

Ruomeng Ding, Chaoyun Zhang, Lu Wang, Yong Xu, Minghua Ma, Wei Zhang, Si Qin, Saravan Rajmohan, Qingwei Lin, and Dongmei Zhang. Everything of thoughts: Defying the law of penrose triangle for thought generation. *arXiv preprint arXiv:2311.04254*, 2023.

Table 55: Overview of web GUI agent benchmarks (Part V).

| Benchmark | Year | Live | Highlight | Data Size | Metric | Measurement | Link |
|---|---|---|---|---|---|---|---|
| WebGames Thomas et al. (2025) | 2025 | Yes | A comprehensive benchmark designed to evaluate the capabilities of general-purpose web-browsing AI agents through 50+ interactive challenges. It uniquely provides a hermetic testing environment with verifiable ground-truth solutions. | 50+ challenges | Task Success Rate | Action Match | `https://github.com/convergence-ai/webgames` |
| SafeArena Tur et al. (2025) | 2025 | Yes | The first benchmark specifically designed to evaluate the deliberate misuse of web agents by testing their ability to complete both safe and harmful tasks. | 500 tasks | Task Success Rate, Completion under Policy, Risk Ratio | Text Match, State Information | `https://safearena.github.io` |
| WABER Kara et al. | 2025 | Yes | Introduces two new evaluation metrics—Efficiency and Reliability—that go beyond standard success rate measurements | 655 tasks | Task Success Rate, Efficiency Score | Action Match, State Information | `https://github.com/SumanKNath/WABER` |
| Online-Mind2Web Xue et al. (2025) | 2025 | Yes | A real-world online evaluation benchmark designed to reflect actual user interactions with live web interfaces | 300 tasks from 136 websites | Task Success Rate, Efficiency Score | Image Match, Action Match, State Information, LLM-as-a-Judge Evaluation | `https://github.com/OSU-NLP-Group/Online-Mind2Web` |
| AgentDAM Zharmagambetov et al. (2025) | 2025 | Yes | The first benchmark to evaluate privacy leakage risks in multimodal, realistic web environments using agentic models | 246 human-annotated test cases | Task Success Rate, Risk Ratio | Action Match, Text Match | `https://github.com/facebookresearch/ai-agent-privacy` |
| AgentReward-Bench Lù et al. (2025) | 2025 | No | The first benchmark to rigorously evaluate LLM-based judges against human expert annotations across multiple web agent tasks | 1,302 trajectories, 351 tasks | Task Success Rate, Completion under Policy | Image Match, Element/State Match | `https://agent-reward-bench.github.io` |

Qingxiu Dong, Lei Li, Damai Dai, Ce Zheng, Jingyuan Ma, Rui Li, Heming Xia, Jingjing Xu, Zhiyong Wu, Tianyu Liu, et al. A survey on in-context learning. *arXiv preprint arXiv:2301.00234*, 2022.

Alexey Dosovitskiy, Lucas Beyer, Alexander Kolesnikov, Dirk Weissenborn, Xiaohua Zhai, Thomas Unterthiner, Mostafa Dehghani, Matthias Minderer, Georg Heigold, Sylvain Gelly, Jakob Uszkoreit, and Neil Houlsby. An image is worth 16x16 words: Transformers for image recognition at scale, 2021. URL `https://arxiv.org/abs/2010.11929`.

Alexandre Drouin, Maxime Gasse, Massimo Caccia, Issam H Laradji, Manuel Del Verme, Tom Marty, Léo Boisvert, Megh Thakkar, Quentin Cappart, David Vazquez, et al. Workarena: How capable are web agents at solving common knowledge work tasks? *arXiv preprint arXiv:2403.07718*, 2024.

Yu Du, Fangyun Wei, and Hongyang Zhang. Anytool: Self-reflective, hierarchical agents for large-scale api calls. *arXiv preprint arXiv:2402.04253*, 2024.

Zhengxiao Du, Yujie Qian, Xiao Liu, Ming Ding, Jiezhong Qiu, Zhilin Yang, and Jie Tang. Glm: General language model pretraining with autoregressive blank infilling. *arXiv preprint arXiv:2103.10360*, 2021.

Table 56: Overview of web GUI agent benchmarks (Part VI).

| Benchmark | Year | Live | Highlight | Data Size | Metric | Measurement | Link |
|---|---|---|---|---|---|---|---|
| RealWebAssist Ye et al. (2025) | 2025 | No | The first benchmark to evaluate long-horizon web assistance using real-world users' sequential instructions expressed in natural and often ambiguous language | 1,885 instructions | Task Success Rate, Step Success Rate, Efficiency Score | Action Match | https://scai.cs.jhu.edu/projects/RealWebAssist/ |
| REAL Garg et al. (2025) | 2025 | Yes | Fully deterministic, high-fidelity replicas of real-world websites (e.g., Airbnb, Amazon, Gmail), enabling safe, reproducible, and configurable testing for multi-turn GUI-based agents | 112 tasks across 11 deterministic websites | Task Success Rate | Text Match, Action Match, State Information Match | https://github.com/agi-inc/agisdk |
| BEARCUBS Song et al. (2025) | 2025 | Yes | Emphasizes interaction with live web pages and includes multimodal tasks (e.g., video, audio, 3D) that cannot be solved by text-only methods, addressing limitations of prior benchmarks relying on static or simulated environments | 111 questions | Task Success Rate, Efficiency Score | Text Match, Action Match | https://bear-cubs.github.io |

Abhimanyu Dubey, Abhinav Jauhri, Abhinav Pandey, Abhishek Kadian, Ahmad Al-Dahle, Aiesha Letman, Akhil Mathur, Alan Schelten, Amy Yang, Angela Fan, et al. The llama 3 herd of models. *arXiv preprint arXiv:2407.21783*, 2024.

Zane Durante, Qiuyuan Huang, Naoki Wake, Ran Gong, Jae Sung Park, Bidipta Sarkar, Rohan Taori, Yusuke Noda, Demetri Terzopoulos, Yejin Choi, et al. Agent ai: Surveying the horizons of multimodal interaction. *arXiv preprint arXiv:2401.03568*, 2024.

Manuel Egele, Christopher Kruegel, Engin Kirda, and Giovanni Vigna. Pios: Detecting privacy leaks in ios applications. In *NDSS*, volume 2011, pp. 18th, 2011.

William Enck, Damien Octeau, Patrick D McDaniel, and Swarat Chaudhuri. A study of android application security. In *USENIX security symposium*, volume 2, 2011.

José Gonzalez Enríquez, Andres Jiménez-Ramírez, Francisco José Domínguez-Mayo, and Julián Alberto García-García. Robotic process automation: a scientific and industrial systematic mapping study. *IEEE Access*, 8:39113–39129, 2020.

Lutfi Eren Erdogan, Nicholas Lee, Sehoon Kim, Suhong Moon, Hiroki Furuta, Gopala Anumanchipalli, Kurt Keutzer, and Amir Gholami. Plan-and-act: Improving planning of agents for long-horizon tasks. *arXiv preprint arXiv:2503.09572*, 2025.

Ivan Evtimov, Arman Zharmagambetov, Aaron Grattafiori, Chuan Guo, and Kamalika Chaudhuri. Wasp: Benchmarking web agent security against prompt injection attacks. *arXiv preprint arXiv:2504.18575*, 2025.

Yue Fan, Lei Ding, Ching-Chen Kuo, Shan Jiang, Yang Zhao, Xinze Guan, Jie Yang, Yi Zhang, and Xin Eric Wang. Read anywhere pointed: Layout-aware gui screen reading with tree-of-lens grounding, 2024. URL https://arxiv.org/abs/2406.19263.

Yue Fan, Handong Zhao, Ruiyi Zhang, Yu Shen, Xin Eric Wang, and Gang Wu. Gui-bee: Align gui action grounding to novel environments via autonomous exploration, 2025. URL https://arxiv.org/abs/2501.13896.

Table 57: Overview of mobile GUI agent benchmarks (Part I).

| Benchmark | Year | Live | Highlight | Data Size | Metric | Measurement | Link |
|---|---|---|---|---|---|---|---|
| AndroidEnv Toyama et al. (2021a) | 2021 | Yes | Provides an open-source platform based on the Android ecosystem with over 100 tasks across approximately 30 apps, focusing on reinforcement learning for various Android interactions. | 100+ tasks | NA | NA | `https://github.com/google-deepmind/android_env` |
| PIXELHELP Li et al. (2020a) | 2020 | No | Includes a corpus of natural language instructions paired with UI actions across four task categories, aiding in grounding language to UI interactions. | 187 multi-step instructions | Step Success Rate | Element Match, Action Match | `https://github.com/google-research/google-research/tree/master/seq2act` |
| Mobile-Env Zhang et al. (2024c) | 2024 | Yes | Comprehensive toolkit for Android GUI benchmarks to enable controlled evaluations of real-world app interactions. | 224 tasks | Task Success Rate, Step Success Rate | Text Match, Element Match, Image Match, State Information | `https://github.com/X-LANCE/Mobile-Env` |
| B-MOCA Lee et al. (2024b) | 2024 | Yes | Benchmarks mobile device control agents on realistic tasks, incorporating UI layout and language randomization to evaluate generalization capabilities. | 131 tasks | Task Success Rate | Element Match, State Information | `https://b-moca.github.io/` |
| AndroidWorld Rawles et al. (2024) | 2024 | Yes | Offers a dynamic Android environment, allowing for diverse natural language instruction testing. | 116 tasks | Task Success Rate | State Information | `https://github.com/google-research/android_world` |

Adrienne Porter Felt, Erika Chin, Steve Hanna, Dawn Song, and David Wagner. Android permissions demystified. In *Proceedings of the 18th ACM conference on Computer and communications security*, pp. 627–638, 2011.

Sidong Feng and Chunyang Chen. Prompting is all you need: Automated android bug replay with large language models. In *Proceedings of the 46th IEEE/ACM International Conference on Software Engineering*, pp. 1–13, 2024.

Sidong Feng, Changhao Du, Huaxiao Liu, Qingnan Wang, Zhengwei Lv, Gang Huo, Xu Yang, and Chunyang Chen. Agent for user: Testing multi-user interactive features in tiktok. *arXiv preprint arXiv:2504.15474*, 2025.

Tao Feng, Chuanyang Jin, Jingyu Liu, Kunlun Zhu, Haoqin Tu, Zirui Cheng, Guanyu Lin, and Jiaxuan You. How far are we from agi: Are llms all we need? *Transactions on Machine Learning Research*.

Xueyang Feng, Zhi-Yuan Chen, Yujia Qin, Yankai Lin, Xu Chen, Zhiyuan Liu, and Ji-Rong Wen. Large language model-based human-agent collaboration for complex task solving. *arXiv preprint arXiv:2402.12914*, 2024.

Moghis Fereidouni and A. B. Siddique. Search beyond queries: Training smaller language models for web interactions via reinforcement learning, 2024. URL `https://arxiv.org/abs/2404.10887`.

Table 58: Overview of mobile GUI agent benchmarks (Part II).

| Benchmark | Year | Live | Highlight | Data Size | Metric | Measurement | Link |
|---|---|---|---|---|---|---|---|
| Mobile-Eval Wang et al. (2024e) | 2024 | Yes | Benchmark based on mainstream Android apps, and designed to test common mobile interactions. | 30 instructions | Task Success Rate, Step Success Rate, Efficiency Score | Text Match, Path Length | `https://github.com/X-PLUG/MobileAgent` |
| DroidTask Wen et al. (2024a) | 2024 | Yes | Android Task Automation benchmark supports exploration and task recording in real apps with corresponding GUI action traces. | 158 tasks | Step Success Rate, Task Success Rate | Element Match, Action Match | `https://github.com/MobileLLM/AutoDroid` |
| AITW Rawles et al. (2023) | 2023 | No | A large-scale dataset, which is partly inspired by PIXELHELP, covering diverse Android interactions. | 715,142 episodes | Task Success Rate, Step Success Rate | Action Match | `https://github.com/google-research/google-research/tree/master/android_in_the_wild` |
| AndroidArena Xing et al. (2024) | 2024 | Yes | Focuses on daily cross-app and constrained tasks within the Android ecosystem, providing single-app and multi-app interaction scenarios. | 221 tasks | Task Success Rate, Step Success Rate, Efficiency Score | Action Match, Path Length | `https://github.com/AndroidArenaAgent/AndroidArena` |
| LearnGUI Liu et al. (2025a) | 2025 | Yes | The first benchmark to systematically study few-shot demonstration-based learning in mobile GUI agents, featuring both offline and online task environments | Offline: 2,252 tasks (k-shot variants) across 44 apps; Online: 101 interactive tasks across 20 apps | Task Success Rate | Action Match | `https://lgy0404.github.io/LearnAct` |

Nádia Fernandes, Rui Lopes, and Luís Carriço. On web accessibility evaluation environments. In *Proceedings of the International Cross-Disciplinary Conference on Web Accessibility*, pp. 1–10, 2011.

Emilio Ferrara. Should chatgpt be biased? challenges and risks of bias in large language models. *arXiv preprint arXiv:2304.03738*, 2023.

Hiroki Furuta, Kuang-Huei Lee, Ofir Nachum, Yutaka Matsuo, Aleksandra Faust, Shixiang Shane Gu, and Izzeddin Gur. Multimodal web navigation with instruction-finetuned foundation models. *arXiv preprint arXiv:2305.11854*, 2023.

Hiroki Furuta, Yutaka Matsuo, Aleksandra Faust, and Izzeddin Gur. Exposing limitations of language model agents in sequential-task compositions on the web. In *ICLR 2024 Workshop on Large Language Model (LLM) Agents*, 2024.

Orazio Gambino, Leonardo Rundo, Vincenzo Cannella, Salvatore Vitabile, and Roberto Pirrone. A framework for data-driven adaptive gui generation based on dicom. *Journal of biomedical informatics*, 88:37–52, 2018.

Table 59: Overview of mobile GUI agent benchmarks (Part III).

| Benchmark | Year | Live | Highlight | Data Size | Metric | Measurement | Link |
|---|---|---|---|---|---|---|---|
| ANDROIDLAB Xu et al. (2024i) | 2024 | Yes | Provides a structured evaluation framework with 138 tasks across nine apps, supporting both text-only and multimodal agent evaluations on Android. | 138 tasks | Task Success Rate, Step Success Rate, Efficiency Score | Element Match, Image Match | `https://github.com/THUDM/Android-Lab` |
| GTArena Zhao et al. (2024b) | 2024 | No | Introduces a Transition Tuple for GUI defects, enabling large-scale defect dataset creation and reproducible, end-to-end automated testing. | 10,000+ GUI display and GUI interactions | Task Success Rate, Step Success Rate | Text Match, Element Match, Action Match | `https://github.com/ZJU-ACES-ISE/ChatUITest` |
| A3 Chai et al. (2025) | 2025 | Yes | Introduces a novel business-level LLM-based evaluation process, significantly reducing human labor and coding expertise requirements. | 201 tasks across 21 widely used apps | Task Success Rate | Element Match, Action Match | `https://yuxiangchai.github.io/Android-Agent-Arena/` |
| LlamaTouch Zhang et al. (2024h) | 2024 | Yes | Enables faithful and scalable evaluations for mobile UI task automation by matching task execution traces against annotated essential states | 496 tasks covering 57 unique Android applications | Task Success Rate, Step Success Rate, Efficiency Score | Text Match, Action Match, State Information Match | `https://github.com/LlamaTouch/LlamaTouch` |
| Mobile-AgentBench Wang et al. (2024i) | 2024 | Yes | Provides a fully autonomous evaluation process on real Android devices and flexibility in judging success conditions across multiple paths to completion | 100 tasks across 10 open-source Android applications | Task Success Rate, Efficiency Score, Latency, Token Cost | State Information (UI State Matching) | `https://mobileagentbench.github.io/` |

Erich Gamma. Design patterns: elements of reusable object-oriented software. *Person Education Inc*, 1995.

Yuyou Gan, Yong Yang, Zhe Ma, Ping He, Rui Zeng, Yiming Wang, Qingming Li, Chunyi Zhou, Songze Li, Ting Wang, Yunjun Gao, Yingcai Wu, and Shouling Ji. Navigating the risks: A survey of security, privacy, and ethics threats in llm-based agents, 2024. URL `https://arxiv.org/abs/2411.09523`.

Difei Gao, Siyuan Hu, Zechen Bai, Qinghong Lin, and Mike Zheng Shou. Assisteditor: Multi-agent collaboration for gui workflow automation in video creation. In *Proceedings of the 32nd ACM International Conference on Multimedia*, pp. 11255–11257, 2024a.

Difei Gao, Lei Ji, Zechen Bai, Mingyu Ouyang, Peiran Li, Dongxing Mao, Qinchen Wu, Weichen Zhang, Peiyi Wang, Xiangwu Guo, et al. Assistgui: Task-oriented pc graphical user interface automation. In *Proceedings of the IEEE/CVF Conference on Computer Vision and Pattern Recognition*, pp. 13289–13298, 2024b.

Ge Gao, Alexey Taymanov, Eduardo Salinas, Paul Mineiro, and Dipendra Misra. Aligning llm agents by learning latent preference from user edits. *arXiv preprint arXiv:2404.15269*, 2024c.

Jie Gao, Simret Araya Gebreegziabher, Kenny Tsu Wei Choo, Toby Jia-Jun Li, Simon Tangi Perrault, and Thomas W Malone. A taxonomy for human-llm interaction modes: An initial exploration. In *Extended Abstracts of the CHI Conference on Human Factors in Computing Systems*, pp. 1–11, 2024d.

Table 60: Overview of mobile GUI agent benchmarks (Part IV).

| Benchmark | Year | Live | Highlight | Data Size | Metric | Measurement | Link |
|---|---|---|---|---|---|---|---|
| Mobile-Bench Deng et al. (2024a) | 2024 | Yes | Supports both UI and API-based actions in multi-app scenarios, testing agents on single and multi-task structures with a checkpoint-based evaluation approach. | 832 entries (200+ tasks) | Task Success Rate, Step Success Rate, Efficiency Score | Action Match, Path Length | `https://github.com/XiaoMi/MobileBench` |
| Mobile Safety Bench Lee et al. (2024a) | 2024 | Yes | Prioritizes safety evaluation in mobile control tasks, with distinct tasks focused on helpfulness, privacy, and legal compliance. | 100 tasks | Task Success Rate, Risk Mitigation Success | Action Match with Safety Considered, Element Match, State Information | `https://mobilesafetybench.github.io/` |
| SPA-BENCH Chen et al. (2024d) | 2024 | Yes | Extensive evaluation framework supporting single-app and cross-app tasks in English and Chinese, providing a plug-and-play structure for diverse task scenarios. | 340 tasks | Task Success Rate, Step Success Rate, Efficiency Score | Action Match, State Information, Time Spent, API Cost | `https://spa-bench.github.io` |
| SPHINX Ran et al. (2025) | 2025 | Yes | Provides a fully automated benchmarking suite and introduces a multi-dimensional evaluation framework. | 284 common tasks | Task Success Rate, Efficiency Score, Completion under Policy, Turn Success Rate | Text Match, Image Match, Element Match, Action Match | / |
| AEIA-MN Chen et al. (2025c) | 2025 | Yes | Introduces the Active Environment Injection Attack (AEIA) framework that manipulates environmental elements (e.g., notifications) to mislead LLM-powered agents. | 61 tasks (Android-World) + 45 tasks (AppAgent) | Task Success Rate, Risk Ratio, Efficiency Score | Text Match, State Info, Action Match | / |
| AutoEval Sun et al. (2025a) | 2025 | Yes | Fully autonomous evaluation framework for mobile agents, removing manual reward-signal definition and evaluation coding. | 93 tasks | Task Success Rate | Action Match, State Info | / |

Longxi Gao, Li Zhang, Shihe Wang, Shangguang Wang, Yuanchun Li, and Mengwei Xu. Mobileviews: A large-scale mobile gui dataset. *arXiv preprint arXiv:2409.14337*, 2024e.

Minghe Gao, Wendong Bu, Bingchen Miao, Yang Wu, Yunfei Li, Juncheng Li, Siliang Tang, Qi Wu, Yueting Zhuang, and Meng Wang. Generalist virtual agents: A survey on autonomous agents across digital platforms. *arXiv preprint arXiv:2411.10943*, 2024f.

Weiwei Gao, Kexin Du, Yujia Luo, Weinan Shi, Chun Yu, and Yuanchun Shi. Easyask: An in-app contextual tutorial search assistant for older adults with voice and touch inputs. *Proceedings of the ACM on Interactive, Mobile, Wearable and Ubiquitous Technologies*, 8(3):1–27, 2024g.

Table 61: Overview of computer GUI agent benchmarks (Part I).

| Benchmark | Year | Live | Highlight | Data Size | Metric | Measurement | Link |
|---|---|---|---|---|---|---|---|
| OSWorld Xie et al. (2024b) | 2024 | Yes | Scalable, real computer environment for multimodal agents, supporting task setup, execution-based evaluation, and interactive learning across Ubuntu, Windows, and macOS. | 369 Ubuntu tasks, 43 Windows tasks | Task Success Rate | Execution-based State Information (such as internal file interpretation, permission management) | `https://os-world.github.io/` |
| Windows Agent Arena Bonatti et al. (2024) | 2024 | Yes | Adaptation of OSWorld focusing exclusively on the Windows OS with diverse multi-step tasks, enabling agents to use a wide range of applications and tools. | 154 tasks | Task Success Rate | Same as OSWorld, scalable with cloud parallelization | `https://microsoft.github.io/WindowsAgentArena` |
| OmniACT Kapoor et al. (2024) | 2024 | No | Assesses agents' capability to generate executable programs for computer tasks across desktop and web applications in various OS environments, prioritizing multimodal challenges. | 9,802 data points | Task Success Rate, Step Success Rate | Action Match | `https://huggingface.co/datasets/Writer/omniact` |
| VideoGUI Lin et al. (2024b) | 2024 | No | Focuses on visual-centric tasks from instructional videos, emphasizing planning and action precision in applications like Adobe Photoshop and Premiere Pro. | 178 tasks, 463 sub-tasks | Task Success Rate | State Information, Action Match | `https://showlab.github.io/videogui` |

Yunfan Gao, Yun Xiong, Xinyu Gao, Kangxiang Jia, Jinliu Pan, Yuxi Bi, Yi Dai, Jiawei Sun, Meng Wang, and Haofen Wang. Retrieval-augmented generation for large language models: A survey. *arXiv preprint arXiv:2312.10997*, 2023.

Divyansh Garg, Shaun VanWeelden, Diego Caples, Andis Draguns, Nikil Ravi, Pranav Putta, Naman Garg, Tomas Abraham, Michael Lara, Federico Lopez, et al. Real: Benchmarking autonomous agents on deterministic simulations of real websites. *arXiv preprint arXiv:2504.11543*, 2025.

Jesse James Garrett et al. Ajax: A new approach to web applications. 2005.

Zhiqi Ge, Juncheng Li, Xinglei Pang, Minghe Gao, Kaihang Pan, Wang Lin, Hao Fei, Wenqiao Zhang, Siliang Tang, and Yueting Zhuang. Iris: Breaking gui complexity with adaptive focus and self-refining, 2024. URL `https://arxiv.org/abs/2412.10342`.

Team GLM, Aohan Zeng, Bin Xu, Bowen Wang, Chenhui Zhang, Da Yin, Diego Rojas, Guanyu Feng, Hanlin Zhao, Hanyu Lai, et al. Chatglm: A family of large language models from glm-130b to glm-4 all tools. *arXiv preprint arXiv:2406.12793*, 2024.

Joshua Gorniak, Yoon Kim, Donglai Wei, and Nam Wook Kim. Vizability: Enhancing chart accessibility with llm-based conversational interaction. In *Proceedings of the 37th Annual ACM Symposium on User Interface Software and Technology*, pp. 1–19, 2024.

Table 62: Overview of computer GUI agent benchmarks (Part II).

| Benchmark | Year | Live | Highlight | Data Size | Metric | Measurement | Link |
|---|---|---|---|---|---|---|---|
| Spider2-V Cao et al. (2024) | 2024 | Yes | Benchmarks agents across data science and engineering workflows in authentic enterprise software environments, covering tasks from data ingestion to visualization. | 494 tasks | Task Success Rate | Action Match, State Information | `https://spider2-v.github.io` |
| Act2Cap Wu et al. (2024b) | 2024 | Yes | Emphasizes GUI action narration using cursor-based prompts in video format, covering a variety of GUI interactions like clicks, typing, and dragging. | 4,189 samples | Step Success Rate | Element Match | `https://showlab.github.io/GUI-Narrator` |
| OFFICEBENCH Wang et al. (2024p) | 2024 | Yes | Tests cross-application automation in office workflows with complex multi-step tasks across applications like Word and Excel, assessing operational integration in realistic scenarios. | 300 tasks | Task Success Rate | Action Match, Text Match, State Information | `https://github.com/zlwang-cs/OfficeBench` |
| AssistGUI Gao et al. (2024b) | 2024 | Yes | The first benchmark focused on task-oriented desktop GUI automation | 100 tasks from 9 popular applications | Task Success Rate, Efficiency Score | Element Match, Action Match | `https://showlab.github.io/assistgui/` |
| WorldGUI Zhao et al. (2025c) | 2025 | Yes | First GUI benchmark designed to evaluate dynamic GUI interactions by incorporating various initial states. | 315 total tasks from 10 Windows applications | Task Success Rate, Efficiency Score | Image Match, Element Match, Action Match | / |
| UI-Vision Nayak et al. (2025) | 2025 | No | The first large-scale benchmark specifically designed for desktop GUI agents | 8,227 query–label pairs in total | Task Success Rate | Action Match, Text Match | `https://uivision.github.io` |
| Computer Agent Arena Wang et al. (2025a) | 2025 | Yes | The first large-scale, open-ended evaluation platform for multimodal LLM-based agents in real desktop computing environments | User-proposed tasks | Task Success Rate | Human evaluators | `https://arena.xlang.ai/` |

Boyu Gou, Ruohan Wang, Boyuan Zheng, Yanan Xie, Cheng Chang, Yiheng Shu, Huan Sun, and Yu Su. Navigating the digital world as humans do: Universal visual grounding for gui agents, 2024. URL `https://arxiv.org/abs/2410.05243`.

Robert Gove and Jorge Faytong. Machine learning and event-based software testing: classifiers for identifying infeasible gui event sequences. In *Advances in computers*, volume 86, pp. 109–135. Elsevier, 2012.

Maria Fernanda Granda, Otto Parra, and Bryan Alba-Sarango. Towards a model-driven testing framework for gui test cases generation from user stories. In *ENASE*, pp. 453–460, 2021.

Table 63: Overview of cross-platform GUI agent benchmarks.

| Benchmark | Platform | Year | Live | Highlight | Data Size | Metric | Measurement | Link |
|---|---|---|---|---|---|---|---|---|
| VisualAgent Bench Liu et al. (2024i) | Web, Android, Game, Virtual Embodied. | 2024 | Yes | First benchmark designed for visual foundation agents across GUI and multi-modal tasks, focusing on vision-centric interactions in Android, web, and game environments. | 4,482 trajectories | Task Success Rate | Text Match | `https://github.com/THUDM/VisualAgentBench/` |
| SPR Benchmark Fan et al. (2024) | Mobile, Web, and Operating Systems | 2024 | Yes | Evaluates GUI screen readers' ability to describe both content and layout information | Includes 650 screenshots annotated with 1,500 target points and regions | Task Success Rate, Efficiency Score | Text Match, Element Match | / |
| AgentStudio Zheng et al. (2024c) | Windows, Linux, macOS | 2024 | Yes | Open toolkit for creating and benchmarking general-purpose virtual agents, supporting complex interactions across diverse software applications. | NA | Step Success Rate | Action Match, State Information and Image Match | `https://computer-agents.github.io/agent-studio/` |
| CRAB Xu et al. (2024g) | Linux, Android | 2024 | Yes | Cross-environment benchmark evaluating agents across mobile and desktop devices, using a graph-based evaluation method to handle multiple correct paths and task flexibility. | 120 tasks | Step Success Rate, Efficiency Score | Action Match | `https://github.com/crab-benchmark` |
| ScreenSpot Cheng et al. (2024a) | iOS, Android, macOS, Windows, Web. | 2024 | No | Vision-based GUI benchmark with pre-trained GUI grounding, assessing agents' ability to interact with GUI elements across mobile, desktop, and web platforms using only screenshots. | 1,200 instructions | Step Success Rate | Action Match | `https://github.com/njucckevin/SeeClick` |

Roger Grosse, Juhan Bae, Cem Anil, Nelson Elhage, Alex Tamkin, Amirhossein Tajdini, Benoit Steiner, Dustin Li, Esin Durmus, Ethan Perez, et al. Studying large language model generalization with influence functions. *arXiv preprint arXiv:2308.03296*, 2023.

Xiaodong Gu, Hongyu Zhang, Dongmei Zhang, and Sunghun Kim. Deep api learning. In *Proceedings of the 2016 24th ACM SIGSOFT international symposium on foundations of software engineering*, pp. 631–642, 2016.

Yu Gu, Boyuan Zheng, Boyu Gou, Kai Zhang, Cheng Chang, Sanjari Srivastava, Yanan Xie, Peng Qi, Huan Sun, and Yu Su. Is your llm secretly a world model of the internet? model-based planning for web agents. *arXiv preprint arXiv:2411.06559*, 2024.

Yanchu Guan, Dong Wang, Zhixuan Chu, Shiyu Wang, Feiyue Ni, Ruihua Song, Longfei Li, Jinjie Gu, and Chenyi Zhuang. Intelligent virtual assistants with llm-based process automation. *ArXiv*, abs/2312.06677, 2023. URL `https://api.semanticscholar.org/CorpusID:266174422`.

Table 64: Overview of GUI-testing with LLM-powered GUI agents (Part I).

| Project | Category | Platform | Model | Perception | Action | Scenario | Highlight | Link |
|---------|----------|----------|-------|------------|--------|----------|-----------|------|
| Daniel and Anne Zimmermann & Koziolek (2023a) | General testing | General-purpose platforms | GPT-3 | GUI structure and state | Standard UI operations | Automates the software testing process using natural language test cases | Applies GPT-3's language understanding capabilities to GUI-based software testing, enabling natural interaction through text-based test case descriptions. | `https://github.com/neuroevolution%2Dai/SoftwareTestingLanguageModels` |
| Daniel and Anne Zimmermann & Koziolek (2023b) | General testing | Web platforms | GPT-4 | HTML DOM structure | Standard UI operations | Automated GUI testing to enhance branch coverage and efficiency | Performs end-to-end GUI testing using GPT-4's natural language understanding and reasoning capabilities. | `https://github.com/SoftwareTestingLLMs/WebtestingWithLLMs` |
| GPTDroid Liu et al. (2024k) | General testing | Mobile Android | GPT-3.5 | UI view hierarchy files | Standard UI operations and compound actions | Automates GUI testing to improve testing coverage and detect bugs efficiently | Formulates GUI testing as a Q&A task, utilizing LLM capabilities to provide human-like interaction. | `https://github.com/franklinbill/GPTDroid` |
| DROID-AGENT Yoon et al. (2024) | General testing | Mobile Android | GPT-3.5, GPT-4 | JSON representation of the GUI state | Standard UI operations, higher-level APIs, and custom actions | Semantic, intent-driven automation of GUI testing | Autonomously generates and executes high-level, realistic tasks for Android GUI testing based on app-specific functionalities. | `https://github.com/coinse/droidagent` |

Yanchu Guan, Dong Wang, Zhixuan Chu, Shiyu Wang, Feiyue Ni, Ruihua Song, and Chenyi Zhuang. Intelligent agents with llm-based process automation. In *Proceedings of the 30th ACM SIGKDD Conference on Knowledge Discovery and Data Mining*, pp. 5018–5027, 2024a.

Yanchu Guan, Dong Wang, Yan Wang, Haiqing Wang, Renen Sun, Chenyi Zhuang, Jinjie Gu, and Zhixuan Chu. Explainable behavior cloning: Teaching large language model agents through learning by demonstration. *arXiv preprint arXiv:2410.22916*, 2024b.

Daya Guo, Qihao Zhu, Dejian Yang, Zhenda Xie, Kai Dong, Wentao Zhang, Guanting Chen, Xiao Bi, Yu Wu, YK Li, et al. Deepseek-coder: When the large language model meets programming–the rise of code intelligence. *arXiv preprint arXiv:2401.14196*, 2024a.

Daya Guo, Dejian Yang, Haowei Zhang, Junxiao Song, Ruoyu Zhang, Runxin Xu, Qihao Zhu, Shirong Ma, Peiyi Wang, Xiao Bi, et al. Deepseek-r1: Incentivizing reasoning capability in llms via reinforcement learning. *arXiv preprint arXiv:2501.12948*, 2025.

Taicheng Guo, Xiuying Chen, Yaqi Wang, Ruidi Chang, Shichao Pei, Nitesh V Chawla, Olaf Wiest, and Xiangliang Zhang. Large language model based multi-agents: A survey of progress and challenges. *arXiv preprint arXiv:2402.01680*, 2024b.

Zhicheng Guo, Sijie Cheng, Hao Wang, Shihao Liang, Yujia Qin, Peng Li, Zhiyuan Liu, Maosong Sun, and Yang Liu. Stabletoolbench: Towards stable large-scale benchmarking on tool learning of large language models, 2024c.

Izzeddin Gur, Hiroki Furuta, Austin Huang, Mustafa Safdari, Yutaka Matsuo, Douglas Eck, and Aleksandra Faust. A real-world webagent with planning, long context understanding, and program synthesis. *arXiv preprint arXiv:2307.12856*, 2023.

Izzeddin Gur, Hiroki Furuta, Austin Huang, Mustafa Safdari, Yutaka Matsuo, Douglas Eck, and Aleksandra Faust. A real-world webagent with planning, long context understanding, and program synthesis, 2024. URL `https://arxiv.org/abs/2307.12856`.

Table 65: Overview of GUI-testing with LLM-powered GUI agents (Part II).

| Project | Category | Platform | Model | Perception | Action | Scenario | Highlight | Link |
|---|---|---|---|---|---|---|---|---|
| AUITest-Agent Hu et al. (2024c) | General testing | Mobile Android | GPT-4 | GUI screenshots, UI hierarchy files, and CV-enhanced techniques like Vision-UI | Standard UI operations | Automated functional testing of GUIs | Features dynamic agent organization for step-oriented testing and a multi-source data extraction strategy for precise function verification. | `https://github.com/bz-lab/AUITestAgent` |
| VisionDroid Liu et al. (2024l) | General testing | Mobile Android | GPT-4 | GUI screenshots with annotated bounding boxes, View hierarchy files | Standard UI operations | Identifies non-crash bugs | Integrates vision-driven prompts and GUI text alignment with vision-language models to enhance understanding of GUI contexts and app logic. | `https://github.com/testtestA6/VisionDroid` |
| AXNav Taeb et al. (2024) | Accessibility testing | iOS mobile devices | GPT-4 | GUI screenshots, UI element detection model, and OCR | Gestures, capturing screenshots, and highlighting potential accessibility issues | Automates accessibility testing workflows, including testing features like VoiceOver, Dynamic Type, Bold Text, and Button Shapes | Adapts to natural language test instructions and generates annotated videos to visually and interactively review accessibility test results. | / |
| LLMigrate Beyzaei et al. (2024) | General testing | Mobile Android | GPT-4o | DOM and screenshots | Standard UI operations | Automates the transfer of usage-based UI tests between Android apps | Leverages multimodal LLMs to perform UI test transfers without requiring source code access | / |
| Cui et al., Cui et al. (2024) | Test input generation | Mobile Android | GPT-3.5, GPT-4 | GUI structures and contextual information | Entering text inputs | Generating and validating text inputs for Android applications | Demonstrates the effectiveness of various LLMs in generating context-aware text inputs, improving UI test coverage, and identifying previously unreported bugs. | / |
| QTypist Liu et al. (2023c) | Test input generation | Mobile Android | GPT-3 | UI hierarchy files | Generates semantic text inputs | Automates mobile GUI testing by generating appropriate text inputs | Formulates text input generation as a cloze-style fill-in-the-blank language task. | / |

Shanshan Han, Qifan Zhang, Yuhang Yao, Weizhao Jin, Zhaozhuo Xu, and Chaoyang He. Llm multi-agent systems: Challenges and open problems. *arXiv preprint arXiv:2402.03578*, 2024.

Hao Hao, Vicky Singh, and Wenliang Du. On the effectiveness of api-level access control using bytecode rewriting in android. In *Proceedings of the 8th ACM SIGSAC symposium on Information, computer and communications security*, pp. 25–36, 2013.

Robert Hardy and Enrico Rukzio. Touch & interact: touch-based interaction of mobile phones with displays. In *Proceedings of the 10th international conference on Human computer interaction with mobile devices and services*, pp. 245–254, 2008.

Feng He, Tianqing Zhu, Dayong Ye, Bo Liu, Wanlei Zhou, and Philip S Yu. The emerged security and privacy of llm agent: A survey with case studies. *arXiv preprint arXiv:2407.19354*, 2024a.

Table 66: Overview of GUI-testing with LLM-powered GUI agents (Part III).

| Project | Category | Platform | Model | Perception | Action | Scenario | Highlight | Link |
|---|---|---|---|---|---|---|---|---|
| Crash-Translator Huang et al. (2024c) | Bug replay | Mobile Android | GPT-3 | Crash-related stack trace information and GUI structure | Standard UI operations | Automates the reproduction of mobile application crashes | Leverages LLMs for iterative GUI navigation and crash reproduction from stack traces, integrating a reinforcement learning-based scoring system to optimize exploration steps. | `https://github.com/wuchiuwong/CrashTranslator` |
| AdbGPT Feng & Chen (2024) | Bug replay | Mobile Android | GPT-3.5 | GUI structure and hierarchy | Standard UI operations | Automates bug reproduction by extracting S2R (Steps to Reproduce) entities | Combines prompt engineering with few-shot learning and chain-of-thought reasoning to leverage LLMs for GUI-based tasks. | `https://github.com/sidongfeng/AdbGPT` |
| MagicWand Ding et al. (2024b) | Verrification | Mobile Android | GPT-4V | UI screenshots and hierarchical UI control tree | Standard UI operations | Automates the verification of "How-to" instructions from a search engine | Features a three-stage process: extracting instructions, executing them in a simulated environment, and reranking search results based on execution outcomes. | / |
| UXAgent Lu et al. (2025a) | Usability testing for web design | Web | Self-designed | Simplified HTML representations | Standard UI operations | Automated usability testing of web applications | Enables LLM-powered automated usability testing by simulating thousands of user interactions, collecting both qualitative and quantitative data, and providing researchers with early feedback before real-user studies. | `https://uxagent.hailab.io` |
| Guardian Ran et al. (2024) | GUI Testing | Mobile Android | GPT-3.5 | GUI structure, Properties | Standard UI operations | Autonomously explores mobile applications, interacting with the UI to validate core functionalities. | Improves LLM-driven UI testing by offloading planning tasks to an external runtime system. | / |
| Test-Agent Li et al. (2024h) | GUI Testing | Android, iOS, Harmony OS | Not Mentioned | GUI screenshots, UI structure information | Standard UI operations | Cross-platform mobile testing | Eliminates the need for pre-written test scripts by leveraging LLMs and multimodal perception to generate and execute test cases automatically. | / |
| VLM-Fuzz Demissie et al. (2025) | GUI Testing | Android (Mobile) | GPT-4o | GUI screenshots and UI structure information | Standard UI operations, system-level actions | Automated Android app testing for detection of crashes and bugs | Integrates vision-language reasoning with heuristic-based depth-first search (DFS) to systematically explore complex Android UIs, achieving significantly higher code coverage | / |

Hongliang He, Wenlin Yao, Kaixin Ma, Wenhao Yu, Yong Dai, Hongming Zhang, Zhenzhong Lan, and Dong Yu. Webvoyager: Building an end-to-end web agent with large multimodal models, 2024b. URL `https://arxiv.org/abs/2401.13919`.

Hongliang He, Wenlin Yao, Kaixin Ma, Wenhao Yu, Hongming Zhang, Tianqing Fang, Zhenzhong Lan, and Dong Yu. Openwebvoyager: Building multimodal web agents via iterative real-world exploration, feedback and optimization, 2024c. URL `https://arxiv.org/abs/2410.19609`.

Table 67: Overview of GUI-testing with LLM-powered GUI agents (Part IV).

| Project | Category | Platform | Model | Perception | Action | Scenario | Highlight | Link |
|---------|----------|----------|-------|------------|--------|----------|-----------|------|
| BugCraft Yapağcı et al. (2025) | Bug Reproduction | Windows Computer | BugCraft based on GPT-4o | GUI screenshots | Standard UI operations | Automatically reproduces crash bugs in Minecraft by reading user-submitted bug reports, generating structured steps, and executing them to cause a crash | First end-to-end framework that automates crash bug reproduction in a complex open-world game (Minecraft) using LLM-driven agents, vision-based UI parsing, and structured action execution | https://bugcraft2025.github.io/ |
| ReuseDroid Li et al. (2025c) | GUI Testing | Mobile Android | ReuseDroid based on GPT-4o | GUI screenshots and widget properties | Standard UI operations | Migrates GUI test cases between Android apps that share similar functionality but differ in operational logic | Leverages visual contexts and dynamic feedback mechanisms to significantly boost migration success rates compared to prior mapping- and LLM-based methods | / |
| SeeAct-ATA and PinATA Chevrot et al. (2025) | GUI Testing | Web | SeeAct Zheng et al. (2024a) | GUI structure and DOM | Standard UI operations | Automates manual end-to-end (E2E) web application testing | First open-source attempt to adapt LLM-powered Autonomous Web Agents into Autonomous Test Agents (ATA) for web testing | / |
| GERALLT Rosenbach et al. (2025) | GUI Testing | Desktop (Windows/Linux) | GPT-4o | GUI screenshots and UI structure information | Standard UI operations | Finds unintuitive behavior, inconsistencies, and functional errors in GUIs without pre-defined test scripts | Pioneers LLM-driven testing on real-world desktop GUI applications (not web or mobile), combining structured GUI parsing with LLM-based control and evaluation | https://github.com/DLR-SC/GERALLT |
| ProphetAgent Kong et al. | GUI Testing | Android Mobile | GPT-4o | XML UI trees | Executable UI test scripts | Automates GUI test case generation from natural language for regression and compatibility testing in mobile apps | Innovatively combines LLM reasoning with a semantically enriched GUI graph (CUTG), significantly improving GUI test synthesis performance and efficiency over state-of-the-art tools | https://github.com/prophetagent/Home |
| Agent for User Feng et al. (2025) | GUI Testing | Android Mobile | GPT-4 | XML view hierarchy | Standard UI operations | Automated testing of multi-user interactive features | Introduces a multi-agent LLM framework where each agent simulates a user on a virtual device | / |

Jiang He, I-Ling Yen, Tu Peng, Jing Dong, and Farokh Bastani. An adaptive user interface generation framework for web services. In *2008 IEEE Congress on Services Part II (services-2 2008)*, pp. 175–182. IEEE, 2008.

Yanheng He, Jiahe Jin, Shijie Xia, Jiadi Su, Runze Fan, Haoyang Zou, Xiangkun Hu, and Pengfei Liu. Pc agent: While you sleep, ai works – a cognitive journey into digital world, 2024d. URL https://arxiv.org/abs/2412.17589.

Zhitao He, Zijun Liu, Peng Li, May Fung, Ming Yan, Ji Zhang, Fei Huang, and Yang Liu. Enhancing language multi-agent learning with multi-agent credit re-assignment for interactive environment generalization. *arXiv preprint arXiv:2502.14496*, 2025.

Table 68: Overview of virtual assistants with LLM-powered GUI agents (Part I).

| Project | Type | Platform | Model | Perception | Action | Scenario | Highlight | Link |
|---|---|---|---|---|---|---|---|---|
| ProAgent Ye et al. (2023) | Research | Web and Desktop | GPT-4 | Task descriptions and structured application data | Standard UI operations and dynamic branching | Automates business processes such as data analysis, report generation, and notifications via GUI-based tools | Introduces dynamic workflows where agents interpret and execute tasks flexibly, surpassing traditional RPA systems | https://github.com/OpenBMB/ProAgent |
| LLMPA Guan et al. (2024a) | Research | Mobile (Android) | AntLLM-10b | UI tree structures, visual modeling, and text extraction modules | Standard UI operations | Automates user interactions within mobile apps, such as ticket booking | Integrates LLM reasoning capabilities with a modular design that supports task decomposition, object detection, and robust action prediction in GUI environments | / |
| VizAbility Gorniak et al. (2024) | Research | Desktop | GPT-4V | Keyboard-navigable tree views | Navigates chart structures and generates answers | Assists blind and low-vision users in exploring and understanding data visualizations | Integrates structured chart navigation with LLM-powered conversational capabilities, enabling visually impaired users to query in natural language | https://dwr.bc.edu/vizability/ |
| GPTVoice-Tasker Vu et al. (2024) | Research | Mobile (Android) | GPT-4 | Android Accessibility Tree | Standard UI operations | Automates user interactions on mobile devices through voice commands | Integrates LLMs for natural command interpretation and real-time GUI interactions, using a graph-based local database to record and replicate interactions | https://github.com/vuminhduc796/GPTVoiceTasker |

Theodore D Hellmann and Frank Maurer. Rule-based exploratory testing of graphical user interfaces. In *2011 Agile Conference*, pp. 107–116. IEEE, 2011.

S Hochreiter. Long short-term memory. *Neural Computation MIT-Press*, 1997.

Nobukatsu Hojo, Kazutoshi Shinoda, Yoshihiro Yamazaki, Keita Suzuki, Hiroaki Sugiyama, Kyosuke Nishida, and Kuniko Saito. Generativegui: Dynamic gui generation leveraging llms for enhanced user interaction on chat interfaces. In *Proceedings of the Extended Abstracts of the CHI Conference on Human Factors in Computing Systems*, pp. 1–9, 2025.

Wenyi Hong, Weihan Wang, Qingsong Lv, Jiazheng Xu, Wenmeng Yu, Junhui Ji, Yan Wang, Zihan Wang, Yuxuan Zhang, Juanzi Li, Bin Xu, Yuxiao Dong, Ming Ding, and Jie Tang. Cogagent: A visual language model for gui agents, 2023. URL https://arxiv.org/abs/2312.08914.

HONOR. Honor introduces magicos 9.0, 2024. URL https://www.fonearena.com/blog/438680/honor-magicos-9-0-features.html. Accessed: 2024-11-16.

Jakub Hoscilowicz, Bartosz Maj, Bartosz Kozakiewicz, Oleksii Tymoshchuk, and Artur Janicki. Clickagent: Enhancing ui location capabilities of autonomous agents. *arXiv preprint arXiv:2410.11872*, 2024.

Yu-Chung Hsiao, Fedir Zubach, Gilles Baechler, Victor Carbune, Jason Lin, Maria Wang, Srinivas Sunkara, Yun Zhu, and Jindong Chen. Screenqa: Large-scale question-answer pairs over mobile app screenshots, 2024. URL https://arxiv.org/abs/2209.08199.

Cheng-Yu Hsieh, Si-An Chen, Chun-Liang Li, Yasuhisa Fujii, Alexander Ratner, Chen-Yu Lee, Ranjay Krishna, and Tomas Pfister. Tool documentation enables zero-shot tool-usage with large language models. *arXiv preprint arXiv:2308.00675*, 2023.

Table 69: Overview of virtual assistants with LLM-powered GUI agents (Part II).

| Project | Type | Platform | Model | Perception | Action | Scenario | Highlight | Link |
|---|---|---|---|---|---|---|---|---|
| AutoTask Pan et al. (2023b) | Research | Mobile (Android) | GPT-4 | Android Accessibility Tree | Standard UI operations | Automates multi-step tasks on mobile devices | Operates without predefined scripts or configurations, autonomously exploring GUI environments | `https://github.com/BowenBryanWang/AutoTask` |
| AssistEditor Gao et al. (2024a) | Research | Windows | UniVTG Lin et al. (2023) | GUI elements, user requirements, and video data | Standard UI operations | Automates video editing workflows | Employs a multi-agent collaboration framework where agents specialize in roles to integrate user requirements into video editing workflows | / |
| PromptRPA Huang et al. (2024a) | Research | Mobile (Android) | GPT-4 and GPT-3.5 Turbo | Layout hierarchy and screenshots with OCR | Standard UI operations and application-level functionalities | Automates smartphone tasks and creates interactive tutorials | Integrates user feedback loops for continuous improvement, addressing interface evolution and task variability | / |
| EasyAsk Gao et al. (2024g) | Research | Mobile (Android) | GPT-4 | Android Accessibility Tree | Highlights specific UI elements for user interaction | Assists older adults in learning and navigating smartphone functions through in-app interactive tutorials | Combines voice and touch inputs, supplementing incomplete or ambiguous queries with in-app contextual information | / |
| WebNav Srinivasan & Patapati (2025) | Research | Web | Gemini 2.0 Flash Thinking | Standard UI operations | GUI screenshots and DOM | Assistive technology for visually impaired users, enabling voice-based navigation of complex websites | Combines a ReAct-style reasoning loop, real-time DOM labeling, and voice-driven interaction to support intelligent web navigation for visually impaired users | / |

Table 70: Overview of virtual assistants with LLM-powered GUI agents (Part III).

| Project | Type | Platform | Model | Perception | Action | Scenario | Highlight | Link |
|---------|------|----------|-------|------------|--------|----------|-----------|------|
| OpenAdapt OpenAdapt AI (2024) | Open-source | Desktop | LLM, VLM (e.g., GPT-4, ACT-1) | Screenshots with CV tools for GUI parsing | Standard UI operations | Automates repetitive tasks across industries | Learns task automation by observing user interactions, eliminating manual scripting | https://github.com/OpenAdaptAI/OpenAdapt |
| AgentSea AgentSea AI (2024) | Open-source | Desktop and Web | LLM, VLM | Screenshots with CV tools for GUI parsing | Standard UI operations | Automates tasks within GUI environments | Offers a modular toolkit adhering to the UNIX philosophy, allowing developers to create custom AI agents for diverse GUI environments | https://www.agentsea.ai/ |
| Open Interpreter Interpreter (2024) | Open-source | Desktop, Web, Mobile (Android) | LLM | System perception via command-line | Shell commands, code, and native APIs | Automates tasks, conducts data analysis, manages files, and controls web browsers for research | Executes code locally, providing full access to system resources and libraries, overcoming limitations of cloud-based services | https://github.com/OpenInterpreter/open-interpreter |
| MultiOn MultiOn AI (2024) | Production | Web | LLM | / | Standard UI operations | Automates web-based tasks | Performs autonomous web actions via natural language commands | https://www.multion.ai/ |
| YOYO Agent in MagicOS HONOR (2024) | Production | Mobile (MagicOS 9.0) | MagicLM | GUI context | Executes in-app and cross-app operations | Automates daily tasks, enhancing productivity | Leverages MagicLM to understand and execute complex tasks across applications, learning user habits to provide personalized assistance | / |
| Power Automate Microsoft (2024) | Production | Windows | LLM, VLM | Records user interactions with the GUI | Standard UI operations | Automates repetitive tasks and streamlines workflows | Translates natural language descriptions of desired automations into executable workflows | https://learn.microsoft.com/en-us/power-automate/desktop-flows/create%2Dflow-using%2Dai-recorder |
| Eko AI (2025) | Production | Web browsers and computer environments | ChatGPT and Claude 3.5 | Visual-Interactive Element Perception (VIEP) technology for interacting with GUI elements. | Standard UI operations. | Automates tasks by handling diverse workflows. | Decomposes natural language task descriptions into executable workflows, enabling seamless integration of natural language and programming logic in agent design. | https://eko.fellou.ai/ |

Gang Hu, Linjie Zhu, and Junfeng Yang. Appflow: using machine learning to synthesize robust, reusable ui tests. In *Proceedings of the 2018 26th ACM Joint Meeting on European Software Engineering Conference and Symposium on the Foundations of Software Engineering*, ESEC/FSE 2018, pp. 269–282, New York, NY, USA, 2018. Association for Computing Machinery. ISBN 9781450355735. doi: 10.1145/3236024.3236055. URL `https://doi.org/10.1145/3236024.3236055`.

Siyuan Hu, Mingyu Ouyang, Difei Gao, and Mike Zheng Shou. The dawn of gui agent: A preliminary case study with claude 3.5 computer use, 2024a. URL `https://arxiv.org/abs/2411.10323`.

Xueyu Hu, Tao Xiong, Biao Yi, Zishu Wei, Ruixuan Xiao, Yurun Chen, Jiasheng Ye, Meiling Tao, Xiangxin Zhou, Ziyu Zhao, et al. Os agents: A survey on mllm-based agents for general computing devices use. 2024b.

Yongxiang Hu, Xuan Wang, Yingchuan Wang, Yu Zhang, Shiyu Guo, Chaoyi Chen, Xin Wang, and Yangfan Zhou. Auitestagent: Automatic requirements oriented gui function testing, 2024c. URL `https://arxiv.org/abs/2407.09018`.

Zhiyuan Hu, Shiyun Xiong, Yifan Zhang, See-Kiong Ng, Anh Tuan Luu, Bo An, Shuicheng Yan, and Bryan Hooi. Guiding vlm agents with process rewards at inference time for gui navigation, 2025. URL `https://arxiv.org/abs/2504.16073`.

Jiaxing Huang and Jingyi Zhang. A survey on evaluation of multimodal large language models. *arXiv preprint arXiv:2408.15769*, 2024.

Jie Huang and Kevin Chen-Chuan Chang. Towards reasoning in large language models: A survey. *arXiv preprint arXiv:2212.10403*, 2022.

Jing Huang, Zhixiong Zeng, Wenkang Han, Yufeng Zhong, Liming Zheng, Shuai Fu, Jingyuan Chen, and Lin Ma. Scaletrack: Scaling and back-tracking automated gui agents, 2025a. URL `https://arxiv.org/abs/2505.00416`.

Tenghao Huang, Kinjal Basu, Ibrahim Abdelaziz, Pavan Kapanipathi, Jonathan May, and Muhao Chen. R2d2: Remembering, reflecting and dynamic decision making for web agents. *arXiv preprint arXiv:2501.12485*, 2025b.

Tian Huang, Chun Yu, Weinan Shi, Zijian Peng, David Yang, Weiqi Sun, and Yuanchun Shi. Prompt2task: Automating ui tasks on smartphones from textual prompts. *ACM Transactions on Computer-Human Interaction*.

Tian Huang, Chun Yu, Weinan Shi, Zijian Peng, David Yang, Weiqi Sun, and Yuanchun Shi. Promptrpa: Generating robotic process automation on smartphones from textual prompts. *arXiv preprint arXiv:2404.02475*, 2024a.

Xiaowei Huang, Wenjie Ruan, Wei Huang, Gao Jin, Yizhen Dong, Changshun Wu, Saddek Bensalem, Ronghui Mu, Yi Qi, Xingyu Zhao, Kaiwen Cai, Yanghao Zhang, Sihao Wu, Peipei Xu, Dengyu Wu, André Freitas, and Mustafa A. Mustafa. A survey of safety and trustworthiness of large language models through the lens of verification and validation. *Artif. Intell. Rev.*, 57:175, 2023. URL `https://api.semanticscholar.org/CorpusID:258823083`.

Xu Huang, Weiwen Liu, Xiaolong Chen, Xingmei Wang, Hao Wang, Defu Lian, Yasheng Wang, Ruiming Tang, and Enhong Chen. Understanding the planning of llm agents: A survey. *arXiv preprint arXiv:2402.02716*, 2024b.

Yuchao Huang, Junjie Wang, Zhe Liu, Yawen Wang, Song Wang, Chunyang Chen, Yuanzhe Hu, and Qing Wang. Crashtranslator: Automatically reproducing mobile application crashes directly from stack trace. In *Proceedings of the 46th IEEE/ACM International Conference on Software Engineering*, pp. 1–13, 2024c.

Zhiyuan Huang, Ziming Cheng, Junting Pan, Zhaohui Hou, and Mingjie Zhan. Spiritsight agent: Advanced gui agent with one look. *arXiv preprint arXiv:2503.03196*, 2025c.

Zheng Hui, Yinheng Li, Dan zhao, Tianyi Chen, Colby Banbury, and Kazuhito Koishida. Winclick: Gui grounding with multimodal large language models, 2025. URL https://arxiv.org/abs/2503.04730.

Aaron Hurst, Adam Lerer, Adam P Goucher, Adam Perelman, Aditya Ramesh, Aidan Clark, AJ Ostrow, Akila Welihinda, Alan Hayes, Alec Radford, et al. Gpt-4o system card. *arXiv preprint arXiv:2410.21276*, 2024.

Open Interpreter. Open interpreter: A natural language interface for computers. GitHub repository, 2024. URL https://github.com/OpenInterpreter/open-interpreter. Accessed: 2024-10-27.

Iat Long Iong, Xiao Liu, Yuxuan Chen, Hanyu Lai, Shuntian Yao, Pengbo Shen, Hao Yu, Yuxiao Dong, and Jie Tang. Openwebagent: An open toolkit to enable web agents on large language models. In *Proceedings of the 62nd Annual Meeting of the Association for Computational Linguistics (Volume 3: System Demonstrations)*, pp. 72–81, 2024.

Lucija Ivančić, Dalia Suša Vugec, and Vesna Bosilj Vukšić. Robotic process automation: systematic literature review. In *Business Process Management: Blockchain and Central and Eastern Europe Forum: BPM 2019 Blockchain and CEE Forum, Vienna, Austria, September 1–6, 2019, Proceedings 17*, pp. 280–295. Springer, 2019.

Aaron Jaech, Adam Kalai, Adam Lerer, Adam Richardson, Ahmed El-Kishky, Aiden Low, Alec Helyar, Aleksander Madry, Alex Beutel, Alex Carney, et al. Openai o1 system card. *arXiv preprint arXiv:2412.16720*, 2024.

Lawrence Jang, Yinheng Li, Charles Ding, Justin Lin, Paul Pu Liang, Dan Zhao, Rogerio Bonatti, and Kazuhito Koishida. Videowebarena: Evaluating long context multimodal agents with video understanding web tasks. *arXiv preprint arXiv:2410.19100*, 2024.

Bernard Jim Jansen. The graphical user interface. *ACM SIGCHI Bull.*, 30:22–26, 1998. URL https://api.semanticscholar.org/CorpusID:18416305.

Susmit Jha, Sumit Kumar Jha, Patrick Lincoln, Nathaniel D. Bastian, Alvaro Velasquez, and Sandeep Neema. Dehallucinating large language models using formal methods guided iterative prompting. *2023 IEEE International Conference on Assured Autonomy (ICAA)*, pp. 149–152, 2023. URL https://api.semanticscholar.org/CorpusID:260810131.

Chengyou Jia, Minnan Luo, Zhuohang Dang, Qiushi Sun, Fangzhi Xu, Junlin Hu, Tianbao Xie, and Zhiyong Wu. Agentstore: Scalable integration of heterogeneous agents as specialized generalist computer assistant. *arXiv preprint arXiv:2410.18603*, 2024.

Albert Q Jiang, Alexandre Sablayrolles, Arthur Mensch, Chris Bamford, Devendra Singh Chaplot, Diego de las Casas, Florian Bressand, Gianna Lengyel, Guillaume Lample, Lucile Saulnier, et al. Mistral 7b. *arXiv preprint arXiv:2310.06825*, 2023.

Wenjia Jiang, Yangyang Zhuang, Chenxi Song, Xu Yang, and Chi Zhang. Appagentx: Evolving gui agents as proficient smartphone users. *arXiv preprint arXiv:2503.02268*, 2025.

Yuxuan Jiang, Chaoyun Zhang, Shilin He, Zhihao Yang, Minghua Ma, Si Qin, Yu Kang, Yingnong Dang, Saravan Rajmohan, Qingwei Lin, et al. Xpert: Empowering incident management with query recommendations via large language models. In *Proceedings of the IEEE/ACM 46th International Conference on Software Engineering*, pp. 1–13, 2024.

Yiqiao Jin, Stefano Petrangeli, Yu Shen, and Gang Wu. Screenllm: Stateful screen schema for efficient action understanding and prediction, 2025.

Kristiina Jokinen. User interaction in mobile navigation applications. In *Map-based Mobile Services: Design, Interaction and Usability*, pp. 168–197. Springer, 2008.

Christoforos Kachris. A survey on hardware accelerators for large language models. *arXiv preprint arXiv:2401.09890*, 2024.

Leslie Pack Kaelbling, Michael L Littman, and Andrew W Moore. Reinforcement learning: A survey. *Journal of artificial intelligence research*, 4:237–285, 1996.

Tomoyuki Kagaya, Thong Jing Yuan, Yuxuan Lou, Jayashree Karlekar, Sugiri Pranata, Akira Kinose, Koki Oguri, Felix Wick, and Yang You. Rap: Retrieval-augmented planning with contextual memory for multimodal llm agents. *arXiv preprint arXiv:2402.03610*, 2024.

Noam Kahlon, Guy Rom, Anatoly Efros, Filippo Galgani, Omri Berkovitch, Sapir Caduri, William E Bishop, Oriana Riva, and Ido Dagan. Agent-initiated interaction in phone ui automation. *arXiv preprint arXiv:2503.19537*, 2025.

Raghav Kapoor, Yash Parag Butala, Melisa Russak, Jing Yu Koh, Kiran Kamble, Waseem Alshikh, and Ruslan Salakhutdinov. Omniact: A dataset and benchmark for enabling multimodal generalist autonomous agents for desktop and web, 2024. URL `https://arxiv.org/abs/2402.17553`.

Su Kara, Fazle Faisal, and Suman Nath. Waber: Web agent benchmarking for efficiency and reliability. In *ICLR 2025 Workshop on Foundation Models in the Wild*.

katalon. Katalon studio: Easy test automation for web, api, mobile, and desktop, 2024. URL `https://katalon.com/katalon-studio`. Accessed: 2024-11-05.

Timo Kaufmann, Paul Weng, Viktor Bengs, and Eyke Hüllermeier. A survey of reinforcement learning from human feedback. *arXiv preprint arXiv:2312.14925*, 2023.

Tushar Khot, Harsh Trivedi, Matthew Finlayson, Yao Fu, Kyle Richardson, Peter Clark, and Ashish Sabharwal. Decomposed prompting: A modular approach for solving complex tasks. *arXiv preprint arXiv:2210.02406*, 2022.

Jihyung Kil, Chan Hee Song, Boyuan Zheng, Xiang Deng, Yu Su, and Wei-Lun Chao. Dual-view visual contextualization for web navigation, 2024. URL `https://arxiv.org/abs/2402.04476`.

Callie Y Kim, Christine P Lee, and Bilge Mutlu. Understanding large-language model (llm)-powered human-robot interaction. In *Proceedings of the 2024 ACM/IEEE International Conference on Human-Robot Interaction*, pp. 371–380, 2024a.

Geunwoo Kim, Pierre Baldi, and Stephen McAleer. Language models can solve computer tasks, 2023. URL `https://arxiv.org/abs/2303.17491`.

Jaekyeom Kim, Dong-Ki Kim, Lajanugen Logeswaran, Sungryull Sohn, and Honglak Lee. Auto-intent: Automated intent discovery and self-exploration for large language model web agents. *arXiv preprint arXiv:2410.22552*, 2024b.

Sein Kim, Hongseok Kang, Seungyoon Choi, Donghyun Kim, Minchul Yang, and Chanyoung Park. Large language models meet collaborative filtering: An efficient all-round llm-based recommender system. In *Proceedings of the 30th ACM SIGKDD Conference on Knowledge Discovery and Data Mining*, pp. 1395–1406, 2024c.

Alexander Kirillov, Eric Mintun, Nikhila Ravi, Hanzi Mao, Chloe Rolland, Laura Gustafson, Tete Xiao, Spencer Whitehead, Alexander C Berg, Wan-Yen Lo, et al. Segment anything. In *Proceedings of the IEEE/CVF International Conference on Computer Vision*, pp. 4015–4026, 2023.

Jing Yu Koh, Robert Lo, Lawrence Jang, Vikram Duvvur, Ming Chong Lim, Po-Yu Huang, Graham Neubig, Shuyan Zhou, Ruslan Salakhutdinov, and Daniel Fried. Visualwebarena: Evaluating multimodal agents on realistic visual web tasks, 2024a. URL `https://arxiv.org/abs/2401.13649`.

Jing Yu Koh, Stephen McAleer, Daniel Fried, and Ruslan Salakhutdinov. Tree search for language model agents. *arXiv preprint arXiv:2407.01476*, 2024b.

Qichao Kong, Zhengwei Lv, Yiheng Xiong, Jingling Sun, Ting Su, Dingchun Wang, Letao Li, Xu Yang, and Gang Huo. Prophetagent: Automatically synthesizing gui tests from test cases in natural language for mobile apps.

Richard Koo and Sam Toueg. Checkpointing and rollback-recovery for distributed systems. *IEEE Transactions on Software Engineering*, SE-13:23–31, 1986. URL https://api.semanticscholar.org/CorpusID:206777989.

Lars Krupp, Daniel Geißler, Paul Lukowicz, and Jakob Karolus. Towards sustainable web agents: A plea for transparency and dedicated metrics for energy consumption. *arXiv preprint arXiv:2502.17903*, 2025.

Weirui Kuang, Bingchen Qian, Zitao Li, Daoyuan Chen, Dawei Gao, Xuchen Pan, Yuexiang Xie, Yaliang Li, Bolin Ding, and Jingren Zhou. Federatedscope-llm: A comprehensive package for fine-tuning large language models in federated learning. In *Proceedings of the 30th ACM SIGKDD Conference on Knowledge Discovery and Data Mining*, pp. 5260–5271, 2024.

Hanyu Lai, Xiao Liu, Iat Long Iong, Shuntian Yao, Yuxuan Chen, Pengbo Shen, Hao Yu, Hanchen Zhang, Xiaohan Zhang, Yuxiao Dong, and Jie Tang. Autowebglm: Bootstrap and reinforce a large language model-based web navigating agent, 2024. URL https://arxiv.org/abs/2404.03648.

Hanyu Lai, Junjie Gao, Xiao Liu, Yifan Xu, Shudan Zhang, Yuxiao Dong, and Jie Tang. Androidgen: Building an android language agent under data scarcity. *arXiv preprint arXiv:2504.19298*, 2025.

Yuanhong Lan, Yifei Lu, Zhong Li, Minxue Pan, Wenhua Yang, Tian Zhang, and Xuandong Li. Deeply reinforcing android gui testing with deep reinforcement learning. In *Proceedings of the 46th IEEE/ACM International Conference on Software Engineering*, pp. 1–13, 2024.

Z Lan. Albert: A lite bert for self-supervised learning of language representations. *arXiv preprint arXiv:1909.11942*, 2019.

Hugo Laurençon, Léo Tronchon, Matthieu Cord, and Victor Sanh. What matters when building vision-language models? *arXiv preprint arXiv:2405.02246*, 2024a.

Hugo Laurençon, Léo Tronchon, and Victor Sanh. Unlocking the conversion of web screenshots into html code with the websight dataset. *arXiv preprint arXiv:2403.09029*, 2024b.

Hansoo Lee, Joonyoung Park, and Uichin Lee. A systematic survey on android api usage for data-driven analytics with smartphones. *ACM Computing Surveys*, 55(5):1–38, 2022.

Jungjae Lee, Dongjae Lee, Chihun Choi, Youngmin Im, Jaeyoung Wi, Kihong Heo, Sangeun Oh, Sunjae Lee, and Insik Shin. Safeguarding mobile gui agent via logic-based action verification. *arXiv preprint arXiv:2503.18492*, 2025.

Juyong Lee, Dongyoon Hahm, June Suk Choi, W Bradley Knox, and Kimin Lee. Mobilesafetybench: Evaluating safety of autonomous agents in mobile device control. *arXiv preprint arXiv:2410.17520*, 2024a.

Juyong Lee, Taywon Min, Minyong An, Changyeon Kim, and Kimin Lee. Benchmarking mobile device control agents across diverse configurations, 2024b. URL https://arxiv.org/abs/2404.16660.

Sunjae Lee, Junyoung Choi, Jungjae Lee, Munim Hasan Wasi, Hojun Choi, Steven Y Ko, Sangeun Oh, and Insik Shin. Explore, select, derive, and recall: Augmenting llm with human-like memory for mobile task automation. *arXiv preprint arXiv:2312.03003*, 2023.

Sunjae Lee, Junyoung Choi, Jungjae Lee, Munim Hasan Wasi, Hojun Choi, Steve Ko, Sangeun Oh, and Insik Shin. Mobilegpt: Augmenting llm with human-like app memory for mobile task automation. In *Proceedings of the 30th Annual International Conference on Mobile Computing and Networking*, pp. 1119–1133, 2024c.

Wonbeom Lee, Jungi Lee, Junghwan Seo, and Jaewoong Sim. {InfiniGen}: Efficient generative inference of large language models with dynamic {KV} cache management. In *18th USENIX Symposium on Operating Systems Design and Implementation (OSDI 24)*, pp. 155–172, 2024d.

Ido Levy, Ben Wiesel, Sami Marreed, Alon Oved, Avi Yaeli, and Segev Shlomov. St-webagentbench: A benchmark for evaluating safety and trustworthiness in web agents. *arXiv preprint arXiv:2410.06703*, 2024.

M Lewis. Bart: Denoising sequence-to-sequence pre-training for natural language generation, translation, and comprehension. *arXiv preprint arXiv:1910.13461*, 2019.

Patrick Lewis, Ethan Perez, Aleksandra Piktus, Fabio Petroni, Vladimir Karpukhin, Naman Goyal, Heinrich Küttler, Mike Lewis, Wen-tau Yih, Tim Rocktäschel, et al. Retrieval-augmented generation for knowledge-intensive nlp tasks. *Advances in Neural Information Processing Systems*, 33:9459–9474, 2020.

Baolin Li, Yankai Jiang, Vijay Gadepally, and Devesh Tiwari. Llm inference serving: Survey of recent advances and opportunities. *arXiv preprint arXiv:2407.12391*, 2024a.

Dongxu Li, Yudong Liu, Haoning Wu, Yue Wang, Zhiqi Shen, Bowen Qu, Xinyao Niu, Guoyin Wang, Bei Chen, and Junnan Li. Aria: An open multimodal native mixture-of-experts model. *arXiv preprint arXiv:2410.05993*, 2024b.

Eric Li and Jim Waldo. Websuite: Systematically evaluating why web agents fail. *arXiv preprint arXiv:2406.01623*, 2024.

Guohao Li, Hasan Abed Al Kader Hammoud, Hani Itani, Dmitrii Khizbullin, and Bernard Ghanem. Camel: Communicative agents for "mind" exploration of large language model society. In *Thirty-seventh Conference on Neural Information Processing Systems*, 2023a.

Hao Li, Chenghao Yang, An Zhang, Yang Deng, Xiang Wang, and Tat-Seng Chua. Hello again! llm-powered personalized agent for long-term dialogue. *arXiv preprint arXiv:2406.05925*, 2024c.

Haoyuan Li, Hao Jiang, Tianke Zhang, Zhelun Yu, Aoxiong Yin, Hao Cheng, Siming Fu, Yuhao Zhang, and Wanggui He. Traineragent: Customizable and efficient model training through llm-powered multi-agent system. *arXiv preprint arXiv:2311.06622*, 2023b.

Hongxin Li, Jingfan Chen, Jingran Su, Yuntao Chen, Qing Li, and Zhaoxiang Zhang. Autogui: Scaling gui grounding with automatic functionality annotations from llms. *arXiv preprint arXiv:2502.01977*, 2025a.

Junnan Li, Dongxu Li, Silvio Savarese, and Steven Hoi. Blip-2: Bootstrapping language-image pre-training with frozen image encoders and large language models. In *International conference on machine learning*, pp. 19730–19742. PMLR, 2023c.

Kaixin Li, Ziyang Meng, Hongzhan Lin, Ziyang Luo, Yuchen Tian, Jing Ma, Zhiyong Huang, and Tat-Seng Chua. Screenspot-pro: Gui grounding for professional high-resolution computer use, 2025b.

Kanglin Li and Mengqi Wu. *Effective GUI testing automation: Developing an automated GUI testing tool*. John Wiley & Sons, 2006.

Lin Li, Guikun Chen, Hanrong Shi, Jun Xiao, and Long Chen. A survey on multimodal benchmarks: In the era of large ai models. *arXiv preprint arXiv:2409.18142*, 2024d.

Tao Li, Gang Li, Zhiwei Deng, Bryan Wang, and Yang Li. A zero-shot language agent for computer control with structured reflection. *arXiv preprint arXiv:2310.08740*, 2023d.

Toby Jia-Jun Li, Lindsay Popowski, Tom Mitchell, and Brad A Myers. Screen2vec: Semantic embedding of gui screens and gui components. In *Proceedings of the 2021 CHI Conference on Human Factors in Computing Systems*, pp. 1–15, 2021.

Wei Li, William Bishop, Alice Li, Chris Rawles, Folawiyo Campbell-Ajala, Divya Tyamagundlu, and Oriana Riva. On the effects of data scale on computer control agents. *arXiv preprint arXiv:2406.03679*, 2024e.

Wei Li, Fu-Lin Hsu, William Bishop, Folawiyo Campbell-Ajala, Max Lin, and Oriana Riva. Uinav: A practical approach to train on-device automation agents. In *Proceedings of the 2024 Conference of the North American Chapter of the Association for Computational Linguistics: Human Language Technologies (Volume 6: Industry Track)*, pp. 36–51, 2024f.

Xiaolei Li, Jialun Cao, Yepang Liu, Shing-Chi Cheung, and Hailong Wang. Reusedroid: A vlm-empowered android ui test migrator boosted by active feedback. *arXiv preprint arXiv:2504.02357*, 2025c.

Xuan Li. Gui testing for android applications: a survey. In *2023 7th International Conference on Computer, Software and Modeling (ICCSM)*, pp. 6–10. IEEE, 2023.

Yanda Li, Chi Zhang, Wanqi Yang, Bin Fu, Pei Cheng, Xin Chen, Ling Chen, and Yunchao Wei. Appagent v2: Advanced agent for flexible mobile interactions, 2024g. URL https://arxiv.org/abs/2408.11824.

Yang Li and Otmar Hilliges. *Artificial intelligence for human computer interaction: a modern approach*. Springer, 2021.

Yang Li, Jiacong He, Xin Zhou, Yuan Zhang, and Jason Baldridge. Mapping natural language instructions to mobile ui action sequences. In *Proceedings of the 58th Annual Meeting of the Association for Computational Linguistics*, pp. 8198–8210, 2020a.

Yang Li, Gang Li, Luheng He, Jingjie Zheng, Hong Li, and Zhiwei Guan. Widget captioning: Generating natural language description for mobile user interface elements. *arXiv preprint arXiv:2010.04295*, 2020b.

Yingji Li, Mengnan Du, Rui Song, Xin Wang, and Ying Wang. A survey on fairness in large language models. *arXiv preprint arXiv:2308.10149*, 2023e.

Youwei Li, Yangyang Li, and Yangzhao Yang. Test-agent: A multimodal app automation testing framework based on the large language model. In *2024 IEEE 4th International Conference on Digital Twins and Parallel Intelligence (DTPI)*, pp. 609–614. IEEE, 2024h.

Yuanchun Li, Ziyue Yang, Yao Guo, and Xiangqun Chen. Humanoid: A deep learning-based approach to automated black-box android app testing. In *2019 34th IEEE/ACM International Conference on Automated Software Engineering (ASE)*, pp. 1070–1073. IEEE, 2019.

Yuanchun Li, Hao Wen, Weijun Wang, Xiangyu Li, Yizhen Yuan, Guohong Liu, Jiacheng Liu, Wenxing Xu, Xiang Wang, Yi Sun, et al. Personal llm agents: Insights and survey about the capability, efficiency and security. *arXiv preprint arXiv:2401.05459*, 2024i.

Zhangheng Li, Keen You, Haotian Zhang, Di Feng, Harsh Agrawal, Xiujun Li, Mohana Prasad Sathya Moorthy, Jeff Nichols, Yinfei Yang, and Zhe Gan. Ferret-ui 2: Mastering universal user interface understanding across platforms. *arXiv preprint arXiv:2410.18967*, 2024j.

Zongxia Li, Xiyang Wu, Hongyang Du, Huy Nghiem, and Guangyao Shi. Benchmark evaluations, applications, and challenges of large vision language models: A survey. *arXiv preprint arXiv:2501.02189*, 2025d.

Zeyi Liao, Lingbo Mo, Chejian Xu, Mintong Kang, Jiawei Zhang, Chaowei Xiao, Yuan Tian, Bo Li, and Huan Sun. Eia: Environmental injection attack on generalist web agents for privacy leakage. *arXiv preprint arXiv:2409.11295*, 2024.

Ji Lin, Jiaming Tang, Haotian Tang, Shang Yang, Wei-Ming Chen, Wei-Chen Wang, Guangxuan Xiao, Xingyu Dang, Chuang Gan, and Song Han. Awq: Activation-aware weight quantization for on-device llm compression and acceleration. *Proceedings of Machine Learning and Systems*, 6:87–100, 2024a.

Kevin Qinghong Lin, Pengchuan Zhang, Joya Chen, Shraman Pramanick, Difei Gao, Alex Jinpeng Wang, Rui Yan, and Mike Zheng Shou. Univtg: Towards unified video-language temporal grounding. In *Proceedings of the IEEE/CVF International Conference on Computer Vision*, pp. 2794–2804, 2023.

Kevin Qinghong Lin, Linjie Li, Difei Gao, Qinchen WU, Mingyi Yan, Zhengyuan Yang, Lijuan Wang, and Mike Zheng Shou. Videogui: A benchmark for gui automation from instructional videos, 2024b. URL https://arxiv.org/abs/2406.10227.

Kevin Qinghong Lin, Linjie Li, Difei Gao, Zhengyuan Yang, Shiwei Wu, Zechen Bai, Weixian Lei, Lijuan Wang, and Mike Zheng Shou. Showui: One vision-language-action model for gui visual agent, 2024c. URL https://arxiv.org/abs/2411.17465.

Luyang Lin, Lingzhi Wang, Jinsong Guo, and Kam-Fai Wong. Investigating bias in llm-based bias detection: Disparities between llms and human perception. *arXiv preprint arXiv:2403.14896*, 2024d.

Evan Zheran Liu, Kelvin Guu, Panupong Pasupat, Tianlin Shi, and Percy Liang. Reinforcement learning on web interfaces using workflow-guided exploration, 2018. URL `https://arxiv.org/abs/1802.08802`.

Guangyi Liu, Pengxiang Zhao, Liang Liu, Zhiming Chen, Yuxiang Chai, Shuai Ren, Hao Wang, Shibo He, and Wenchao Meng. Learnact: Few-shot mobile gui agent with a unified demonstration benchmark, 2025a. URL `https://arxiv.org/abs/2504.13805`.

Guangyi Liu, Pengxiang Zhao, Liang Liu, Yaxuan Guo, Han Xiao, Weifeng Lin, Yuxiang Chai, Yue Han, Shuai Ren, Hao Wang, et al. Llm-powered gui agents in phone automation: Surveying progress and prospects. *arXiv preprint arXiv:2504.19838*, 2025b.

Haotian Liu, Chunyuan Li, Yuheng Li, and Yong Jae Lee. Improved baselines with visual instruction tuning. In *Proceedings of the IEEE/CVF Conference on Computer Vision and Pattern Recognition*, pp. 26296–26306, 2024a.

Haotian Liu, Chunyuan Li, Qingyang Wu, and Yong Jae Lee. Visual instruction tuning. *Advances in neural information processing systems*, 36, 2024b.

Haowei Liu, Xi Zhang, Haiyang Xu, Yuyang Wanyan, Junyang Wang, Ming Yan, Ji Zhang, Chunfeng Yuan, Changsheng Xu, Weiming Hu, et al. Pc-agent: A hierarchical multi-agent collaboration framework for complex task automation on pc. *arXiv preprint arXiv:2502.14282*, 2025c.

Jiarun Liu, Jia Hao, Chunhong Zhang, and Zheng Hu. Wepo: Web element preference optimization for llm-based web navigation, 2024c. URL `https://arxiv.org/abs/2412.10742`.

Jun Liu, Chaoyun Zhang, Jiaxu Qian, Minghua Ma, Si Qin, Chetan Bansal, Qingwei Lin, Saravan Rajmohan, and Dongmei Zhang. Large language models can deliver accurate and interpretable time series anomaly detection. *arXiv preprint arXiv:2405.15370*, 2024d.

Junpeng Liu, Tianyue Ou, Yifan Song, Yuxiao Qu, Wai Lam, Chenyan Xiong, Wenhu Chen, Graham Neubig, and Xiang Yue. Harnessing webpage uis for text-rich visual understanding. *arXiv preprint arXiv:2410.13824*, 2024e.

Junpeng Liu, Yifan Song, Bill Yuchen Lin, Wai Lam, Graham Neubig, Yuanzhi Li, and Xiang Yue. Visual-webbench: How far have multimodal llms evolved in web page understanding and grounding?, 2024f. URL `https://arxiv.org/abs/2404.05955`.

Junwei Liu, Kaixin Wang, Yixuan Chen, Xin Peng, Zhenpeng Chen, Lingming Zhang, and Yiling Lou. Large language model-based agents for software engineering: A survey. *arXiv preprint arXiv:2409.02977*, 2024g.

Shilong Liu, Zhaoyang Zeng, Tianhe Ren, Feng Li, Hao Zhang, Jie Yang, Qing Jiang, Chunyuan Li, Jianwei Yang, Hang Su, et al. Grounding dino: Marrying dino with grounded pre-training for open-set object detection. *arXiv preprint arXiv:2303.05499*, 2023a.

Xiao Liu, Hao Yu, Hanchen Zhang, Yifan Xu, Xuanyu Lei, Hanyu Lai, Yu Gu, Hangliang Ding, Kaiwen Men, Kejuan Yang, et al. Agentbench: Evaluating llms as agents. *arXiv preprint arXiv:2308.03688*, 2023b.

Xiao Liu, Bo Qin, Dongzhu Liang, Guang Dong, Hanyu Lai, Hanchen Zhang, Hanlin Zhao, Iat Long Iong, Jiadai Sun, Jiaqi Wang, et al. Autoglm: Autonomous foundation agents for guis. *arXiv preprint arXiv:2411.00820*, 2024h.

Xiao Liu, Tianjie Zhang, Yu Gu, Iat Long Iong, Yifan Xu, Xixuan Song, Shudan Zhang, Hanyu Lai, Xinyi Liu, Hanlin Zhao, Jiadai Sun, Xinyue Yang, Yu Yang, Zehan Qi, Shuntian Yao, Xueqiao Sun, Siyi Cheng, Qinkai Zheng, Hao Yu, Hanchen Zhang, Wenyi Hong, Ming Ding, Lihang Pan, Xiaotao Gu, Aohan Zeng, Zhengxiao Du, Chan Hee Song, Yu Su, Yuxiao Dong, and Jie Tang. Visualagentbench: Towards large multimodal models as visual foundation agents, 2024i. URL `https://arxiv.org/abs/2408.06327`.

Xinyi Liu, Xiaoyi Zhang, Ziyun Zhang, and Yan Lu. Ui-e2i-synth: Advancing gui grounding with large-scale instruction synthesis. *arXiv preprint arXiv:2504.11257*, 2025d.

Yinhan Liu. Roberta: A robustly optimized bert pretraining approach. *arXiv preprint arXiv:1907.11692*, 364, 2019.

Yuhang Liu, Pengxiang Li, Zishu Wei, Congkai Xie, Xueyu Hu, Xinchen Xu, Shengyu Zhang, Xiaotian Han, Hongxia Yang, and Fei Wu. Infiguiagent: A multimodal generalist gui agent with native reasoning and reflection, 2025e. URL `https://arxiv.org/abs/2501.04575`.

Yuhang Liu, Pengxiang Li, Congkai Xie, Xavier Hu, Xiaotian Han, Shengyu Zhang, Hongxia Yang, and Fei Wu. Infigui-r1: Advancing multimodal gui agents from reactive actors to deliberative reasoners, 2025f. URL `https://arxiv.org/abs/2504.14239`.

Yuxuan Liu, Hongda Sun, Wei Liu, Jian Luan, Bo Du, and Rui Yan. Mobilesteward: Integrating multiple app-oriented agents with self-evolution to automate cross-app instructions. *arXiv preprint arXiv:2502.16796*, 2025g.

Ze Liu, Yutong Lin, Yue Cao, Han Hu, Yixuan Wei, Zheng Zhang, Stephen Lin, and Baining Guo. Swin transformer: Hierarchical vision transformer using shifted windows. In *Proceedings of the IEEE/CVF international conference on computer vision*, pp. 10012–10022, 2021.

Zechun Liu, Changsheng Zhao, Forrest Iandola, Chen Lai, Yuandong Tian, Igor Fedorov, Yunyang Xiong, Ernie Chang, Yangyang Shi, Raghuraman Krishnamoorthi, et al. Mobilellm: Optimizing sub-billion parameter language models for on-device use cases. *arXiv preprint arXiv:2402.14905*, 2024j.

Zhe Liu, Chunyang Chen, Junjie Wang, Xing Che, Yuekai Huang, Jun Hu, and Qing Wang. Fill in the blank: Context-aware automated text input generation for mobile gui testing. In *2023 IEEE/ACM 45th International Conference on Software Engineering (ICSE)*, pp. 1355–1367. IEEE, 2023c.

Zhe Liu, Chunyang Chen, Junjie Wang, Mengzhuo Chen, Boyu Wu, Xing Che, Dandan Wang, and Qing Wang. Make llm a testing expert: Bringing human-like interaction to mobile gui testing via functionality-aware decisions. In *Proceedings of the IEEE/ACM 46th International Conference on Software Engineering*, pp. 1–13, 2024k.

Zhe Liu, Cheng Li, Chunyang Chen, Junjie Wang, Boyu Wu, Yawen Wang, Jun Hu, and Qing Wang. Vision-driven automated mobile gui testing via multimodal large language model, 2024l. URL `https://arxiv.org/abs/2407.03037`.

Zhuang Liu, Hanzi Mao, Chao-Yuan Wu, Christoph Feichtenhofer, Trevor Darrell, and Saining Xie. A convnet for the 2020s. In *Proceedings of the IEEE/CVF conference on computer vision and pattern recognition*, pp. 11976–11986, 2022.

Fanbin Lu, Zhisheng Zhong, Ziqin Wei, Shu Liu, Chi-Wing Fu, and Jiaya Jia. Steve: Astep verification pipeline for computer-use agent training. *arXiv preprint arXiv:2503.12532*, 2025.

Junru Lu, Siyu An, Mingbao Lin, Gabriele Pergola, Yulan He, Di Yin, Xing Sun, and Yunsheng Wu. Memochat: Tuning llms to use memos for consistent long-range open-domain conversation. *arXiv preprint arXiv:2308.08239*, 2023.

Junting Lu, Zhiyang Zhang, Fangkai Yang, Jue Zhang, Lu Wang, Chao Du, Qingwei Lin, Saravan Rajmohan, Dongmei Zhang, and Qi Zhang. Turn every application into an agent: Towards efficient human-agent-computer interaction with api-first llm-based agents. *arXiv preprint arXiv:2409.17140*, 2024a.

Quanfeng Lu, Wenqi Shao, Zitao Liu, Fanqing Meng, Boxuan Li, Botong Chen, Siyuan Huang, Kaipeng Zhang, Yu Qiao, and Ping Luo. Gui odyssey: A comprehensive dataset for cross-app gui navigation on mobile devices, 2024b. URL `https://arxiv.org/abs/2406.08451`.

Xing Han Lu, Zdeněk Kasner, and Siva Reddy. Weblinx: Real-world website navigation with multi-turn dialogue. In *International Conference on Machine Learning*, pp. 33007–33056. PMLR, 2024c.

Xing Han Lù, Amirhossein Kazemnejad, Nicholas Meade, Arkil Patel, Dongchan Shin, Alejandra Zambrano, Karolina Stańczak, Peter Shaw, Christopher J Pal, and Siva Reddy. Agentrewardbench: Evaluating automatic evaluations of web agent trajectories. *arXiv preprint arXiv:2504.08942*, 2025.

Yadong Lu, Jianwei Yang, Yelong Shen, and Ahmed Awadallah. Omniparser for pure vision based gui agent. *arXiv preprint arXiv:2408.00203*, 2024d.

Yuwen Lu, Yuewen Yang, Qinyi Zhao, Chengzhi Zhang, and Toby Jia-Jun Li. Ai assistance for ux: A literature review through human-centered ai. *arXiv preprint arXiv:2402.06089*, 2024e.

Yuxuan Lu, Bingsheng Yao, Hansu Gu, Jing Huang, Jessie Wang, Laurence Li, Jiri Gesi, Qi He, Toby Jia-Jun Li, and Dakuo Wang. Uxagent: An llm agent-based usability testing framework for web design. *arXiv preprint arXiv:2502.12561*, 2025a.

Zhengxi Lu, Yuxiang Chai, Yaxuan Guo, Xi Yin, Liang Liu, Hao Wang, Guanjing Xiong, and Hongsheng Li. Ui-r1: Enhancing action prediction of gui agents by reinforcement learning. *arXiv preprint arXiv:2503.21620*, 2025b.

Dezhao Luo, Bohan Tang, Kang Li, Georgios Papoudakis, Jifei Song, Shaogang Gong, Jianye Hao, Jun Wang, and Kun Shao. Vimo: A generative visual gui world model for app agent, 2025a. URL `https://arxiv.org/abs/2504.13936`.

Tiange Luo, Lajanugen Logeswaran, Justin Johnson, and Honglak Lee. Visual test-time scaling for gui agent grounding, 2025b. URL `https://arxiv.org/abs/2505.00684`.

Yang Luo, Qixun Zhang, Qingni Shen, Hongzhi Liu, and Zhonghai Wu. Android multi-level system permission management approach. *ArXiv*, abs/1712.02217, 2017. URL `https://api.semanticscholar.org/CorpusID:20909985`.

Michael Lutaaya. Rethinking app permissions on ios. In *Extended Abstracts of the 2018 CHI Conference on Human Factors in Computing Systems*, pp. 1–6, 2018.

Chang Ma, Junlei Zhang, Zhihao Zhu, Cheng Yang, Yujiu Yang, Yaohui Jin, Zhenzhong Lan, Lingpeng Kong, and Junxian He. Agentboard: An analytical evaluation board of multi-turn llm agents. *arXiv preprint arXiv:2401.13178*, 2024a.

Kaixin Ma, Hongming Zhang, Hongwei Wang, Xiaoman Pan, Wenhao Yu, and Dong Yu. Laser: Llm agent with state-space exploration for web navigation. *arXiv preprint arXiv:2309.08172*, 2023.

Kaixin Ma, Hongming Zhang, Hongwei Wang, Xiaoman Pan, Wenhao Yu, and Dong Yu. Laser: Llm agent with state-space exploration for web navigation, 2024b. URL `https://arxiv.org/abs/2309.08172`.

Xinbei Ma, Yiting Wang, Yao Yao, Tongxin Yuan, Aston Zhang, Zhuosheng Zhang, and Hai Zhao. Caution for the environment: Multimodal agents are susceptible to environmental distractions, 2024c. URL `https://arxiv.org/abs/2408.02544`.

Xinbei Ma, Zhuosheng Zhang, and Hai Zhao. Coco-agent: A comprehensive cognitive mllm agent for smartphone gui automation, 2024d. URL `https://arxiv.org/abs/2402.11941`.

Peihua Mai, Ran Yan, Zhe Huang, Youjia Yang, and Yan Pang. Split-and-denoise: Protect large language model inference with local differential privacy. *arXiv preprint arXiv:2310.09130*, 2023.

Ben Mann, N Ryder, M Subbiah, J Kaplan, P Dhariwal, A Neelakantan, P Shyam, G Sastry, A Askell, S Agarwal, et al. Language models are few-shot learners. *arXiv preprint arXiv:2005.14165*, 1, 2020.

Pedro Martins, Filipe Sá, Francisco Morgado, and Carlos Cunha. Using machine learning for cognitive robotic process automation (rpa). In *2020 15th Iberian Conference on Information Systems and Technologies (CISTI)*, pp. 1–6. IEEE, 2020.

Tula Masterman, Sandi Besen, Mason Sawtell, and Alex Chao. The landscape of emerging ai agent architectures for reasoning, planning, and tool calling: A survey. *arXiv preprint arXiv:2404.11584*, 2024.

Sahisnu Mazumder and Oriana Riva. Flin: A flexible natural language interface for web navigation. *arXiv preprint arXiv:2010.12844*, 2020.

Larry R Medsker, Lakhmi Jain, et al. Recurrent neural networks. *Design and Applications*, 5(64-67):2, 2001.

A. Memon, I. Banerjee, N. Hashmi, and A. Nagarajan. Dart: a framework for regression testing "nightly/daily builds" of gui applications. In *International Conference on Software Maintenance, 2003. ICSM 2003. Proceedings.*, pp. 410–419, 2003a. doi: 10.1109/ICSM.2003.1235451.

Atif M Memon, Martha E Pollack, and Mary Lou Soffa. Hierarchical gui test case generation using automated planning. *IEEE transactions on software engineering*, 27(2):144–155, 2001.

Atif M Memon, Ishan Banerjee, and Adithya Nagarajan. Gui ripping: reverse engineering of graphical user interfaces for testing. In *WCRE*, volume 3, pp. 260, 2003b.

Ziyang Meng, Yu Dai, Zezheng Gong, Shaoxiong Guo, Minglong Tang, and Tongquan Wei. Vga: Vision gui assistant – minimizing hallucinations through image-centric fine-tuning, 2024. URL `https://arxiv.org/abs/2406.14056`.

Rafał Michalski, Jerzy Grobelny, and Waldemar Karwowski. The effects of graphical interface design characteristics on human-computer interaction task efficiency. *ArXiv*, abs/1211.6712, 2006. URL `https://api.semanticscholar.org/CorpusID:14695409`.

Microsoft. Create desktop flows using record with copilot (preview), 2024. URL `https://learn.microsoft.com/en-us/power-automate/desktop-flows/create-flow-using-ai-recorder`. Accessed: 2024-11-16.

Sushmita Mitra and Tinku Acharya. Gesture recognition: A survey. *IEEE Transactions on Systems, Man, and Cybernetics, Part C (Applications and Reviews)*, 37(3):311–324, 2007.

Kevin Moran, Cody Watson, John Hoskins, George Purnell, and Denys Poshyvanyk. Detecting and summarizing gui changes in evolving mobile apps. In *Proceedings of the 33rd ACM/IEEE international conference on automated software engineering*, pp. 543–553, 2018.

MosaicML. Mosaicml: Mpt-7b, 2023. URL `https://www.mosaicml.com/blog/mpt-7b`. Accessed: 2024-11-19.

Thiago Santos de Moura, Everton LG Alves, Hugo Feitosa de Figueirêdo, and Cláudio de Souza Baptista. Cytestion: Automated gui testing for web applications. In *Proceedings of the XXXVII Brazilian Symposium on Software Engineering*, pp. 388–397, 2023.

MultiOn AI. Multion ai: Ai agents that act on your behalf, 2024. URL `https://www.multion.ai/`. Accessed: 2024-10-26.

Shikhar Murty, Dzmitry Bahdanau, and Christopher D Manning. Nnetscape navigator: Complex demonstrations for web agents without a demonstrator. *arXiv preprint arXiv:2410.02907*, 2024.

Michel Nass, Emil Alégroth, and Robert Feldt. Why many challenges with gui test automation (will) remain. *Information and Software Technology*, 138:106625, 2021.

Humza Naveed, Asad Ullah Khan, Shi Qiu, Muhammad Saqib, Saeed Anwar, Muhammad Usman, Naveed Akhtar, Nick Barnes, and Ajmal Mian. A comprehensive overview of large language models. *arXiv preprint arXiv:2307.06435*, 2023.

Shravan Nayak, Xiangru Jian, Kevin Qinghong Lin, Juan A Rodriguez, Montek Kalsi, Rabiul Awal, Nicolas Chapados, M Tamer Özsu, Aishwarya Agrawal, David Vazquez, et al. Ui-vision: A desktop-centric gui benchmark for visual perception and interaction. *arXiv preprint arXiv:2503.15661*, 2025.

Anthony Nguyen. Improved gui grounding via iterative narrowing, 2024. URL https://arxiv.org/abs/2411.13591.

Dang Nguyen, Jian Chen, Yu Wang, Gang Wu, Namyong Park, Zhengmian Hu, Hanjia Lyu, Junda Wu, Ryan Aponte, Yu Xia, Xintong Li, Jing Shi, Hongjie Chen, Viet Dac Lai, Zhouhang Xie, Sungchul Kim, Ruiyi Zhang, Tong Yu, Mehrab Tanjim, Nesreen K. Ahmed, Puneet Mathur, Seunghyun Yoon, Lina Yao, Branislav Kveton, Thien Huu Nguyen, Trung Bui, Tianyi Zhou, Ryan A. Rossi, and Franck Dernoncourt. Gui agents: A survey, 2024. URL https://arxiv.org/abs/2412.13501.

Liangbo Ning, Ziran Liang, Zhuohang Jiang, Haohao Qu, Yujuan Ding, Wenqi Fan, Xiao-yong Wei, Shanru Lin, Hui Liu, Philip S Yu, et al. A survey of webagents: Towards next-generation ai agents for web automation with large foundation models. *arXiv preprint arXiv:2503.23350*, 2025.

Runliang Niu, Jindong Li, Shiqi Wang, Yali Fu, Xiyu Hu, Xueyuan Leng, He Kong, Yi Chang, and Qi Wang. Screenagent: A vision language model-driven computer control agent, 2024. URL https://arxiv.org/abs/2402.07945.

Songqin Nong, Jiali Zhu, Rui Wu, Jiongchao Jin, Shuo Shan, Xiutian Huang, and Wenhao Xu. Mobileflow: A multimodal llm for mobile gui agent, 2024. URL https://arxiv.org/abs/2407.04346.

Juho-Jaakko Oksanen. Test automation for windows gui application. 2023.

OpenAdapt AI. OpenAdapt: Open Source Generative Process Automation, 2024. URL https://github.com/OpenAdaptAI/OpenAdapt. Accessed: 2024-10-26.

OpenAI. Gpt-4v(ision) system card. Technical report, OpenAI, September 2023. URL https://cdn.openai.com/papers/GPTV_System_Card.pdf.

OpenAI. Computer-using agent: Introducing a universal interface for ai to interact with the digital world. 2025a. URL https://openai.com/index/computer-using-agent.

OpenAI. Operator system card, January 2025b. Released on January 23, 2025.

Maxime Oquab, Timothée Darcet, Théo Moutakanni, Huy Vo, Marc Szafraniec, Vasil Khalidov, Pierre Fernandez, Daniel Haziza, Francisco Massa, Alaaeldin El-Nouby, et al. Dinov2: Learning robust visual features without supervision. *arXiv preprint arXiv:2304.07193*, 2023.

Long Ouyang, Jeffrey Wu, Xu Jiang, Diogo Almeida, Carroll Wainwright, Pamela Mishkin, Chong Zhang, Sandhini Agarwal, Katarina Slama, Alex Ray, et al. Training language models to follow instructions with human feedback. *Advances in neural information processing systems*, 35:27730–27744, 2022.

Vardaan Pahuja, Yadong Lu, Corby Rosset, Boyu Gou, Arindam Mitra, Spencer Whitehead, Yu Su, and Ahmed Awadallah. Explorer: Scaling exploration-driven web trajectory synthesis for multimodal web agents. *arXiv preprint arXiv:2502.11357*, 2025.

Jiayi Pan, Yichi Zhang, Nicholas Tomlin, Yifei Zhou, Sergey Levine, and Alane Suhr. Autonomous evaluation and refinement of digital agents. In *First Conference on Language Modeling*, 2024a.

Liangming Pan, Michael Stephen Saxon, Wenda Xu, Deepak Nathani, Xinyi Wang, and William Yang Wang. Automatically correcting large language models: Surveying the landscape of diverse self-correction strategies. *ArXiv*, abs/2308.03188, 2023a. URL https://api.semanticscholar.org/CorpusID:260682695.

Lihang Pan, Bowen Wang, Chun Yu, Yuxuan Chen, Xiangyu Zhang, and Yuanchun Shi. Autotask: Executing arbitrary voice commands by exploring and learning from mobile gui. *arXiv preprint arXiv:2312.16062*, 2023b.

Yichen Pan, Dehan Kong, Sida Zhou, Cheng Cui, Yifei Leng, Bing Jiang, Hangyu Liu, Yanyi Shang, Shuyan Zhou, Tongshuang Wu, and Zhengyang Wu. Webcanvas: Benchmarking web agents in online environments, 2024b. URL https://arxiv.org/abs/2406.12373.

Georgios Papoudakis, Thomas Coste, Zhihao Wu, Jianye Hao, Jun Wang, and Kun Shao. Appvlm: A lightweight vision language model for online app control. *arXiv preprint arXiv:2502.06395*, 2025.

Pawel Pawlowski, Krystian Zawistowski, Wojciech Lapacz, Marcin Skorupa, Adam Wiacek, Sebastien Postansque, and Jakub Hoscilowicz. Tinyclick: Single-turn agent for empowering gui automation. *arXiv preprint arXiv:2410.11871*, 2024.

Andrés Piñeiro-Martín, Carmen García-Mateo, Laura Docío-Fernández, and Maria Del Carmen Lopez-Perez. Ethical challenges in the development of virtual assistants powered by large language models. *Electronics*, 12(14):3170, 2023.

Martin L Puterman. Markov decision processes. *Handbooks in operations research and management science*, 2:331–434, 1990.

Pranav Putta, Edmund Mills, Naman Garg, Sumeet Motwani, Chelsea Finn, Divyansh Garg, and Rafael Rafailov. Agent q: Advanced reasoning and learning for autonomous ai agents. *arXiv preprint arXiv:2408.07199*, 2024.

Zehan Qi, Xiao Liu, Iat Long Iong, Hanyu Lai, Xueqiao Sun, Xinyue Yang, Jiadai Sun, Yu Yang, Shuntian Yao, Tianjie Zhang, Wei Xu, Jie Tang, and Yuxiao Dong. Webrl: Training llm web agents via self-evolving online curriculum reinforcement learning, 2024. URL https://arxiv.org/abs/2411.02337.

Ju Qian, Zhengyu Shang, Shuoyan Yan, Yan Wang, and Lin Chen. Roscript: A visual script driven truly non-intrusive robotic testing system for touch screen applications. In *2020 IEEE/ACM 42nd International Conference on Software Engineering (ICSE)*, pp. 297–308, 2020.

Ju Qian, Yingwei Ma, Chenghao Lin, and Lin Chen. Accelerating ocr-based widget localization for test automation of gui applications. In *Proceedings of the 37th IEEE/ACM International Conference on Automated Software Engineering*, pp. 1–13, 2022.

Yijun Qian, Yujie Lu, Alexander Hauptmann, and Oriana Riva. Visual grounding for user interfaces. In Yi Yang, Aida Davani, Avi Sil, and Anoop Kumar (eds.), *Proceedings of the 2024 Conference of the North American Chapter of the Association for Computational Linguistics: Human Language Technologies (Volume 6: Industry Track)*, pp. 97–107, Mexico City, Mexico, June 2024. Association for Computational Linguistics. doi: 10.18653/v1/2024.naacl-industry.9. URL https://aclanthology.org/2024.naacl-industry.9.

Bo Qiao, Liqun Li, Xu Zhang, Shilin He, Yu Kang, Chaoyun Zhang, Fangkai Yang, Hang Dong, Jue Zhang, Lu Wang, et al. Taskweaver: A code-first agent framework. *arXiv preprint arXiv:2311.17541*, 2023.

Yujia Qin, Yining Ye, Junjie Fang, Haoming Wang, Shihao Liang, Shizuo Tian, Junda Zhang, Jiahao Li, Yunxin Li, Shijue Huang, Wanjun Zhong, Kuanye Li, Jiale Yang, Yu Miao, Woyu Lin, Longxiang Liu, Xu Jiang, Qianli Ma, Jingyu Li, Xiaojun Xiao, Kai Cai, Chuang Li, Yaowei Zheng, Chaolin Jin, Chen Li, Xiao Zhou, Minchao Wang, Haoli Chen, Zhaojian Li, Haihua Yang, Haifeng Liu, Feng Lin, Tao Peng, Xin Liu, and Guang Shi. Ui-tars: Pioneering automated gui interaction with native agents, 2025. URL https://arxiv.org/abs/2501.12326.

Guanqiao Qu, Qiyuan Chen, Wei Wei, Zheng Lin, Xianhao Chen, and Kaibin Huang. Mobile edge intelligence for large language models: A contemporary survey. *arXiv preprint arXiv:2407.18921*, 2024.

Alec Radford. Improving language understanding by generative pre-training. 2018.

Alec Radford, Jeffrey Wu, Rewon Child, David Luan, Dario Amodei, Ilya Sutskever, et al. Language models are unsupervised multitask learners. *OpenAI blog*, 1(8):9, 2019.

Alec Radford, Jong Wook Kim, Chris Hallacy, Aditya Ramesh, Gabriel Goh, Sandhini Agarwal, Girish Sastry, Amanda Askell, Pamela Mishkin, Jack Clark, et al. Learning transferable visual models from natural language supervision. In *International conference on machine learning*, pp. 8748–8763. PMLR, 2021.

Rafael Rafailov, Archit Sharma, Eric Mitchell, Christopher D Manning, Stefano Ermon, and Chelsea Finn. Direct preference optimization: Your language model is secretly a reward model. *Advances in Neural Information Processing Systems*, 36, 2024.

Colin Raffel, Noam Shazeer, Adam Roberts, Katherine Lee, Sharan Narang, Michael Matena, Yanqi Zhou, Wei Li, and Peter J Liu. Exploring the limits of transfer learning with a unified text-to-text transformer. *Journal of machine learning research*, 21(140):1–67, 2020.

Abdur Rahman, Rajat Chawla, Muskaan Kumar, Arkajit Datta, Adarsh Jha, Mukunda NS, and Ishaan Bhola. V-zen: Efficient gui understanding and precise grounding with a novel multimodal llm. *arXiv preprint arXiv:2405.15341*, 2024.

Aditya Ramesh, Mikhail Pavlov, Gabriel Goh, Scott Gray, Chelsea Voss, Alec Radford, Mark Chen, and Ilya Sutskever. Zero-shot text-to-image generation. In *International conference on machine learning*, pp. 8821–8831. Pmlr, 2021.

Dezhi Ran, Hao Wang, Zihe Song, Mengzhou Wu, Yuan Cao, Ying Zhang, Wei Yang, and Tao Xie. Guardian: A runtime framework for llm-based ui exploration. In *Proceedings of the 33rd ACM SIGSOFT International Symposium on Software Testing and Analysis*, pp. 958–970, 2024.

Dezhi Ran, Mengzhou Wu, Hao Yu, Yuetong Li, Jun Ren, Yuan Cao, Xia Zeng, Haochuan Lu, Zexin Xu, Mengqian Xu, et al. Beyond pass or fail: A multi-dimensional benchmark for mobile ui navigation. *arXiv preprint arXiv:2501.02863*, 2025.

ranorex. Ranorex studio: Test automation for gui testing, 2024. URL `https://www.ranorex.com/`. Accessed: 2024-11-05.

Christopher Rawles, Alice Li, Daniel Rodriguez, Oriana Riva, and Timothy Lillicrap. Androidinthewild: A large-scale dataset for android device control. *Advances in Neural Information Processing Systems*, 36: 59708–59728, 2023.

Christopher Rawles, Sarah Clinckemaillie, Yifan Chang, Jonathan Waltz, Gabrielle Lau, Marybeth Fair, Alice Li, William Bishop, Wei Li, Folawiyo Campbell-Ajala, Daniel Toyama, Robert Berry, Divya Tyamagundlu, Timothy Lillicrap, and Oriana Riva. Androidworld: A dynamic benchmarking environment for autonomous agents, 2024. URL `https://arxiv.org/abs/2405.14573`.

Dillon Reis, Jordan Kupec, Jacqueline Hong, and Ahmad Daoudi. Real-time flying object detection with yolov8. *arXiv preprint arXiv:2305.09972*, 2023.

Matthew Renze and Erhan Guven. Self-reflection in llm agents: Effects on problem-solving performance. *arXiv preprint arXiv:2405.06682*, 2024.

Jorge Ribeiro, Rui Lima, Tiago Eckhardt, and Sara Paiva. Robotic process automation and artificial intelligence in industry 4.0–a literature review. *Procedia Computer Science*, 181:51–58, 2021.

Fernando Pastor Ricós, Rick Neeft, Beatriz Marín, Tanja EJ Vos, and Pekka Aho. Using gui change detection for delta testing. In *International Conference on Research Challenges in Information Science*, pp. 509–517. Springer, 2023.

Olivia Rodríguez-Valdés, Tanja EJ Vos, Pekka Aho, and Beatriz Marín. 30 years of automated gui testing: a bibliometric analysis. In *Quality of Information and Communications Technology: 14th International Conference, QUATIC 2021, Algarve, Portugal, September 8–11, 2021, Proceedings 14*, pp. 473–488. Springer, 2021.

Robin Rombach, Andreas Blattmann, Dominik Lorenz, Patrick Esser, and Björn Ommer. High-resolution image synthesis with latent diffusion models. In *Proceedings of the IEEE/CVF conference on computer vision and pattern recognition*, pp. 10684–10695, 2022.

Tim Rosenbach, David Heidrich, and Alexander Weinert. Automated testing of the gui of a real-life engineering software using large language models. In *2025 IEEE International Conference on Software Testing, Verification and Validation Workshops (ICSTW)*, pp. 103–110. IEEE, 2025.

Baptiste Rozière, Jonas Gehring, Fabian Gloeckle, Sten Sootla, Itai Gat, Xiaoqing Ellen Tan, Yossi Adi, Jingyu Liu, Romain Sauvestre, Tal Remez, Jérémy Rapin, Artyom Kozhevnikov, Ivan Evtimov, Joanna Bitton, Manish Bhatt, Cristian Canton Ferrer, Aaron Grattafiori, Wenhan Xiong, Alexandre Défossez, Jade Copet, Faisal Azhar, Hugo Touvron, Louis Martin, Nicolas Usunier, Thomas Scialom, and Gabriel Synnaeve. Code llama: Open foundation models for code, 2024. URL `https://arxiv.org/abs/2308.12950`.

Nicole Rupp, Katrin Peschke, Michael Köppl, David Drissner, and Thole Zuchner. Establishment of low-cost laboratory automation processes using autoit and 4-axis robots. *SLAS technology*, 27(5):312–318, 2022.

Pascal J Sager, Benjamin Meyer, Peng Yan, Rebekka von Wartburg-Kottler, Layan Etaiwi, Aref Enayati, Gabriel Nobel, Ahmed Abdulkadir, Benjamin F Grewe, and Thilo Stadelmann. Ai agents for computer use: A review of instruction-based computer control, gui automation, and operator assistants. *arXiv preprint arXiv:2501.16150*, 2025.

Kabir S Said, Liming Nie, Adekunle A Ajibode, and Xueyi Zhou. Gui testing for mobile applications: objectives, approaches and challenges. In *Proceedings of the 12th Asia-Pacific Symposium on Internetware*, pp. 51–60, 2020.

Harini Sampath, Alice Merrick, and Andrew Peter Macvean. Accessibility of command line interfaces. *Proceedings of the 2021 CHI Conference on Human Factors in Computing Systems*, 2021. URL `https://api.semanticscholar.org/CorpusID:233987139`.

Iqbal H Sarker. Llm potentiality and awareness: a position paper from the perspective of trustworthy and responsible ai modeling. *Discover Artificial Intelligence*, 4(1):40, 2024.

selenium. Selenium: Browser automation, 2024. URL `https://www.selenium.dev/`. Accessed: 2024-11-05.

Mobina Shahbandeh, Parsa Alian, Noor Nashid, and Ali Mesbah. Naviqate: Functionality-guided web application navigation. *arXiv preprint arXiv:2409.10741*, 2024.

Claude E Shannon. Prediction and entropy of printed english. *Bell system technical journal*, 30(1):50–64, 1951.

Zhihong Shao, Peiyi Wang, Qihao Zhu, Runxin Xu, Junxiao Song, Xiao Bi, Haowei Zhang, Mingchuan Zhang, YK Li, Y Wu, et al. Deepseekmath: Pushing the limits of mathematical reasoning in open language models. *arXiv preprint arXiv:2402.03300*, 2024.

Sarath Shekkizhar and Romain Cosentino. Agi is coming... right after ai learns to play wordle, 2025. URL `https://arxiv.org/abs/2504.15434`.

Huawen Shen, Chang Liu, Gengluo Li, Xinlong Wang, Yu Zhou, Can Ma, and Xiangyang Ji. Falcon-ui: Understanding gui before following user instructions. *arXiv preprint arXiv:2412.09362*, 2024a.

Junhong Shen, Atishay Jain, Zedian Xiao, Ishan Amlekar, Mouad Hadji, Aaron Podolny, and Ameet Talwalkar. Scribeagent: Towards specialized web agents using production-scale workflow data, 2024b. URL `https://arxiv.org/abs/2411.15004`.

Zhuocheng Shen. Llm with tools: A survey. *arXiv preprint arXiv:2409.18807*, 2024.

Tianlin Shi, Andrej Karpathy, Linxi Fan, Jonathan Hernandez, and Percy Liang. World of bits: An open-domain platform for web-based agents. In Doina Precup and Yee Whye Teh (eds.), *Proceedings of the 34th International Conference on Machine Learning*, volume 70 of *Proceedings of Machine Learning Research*, pp. 3135–3144. PMLR, 06–11 Aug 2017. URL `https://proceedings.mlr.press/v70/shi17a.html`.

Yucheng Shi, Wenhao Yu, Wenlin Yao, Wenhu Chen, and Ninghao Liu. Towards trustworthy gui agents: A survey. *arXiv preprint arXiv:2503.23434*, 2025.

Noah Shinn, Federico Cassano, Ashwin Gopinath, Karthik Narasimhan, and Shunyu Yao. Reflexion: Language agents with verbal reinforcement learning. *Advances in Neural Information Processing Systems*, 36, 2024.

Brian Sierkowski. Achieving web accessibility. In *Proceedings of the 30th annual ACM SIGUCCS conference on User services*, pp. 288–291, 2002.

Kunal Singh, Shreyas Singh, and Mukund Khanna. Trishul: Towards region identification and screen hierarchy understanding for large vlm based gui agents, 2025. URL `https://arxiv.org/abs/2502.08226`.

smartbear. Testcomplete: Automated ui testing tool, 2024. URL `https://smartbear.com/product/testcomplete/`. Accessed: 2024-11-05.

Yifan Song, Weimin Xiong, Xiutian Zhao, Dawei Zhu, Wenhao Wu, Ke Wang, Cheng Li, Wei Peng, and Sujian Li. Agentbank: Towards generalized llm agents via fine-tuning on 50000+ interaction trajectories. *arXiv preprint arXiv:2410.07706*, 2024a.

Yixiao Song, Katherine Thai, Chau Minh Pham, Yapei Chang, Mazin Nadaf, and Mohit Iyyer. Bearcubs: A benchmark for computer-using web agents. *arXiv preprint arXiv:2503.07919*, 2025.

Yueqi Song, Frank Xu, Shuyan Zhou, and Graham Neubig. Beyond browsing: Api-based web agents. *arXiv preprint arXiv:2410.16464*, 2024b.

Yunpeng Song, Yiheng Bian, Yongtao Tang, Guiyu Ma, and Zhongmin Cai. Visiontasker: Mobile task automation using vision based ui understanding and llm task planning. In *Proceedings of the 37th Annual ACM Symposium on User Interface Software and Technology*, UIST '24, pp. 1–17. ACM, October 2024c. doi: 10.1145/3654777.3676386. URL `http://dx.doi.org/10.1145/3654777.3676386`.

Zirui Song, Yaohang Li, Meng Fang, Zhenhao Chen, Zecheng Shi, Yuan Huang, and Ling Chen. Mmac-copilot: Multi-modal agent collaboration operating system copilot. *arXiv preprint arXiv:2404.18074*, 2024d.

Trisanth Srinivasan and Santosh Patapati. Webnav: An intelligent agent for voice-controlled web navigation. *arXiv preprint arXiv:2503.13843*, 2025.

Zinovia Stefanidi, George Margetis, Stavroula Ntoa, and George Papagiannakis. Real-time adaptation of context-aware intelligent user interfaces, for enhanced situational awareness. *IEEE Access*, 10:23367–23393, 2022.

John Steven, Pravir Chandra, Bob Fleck, and Andy Podgurski. jrapture: A capture/replay tool for observation-based testing. *SIGSOFT Softw. Eng. Notes*, 25(5):158–167, August 2000. ISSN 0163-5948. doi: 10.1145/347636.348993. URL `https://doi.org/10.1145/347636.348993`.

Hongjin Su, Ruoxi Sun, Jinsung Yoon, Pengcheng Yin, Tao Yu, and Sercan O. Arık. Learn-by-interact: A data-centric framework for self-adaptive agents in realistic environments, 2025. URL `https://arxiv.org/abs/2501.10893`.

Chuanneng Sun, Songjun Huang, and Dario Pompili. Llm-based multi-agent reinforcement learning: Current and future directions. *arXiv preprint arXiv:2405.11106*, 2024a.

Jiahui Sun, Zhichao Hua, and Yubin Xia. Autoeval: A practical framework for autonomous evaluation of mobile agents. *arXiv preprint arXiv:2503.02403*, 2025a.

Liangtai Sun, Xingyu Chen, Lu Chen, Tianle Dai, Zichen Zhu, and Kai Yu. Meta-gui: Towards multi-modal conversational agents on mobile gui, 2022. URL `https://arxiv.org/abs/2205.11029`.

Qiushi Sun, Kanzhi Cheng, Zichen Ding, Chuanyang Jin, Yian Wang, Fangzhi Xu, Zhenyu Wu, Chengyou Jia, Liheng Chen, Zhoumianze Liu, et al. Os-genesis: Automating gui agent trajectory construction via reverse task synthesis. *arXiv preprint arXiv:2412.19723*, 2024b.

Quan Sun, Yuxin Fang, Ledell Wu, Xinlong Wang, and Yue Cao. Eva-clip: Improved training techniques for clip at scale. *arXiv preprint arXiv:2303.15389*, 2023.

Yuchen Sun, Shanhui Zhao, Tao Yu, Hao Wen, Samith Va, Mengwei Xu, Yuanchun Li, and Chongyang Zhang. Gui-xplore: Empowering generalizable gui agents with one exploration. *arXiv preprint arXiv:2503.17709*, 2025b.

Al Sweigart. Pyautogui: A cross-platform gui automation python module. GitHub repository, 2024. URL `https://github.com/asweigart/pyautogui`. Accessed: 2024-10-27.

Rehan Syed, Suriadi Suriadi, Michael Adams, Wasana Bandara, Sander JJ Leemans, Chun Ouyang, Arthur HM Ter Hofstede, Inge Van De Weerd, Moe Thandar Wynn, and Hajo A Reijers. Robotic process automation: contemporary themes and challenges. *Computers in Industry*, 115:103162, 2020.

Jihoon Tack, Jaehyung Kim, Eric Mitchell, Jinwoo Shin, Yee Whye Teh, and Jonathan Richard Schwarz. Online adaptation of language models with a memory of amortized contexts. *arXiv preprint arXiv:2403.04317*, 2024.

Maryam Taeb, Amanda Swearngin, Eldon Schoop, Ruijia Cheng, Yue Jiang, and Jeffrey Nichols. Axnav: Replaying accessibility tests from natural language. In *Proceedings of the CHI Conference on Human Factors in Computing Systems*, pp. 1–16, 2024.

Wrick Talukdar and Anjanava Biswas. Improving large language model (llm) fidelity through context-aware grounding: A systematic approach to reliability and veracity. *arXiv preprint arXiv:2408.04023*, 2024.

Weihao Tan, Wentao Zhang, Xinrun Xu, Haochong Xia, Ziluo Ding, Boyu Li, Bohan Zhou, Junpeng Yue, Jiechuan Jiang, Yewen Li, Ruyi An, Molei Qin, Chuqiao Zong, Longtao Zheng, Yujie Wu, Xiaoqiang Chai, Yifei Bi, Tianbao Xie, Pengjie Gu, Xiyun Li, Ceyao Zhang, Long Tian, Chaojie Wang, Xinrun Wang, Börje F. Karlsson, Bo An, Shuicheng Yan, and Zongqing Lu. Cradle: Empowering foundation agents towards general computer control, 2024a. URL `https://arxiv.org/abs/2403.03186`.

Zhaoxuan Tan and Meng Jiang. User modeling in the era of large language models: Current research and future directions. *arXiv preprint arXiv:2312.11518*, 2023.

Zhen Tan, Dawei Li, Song Wang, Alimohammad Beigi, Bohan Jiang, Amrita Bhattacharjee, Mansooreh Karami, Jundong Li, Lu Cheng, and Huan Liu. Large language models for data annotation: A survey. *arXiv preprint arXiv:2402.13446*, 2024b.

Brian Tang and Kang G Shin. Steward: Natural language web automation. *arXiv preprint arXiv:2409.15441*, 2024.

Fei Tang, Yongliang Shen, Hang Zhang, Siqi Chen, Guiyang Hou, Wenqi Zhang, Wenqiao Zhang, Kaitao Song, Weiming Lu, and Yueting Zhuang. Think twice, click once: Enhancing gui grounding via fast and slow systems. *arXiv preprint arXiv:2503.06470*, 2025a.

Fei Tang, Haolei Xu, Hang Zhang, Siqi Chen, Xingyu Wu, Yongliang Shen, Wenqi Zhang, Guiyang Hou, Zeqi Tan, Yuchen Yan, Kaitao Song, Jian Shao, Weiming Lu, Jun Xiao, and Yueting Zhuang. A survey on (m)llm-based gui agents, 2025b. URL `https://arxiv.org/abs/2504.13865`.

Zhengwei Tao, Ting-En Lin, Xiancai Chen, Hangyu Li, Yuchuan Wu, Yongbin Li, Zhi Jin, Fei Huang, Dacheng Tao, and Jingren Zhou. A survey on self-evolution of large language models. *arXiv preprint arXiv:2404.14387*, 2024.

Gemini Team, Rohan Anil, Sebastian Borgeaud, Jean-Baptiste Alayrac, Jiahui Yu, Radu Soricut, Johan Schalkwyk, Andrew M Dai, Anja Hauth, Katie Millican, et al. Gemini: a family of highly capable multimodal models. *arXiv preprint arXiv:2312.11805*, 2023.

Gemma Team, Thomas Mesnard, Cassidy Hardin, Robert Dadashi, Surya Bhupatiraju, Shreya Pathak, Laurent Sifre, Morgane Rivière, Mihir Sanjay Kale, Juliette Love, et al. Gemma: Open models based on gemini research and technology. *arXiv preprint arXiv:2403.08295*, 2024.

Lucas-Andrei Thil, Mirela Popa, and Gerasimos Spanakis. Navigating webai: Training agents to complete web tasks with large language models and reinforcement learning. In *Proceedings of the 39th ACM/SIGAPP Symposium on Applied Computing*, volume 30 of *SAC '24*, pp. 866–874. ACM, April 2024. doi: 10.1145/ 3605098.3635903. URL http://dx.doi.org/10.1145/3605098.3635903.

George Thomas, Alex J Chan, Jikun Kang, Wenqi Wu, Filippos Christianos, Fraser Greenlee, Andy Toulis, and Marvin Purtorab. Webgames: Challenging general-purpose web-browsing ai agents. *arXiv preprint arXiv:2502.18356*, 2025.

Daniel Toyama, Philippe Hamel, Anita Gergely, Gheorghe Comanici, Amelia Glaese, Zafarali Ahmed, Tyler Jackson, Shibl Mourad, and Doina Precup. Androidenv: A reinforcement learning platform for android, 2021a. URL https://arxiv.org/abs/2105.13231.

Daniel Toyama, Philippe Hamel, Anita Gergely, Gheorghe Comanici, Amelia Glaese, Zafarali Ahmed, Tyler Jackson, Shibl Mourad, and Doina Precup. Androidenv: A reinforcement learning platform for android. *arXiv preprint arXiv:2105.13231*, 2021b.

Brandon Trabucco, Gunnar Sigurdsson, Robinson Piramuthu, and Ruslan Salakhutdinov. Towards internet-scale training for agents. *arXiv preprint arXiv:2502.06776*, 2025.

Ada Defne Tur, Nicholas Meade, Xing Han Lù, Alejandra Zambrano, Arkil Patel, Esin Durmus, Spandana Gella, Karolina Stańczak, and Siva Reddy. Safearena: Evaluating the safety of autonomous web agents, 2025. URL https://arxiv.org/abs/2503.04957.

Iulia Turc, Ming-Wei Chang, Kenton Lee, and Kristina Toutanova. Well-read students learn better: On the importance of pre-training compact models. *arXiv preprint arXiv:1908.08962*, 2019.

A Vaswani. Attention is all you need. *Advances in Neural Information Processing Systems*, 2017.

Gaurav Verma, Rachneet Kaur, Nishan Srishankar, Zhen Zeng, Tucker Balch, and Manuela Veloso. Adaptagent: Adapting multimodal web agents with few-shot learning from human demonstrations. *arXiv preprint arXiv:2411.13451*, 2024.

Minh Duc Vu, Han Wang, Zhuang Li, Jieshan Chen, Shengdong Zhao, Zhenchang Xing, and Chunyang Chen. Gptvoicetasker: Llm-powered virtual assistant for smartphone. *arXiv preprint arXiv:2401.14268*, 2024.

Abdul Wali, Saipunidzam Mahamad, and Suziah Sulaiman. Task automation intelligent agents: A review. *Future Internet*, 15(6):196, 2023.

Zhongwei Wan, Xin Wang, Che Liu, Samiul Alam, Yu Zheng, Jiachen Liu, Zhongnan Qu, Shen Yan, Yi Zhu, Quanlu Zhang, et al. Efficient large language models: A survey. *arXiv preprint arXiv:2312.03863*, 2023.

Bowen Wang, Xinyuan Wang, Jiaqi Deng, Tianbao Xie, Ryan Li, Yanzhe Zhang, Gavin Li, Toh Jing Hua, Ion Stoica, Wei-Lin Chiang, Diyi Yang, Yu Su, Yi Zhang, Zhiguo Wang, Victor Zhong, and Tao Yu. Computer agent arena: Compare & test computer use agents on crowdsourced real-world tasks, 2025a.

Bryan Wang, Gang Li, Xin Zhou, Zhourong Chen, Tovi Grossman, and Yang Li. Screen2words: Automatic mobile ui summarization with multimodal learning. In *The 34th Annual ACM Symposium on User Interface Software and Technology*, pp. 498–510, 2021.

Bryan Wang, Gang Li, and Yang Li. Enabling conversational interaction with mobile ui using large language models. In *Proceedings of the 2023 CHI Conference on Human Factors in Computing Systems*, pp. 1–17, 2023a.

Fali Wang, Zhiwei Zhang, Xianren Zhang, Zongyu Wu, Tzuhao Mo, Qiuhao Lu, Wanjing Wang, Rui Li, Junjie Xu, Xianfeng Tang, Qi He, Yao Ma, Ming Huang, and Suhang Wang. A comprehensive survey of small language models in the era of large language models: Techniques, enhancements, applications, collaboration with llms, and trustworthiness, 2024a. URL https://arxiv.org/abs/2411.03350.

Jiaqi Wang, Zhengliang Liu, Lin Zhao, Zihao Wu, Chong Ma, Sigang Yu, Haixing Dai, Qiushi Yang, Yiheng Liu, Songyao Zhang, et al. Review of large vision models and visual prompt engineering. *Meta-Radiology*, pp. 100047, 2023b.

Jiayin Wang, Weizhi Ma, Peijie Sun, Min Zhang, and Jian-Yun Nie. Understanding user experience in large language model interactions. *arXiv preprint arXiv:2401.08329*, 2024b.

Junjie Wang, Yuchao Huang, Chunyang Chen, Zhe Liu, Song Wang, and Qing Wang. Software testing with large language models: Survey, landscape, and vision. *IEEE Transactions on Software Engineering*, 2024c.

Junyang Wang, Haiyang Xu, Haitao Jia, Xi Zhang, Ming Yan, Weizhou Shen, Ji Zhang, Fei Huang, and Jitao Sang. Mobile-agent-v2: Mobile device operation assistant with effective navigation via multi-agent collaboration, 2024d. URL https://arxiv.org/abs/2406.01014.

Junyang Wang, Haiyang Xu, Jiabo Ye, Ming Yan, Weizhou Shen, Ji Zhang, Fei Huang, and Jitao Sang. Mobile-agent: Autonomous multi-modal mobile device agent with visual perception, 2024e. URL https://arxiv.org/abs/2401.16158.

Junyang Wang, Haiyang Xu, Xi Zhang, Ming Yan, Ji Zhang, Fei Huang, and Jitao Sang. Mobile-agent-v: Learning mobile device operation through video-guided multi-agent collaboration. *arXiv preprint arXiv:2502.17110*, 2025b.

Ke Wang, Tianyu Xia, Zhangxuan Gu, Yi Zhao, Shuheng Shen, Changhua Meng, Weiqiang Wang, and Ke Xu. E-ant: A large-scale dataset for efficient automatic gui navigation, 2024f. URL https://arxiv.org/abs/2406.14250.

Lei Wang, Chen Ma, Xueyang Feng, Zeyu Zhang, Hao Yang, Jingsen Zhang, Zhiyuan Chen, Jiakai Tang, Xu Chen, Yankai Lin, et al. A survey on large language model based autonomous agents. *Frontiers of Computer Science*, 18(6):186345, 2024g.

Lu Wang, Fangkai Yang, Chaoyun Zhang, Junting Lu, Jiaxu Qian, Shilin He, Pu Zhao, Bo Qiao, Ray Huang, Si Qin, Qisheng Su, Jiayi Ye, Yudi Zhang, Jian-Guang Lou, Qingwei Lin, Saravan Rajmohan, Dongmei Zhang, and Qi Zhang. Large action models: From inception to implementation, 2024h. URL https://arxiv.org/abs/2412.10047.

Luyuan Wang, Yongyu Deng, Yiwei Zha, Guodong Mao, Qinmin Wang, Tianchen Min, Wei Chen, and Shoufa Chen. Mobileagentbench: An efficient and user-friendly benchmark for mobile llm agents, 2024i. URL https://arxiv.org/abs/2406.08184.

Peng Wang, Shuai Bai, Sinan Tan, Shijie Wang, Zhihao Fan, Jinze Bai, Keqin Chen, Xuejing Liu, Jialin Wang, Wenbin Ge, Yang Fan, Kai Dang, Mengfei Du, Xuancheng Ren, Rui Men, Dayiheng Liu, Chang Zhou, Jingren Zhou, and Junyang Lin. Qwen2-vl: Enhancing vision-language model's perception of the world at any resolution, 2024j. URL https://arxiv.org/abs/2409.12191.

Shuai Wang, Weiwen Liu, Jingxuan Chen, Weinan Gan, Xingshan Zeng, Shuai Yu, Xinlong Hao, Kun Shao, Yasheng Wang, and Ruiming Tang. Gui agents with foundation models: A comprehensive survey, 2024k. URL https://arxiv.org/abs/2411.04890.

Taiyi Wang, Zhihao Wu, Jianheng Liu, Jianye Hao, Jun Wang, and Kun Shao. Distrl: An asynchronous distributed reinforcement learning framework for on-device control agents. *arXiv preprint arXiv:2410.14803*, 2024l.

Weihan Wang, Qingsong Lv, Wenmeng Yu, Wenyi Hong, Ji Qi, Yan Wang, Junhui Ji, Zhuoyi Yang, Lei Zhao, Xixuan Song, Jiazheng Xu, Bin Xu, Juanzi Li, Yuxiao Dong, Ming Ding, and Jie Tang. Cogvlm: Visual expert for pretrained language models, 2024m. URL https://arxiv.org/abs/2311.03079.

Weizhi Wang, Li Dong, Hao Cheng, Xiaodong Liu, Xifeng Yan, Jianfeng Gao, and Furu Wei. Augmenting language models with long-term memory. *Advances in Neural Information Processing Systems*, 36, 2024n.

Wenhao Wang, Zijie Yu, William Liu, Rui Ye, Tian Jin, Siheng Chen, and Yanfeng Wang. Fedmobileagent: Training mobile agents using decentralized self-sourced data from diverse users. *arXiv preprint arXiv:2502.02982*, 2025c.

Wenhao Wang, Zijie Yu, Rui Ye, Jianqing Zhang, Siheng Chen, and Yanfeng Wang. Fedmabench: Benchmarking mobile agents on decentralized heterogeneous user data. *arXiv preprint arXiv:2503.05143*, 2025d.

Xiaoqiang Wang and Bang Liu. Oscar: Operating system control via state-aware reasoning and re-planning. *arXiv preprint arXiv:2410.18963*, 2024.

Yiqin Wang, Haoji Zhang, Jingqi Tian, and Yansong Tang. Ponder & press: Advancing visual gui agent towards general computer control, 2024o. URL https://arxiv.org/abs/2412.01268.

Yufei Wang, Wanjun Zhong, Liangyou Li, Fei Mi, Xingshan Zeng, Wenyong Huang, Lifeng Shang, Xin Jiang, and Qun Liu. Aligning large language models with human: A survey. *arXiv preprint arXiv:2307.12966*, 2023c.

Zhenhailong Wang, Haiyang Xu, Junyang Wang, Xi Zhang, Ming Yan, Ji Zhang, Fei Huang, and Heng Ji. Mobile-agent-e: Self-evolving mobile assistant for complex tasks, 2025e. URL https://arxiv.org/abs/2501.11733.

Ziheng Wang, Jeremy Wohlwend, and Tao Lei. Structured pruning of large language models. *arXiv preprint arXiv:1910.04732*, 2019.

Zilong Wang, Yuedong Cui, Li Zhong, Zimin Zhang, Da Yin, Bill Yuchen Lin, and Jingbo Shang. Officebench: Benchmarking language agents across multiple applications for office automation, 2024p. URL https://arxiv.org/abs/2407.19056.

Ziwei Wang, Weizhi Chen, Leyang Yang, Sheng Zhou, Shengchu Zhao, Hanbei Zhan, Jiongchao Jin, Liangcheng Li, Zirui Shao, and Jiajun Bu. Mp-gui: Modality perception with mllms for gui understanding. *arXiv preprint arXiv:2503.14021*, 2025f.

Zora Zhiruo Wang, Apurva Gandhi, Graham Neubig, and Daniel Fried. Inducing programmatic skills for agentic tasks. *arXiv preprint arXiv:2504.06821*, 2025g.

Jason Wei, Maarten Bosma, Vincent Y Zhao, Kelvin Guu, Adams Wei Yu, Brian Lester, Nan Du, Andrew M Dai, and Quoc V Le. Finetuned language models are zero-shot learners. *arXiv preprint arXiv:2109.01652*, 2021.

Jason Wei, Xuezhi Wang, Dale Schuurmans, Maarten Bosma, Fei Xia, Ed Chi, Quoc V Le, Denny Zhou, et al. Chain-of-thought prompting elicits reasoning in large language models. *Advances in neural information processing systems*, 35:24824–24837, 2022.

Karl Weiss, Taghi M Khoshgoftaar, and DingDing Wang. A survey of transfer learning. *Journal of Big data*, 3:1–40, 2016.

Hao Wen, Yuanchun Li, Guohong Liu, Shanhui Zhao, Tao Yu, Toby Jia-Jun Li, Shiqi Jiang, Yunhao Liu, Yaqin Zhang, and Yunxin Liu. Autodroid: Llm-powered task automation in android. In *Proceedings of the 30th Annual International Conference on Mobile Computing and Networking*, pp. 543–557, 2024a.

Hao Wen, Shizuo Tian, Borislav Pavlov, Wenjie Du, Yixuan Li, Ge Chang, Shanhui Zhao, Jiacheng Liu, Yunxin Liu, Ya-Qin Zhang, and Yuanchun Li. Autodroid-v2: Boosting slm-based gui agents via code generation, 2024b. URL https://arxiv.org/abs/2412.18116.

Hao Wen, Hongming Wang, Jiaxuan Liu, and Yuanchun Li. Droidbot-gpt: Gpt-powered ui automation for android, 2024c. URL https://arxiv.org/abs/2304.07061.

Joel Wester, Tim Schrills, Henning Pohl, and Niels van Berkel. "as an ai language model, i cannot": Investigating llm denials of user requests. In *Proceedings of the CHI Conference on Human Factors in Computing Systems*, pp. 1–14, 2024.

Thomas Wetzlmaier, Rudolf Ramler, and Werner Putschögl. A framework for monkey gui testing. In *2016 IEEE international conference on software testing, verification and validation (ICST)*, pp. 416–423. IEEE, 2016.

Thomas D White, Gordon Fraser, and Guy J Brown. Improving random gui testing with image-based widget detection. In *Proceedings of the 28th ACM SIGSOFT international symposium on software testing and analysis*, pp. 307–317, 2019.

Josephine Wolff, William Lehr, and Christopher S Yoo. Lessons from gdpr for ai policymaking. *Virginia Journal of Law & Technology*, 27(4):2, 2024.

Michael Wornow, Avanika Narayan, Benjamin Viggiano, Ishan S Khare, Tathagat Verma, Tibor Thompson, Miguel Angel Fuentes Hernandez, Sudharsan Sundar, Chloe Trujillo, Krrish Chawla, et al. Wonderbread: A benchmark for evaluating multimodal foundation models on business process management tasks. In *The Thirty-eight Conference on Neural Information Processing Systems Datasets and Benchmarks Track*.

Michael Wornow, Avanika Narayan, Krista Opsahl-Ong, Quinn McIntyre, Nigam Shah, and Christopher Re. Automating the enterprise with foundation models. *Proceedings of the VLDB Endowment*, 17(11): 2805–2812, 2024.

Biao Wu, Yanda Li, Meng Fang, Zirui Song, Zhiwei Zhang, Yunchao Wei, and Ling Chen. Foundations and recent trends in multimodal mobile agents: A survey. *arXiv preprint arXiv:2411.02006*, 2024a.

Bingyang Wu, Yinmin Zhong, Zili Zhang, Gang Huang, Xuanzhe Liu, and Xin Jin. Fast distributed inference serving for large language models. *arXiv preprint arXiv:2305.05920*, 2023a.

Jason Wu, Siyan Wang, Siman Shen, Yi-Hao Peng, Jeffrey Nichols, and Jeffrey P Bigham. Webui: A dataset for enhancing visual ui understanding with web semantics. In *Proceedings of the 2023 CHI Conference on Human Factors in Computing Systems*, pp. 1–14, 2023b.

Jialong Wu, Wenbiao Yin, Yong Jiang, Zhenglin Wang, Zekun Xi, Runnan Fang, Deyu Zhou, Pengjun Xie, and Fei Huang. Webwalker: Benchmarking llms in web traversal, 2025a. URL https://arxiv.org/abs/2501.07572.

Qinchen Wu, Difei Gao, Kevin Qinghong Lin, Zhuoyu Wu, Xiangwu Guo, Peiran Li, Weichen Zhang, Hengxu Wang, and Mike Zheng Shou. Gui action narrator: Where and when did that action take place?, 2024b. URL https://arxiv.org/abs/2406.13719.

Qingyuan Wu, Jianheng Liu, Jianye Hao, Jun Wang, and Kun Shao. Vsc-rl: Advancing autonomous vision-language agents with variational subgoal-conditioned reinforcement learning. *arXiv preprint arXiv:2502.07949*, 2025b.

Qinzhuo Wu, Weikai Xu, Wei Liu, Tao Tan, Jianfeng Liu, Ang Li, Jian Luan, Bin Wang, and Shuo Shang. Mobilevlm: A vision-language model for better intra- and inter-ui understanding, 2024c. URL https://arxiv.org/abs/2409.14818.

Qinzhuo Wu, Wei Liu, Jian Luan, and Bin Wang. Reachagent: Enhancing mobile agent via page reaching and operation. *arXiv preprint arXiv:2502.02955*, 2025c.

Tianyu Wu, Shizhu He, Jingping Liu, Siqi Sun, Kang Liu, Qing-Long Han, and Yang Tang. A brief overview of chatgpt: The history, status quo and potential future development. *IEEE/CAA Journal of Automatica Sinica*, 10(5):1122–1136, 2023c.

Xiongfei Wu, Jiaming Ye, Ke Chen, Xiaofei Xie, Yujing Hu, Ruochen Huang, Lei Ma, and Jianjun Zhao. Widget detection-based testing for industrial mobile games. In *2023 IEEE/ACM 45th International Conference on Software Engineering: Software Engineering in Practice (ICSE-SEIP)*, pp. 173–184. IEEE, 2023d.

Xuansheng Wu, Haiyan Zhao, Yaochen Zhu, Yucheng Shi, Fan Yang, Tianming Liu, Xiaoming Zhai, Wenlin Yao, Jundong Li, Mengnan Du, et al. Usable xai: 10 strategies towards exploiting explainability in the llm era. *arXiv preprint arXiv:2403.08946*, 2024d.

Zhiyong Wu, Chengcheng Han, Zichen Ding, Zhenmin Weng, Zhoumianze Liu, Shunyu Yao, Tao Yu, and Lingpeng Kong. Os-copilot: Towards generalist computer agents with self-improvement, 2024e. URL `https://arxiv.org/abs/2402.07456`.

Zhiyong Wu, Zhenyu Wu, Fangzhi Xu, Yian Wang, Qiushi Sun, Chengyou Jia, Kanzhi Cheng, Zichen Ding, Liheng Chen, Paul Pu Liang, et al. Os-atlas: A foundation action model for generalist gui agents. *arXiv preprint arXiv:2410.23218*, 2024f.

Zongru Wu, Pengzhou Cheng, Zheng Wu, Tianjie Ju, Zhuosheng Zhang, and Gongshen Liu. Smoothing grounding and reasoning for mllm-powered gui agents with query-oriented pivot tasks. *arXiv preprint arXiv:2503.00401*, 2025d.

Zhiheng Xi, Wenxiang Chen, Xin Guo, Wei He, Yiwen Ding, Boyang Hong, Ming Zhang, Junzhe Wang, Senjie Jin, Enyu Zhou, et al. The rise and potential of large language model based agents: A survey. *arXiv preprint arXiv:2309.07864*, 2023.

Xiaobo Xia and Run Luo. Gui-r1: A generalist r1-style vision-language action model for gui agents. *arXiv preprint arXiv:2504.10458*, 2025.

Zhen Xiang, Linzhi Zheng, Yanjie Li, Junyuan Hong, Qinbin Li, Han Xie, Jiawei Zhang, Zidi Xiong, Chulin Xie, Carl Yang, et al. Guardagent: Safeguard llm agents by a guard agent via knowledge-enabled reasoning. *arXiv preprint arXiv:2406.09187*, 2024.

Bin Xiao, Haiping Wu, Weijian Xu, Xiyang Dai, Houdong Hu, Yumao Lu, Michael Zeng, Ce Liu, and Lu Yuan. Florence-2: Advancing a unified representation for a variety of vision tasks, 2023. URL `https://arxiv.org/abs/2311.06242`.

Xiaokui Xiao and Yufei Tao. Personalized privacy preservation. In *Proceedings of the 2006 ACM SIGMOD international conference on Management of data*, pp. 229–240, 2006.

Junlin Xie, Zhihong Chen, Ruifei Zhang, Xiang Wan, and Guanbin Li. Large multimodal agents: A survey. *arXiv preprint arXiv:2402.15116*, 2024a.

Mulong Xie, Sidong Feng, Zhenchang Xing, Jieshan Chen, and Chunyang Chen. Uied: a hybrid tool for gui element detection. In *Proceedings of the 28th ACM Joint Meeting on European Software Engineering Conference and Symposium on the Foundations of Software Engineering*, pp. 1655–1659, 2020.

Tianbao Xie, Fan Zhou, Zhoujun Cheng, Peng Shi, Luoxuan Weng, Yitao Liu, Toh Jing Hua, Junning Zhao, Qian Liu, Che Liu, Leo Z. Liu, Yiheng Xu, Hongjin Su, Dongchan Shin, Caiming Xiong, and Tao Yu. Openagents: An open platform for language agents in the wild, 2023. URL `https://arxiv.org/abs/2310.10634`.

Tianbao Xie, Danyang Zhang, Jixuan Chen, Xiaochuan Li, Siheng Zhao, Ruisheng Cao, Toh Jing Hua, Zhoujun Cheng, Dongchan Shin, Fangyu Lei, Yitao Liu, Yiheng Xu, Shuyan Zhou, Silvio Savarese, Caiming Xiong, Victor Zhong, and Tao Yu. Osworld: Benchmarking multimodal agents for open-ended tasks in real computer environments, 2024b. URL `https://arxiv.org/abs/2404.07972`.

Mingzhe Xing, Rongkai Zhang, Hui Xue, Qi Chen, Fan Yang, and Zhen Xiao. Understanding the weakness of large language model agents within a complex android environment. In *Proceedings of the 30th ACM SIGKDD Conference on Knowledge Discovery and Data Mining*, pp. 6061–6072, 2024.

Chejian Xu, Mintong Kang, Jiawei Zhang, Zeyi Liao, Lingbo Mo, Mengqi Yuan, Huan Sun, and Bo Li. Advweb: Controllable black-box attacks on vlm-powered web agents. *arXiv preprint arXiv:2410.17401*, 2024a.

Hai-Ming Xu, Qi Chen, Lei Wang, and Lingqiao Liu. Attention-driven gui grounding: Leveraging pretrained multimodal large language models without fine-tuning, 2024b. URL `https://arxiv.org/abs/2412.10840`.

Jia Xu, Weilin Du, Xiao Liu, and Xuejun Li. Llm4workflow: An llm-based automated workflow model generation tool. *Proceedings of the 39th IEEE/ACM International Conference on Automated Software Engineering*, 2024c. URL `https://api.semanticscholar.org/CorpusID:273465368`.

Jiajun Xu, Zhiyuan Li, Wei Chen, Qun Wang, Xin Gao, Qi Cai, and Ziyuan Ling. On-device language models: A comprehensive review. *arXiv preprint arXiv:2409.00088*, 2024d.

Kevin Xu, Yeganeh Kordi, Tanay Nayak, Ado Asija, Yizhong Wang, Kate Sanders, Adam Byerly, Jingyu Zhang, Benjamin Van Durme, and Daniel Khashabi. Tur [k] ingbench: A challenge benchmark for web agents. *arXiv preprint arXiv:2403.11905*, 2024e.

Mengwei Xu, Wangsong Yin, Dongqi Cai, Rongjie Yi, Daliang Xu, Qipeng Wang, Bingyang Wu, Yihao Zhao, Chen Yang, Shihe Wang, et al. A survey of resource-efficient llm and multimodal foundation models. *arXiv preprint arXiv:2401.08092*, 2024f.

Nancy Xu, Sam Masling, Michael Du, Giovanni Campagna, Larry Heck, James Landay, and Monica S Lam. Grounding open-domain instructions to automate web support tasks, 2021. URL `https://arxiv.org/abs/2103.16057`.

Tianqi Xu, Linyao Chen, Dai-Jie Wu, Yanjun Chen, Zecheng Zhang, Xiang Yao, Zhiqiang Xie, Yongchao Chen, Shilong Liu, Bochen Qian, Philip Torr, Bernard Ghanem, and Guohao Li. Crab: Cross-environment agent benchmark for multimodal language model agents, 2024g. URL `https://arxiv.org/abs/2407.01511`.

Xiaohan Xu, Ming Li, Chongyang Tao, Tao Shen, Reynold Cheng, Jinyang Li, Can Xu, Dacheng Tao, and Tianyi Zhou. A survey on knowledge distillation of large language models. *arXiv preprint arXiv:2402.13116*, 2024h.

Yibin Xu, Liang Yang, Hao Chen, Hua Wang, Zhi Chen, and Yaohua Tang. Deskvision: Large scale desktop region captioning for advanced gui agents. *arXiv preprint arXiv:2503.11170*, 2025.

Yifan Xu, Xiao Liu, Xueqiao Sun, Siyi Cheng, Hao Yu, Hanyu Lai, Shudan Zhang, Dan Zhang, Jie Tang, and Yuxiao Dong. Androidlab: Training and systematic benchmarking of android autonomous agents, 2024i. URL `https://arxiv.org/abs/2410.24024`.

Yiheng Xu, Dunjie Lu, Zhennan Shen, Junli Wang, Zekun Wang, Yuchen Mao, Caiming Xiong, and Tao Yu. Agenttrek: Agent trajectory synthesis via guiding replay with web tutorials, 2024j. URL `https://arxiv.org/abs/2412.09605`.

Yiheng Xu, Zekun Wang, Junli Wang, Dunjie Lu, Tianbao Xie, Amrita Saha, Doyen Sahoo, Tao Yu, and Caiming Xiong. Aguvis: Unified pure vision agents for autonomous gui interaction, 2024k. URL `https://arxiv.org/abs/2412.04454`.

Tianci Xue, Weijian Qi, Tianneng Shi, Chan Hee Song, Boyu Gou, Dawn Song, Huan Sun, and Yu Su. An illusion of progress? assessing the current state of web agents. *arXiv preprint arXiv:2504.01382*, 2025.

An Yan, Zhengyuan Yang, Wanrong Zhu, Kevin Lin, Linjie Li, Jianfeng Wang, Jianwei Yang, Yiwu Zhong, Julian McAuley, Jianfeng Gao, Zicheng Liu, and Lijuan Wang. Gpt-4v in wonderland: Large multimodal models for zero-shot smartphone gui navigation, 2023a. URL `https://arxiv.org/abs/2311.07562`.

An Yan, Zhengyuan Yang, Wanrong Zhu, Kevin Lin, Linjie Li, Jianfeng Wang, Jianwei Yang, Yiwu Zhong, Julian McAuley, Jianfeng Gao, et al. Gpt-4v in wonderland: Large multimodal models for zero-shot smartphone gui navigation. *arXiv preprint arXiv:2311.07562*, 2023b.

Jianwei Yang, Hao Zhang, Feng Li, Xueyan Zou, Chunyuan Li, and Jianfeng Gao. Set-of-mark prompting unleashes extraordinary visual grounding in gpt-4v. *arXiv preprint arXiv:2310.11441*, 2023.

Jianwei Yang, Reuben Tan, Qianhui Wu, Ruijie Zheng, Baolin Peng, Yongyuan Liang, Yu Gu, Mu Cai, Seonghyeon Ye, Joel Jang, et al. Magma: A foundation model for multimodal ai agents. *arXiv preprint arXiv:2502.13130*, 2025a.

Jiaxi Yang and Haowen Hou. Rwkv-ui: Ui understanding with enhanced perception and reasoning. *arXiv preprint arXiv:2502.03971*, 2025.

Jingkang Yang, Yuhao Dong, Shuai Liu, Bo Li, Ziyue Wang, Haoran Tan, Chencheng Jiang, Jiamu Kang, Yuanhan Zhang, Kaiyang Zhou, et al. Octopus: Embodied vision-language programmer from environmental feedback. In *European Conference on Computer Vision*, pp. 20–38. Springer, 2025b.

Ke Yang, Yao Liu, Sapana Chaudhary, Rasool Fakoor, Pratik Chaudhari, George Karypis, and Huzefa Rangwala. Agentoccam: A simple yet strong baseline for llm-based web agents, 2024a. URL `https://arxiv.org/abs/2410.13825`.

Qi Yang, Weichen Bi, Haiyang Shen, Yaoqi Guo, and Yun Ma. Pixelweb: The first web gui dataset with pixel-wise labels, 2025c. URL `https://arxiv.org/abs/2504.16419`.

Yuhao Yang, Yue Wang, Dongxu Li, Ziyang Luo, Bei Chen, Chao Huang, and Junnan Li. Aria-ui: Visual grounding for gui instructions, 2024b. URL `https://arxiv.org/abs/2412.16256`.

Yulong Yang, Xinshan Yang, Shuaidong Li, Chenhao Lin, Zhengyu Zhao, Chao Shen, and Tianwei Zhang. Security matrix for multimodal agents on mobile devices: A systematic and proof of concept study. *arXiv preprint arXiv:2407.09295*, 2024c.

Yulong Yang, Xinshan Yang, Shuaidong Li, Chenhao Lin, Zhengyu Zhao, Chao Shen, and Tianwei Zhang. Systematic categorization, construction and evaluation of new attacks against multi-modal mobile gui agents, 2025d. URL `https://arxiv.org/abs/2407.09295`.

Shunyu Yao, Howard Chen, John Yang, and Karthik Narasimhan. Webshop: Towards scalable real-world web interaction with grounded language agents. *Advances in Neural Information Processing Systems*, 35: 20744–20757, 2022a.

Shunyu Yao, Jeffrey Zhao, Dian Yu, Nan Du, Izhak Shafran, Karthik Narasimhan, and Yuan Cao. React: Synergizing reasoning and acting in language models. *arXiv preprint arXiv:2210.03629*, 2022b.

Shunyu Yao, Howard Chen, John Yang, and Karthik Narasimhan. Webshop: Towards scalable real-world web interaction with grounded language agents, 2023. URL `https://arxiv.org/abs/2207.01206`.

Shunyu Yao, Dian Yu, Jeffrey Zhao, Izhak Shafran, Tom Griffiths, Yuan Cao, and Karthik Narasimhan. Tree of thoughts: Deliberate problem solving with large language models. *Advances in Neural Information Processing Systems*, 36, 2024.

Eray Yapağcı, Yavuz Alp Sencer Öztürk, and Eray Tüzün. Bugcraft: End-to-end crash bug reproduction using llm agents in minecraft. *arXiv preprint arXiv:2503.20036*, 2025.

Faraz YazdaniBanafsheDaragh and Sam Malek. Deep gui: Black-box gui input generation with deep learning. In *2021 36th IEEE/ACM International Conference on Automated Software Engineering (ASE)*, pp. 905–916. IEEE, 2021.

Jiaming Ye, Ke Chen, Xiaofei Xie, Lei Ma, Ruochen Huang, Yingfeng Chen, Yinxing Xue, and Jianjun Zhao. An empirical study of gui widget detection for industrial mobile games. In *Proceedings of the 29th ACM Joint Meeting on European Software Engineering Conference and Symposium on the Foundations of Software Engineering*, pp. 1427–1437, 2021.

Suyu Ye, Haojun Shi, Darren Shih, Hyokun Yun, Tanya Roosta, and Tianmin Shu. Realwebassist: A benchmark for long-horizon web assistance with real-world users. *arXiv preprint arXiv:2504.10445*, 2025.

Yining Ye, Xin Cong, Shizuo Tian, Jiannan Cao, Hao Wang, Yvu2024gptvoicetaskerujia Qin, Yaxi Lu, Heyang Yu, Huadong Wang, Yankai Lin, et al. Proagent: From robotic process automation to agentic process automation. *arXiv preprint arXiv:2311.10751*, 2023.

Tom Yeh, Tsung-Hsiang Chang, and Robert C Miller. Sikuli: using gui screenshots for search and automation. In *Proceedings of the 22nd annual ACM symposium on User interface software and technology*, pp. 183–192, 2009.

Shukang Yin, Chaoyou Fu, Sirui Zhao, Ke Li, Xing Sun, Tong Xu, and Enhong Chen. A survey on multimodal large language models. *arXiv preprint arXiv:2306.13549*, 2023.

Yiwen Yin, Yu Mei, Chun Yu, Toby Jia-Jun Li, Aamir Khan Jadoon, Sixiang Cheng, Weinan Shi, Mohan Chen, and Yuanchun Shi. From operation to cognition: Automatic modeling cognitive dependencies from user demonstrations for gui task automation. In *Proceedings of the 2025 CHI Conference on Human Factors in Computing Systems*, pp. 1–24, 2025.

Juyeon Yoon, Robert Feldt, and Shin Yoo. Intent-driven mobile gui testing with autonomoufs large language model agents. In *2024 IEEE Conference on Software Testing, Verification and Validation (ICST)*, pp. 129–139. IEEE, 2024.

Haoxuan You, Haotian Zhang, Zhe Gan, Xianzhi Du, Bowen Zhang, Zirui Wang, Liangliang Cao, Shih-Fu Chang, and Yinfei Yang. Ferret: Refer and ground anything anywhere at any granularity. *arXiv preprint arXiv:2310.07704*, 2023.

Keen You, Haotian Zhang, Eldon Schoop, Floris Weers, Amanda Swearngin, Jeffrey Nichols, Yinfei Yang, and Zhe Gan. Ferret-ui: Grounded mobile ui understanding with multimodal llms. In *European Conference on Computer Vision*, pp. 240–255. Springer, 2025.

Shengcheng Yu, Chunrong Fang, Ziyuan Tuo, Quanjun Zhang, Chunyang Chen, Zhenyu Chen, and Zhendong Su. Vision-based mobile app gui testing: A survey. *arXiv preprint arXiv:2310.13518*, 2023.

Yue Yu, Yuchen Zhuang, Jieyu Zhang, Yu Meng, Alexander J Ratner, Ranjay Krishna, Jiaming Shen, and Chao Zhang. Large language model as attributed training data generator: A tale of diversity and bias. *Advances in Neural Information Processing Systems*, 36, 2024.

Tongxin Yuan, Zhiwei He, Lingzhong Dong, Yiming Wang, Ruijie Zhao, Tian Xia, Lizhen Xu, Binglin Zhou, Fangqi Li, Zhuosheng Zhang, et al. R-judge: Benchmarking safety risk awareness for llm agents. *arXiv preprint arXiv:2401.10019*, 2024.

Sukmin Yun, Haokun Lin, Rusiru Thushara, Mohammad Qazim Bhat, Yongxin Wang, Zutao Jiang, Mingkai Deng, Jinhong Wang, Tianhua Tao, Junbo Li, et al. Web2code: A large-scale webpage-to-code dataset and evaluation framework for multimodal llms. *arXiv preprint arXiv:2406.20098*, 2024.

Xia Zeng, Dengfeng Li, Wujie Zheng, Fan Xia, Yuetang Deng, Wing Lam, Wei Yang, and Tao Xie. Automated test input generation for android: are we really there yet in an industrial case? In *Proceedings of the 2016 24th ACM SIGSOFT International Symposium on Foundations of Software Engineering*, FSE 2016, pp. 987–992, New York, NY, USA, 2016. Association for Computing Machinery. ISBN 9781450342186. doi: 10.1145/2950290.2983958. URL https://doi.org/10.1145/2950290.2983958.

Xiaohua Zhai, Basil Mustafa, Alexander Kolesnikov, and Lucas Beyer. Sigmoid loss for language image pre-training. In *Proceedings of the IEEE/CVF International Conference on Computer Vision*, pp. 11975–11986, 2023.

Yuexiang Zhai, Hao Bai, Zipeng Lin, Jiayi Pan, Shengbang Tong, Yifei Zhou, Alane Suhr, Saining Xie, Yann LeCun, Yi Ma, et al. Fine-tuning large vision-language models as decision-making agents via reinforcement learning. *arXiv preprint arXiv:2405.10292*, 2024.

Xian Zhan, Tianming Liu, Lingling Fan, Li Li, Sen Chen, Xiapu Luo, and Yang Liu. Research on third-party libraries in android apps: A taxonomy and systematic literature review. *IEEE Transactions on Software Engineering*, 48(10):4181–4213, 2021.

Chaoyun Zhang, Paul Patras, and Hamed Haddadi. Deep learning in mobile and wireless networking: A survey. *IEEE Communications surveys & tutorials*, 21(3):2224–2287, 2019.

Chaoyun Zhang, Liqun Li, Shilin He, Xu Zhang, Bo Qiao, Si Qin, Minghua Ma, Yu Kang, Qingwei Lin, Saravan Rajmohan, Dongmei Zhang, and Qi Zhang. UFO: A UI-Focused Agent for Windows OS Interaction. *arXiv preprint arXiv:2402.07939*, 2024a.

Chaoyun Zhang, Zicheng Ma, Yuhao Wu, Shilin He, Si Qin, Minghua Ma, Xiaoting Qin, Yu Kang, Yuyi Liang, Xiaoyu Gou, et al. Allhands: Ask me anything on large-scale verbatim feedback via large language models. *arXiv preprint arXiv:2403.15157*, 2024b.

Chaoyun Zhang, Shilin He, Liqun Li, Si Qin, Yu Kang, Qingwei Lin, and Dongmei Zhang. Api agents vs. gui agents: Divergence and convergence. *arXiv preprint arXiv:2503.11069*, 2025a.

Chaoyun Zhang, He Huang, Chiming Ni, Jian Mu, Si Qin, Shilin He, Lu Wang, Fangkai Yang, Pu Zhao, Chao Du, et al. Ufo2: The desktop agentos. *arXiv preprint arXiv:2504.14603*, 2025b.

Chi Zhang, Zhao Yang, Jiaxuan Liu, Yucheng Han, Xin Chen, Zebiao Huang, Bin Fu, and Gang Yu. Appagent: Multimodal agents as smartphone users, 2023a. URL `https://arxiv.org/abs/2312.13771`.

Danqing Zhang, Balaji Rama, Jingyi Ni, Shiying He, Fu Zhao, Kunyu Chen, Arnold Chen, and Junyu Cao. Litewebagent: The open-source suite for vlm-based web-agent applications. *arXiv preprint arXiv:2503.02950*, 2025c.

Danyang Zhang, Zhennan Shen, Rui Xie, Situo Zhang, Tianbao Xie, Zihan Zhao, Siyuan Chen, Lu Chen, Hongshen Xu, Ruisheng Cao, and Kai Yu. Mobile-env: Building qualified evaluation benchmarks for llm-gui interaction, 2024c. URL `https://arxiv.org/abs/2305.08144`.

Jianguo Zhang, Tian Lan, Ming Zhu, Zuxin Liu, Thai Hoang, Shirley Kokane, Weiran Yao, Juntao Tan, Akshara Prabhakar, Haolin Chen, et al. xlam: A family of large action models to empower ai agent systems. *arXiv preprint arXiv:2409.03215*, 2024d.

Jiayi Zhang, Chuang Zhao, Yihan Zhao, Zhaoyang Yu, Ming He, and Jianping Fan. Mobileexperts: A dynamic tool-enabled agent team in mobile devices, 2024e. URL `https://arxiv.org/abs/2407.03913`.

Jiwen Zhang, Jihao Wu, Yihua Teng, Minghui Liao, Nuo Xu, Xiao Xiao, Zhongyu Wei, and Duyu Tang. Android in the zoo: Chain-of-action-thought for gui agents. *arXiv preprint arXiv:2403.02713*, 2024f.

Jiwen Zhang, Jihao Wu, Yihua Teng, Minghui Liao, Nuo Xu, Xiao Xiao, Zhongyu Wei, and Duyu Tang. Android in the zoo: Chain-of-action-thought for gui agents, 2024g. URL `https://arxiv.org/abs/2403.02713`.

Junlei Zhang, Zichen Ding, Chang Ma, Zijie Chen, Qiushi Sun, Zhenzhong Lan, and Junxian He. Breaking the data barrier–building gui agents through task generalization. *arXiv preprint arXiv:2504.10127*, 2025d.

Li Zhang, Shihe Wang, Xianqing Jia, Zhihan Zheng, Yunhe Yan, Longxi Gao, Yuanchun Li, and Mengwei Xu. Llamatouch: A faithful and scalable testbed for mobile ui task automation, 2024h. URL `https://arxiv.org/abs/2404.16054`.

Liang Zhang, Qin Jin, Haoyang Huang, Dongdong Zhang, and Furu Wei. Respond in my language: Mitigating language inconsistency in response generation based on large language models. In *Proceedings of the 62nd Annual Meeting of the Association for Computational Linguistics (Volume 1: Long Papers)*, pp. 4177–4192, 2024i.

Quanjun Zhang, Tongke Zhang, Juan Zhai, Chunrong Fang, Bo-Chen Yu, Weisong Sun, and Zhenyu Chen. A critical review of large language model on software engineering: An example from chatgpt and automated program repair. *ArXiv*, abs/2310.08879, 2023b. URL `https://api.semanticscholar.org/CorpusID:264127977`.

Ruichen Zhang, Mufan Qiu, Zhen Tan, Mohan Zhang, Vincent Lu, Jie Peng, Kaidi Xu, Leandro Z Agudelo, Peter Qian, and Tianlong Chen. Symbiotic cooperation for web agents: Harnessing complementary strengths of large and small llms. *arXiv preprint arXiv:2502.07942*, 2025e.

Shaoqing Zhang, Zhuosheng Zhang, Kehai Chen, Xinbe Ma, Muyun Yang, Tiejun Zhao, and Min Zhang. Dynamic planning for llm-based graphical user interface automation. *arXiv preprint arXiv:2410.00467*, 2024j.

Shengyu Zhang, Linfeng Dong, Xiaoya Li, Sen Zhang, Xiaofei Sun, Shuhe Wang, Jiwei Li, Runyi Hu, Tianwei Zhang, Fei Wu, et al. Instruction tuning for large language models: A survey. *arXiv preprint arXiv:2308.10792*, 2023c.

Xingxuan Zhang, Jiansheng Li, Wenjing Chu, Junjia Hai, Renzhe Xu, Yuqing Yang, Shikai Guan, Jiazheng Xu, and Peng Cui. On the out-of-distribution generalization of multimodal large language models. *arXiv preprint arXiv:2402.06599*, 2024k.

Xinyu Zhang, Huiyu Xu, Zhongjie Ba, Zhibo Wang, Yuan Hong, Jian Liu, Zhan Qin, and Kui Ren. Privacyasst: Safeguarding user privacy in tool-using large language model agents. *IEEE Transactions on Dependable and Secure Computing*, 2024l.

Yanzhe Zhang, Tao Yu, and Diyi Yang. Attacking vision-language computer agents via pop-ups, 2024m. URL `https://arxiv.org/abs/2411.02391`.

Yao Zhang, Zijian Ma, Yunpu Ma, Zhen Han, Yu Wu, and Volker Tresp. Webpilot: A versatile and autonomous multi-agent system for web task execution with strategic exploration. *arXiv preprint arXiv:2408.15978*, 2024n.

Yi-Fan Zhang, Qingsong Wen, Chaoyou Fu, Xue Wang, Zhang Zhang, Liang Wang, and Rong Jin. Beyond llava-hd: Diving into high-resolution large multimodal models. *arXiv preprint arXiv:2406.08487*, 2024o.

Yudi Zhang, Pei Xiao, Lu Wang, Chaoyun Zhang, Meng Fang, Yali Du, Yevgeniy Puzyrev, Randolph Yao, Si Qin, Qingwei Lin, Mykola Pechenizkiy, Dongmei Zhang, Saravan Rajmohan, and Qi Zhang. Ruag: Learned-rule-augmented generation for large language models, 2024p. URL `https://arxiv.org/abs/2411.03349`.

Yue Zhang, Yafu Li, Leyang Cui, Deng Cai, Lemao Liu, Tingchen Fu, Xinting Huang, Enbo Zhao, Yu Zhang, Yulong Chen, Longyue Wang, Anh Tuan Luu, Wei Bi, Freda Shi, and Shuming Shi. Siren's song in the ai ocean: A survey on hallucination in large language models, 2023d. URL `https://arxiv.org/abs/2309.01219`.

Zeyu Zhang, Xiaohe Bo, Chen Ma, Rui Li, Xu Chen, Quanyu Dai, Jieming Zhu, Zhenhua Dong, and Ji-Rong Wen. A survey on the memory mechanism of large language model based agents. *arXiv preprint arXiv:2404.13501*, 2024q.

Zhiping Zhang, Michelle Jia, Hao-Ping Lee, Bingsheng Yao, Sauvik Das, Ada Lerner, Dakuo Wang, and Tianshi Li. "it's a fair game", or is it? examining how users navigate disclosure risks and benefits when using llm-based conversational agents. In *Proceedings of the CHI Conference on Human Factors in Computing Systems*, pp. 1–26, 2024r.

Zhisong Zhang, Tianqing Fang, Kaixin Ma, Wenhao Yu, Hongming Zhang, Haitao Mi, and Dong Yu. Enhancing web agents with explicit rollback mechanisms. *arXiv preprint arXiv:2504.11788*, 2025f.

Zhizheng Zhang, Wenxuan Xie, Xiaoyi Zhang, and Yan Lu. Reinforced ui instruction grounding: Towards a generic ui task automation api, 2023e. URL `https://arxiv.org/abs/2310.04716`.

Zhuohao Zhang, Eldon Schoop, Jeffrey Nichols, Anuj Mahajan, and Amanda Swearngin. From interaction to impact: Towards safer ai agent through understanding and evaluating mobile ui operation impacts. In *Proceedings of the 30th International Conference on Intelligent User Interfaces*, pp. 727–744, 2025g.

Zhuosheng Zhang and Aston Zhang. You only look at screens: Multimodal chain-of-action agents, 2024. URL https://arxiv.org/abs/2309.11436.

Ziniu Zhang, Shulin Tian, Liangyu Chen, and Ziwei Liu. Mmina: Benchmarking multihop multimodal internet agents, 2024s. URL https://arxiv.org/abs/2404.09992.

Andrew Zhao, Daniel Huang, Quentin Xu, Matthieu Lin, Yong-Jin Liu, and Gao Huang. Expel: Llm agents are experiential learners. In *Proceedings of the AAAI Conference on Artificial Intelligence*, volume 38, pp. 19632–19642, 2024a.

Di Zhao, Longhui Ma, Siwei Wang, Miao Wang, and Zhao Lv. Cola: A scalable multi-agent framework for windows ui task automation. *arXiv preprint arXiv:2503.09263*, 2025a.

Haoren Zhao, Tianyi Chen, and Zhen Wang. On the robustness of gui grounding models against image attacks. *arXiv preprint arXiv:2504.04716*, 2025b.

Henry Hengyuan Zhao, Difei Gao, and Mike Zheng Shou. Worldgui: Dynamic testing for comprehensive desktop gui automation, 2025c. URL https://arxiv.org/abs/2502.08047.

Kangjia Zhao, Jiahui Song, Leigang Sha, HaoZhan Shen, Zhi Chen, Tiancheng Zhao, Xiubo Liang, and Jianwei Yin. Gui testing arena: A unified benchmark for advancing autonomous gui testing agent. *arXiv preprint arXiv:2412.18426*, 2024b.

Pengyu Zhao, Zijian Jin, and Ning Cheng. An in-depth survey of large language model-based artificial intelligence agents. *arXiv preprint arXiv:2309.14365*, 2023a.

Wayne Xin Zhao, Kun Zhou, Junyi Li, Tianyi Tang, Xiaolei Wang, Yupeng Hou, Yingqian Min, Beichen Zhang, Junjie Zhang, Zican Dong, et al. A survey of large language models. *arXiv preprint arXiv:2303.18223*, 2023b.

Arman Zharmagambetov, Chuan Guo, Ivan Evtimov, Maya Pavlova, Ruslan Salakhutdinov, and Kamalika Chaudhuri. Agentdam: Privacy leakage evaluation for autonomous web agents. *arXiv preprint arXiv:2503.09780*, 2025.

Boyuan Zheng, Zeyuan Liu, Scott Salisbury, Zheng Du, Xuyan Huang, Qinyuan Zheng, Lee Davis, Michael Lin, Xiaolong Jin, Huan Sun, et al. Agentmonitor: Towards a generalist guardrail for web agent.

Boyuan Zheng, Boyu Gou, Jihyung Kil, Huan Sun, and Yu Su. Gpt-4v(ision) is a generalist web agent, if grounded, 2024a. URL https://arxiv.org/abs/2401.01614.

Boyuan Zheng, Boyu Gou, Scott Salisbury, Zheng Du, Huan Sun, and Yu Su. Webolympus: An open platform for web agents on live websites. In *Proceedings of the 2024 Conference on Empirical Methods in Natural Language Processing: System Demonstrations*, pp. 187–197, 2024b.

Boyuan Zheng, Michael Y Fatemi, Xiaolong Jin, Zora Zhiruo Wang, Apurva Gandhi, Yueqi Song, Yu Gu, Jayanth Srinivasa, Gaowen Liu, Graham Neubig, et al. Skillweaver: Web agents can self-improve by discovering and honing skills. *arXiv preprint arXiv:2504.07079*, 2025a.

Jiani Zheng, Lu Wang, Fangkai Yang, Chaoyun Zhang, Lingrui Mei, Wenjie Yin, Qingwei Lin, Dongmei Zhang, Saravan Rajmohan, and Qi Zhang. Vem: Environment-free exploration for training gui agent with value environment model. *arXiv preprint arXiv:2502.18906*, 2025b.

Junhao Zheng, Chengming Shi, Xidi Cai, Qiuke Li, Duzhen Zhang, Chenxing Li, Dong Yu, and Qianli Ma. Lifelong learning of large language model based agents: A roadmap. *arXiv preprint arXiv:2501.07278*, 2025c.

Longtao Zheng, Zhiyuan Huang, Zhenghai Xue, Xinrun Wang, Bo An, and Shuicheng Yan. Agentstudio: A toolkit for building general virtual agents, 2024c. URL https://arxiv.org/abs/2403.17918.

Longtao Zheng, Rundong Wang, Xinrun Wang, and Bo An. Synapse: Trajectory-as-exemplar prompting with memory for computer control, 2024d. URL `https://arxiv.org/abs/2306.07863`.

Li Zhong and Zilong Wang. A study on robustness and reliability of large language model code generation. *arXiv preprint arXiv:2308.10335*, 2023.

Shuyan Zhou, Frank F Xu, Hao Zhu, Xuhui Zhou, Robert Lo, Abishek Sridhar, Xianyi Cheng, Tianyue Ou, Yonatan Bisk, Daniel Fried, et al. Webarena: A realistic web environment for building autonomous agents. In *The Twelfth International Conference on Learning Representations*.

Yifei Zhou, Qianlan Yang, Kaixiang Lin, Min Bai, Xiong Zhou, Yu-Xiong Wang, Sergey Levine, and Erran Li. Proposer-agent-evaluator (pae): Autonomous skill discovery for foundation model internet agents, 2024a. URL `https://arxiv.org/abs/2412.13194`.

Yuqi Zhou, Shuai Wang, Sunhao Dai, Qinglin Jia, Zhaocheng Du, Zhenhua Dong, and Jun Xu. Chop: Mobile operating assistant with constrained high-frequency optimized subtask planning. *arXiv preprint arXiv:2503.03743*, 2025.

Zixuan Zhou, Xuefei Ning, Ke Hong, Tianyu Fu, Jiaming Xu, Shiyao Li, Yuming Lou, Luning Wang, Zhihang Yuan, Xiuhong Li, et al. A survey on efficient inference for large language models. *arXiv preprint arXiv:2404.14294*, 2024b.

Xizhou Zhu, Yuntao Chen, Hao Tian, Chenxin Tao, Weijie Su, Chenyu Yang, Gao Huang, Bin Li, Lewei Lu, Xiaogang Wang, et al. Ghost in the minecraft: Generally capable agents for open-world environments via large language models with text-based knowledge and memory. *arXiv preprint arXiv:2305.17144*, 2023a.

Yuqi Zhu, Shuofei Qiao, Yixin Ou, Shumin Deng, Ningyu Zhang, Shiwei Lyu, Yue Shen, Lei Liang, Jinjie Gu, and Huajun Chen. Knowagent: Knowledge-augmented planning for llm-based agents. *arXiv preprint arXiv:2403.03101*, 2024a.

Zhaocheng Zhu, Yuan Xue, Xinyun Chen, Denny Zhou, Jian Tang, Dale Schuurmans, and Hanjun Dai. Large language models can learn rules. *arXiv preprint arXiv:2310.07064*, 2023b.

Zichen Zhu, Hao Tang, Yansi Li, Kunyao Lan, Yixuan Jiang, Hao Zhou, Yixiao Wang, Situo Zhang, Liangtai Sun, Lu Chen, et al. Moba: A two-level agent system for efficient mobile task automation. *arXiv preprint arXiv:2410.13757*, 2024b.

Mingchen Zhuge, Changsheng Zhao, Dylan R. Ashley, Wenyi Wang, Dmitrii Khizbullin, Yunyang Xiong, Zechun Liu, Ernie Chang, Raghuraman Krishnamoorthi, Yuandong Tian, Yangyang Shi, Vikas Chandra, and Jurgen Schmidhuber. Agent-as-a-judge: Evaluate agents with agents. 2024. URL `https://api.semanticscholar.org/CorpusID:273350802`.

Daniel Zimmermann and Anne Koziolek. Automating gui-based software testing with gpt-3. In *2023 IEEE International Conference on Software Testing, Verification and Validation Workshops (ICSTW)*, pp. 62–65, 2023a. doi: 10.1109/ICSTW58534.2023.00022.

Daniel Zimmermann and Anne Koziolek. Gui-based software testing: An automated approach using gpt-4 and selenium webdriver. In *2023 38th IEEE/ACM International Conference on Automated Software Engineering Workshops (ASEW)*, pp. 171–174. IEEE, 2023b.

Zhengxia Zou, Keyan Chen, Zhenwei Shi, Yuhong Guo, and Jieping Ye. Object detection in 20 years: A survey. *Proceedings of the IEEE*, 111(3):257–276, 2023.

