# OpenReview forum: "Large Language Model-Brained GUI Agents: A Survey"
_TMLR — Accepted by TMLR_

### Review · Reviewer_Citg · 2025-03-09

**Summary Of Contributions:**

# contributions
The paper is a survey and focuses on documenting existing efforts across various applications of LLMs and VLMs to interacting with varying GUI enviroments: web interfaces, mobile devices, and desktop applications, including cross-platform. The survey touches on the history of systems to interact with GUIs prior to the use of LLMs or VLMs, the components of these systems, datasets used to train them, evaluation metrics and frameworks, and real-world applications. The conclusion summarizes known challenges and future areas of work.

Overall the paper contributes a collection of references to the key areas and a taxonomy of shared features in each of the topics above.


# new knowledge
As a survey new knowledge is not a major focus of the work. The primary contribution is offering a shared resource and capturing trends in specific areas.

**Audience:**

Yes

**Broader Impact Concerns:**

Section 11 on limitations and challenges addresses several relevant topics including privacy, safety, and ethical and regulatory challenges. The points there are reasonable for the topic. The focus is on user-facing applications, so it is reasonable to focus on the end consumer influences.

**Claims And Evidence:**

Yes

**Requested Changes:**

# critical
- Which citations are archival publications?
	- The paper offers many references to other works and other surveys. But many of these are pre-publication (on arxiv, or at least the citations only reference the arxiv version).
	- I worry that I cannot be sure if the systems cited really work, which makes the value of the review lower. At the least the survey should distinguish archival vs non-archival references for readers to make their own informed choice on trusting the references. This could perhaps be through a color code or other symbol that is easily recognized in the text.
	- This is a major barrier to using the work as a canonical reference on the state of the field. Unpublished work may not meet peer review standards. For industry systems (like GPT) it is understandable to not have peer-reviewed references as these systems have passed a "user test" for quality. The same is not true from non-archival academic systems.
- The paper should provide a clearer set of taxonomies or sub-categories to orient the reader from the beginning of the paper and each section. It was hard to follow the organization and often only clear after reaching the take-aways at the end of sections.
- What are the impacts of the growth of more OS-level LLMs and VLMs?
	- Siri / Apple Intelligence and Microsoft CoPilot (as an application) are not addressed, despite their ubiquity and growing capabilities.
	- How will this impact where LLMs / VLMs can be developed or deployed when systems have pre-existing integrated capabilities?
	- This is a pressing topic given the survey focus. At the least these trends should be acknowledge and incorporated into discussions of future directions.
- (tied to the above) What trends are robust to improving LLMs or VLMs?
	- The paper highlights many trends, yet does little to contextualize them to ongoing rapid progress in LLMs and VLMs. It is important to offer some insight on how trends in increasingly baseline VLM or LLM capabilities are solving some problems and not others. This is an important aspect to trends in the area I was seeking from a survey (and part of the TMLR call for surveys).
	- Another trend worth discussion is the rise of inference-time computing and reasoning models. I recognize this is a newer development, but it plays into questions about rising baseline model capabilities.


# strengthen
- Are there datasets that include simulators / environments that agents can interact with?
	- The evaluation section would benefit from highlighting this particular distinction, as evaluation on offline data is often hard to translate to live systems.
- The historical review was fairly light and did not offer much insight. It could be cut without any harm to the main paper.
- The challenges and roadmap would benefit from specifying which areas are unique to GUI agents, as opposed to general to LLMs or VLMs. Many stated areas are general to the LLM and VLM literature.
- (minor) Many of the summary tables (ex: Table 47, 'Project' column) have text overflowing the boundaries. This should be corrected to be more readable.
- (minor) The citations seem to be in the wrong format, where names are not in the appropriate parentheses.
	- This applies to all the in-text references. Here's a clear case from page 8 (section 2.2 Surveys on LLM Agents): "Surveys by Guo et al., Guo et al. (2024b) and Han et al., Han et al. (2024) provide comprehensive overviews of the current landscape, challenges, and future directions in this area."
		- "Guo et al., Guo et al. (2024b)" should be something like "Guo et al. (2024b),"

**Strengths And Weaknesses:**

# strengths
- Covers citations to many systems and prior surveys in the area.
- Provides examples and visuals to make specific areas and applications clear and concrete for readers.


# weaknesses
- It was not possible to easily tell which work is peer-reviewed vs what is pre-publication. This makes it difficult to assess the strength of research in particular topics.
- The sections often read more as a sequence of compressed abstracts, instead of offering much insight or perspective. I found the takeaways to be the most helpful parts, yet these were often fairly high level.
	- Structurally the paper would benefit from offering the key insights or organizing takeaways at the opening of sections, before offering the literature that supports these claims. It was difficult to read as I was struggling to understand the narrative until reaching the end of each section.
- The historical review was fairly light and did not offer much insight. It could be cut without any harm to the main paper.

---

> ### Author Response · Authors · 2025-04-19
>
> Thank you very much for your thorough and insightful feedback. We greatly appreciate your careful review, which has significantly helped us enhance the manuscript. We have thoroughly revised our paper in response to your comments, clearly marking major changes in blue. Below, we provide detailed, point-by-point responses addressing each of your concerns:
>
> 0. **Clarifying Section Structure and Providing Early Insights:**
>
> We agree with your valuable observation regarding readability and narrative clarity. To address this, we have moved the "Takeaways" subsection to precede detailed discussions within each section (5-9). This restructuring allows readers to clearly grasp key insights and overarching narratives upfront, thus facilitating easier comprehension of subsequent detailed analyses.
>
> 1. **Differentiating Peer-Reviewed and Pre-publication Works:**
>
> We sincerely thank the reviewer for this thoughtful and constructive suggestion. We fully agree that distinguishing between peer-reviewed and non-archival works is important for readers who wish to assess the reliability and maturity of different contributions in this rapidly evolving field.
>
> That said, we would like to offer some clarification on our design choices. The domain of LLM-powered GUI agents is advancing at a fast pace, and many influential works in this space have initially appeared as arXiv preprints or technical reports prior to formal publication. In fact, several foundational systems that have already shaped the trajectory of this area (e.g., UFO) are still in the pre-publication stage but have demonstrated strong real-world influence—measured through community adoption, open-source engagement (e.g., GitHub stars), and active downstream usage. Therefore, we believe that excluding such works—or heavily de-emphasizing them purely based on their publication status might obscure key developments in the field.
>
> To address the reviewer’s concern, we deliberately limit in-depth discussions in the main text (Sections 5–9) to a carefully selected set of representative works. The selection criteria include demonstrated impact (e.g., citations, GitHub popularity), methodological novelty, and relevance to the core taxonomy. Works that are less mature or of narrower scope, regardless of their publication status are instead listed in summary tables, which have now been relocated to the appendix to enhance readability. This layered presentation ensures both comprehensiveness and clarity, while also implicitly signaling which contributions the community has found to be most impactful thus far. We hope this revised strategy offers a more nuanced and effective way to help readers gauge the trustworthiness and significance of cited systems—beyond the archival/non-archival binary—and aligns with the reviewer’s goal of making the survey a reliable and canonical reference in the field.
>
> 2. **Clearer Taxonomy and Organization:**
>
> We appreciate your suggestion to enhance readability and orientation through clearer taxonomies. To this end, we have added introductory taxonomy tables (Table 9-13) at the beginning of Sections 5-9, classifying surveyed works by platform and application type. Additionally, Figure 2 provides a structured visual overview of the survey's organizational framework, reinforcing clarity and coherence throughout the paper.
>
> 3. **Acknowledging OS-Level LLM and VLM Developments (e.g., Siri, Apple Intelligence, Microsoft Copilot):**
>
> We acknowledge your critical point regarding the rise of integrated OS-level systems such as Siri and Microsoft Copilot. These agents primarily rely on API-based interactions, limiting their flexibility when applications do not expose sufficient APIs. GUI-based interactions offer broader generalizability and versatility, providing significant enhancement opportunities for these existing OS-level systems. We have now introduced a dedicated subsection (Section 3.4.2) explicitly discussing the interplay between GUI-based agents and existing OS-level integrated agents, examining implications and future development paths.
>
> 4. **Contextualizing Trends Relative to LLM and VLM Improvements:**
>
> Your observation regarding contextualizing trends within rapid advancements in LAMs is very insightful. To address this, we have enriched Section 7 with an additional subsection (Section 7.7) discussing robust and emerging trends related to ongoing improvements in baseline LAMs capabilities, including data-driven enhancements, reinforcement learning approaches, and inference-time computing and reasoning models. This contextual analysis highlights the evolving capabilities and continuing challenges within GUI agent development.

---

> > ### Author Response · Authors · 2025-04-19
> >
> > 5. **Highlighting Live vs. Offline Evaluation Datasets and Environments:**
> >
> > This distinction is indeed important. We have updated both the main text and summary tables (i.e. Table 38-48) to explicitly identify and discuss datasets and evaluation frameworks involving interactive simulators or live environments. This update emphasizes the practical relevance and applicability of evaluation methods discussed.
> >
> > 6. **Streamlining Historical Review:**
> >
> > Following your recommendation, we have condensed the historical review section by integrating key historical context directly into the relevant sections of the manuscript and move extraneous historical details to Appendix Section A that do not contribute directly to the paper's core focus.
> >
> > 7. **Specifying GUI Agent-specific Challenges:**
> >
> > Thank you for highlighting this crucial distinction. While the subsection titles in the Challenges and Roadmap section may appear general, the content is already deeply grounded in the context of GUI agents, supported by specific examples and targeted solutions. To further improve clarity, we have thoroughly revised Section 10 to more explicitly identify and articulate challenges that are unique to GUI agents, clearly differentiating them from broader challenges faced by LLMs and VLMs.
> >
> > 8. **Improving Table Formatting:**
> >
> > We appreciate your careful attention to detail. All tables have been reformatted to ensure clarity, readability, and alignment within the page boundaries.
> >
> > 9. **Correcting Citation Formatting:**
> >
> > Thank you for pointing out the citation formatting inconsistencies. We acknowledge that the discrepancy arose from mixing manual mentions of author names (e.g., xx et al.) with the auto-formatted `\cite` commands, which rendered differently under the TMLR style. To avoid confusion and ensure consistency, we have systematically revised all in-text citations to appear at the end of sentences, removing inline author mentions. This aligns with proper formatting standards and ensures that author names and publication years are consistently presented within parentheses.
> >
> > We sincerely thank you again for your detailed and valuable feedback. We trust these revisions fully address your concerns and significantly improve the manuscript's clarity, accuracy, and overall contribution to the field.

---

### Review · Reviewer_122h · 2025-03-19

**Summary Of Contributions:**

* Structured Comprehensive Survey: The paper defines eight guiding research questions (RQ1–RQ8) ranging from the historical development of GUI agents, architectural components, representative frameworks, datasets for training, specialized “Large Action Models” (LAMs), evaluation metrics, real-world applications, and future research directions.

* Practical "cookbook": the survey distills common design patterns and components of LLM-driven GUI agents. It presents a reference architecture and workflow, as well as enumerates the existing work for data and benchmarks across diverse platforms.

* Analysis of Applications and Impact: The paper explores real-world applications of LLM-brained GUI agents (Section 10), notably in GUI testing and virtual assistants. The survey’s analysis of several applications highlights how LLM-based GUI agents have the potential to democratize GUI automation.

**Audience:**

Yes

**Broader Impact Concerns:**

While the paper discusses the limitations in Section 11 and list about the several papers for each topic on privacy and ethical concerns, it will benefit from a more detailed discussion of the implications Autonomous GUI agents performing critical operations.
Particularly, the survey should to highlight different scenarios where we need accountability (if an agent performs an unintended action (e.g., deleting data), it becomes essential to track decisions and provide an explainable audit trail), or privacy (there is a risk that data might be inadvertently exposed during processing, especially when data is sent to external APIs).

Currently, they are not addressed in the Broader Impact Statement section.

**Claims And Evidence:**

No

**Requested Changes:**

[Critical] Please Clarify/Update:

0. Include a high-level comparison of agent performance or capabilities. add a small summary (in text or a new table) aggregating results from the literature. This is important for the technical contribution, and to show not just what methods exist but how effective they are.
1. Choose a better name than "LLM-brained" that reflects the multimodality of the GUI agents in this survey.
2. Provide clarity when the term "UI" used in the canonical sense, and when it is repurposed for the GUI agent (in sec 5.1 with description of environment, prompt engineering, etc).
3. Provide a cross reference for such phrases if they appear early without prior definition. For example, the term “Large Action Model (LAM)” is explained (an LLM fine-tuned for GUI actions) later in Section 8.2., it is introduced much earlier and also highlighted in the research questions​. The research questions could have a link to the sections where they are addressed.
4. Explain the difference between “Agent Instruction” and “System Prompt” in Section 5.3.
5. Clarify differences between planning, complementary outputs
6. In Section 5.4. Consider renaming “Action Inference” to “Action Prediction” if that better aligns with the underlying process.
7. Consider including tools and function calling schemas, perhaps as part of Complementary Information / miscellaneous part of the prompt.
8. Is Environment Perception in Sec 7.1.1. and Environment Feedback in Sec 5.x the same thing?
9. Provide what context is not considered in the literature yet, say for deployment considerations and practical integration. for e.g., GUIs change frequently with software updates; an agent might break if a button moves. How do frameworks handle this (do they retrain, rely on robust vision, etc.)?

[good to have] Structural fixes:

1. It'll be better to have a few salient works in main discussion and refer to the detailed table in the appendix. Pruning the papers in each category for the main text will also highlight any key takeaways or insights. The goal should be to maintain comprehensiveness but improve readability by trimming excess detail.
2.  In Fig 4, update the architecture workflow so that it shows how the user interacts with the agent, including aspects like request clarification, error recovery, and user control resumption. Specify how the agent prompts users for clarification and how it resumes after user intervention.
3. In Fig 5, provide insights for the readers to understand what challenges and action space each GUI offers, what set of tasks are considered in each setting?


[good to have] Minor typos and fixes:

1. In Sec1.3 (Survey Structure): “Section ~2~ 4 for the evolution of LLM-powered GUI agents”​.
2. In Table 1, ~Trem~ Term for STM
3. “challenges” is misspelled as “chanllenges”
4. Formatting issues, like in tables, excessive length due to hyperlinks, the content goes beyond the margin, why have the "Platform" column when all are agents for that platform in the table?
5. Improve figure and table captions, for e.g., Figure 3 (the evolution timeline) could add that " by year and platform​" and any key takeaway from it.

**Strengths And Weaknesses:**

Strengths:
* overall comprehensiveness
* wide range of papers with up-to-date coverage
* thorough discussion of challenges and future work

Weaknesses:
* lack of clear definitions and connections: While superficially the topics seem to be carefully outlined, there are gaps and confusing repetitions. “LLM-Brained” is an ambiguous term and “LLM-powered” could encompass models with visual capabilities, tool use, and more. The terms introduced in the beginning of the paper do not connect consistently with those in the following sections.
* limited insights on quantitative findings: Evaluation section (Section 9) describes metrics and benchmarks, and the text cites certain papers’ evaluation setups, but the survey does not aggregating how well current agents perform on those benchmarks. For example, it would have been informative to report ranges of success rates on common tasks (e.g. web form filling or app navigation).
* assumption on reader familiarity: For instance, the term “Large Action Model (LAM)” is introduced as a key idea in Section 8 and highlighted in the research questions​. The authors do explain it (an LLM fine-tuned for GUI actions) later in Section 8.2, but the phrase appears early (even in the abstract​) without prior definition.
* overwhelmingly lengthy and redundant: The paper is detailed, but it is hard to find relevant information. The paper repeatedly uses terms like “foundation” and “crucial” without sufficient evidence. There is lack of the clarity for the readers, especially with several long tables interspersed in main content.

---

> ### Author Response · Authors · 2025-04-19
>
> Thank you very much for your thorough and insightful feedback, which significantly helped us improve the manuscript. We have carefully revised our paper in line with your suggestions, highlighting all major changes in blue. Below, we provide detailed point-by-point responses to each of your comments:
>
> 1. **High-Level Comparison of Agent Performance:**
>
> Thank you for highlighting the importance of demonstrating the effectiveness of the surveyed agents. Given the diversity of benchmarks and evaluation methods across different agents, direct numerical comparison in a single unified table is challenging. Instead, we have enhanced Section 6 by adding concise summaries for each agent at the end of each category, emphasizing key results, features and trends, thus clearly conveying their practical effectiveness.
>
> 2. **Clarification of Terminology ("LLM-brained" → "LLM-powered"):**
>
> We agree with your suggestion. The term "LLM-brained" can indeed appear ambiguous and potentially misleading. Following your recommendation, we have consistently replaced "LLM-brained" with the clearer term "LLM-powered" throughout the manuscript, including in the title. Additionally, we've clarified the definitions and ensured consistent terminology usage in all subsequent sections.
>
> 3. **Clarifying Usage of "UI" vs. "GUI":**
>
> This was an important clarification. We've thoroughly reviewed the manuscript, specifically original Section 5.1 (now-renamed Section 4.1), and standardized all instances to consistently use "GUI" to clearly differentiate it from general UI concepts. This should reduce ambiguity and enhance readability.
>
> 4. **Clarifying "Large Action Model (LAM)":**
>
> We acknowledge your concern regarding early mentions of the "Large Action Model (LAM)" without sufficient prior explanation. We have now introduced and clearly defined LAM earlier in footnote, specifically in the introduction (Section 1.2). To avoid confusion, we've also clarified the term’s usage in the abstract and linked relevant research questions directly to their detailed discussions.
>
> 5. **Distinction Between "Agent Instruction" and "System Prompt":**
>
> Thank you for pointing this out. We clarified in original Section 5.3 (now-renamed Section 4.3) that "Agent Instruction" typically encompasses detailed operational guidance tailored to specific agent tasks, whereas the "System Prompt" usually refers to initial context-setting prompts provided directly to the LLM. We further elaborated on scenarios where instructions can dynamically update or augment the system prompt based on environmental feedback or adaptive contexts.
>
> 6. **Clarifying Planning vs. Complementary Outputs:**
>
> Your observation here is well-noted. We expanded original Section 5.4.3 (now-renamed Section 4.4.3) to clearly differentiate "planning," which involves sequentially structured future action predictions to manage complex multi-step tasks, from "complementary outputs," which represent immediate reasoning, intermediate thoughts, or messages directed toward users or other system components.

---

> > ### Author Response · Authors · 2025-04-19
> >
> > 7. **"Action Inference" Renamed to "Action Prediction":**
> >
> > We appreciate your suggestion to use "Action Prediction," as it better aligns with the underlying task. We have updated this terminology throughout original Section 5.4 (now-renamed Section 4.4) to reflect a clearer description of the involved process.
> >
> > 8. **Including Tools and Function Calling Schemas:**
> > Thank you for pointing this out. We believe that Point 4 in Section original Section 5.3 (now-renamed Section 4.3) of the "Action Document" already conveys a similar idea. To improve clarity, we have revised the text to more explicitly highlight the role of tools and function-calling schemas, emphasizing their importance in both GUI agent functionality and prompt construction.
> >
> >
> > 9. **Clarifying "Environment Perception" vs. "Environment Feedback":**
> >
> > Indeed, these terms needed clearer differentiation. In original Section 7.1.1 (now-renamed Section 6.1.1), we clarified that "Environment Perception" broadly covers both the static assessment of environment states (Section 5.2.2) and dynamic "Environment Feedback" capturing changes post-action (Section 5.2.3). This clarification highlights their complementary roles in data collection and model training.
> >
> > 10. **Highlighting Underexplored Context in Literature:**
> >
> > Your suggestion is valuable for emphasizing practical considerations. We've expanded Section original Section 11.7 (now-renamed Section 10.7) to discuss deployment challenges, such as handling frequent GUI changes due to software updates. Specifically, we examine existing approaches involving robust visual models, retraining strategies, and incorporation of software documentation and update logs to maintain agent robustness and adaptability.
> >
> > 11. **Improving Readability and Reducing Redundancy:**
> >
> > Thank you for your insightful suggestion. In Sections 5–9, while we strive for comprehensiveness, we only provide detailed textual discussions of the most representative works. The selection is based on several factors, including demonstrated impact, GitHub popularity (e.g., stars), publication venues, media coverage, and overall visibility within the community. Other works are presented only at a high level in tables, intended primarily for quick reference and lookup.
> >
> > While there is no universally accepted threshold for what constitutes a “foundational” or “crucial” contribution, we have carefully curated the works discussed in the main text based on the authors’ best judgment and the aforementioned criteria. To enhance transparency, we now explicitly state these selection principles in Section 4.8. Additionally, to improve readability and reduce clutter in the main text, we have moved the longer reference tables in these sections to the appendix.
> >
> > 12. **Enhanced Interaction Flow in Fig. 4:**
> >
> > Thank you for highlighting this omission. We revised now-renamed Figure 5 to explicitly depict bidirectional user-agent interaction, clearly showing how the agent prompts for user clarification, manages error recovery, and resumes task execution following user interventions. These changes are further explained in the corresponding narrative in Section 5.1.
> >
> > 13. **Insights on Challenges and Tasks in Fig. 5 (now-renamed Fig. 6):**
> >
> > We agree additional context is beneficial here. We introduced a new summary table (Table 3 in Section 4.2.1), clearly enumerating platform-specific challenges, action spaces, and representative task categories. This provides clearer insights into the distinct GUI contexts and requirements across different platforms.
> >
> > 14. **Minor Typos and Formatting Issues:**
> >
> > We carefully addressed all the minor errors, typos, and formatting concerns you've identified, including corrections such as the "STM" typo, misspelled "challenges," and margin adjustments in tables. Additionally, figure captions now succinctly convey key insights, such as specifying the timeline "by year and platform" in Figure 3.
> >
> > 15. **Detailed Discussion on Privacy and Accountability:**
> > We fully recognize the significance of this concern. In response, we significantly enriched Section 11, adding detailed discussions on critical operational scenarios that demand robust accountability (e.g., audit trails for unintended actions) and privacy considerations (risks of inadvertent data exposure through external API interactions). We also highlight emerging solutions and best practices to address these critical issues, strengthening our survey's practical implications.
> >
> > ---
> >
> >
> > Thank you once again for your detailed and valuable feedback, which has substantially improved our manuscript. We sincerely hope our revisions and explanations adequately address your concerns.

---

### Review · Reviewer_MvVq · 2025-04-05

**Summary Of Contributions:**

This work surveys large language model brained GUI agents. It covers GUI automation progression and history over the years, contains detailed introduction and survey for each component of LLM brained GUI agents, surveys exemplar LLM GUI agents, data collection methods, available data, key techniques related to build LLM GUI agents, evaluation metrics and benchmarks, application of LLM GUI agents, limitation of current works. It provides summarizing insights for each section, provided authors' suggestion about current limitation.

**Audience:**

Yes

**Broader Impact Concerns:**

The survey already touched ethical concern of LLM brained GUI agents and provided suggestions for solution. The survey itself does not raise any concern.

**Claims And Evidence:**

Yes

**Requested Changes:**

1. Multi-platform LLM brained GUI agents is surveyed throughout each section, it would be good to highlight this earlier on more in introduction and motivation (even in title) for the survey to be more useful for people.
2. 1.3 seems redundant given 1.2
3. I think it make sense to provide a slightly more detailed compare and contrast between this survey and other surveys that addresses bot LLM agent and GUI automation as indicated by Table 2. This should help highlight the value of this survey. (a motivation of why this particular survey is needed)
4. In 5.7.4, slightly more detail could be beneficial, for example, how is guidance and rules extracted and represented in LLM based GUI system?
5. Motivation section is cluttered, a more structural motivation section could be more helpful, with each paragraph highlight one aspect.

**Strengths And Weaknesses:**

# Strengths
1. Highly comprehensive, with textbook level breath and reference.
2. It provides takeaways for people to quickly draw insights.
3. It discussed current works' limitations from multi-perspective.
4. It provides some suggestions for current works' limitations, thus, points out future direction.

# Weaknesses

(see requested changes)

---

> ### Author Response · Authors · 2025-04-19
>
> **Response to Reviewer**
>
> Thank you very much for your thoughtful and constructive feedback. We greatly appreciate your insightful suggestions, which have helped us improve the clarity, coherence, and utility of our survey. We have revised the manuscript accordingly and highlighted the changes in blue. Below, we provide point-by-point responses to your comments:
>
> **1. Emphasizing Multi-Platform LLM-Brained GUI Agents in the Introduction**
>
> We appreciate your suggestion to foreground the importance of multi-platform GUI agents earlier in the paper. Indeed, cross-platform generalization is a key challenge and motivation in this domain. To address this, we have revised the introduction (Sections 1.2) to explicitly emphasize the multi-platform nature of modern GUI agents and the unique challenges they pose. We also clarified the significance of supporting diverse operating systems and environments—Windows, macOS, Linux, mobile platforms, and the web—thereby enhancing the survey’s relevance and utility for a broader audience. We have also reflected this theme more clearly in the abstract and keywords to make it immediately visible to readers.
>
> **2. Redundancy Between Sections 1.2 and 1.3**
>
> Thank you for pointing out the redundancy between Sections 1.2 and 1.3. Upon your suggestion, we carefully reviewed the two subsections and found that their content can be better integrated. We have merged Section 1.3 into the latter part of Section 1.2 to streamline the discussion and eliminate duplication, resulting in a more concise and cohesive introduction.
>
> **3. Distinguishing This Survey from Related Work (e.g., Table 2)**
>
> This is an excellent point. To better clarify the novelty and value of our survey, we have substantially expanded the discussion comparing our work with existing surveys (in Section 2.2). Specifically:
> - We provide a more comprehensive and up-to-date overview of the field with over 500 references, spanning foundation models, data sources, system frameworks, benchmarks, evaluation methods, and practical deployments. In contrast, prior surveys often focus on narrower aspects such as web agents or mobile agent only.
> - In addition to narrative summaries, we offer consolidated reference tables for each subdomain, allowing readers to quickly classify and look up key works across platforms and research topics—functioning as a practical "handbook" for researchers and practitioners.
> - We also include foundational background and evaluation taxonomies tailored for newcomers to the field, which are largely missing in prior work.
>
> We believe this makes our survey a more comprehensive, structured, and accessible entry point for the community, and we now make this contribution more explicit in the revised manuscript.
>
> **4. Clarifying Section 5.7.4 on Guidance and Rules in LLM-based GUI Agents**
>
> We agree that original Section 5.7.4 could benefit from deeper technical detail. To improve clarity, we have revised each subsection of the now-renamed Section 4.7.4 to include additional sentences describing how guidance or rules are extracted, structured, and represented. For instance, we now specify whether heuristics are hard-coded, learned from demonstration, or retrieved from documentation, and we cite corresponding methods to support each point. These enhancements provide a more informative view of how LLM-powered agents can self-improve through rule extraction and incorporation.
>
> **5. Improving the Motivation Section (Section 1.1)**
>
> Thank you for highlighting the need for a clearer structure in the motivation section. We have restructured Section 1.1 to better articulate the core motivations behind LLM-powered GUI agents. Each paragraph now focuses on a distinct aspect: (i) the growing complexity of desktop and multi-app workflows, (ii) the limitations of traditional automation tools, (iii) the emergence of LLMs as general-purpose agents, and (iv) the open challenges of robustness, interactivity, and cross-platform generalization. This clearer structure helps readers understand the motivation more systematically from both technical and practical perspectives.
>
> ---
>
> Thank you again for your valuable feedback. We hope our revisions address your concerns and improve the overall quality and clarity of the survey.

---

### Decision · Action_Editor_joqA · 2025-05-31

**Recommendation:** Accept with minor revision

**Comment:**

Reviewers generally agree that this is a timely survey on an important topic. After the rebuttal and revision, the paper is more readable and better presented. The authors should incorporate the reviewer comments into their camera ready version.

**Audience:**

Yes, the AI agent research community would be generally interested.

**Claims And Evidence:**

Yes.